# FULL-GRAPH VS. MINI-BATCH TRAINING: COMPREHENSIVE ANALYSIS FROM A BATCH SIZE AND FAN-OUT SIZE PERSPECTIVE

**Mengfan Liu**[1], **Da Zheng**[2], **Junwei Su**[1*], **Chuan Wu**[1*]
[1]The University of Hong Kong, [2]Ant Group
`ml621@connect.hku.hk, zhengda.zheng@antgroup.com,`
`junweisu.cs@gmail.com, cwu@cs.hku.hk`

## ABSTRACT

Full-graph and mini-batch Graph Neural Network (GNN) training approaches have distinct system design demands, making it crucial to choose the appropriate approach to develop. A core challenge in comparing these two GNN training approaches lies in characterizing their model performance (i.e., convergence and generalization) and computational efficiency. While a batch size has been an effective lens in analyzing such behaviors in deep neural networks (DNNs), GNNs extend this lens by introducing a fan-out size, as full-graph training can be viewed as mini-batch training with the largest possible batch size and fan-out size. However, the impact of the batch and fan-out size for GNNs remains insufficiently explored. To this end, this paper systematically compares full-graph vs. mini-batch training of GNNs through empirical and theoretical analyses from the view points of the batch size and fan-out size. Our key contributions include: 1) We provide a novel generalization analysis using the Wasserstein distance to study the impact of the graph structure, especially the fan-out size. 2) We uncover the non-isotropic effects of the batch size and the fan-out size in GNN convergence and generalization, providing practical guidance for tuning these hyperparameters under resource constraints. Finally, full-graph training does not always yield better model performance or computational efficiency than well-tuned smaller mini-batch settings. The implementation can be found in the github link: `https://github.com/LIUMENGFAN-gif/GNN_fullgraph_minibatch_training`.

## 1 INTRODUCTION

Graph neural networks (GNNs) have demonstrated exceptional performance across diverse machine learning tasks involving graph-structured data (Zhang & Chen, 2018; Xu et al., 2018; Gilmer et al., 2017; Su & Wu, 2025a). A defining characteristic of GNNs is their reliance on the graph structure to facilitate message-passing, enabling the learning of rich node representations from both structural and feature information (Gilmer et al., 2017). Consequently, the computational patterns of GNNs depend strongly on the underlying graph structure, leading to two prominent and distinct paradigms for training GNNs: *full-graph* and *mini-batch* training (Bajaj et al., 2024; Hamilton et al., 2017; Zheng et al., 2022).

Full-graph training and mini-batch training are distinct GNN training paradigms. In full-graph training, the entire graph is processed simultaneously, and each node aggregates information from its neighbors across multiple message-passing layers. In contrast, mini-batch training divides the graph into smaller subgraphs or batches, training the model iteratively on subsets of nodes and their (sampled) local neighborhoods. These paradigms exhibit fundamentally different computational patterns, each requiring distinct system designs, training pipelines, and optimization strategies. For example, full-graph training necessitates efficient communication mechanisms to synchronize aggregations over the entire graph (Md et al., 2021; Peng et al., 2022), whereas mini-batch training demands careful optimizations of CPU-GPU data loading to accommodate frequent batch processing (Chen et al., 2018; Zhu et al., 2019; Liu et al., 2023) . *Understanding the differences between these*

---
*Corresponding authors.

*two paradigms is essential for identifying suitable training methods in specific scenarios and guiding the design of optimised training systems.*

**Existing Gaps.** To systematically investigate the differences between full-graph and mini-batch training, the hyperparameters *batch size* (the number of sampled nodes) and *fan-out size* (the number of neighbors chosen per node at each hop (Hamilton et al., 2017)) offer critical lenses for analyzing GNN performance and computational efficiency, as full-graph training can be viewed as a special case of mini-batch training with maximum batch and fan-out sizes. However, despite increasing attention in the literature, the impact of these hyperparameters remains insufficiently understood. Existing studies typically focus on individual parameters (e.g., batch size or fan-out size independently) (Hu et al., 2021; Yuan et al., 2023) or singular aspects of evaluation (e.g., convergence (Yang et al., 2023; Awasthi et al., 2021), accuracy (Tang & Liu, 2023; Verma & Zhang, 2019; Su & Wu, 2024) , or system efficiency (Naman & Simmhan, 2024)), providing limited insights into the holistic trade-offs between the two paradigms (see Sec. 6 for further discussions). Although recent empirical studies, such as (Bajaj et al., 2024), have attempted comparisons between full-graph and mini-batch training, their results are largely observational and hardware- or environment-dependent, limiting their generalizability. Meanwhile, most of the existing GNN analyses typically rely on strong simplifications, such as infinite-width assumptions that average out per-neuron gradient noise (Yadati, 2022) or linear models with convex losses that remove local optima (Yang et al., 2023; Lin et al., 2023), which obscure the effects of batch sizes or fan-out sizes on training dynamics. Thus, *a critical open question remains: How do the batch size and fan-out size influence the optimization dynamics, generalization capabilities, and computational efficiency of GNN training, particularly when comparing full-graph and mini-batch training paradigms?*

**Challenges.** Comparing full-graph and mini-batch GNN training paradigms presents multiple intertwined challenges. First, while the batch size and fan-out size are useful for analyzing differences between these paradigms, their impacts on model performance and system efficiency inherently depend on the hardware environment used. Therefore, meaningful comparisons necessitate measurement frameworks that are hardware-agnostic and supported by rigorous theoretical analyses. Second, both the computational dynamics of GNNs and the statistical properties of graph data are intrinsically tied to the underlying graph structure, which is directly influenced by choices of batch size and fan-out size. Altering these hyperparameters thus introduces complex interactions, highlighting the need for flexible analytical frameworks that can accurately capture these dynamics. Finally, comprehensively understanding the trade-offs between full-graph and mini-batch training demands frameworks capable of jointly evaluating model efficiency and generalization, ultimately guiding the development of practically optimized systems.

**Contribution.** To address the aforementioned research gap, in this paper, we conduct a systematic study of full-graph and mini-batch GNN training under different batch sizes and fan-out sizes on transductive node classification tasks. The contributions are highlighted as follows.

▷ We characterize the role of the batch size and fan-out size in GNN optimization dynamic analysis (Theorem 1 and 2), extending the settings to irregular graphs and GNNs with non-linear activations, better aligning with the practice. We also provide a novel GNN generalization analysis (Theorem 3) using the Wasserstein distance to investigate the impact of graph structures, especially the fan-out size, where this distance can quantify graph structure differences between training and testing datasets.

▷ We theoretically uncover the non-isotropic impacts of the batch size and the fan-out size in GNN convergence and generalization, where the batch size has a greater impact on GNN optimization dynamics (Obs.1), while the fan-out size more strongly affects GNN generalization (Obs.2). These findings suggest that, under memory constraints, adjusting the batch size is preferable when generalization is the priority, given its more stable effect on generalization. In contrast, tuning the fan-out size is preferable when convergence is the concern, given its more consistent impact on convergence compared to batch size, while setting the fan-out size to moderate values balances convergence and computational efficiency as the magnitude of its impact on convergence decreases with larger values.

▷ We empirically use additional iteration-based convergence metrics for hardware-agnostic comparisons, rather than relying solely on time-based metrics. Experiments on four real-world datasets (Hamilton et al., 2017; Hu et al., 2020) and three GNN models (Zhang et al., 2019; Hamilton et al., 2017; Veličković et al., 2017) validate our theoretical findings. We recommend keeping batch size

below half of the training nodes and the fan-out size under 15 for sparse graphs (Hamilton et al., 2017; Hu et al., 2020) to balance the model performance and computational efficiency.

Our theoretical and empirical findings support that full-graph training does not always yield superior model performance or computational efficiency compared to smaller mini-batch settings. Instead, carefully tuning the batch size and fan-out size in mini-batch settings often leads to better trade-offs, such as faster convergence or improved generalization under resource constraints. These findings provide practical guidance for selecting training paradigms under specific task requirements.

## 2 PRELIMINARIES

**Graph.** Given a homogeneous undirected graph with total $n$ nodes and the maximal degree $d_{\max} \leq n$, set $n_{\text{train}}$ nodes in the training set and $n_{\text{test}}$ nodes in the testing set, with $n = n_{\text{train}} + n_{\text{test}}$. We allow arbitrary subsets of nodes to be selected as the training and testing sets. Let $b \leq n_{\text{train}}$ be the batch size and $\beta \leq d_{\max}$ be the fan-out size in mini-batch training, where uniform neighbor sampling is employed to select neighbors.

Each node is an instance $(\mathbf{x}_i, y_i)$ with feature $\mathbf{x}_i$ and label $y_i$. Let $\mathbf{X} \in \mathbb{R}^{n \times r}$ be the feature matrix, where $\mathbf{x}_i$ is the $i$-th row of $\mathbf{X}$ and $r$ is the feature size. In the transductive learning setting, our task is to predict the labels of nodes $\{\mathbf{x}_i\}_{i=n_{\text{train}}+1}^n$ by the GNN model trained on $\{\mathbf{x}_i\}_{i=1}^n \cup \{y_i\}_{i=1}^{n_{\text{train}}}$. We assume that node features are fixed, and node labels are independently sampled from distributions conditioned on node features, which is widely adopted in the node classification task.

Let $\mathbf{A}$ represent the adjacency matrix of graph. We define $\mathbf{A}_{\text{train}}^{\text{full}} \in \mathbb{R}^{n_{\text{train}} \times n}$ for full-graph training, $\mathbf{A}_{\text{train}}^{\text{mini}} \in \mathbb{R}^{b \times n}$ for mini-batch training, and $\mathbf{A}_{\text{test}} \in \mathbb{R}^{n_{\text{test}} \times n}$ for inference, where $\mathbf{A}_{\text{train}}^{\text{mini}}$ is a submatrix of $\mathbf{A}_{\text{train}}^{\text{full}}$. Let $\mathbf{D}^{\text{in}}$ denote a diagonal in-degree matrix with $\mathbf{D}_{ii}^{\text{in}}$ representing the number of incoming edges to node $i$. We define $\mathbf{D}_{\text{train}}^{\text{in,full}} \in \mathbb{R}^{n_{\text{train}} \times n_{\text{train}}}$ for full-graph training, $\mathbf{D}_{\text{train}}^{\text{in,mini}} \in \mathbb{R}^{b \times b}$ for mini-batch training, and $\mathbf{D}_{\text{test}}^{\text{in}} \in \mathbb{R}^{n_{\text{test}} \times n_{\text{test}}}$ for testing. $\mathbf{D}^{\text{out}} \in \mathbb{R}^{n \times n}$ denotes the respective diagonal out-degree matrix. $\tilde{\mathbf{A}} = \left(\mathbf{D}^{\text{in}} + \mathbf{I}\right)^{-\frac{1}{2}} \left(\mathbf{A} + \mathbf{I}\right) \left(\mathbf{D}^{\text{out}} + \mathbf{I}\right)^{-\frac{1}{2}}$ is the respective normalized adjacency matrix with self-loops, where self-loops ensure that each node retains its own features during aggregation, improving the model's learning ability. Here $\tilde{\mathbf{a}}_i$ denotes the $i$-th row of $\tilde{\mathbf{A}}$.

**GNN model.** Motivated by recent theoretical advances in understanding GNNs (Su & Wu, 2025b; Awasthi et al., 2021), we analyze the training dynamics using a one-layer GNN model. This model serves as a powerful and well-established testbed for capturing phenomena arising from finite width and nonlinearity of GNNs. Its simplicity in model depth provides the analytical flexibility necessary to precisely characterize how batch size and fan-out size affect GNN training dynamics. In Appendix H, we further discuss how our analyses and results generalize to multi-layer settings. Concretely, let $\mathbf{W} \in \mathbb{R}^{h \times r}$ be the learnable model parameters of the GNN model and $\mathbf{W}^* \in \mathbb{R}^{h \times r}$ be the ground truth of $\mathbf{W}$, where $\mathbf{w}_i$ is the $i$-th row of $\mathbf{W}$ and $h$ is the finite hidden dimension. We study a one-layer GNN with the ReLU activation, and define the output immediately after the first layer as $\mathbf{z}_i = \sigma\left(\tilde{\mathbf{a}}_{\text{train},i} \mathbf{X} \mathbf{W}^\top\right), \forall i \in$ training set, where $\sigma(x) = \max(x, 0)$ is the ReLU activation function, and the term $\tilde{\mathbf{a}}_{\text{train},i} \mathbf{X}$ represents the embedding aggregation on node $i$. This first-layer output may be followed by task-specific post-processing (e.g., a linear projection in binary classification). Similarly, during inference, the output of the first layer is given by $\mathbf{z}_i = \sigma\left(\tilde{\mathbf{a}}_{\text{test},i} \mathbf{X} \mathbf{W}^\top\right), \forall i \in$ testing set.

In this paper, we use $\|\cdot\|_2, \|\cdot\|$ and $\|\cdot\|_F$ to denote the 2-norm of vector, spectral norm of matrix and Frobenius norm of vector, respectively. For two sequences $\{p_n\}$ and $\{q_n\}$, we use $p_n = O(q_n)$ to denote that $p_n \leq C_1 q_n$ for some absolute constant $C_1 > 0$. The notation table is in Appendix A.

## 3 OPTIMIZATION DYNAMIC

We present our theoretical studies on the GNN optimization dynamics. First, the optimization setup is introduced, representing how to handle interactions between batch size and fan-out size in optimization dynamics (Sec. 3.1). Next, we show the convergence results, answering our research question in GNN optimization dynamic. We then reveal an interesting observation, yielding actionable implications for accelerating convergence under memory constraints (Sec. 3.2).

### 3.1 OPTIMIZATION SETUP

**Optimization algorithms.** We aim to minimize the empirical risk $\hat{L}_{\text{train}}\left(\mathbf{W}, \tilde{\mathbf{A}}_{\text{train}}\right) = \frac{1}{n_{\text{train}}} \sum_{i \in \text{training set}} l\left(\mathbf{W}, \tilde{\mathbf{a}}_{\text{train},i}\right)$, where $l(\cdot)$ denotes the loss function. In practice, Cross-Entropy

(CE) and Mean Squared Error (MSE) are the most commonly used losses. Under *full-graph* training settings, the model parameters are updated via gradient descent (GD) as $\mathbf{W}_{t+1}^{full} = \mathbf{W}_t^{full} - \eta_t \nabla_{\mathbf{W}_t^{full}} \hat{L}_{train} \left( \mathbf{W}_t^{full}, \mathbf{A}_{train}^{full} \right)$, where $\eta_t > 0$ is the learning rate at the $t$-th training iteration. Under *mini-batch* training settings, the model parameters are updated via stochastic gradient descent (SGD) as $\mathbf{W}_{t+1}^{mini} = \mathbf{W}_t^{mini} - \eta_t \hat{\mathbf{G}}_t$, where $\hat{\mathbf{G}}_t = \frac{1}{b} \sum_{i \in \text{sampled nodes}} \nabla_{\mathbf{W}_{t+1}^{mini}} l \left( \mathbf{W}_t^{mini}, \tilde{\mathbf{a}}_{train,i}^{mini} \right)$ denotes the stochastic gradient at the $t$-th training iteration.

**Handling interactions between batch size and fan-out size in optimization dynamic.** To handle these interactions, we isolate the impact of the graph structure in the loss and gradient expressions. A key challenge is that the nonlinear activation (e.g., ReLU) processes aggregated node features as input, making these expressions analytically intractable. To overcome this, we decouple the aggregated node features from the activation function. For instance, we extract the aggregation from the ReLU function by reformulating squared loss terms, or rewrite the ReLU function using a position-wise 0/1 indicator matrix that can directly multiply the aggregated node features.

## 3.2 CONVERGENCE RESULTS

Building on the aforementioned setup in Sec 3.1, we study GNN convergence results under suitable assumptions on the distribution of node features as well as the boundedness of the feature matrix norm, the ground truth parameter norm and the separation between aggregated node features with different labels in the training data (see Assumptions B.1.-B.2. in Appendix B and Assumption E.1. in Appendix E), with detailed proofs provided in Appendix B-E.

**Theorem 1.** *(Convergence of Mini-batch Training with MSE) Suppose $\mathbf{W}^{mini}$ are generated by Gaussian initialization. Under Assumptions B.1. and B.2, if the fan-out size satisfies $C_1^{mini} \leq \beta \leq C_2^{mini} b^{\frac{3}{4}}$ for constants $C_1^{mini}, C_2^{mini} \in (0,1)$ to ensure a sparser adjacency than a fully connected graph, then with high probability, $L_{train} \left( \mathbf{W}_T^{mini}, \mathbf{A}_{train}^{mini} \right) \leq \epsilon$ for any $\epsilon \in (0,1)$, provided that the number of iterations $T = O \left( n_{train} h^2 b^{\frac{5}{2}} \beta^{-\frac{1}{2}} \epsilon^{-1} \log \left( h^2 \epsilon^{-1} \right) \right)$ under the mini-batch training.*

**Theorem 2.** *(Convergence of Mini-batch Training with CE) Suppose $\mathbf{W}^{mini}$ are generated by Gaussian initialization. Under Assumptions B.1. and E.1, if the hidden dimension of a one-round GNN satisfies $h = \Omega \left( \log \left( n_{train} \right) \beta^{-1} \left( n_{train}^2 + \epsilon^{-1} \right) \right)$ to ensure the finite width, then with high probability, $\hat{L}_{train} \left( \mathbf{W}_T^{mini}, \mathbf{A}_{train}^{mini} \right) \leq \epsilon$ for any $\epsilon \geq 0$, provided that the number of iterations $T = O \left( n_{train}^2 \left( \log \left( n_{train} \right) \right)^{\frac{1}{2}} \alpha^{-2} b^{-1} \beta^{-\frac{5}{2}} \left( n_{train}^2 + \epsilon^{-1} \right) \right)$ under the mini-batch training.*

When the fan-out size $\beta$ reaches $d_{\max}$ and the batch size $b$ reaches $n_{\text{train}}$, the upper bound on the number of iterations to convergence in mini-batch training matches that of full-graph training (see Theorem B.4. under MSE in Appendix B and Theorem D.2. under CE in Appendix D).

*Remark 3.1.* Our theoretical results show that increasing the batch size $b$ for a fixed fan-out size leads to more iterations to convergence under MSE (Theorem 1), but fewer iterations under CE (Theorem 2) in the mini-batch setting of one-round GNNs, different from DNN training. In contrast, increasing the fan-out size $\beta$ under a fixed batch size consistently reduces the number of iterations required for convergence under both MSE (Theorem 1) and CE (Theorem 2).

*Remark 3.2.* Our theoretical analysis reveals that the magnitude of the fan-out size's impact on GNN convergence jointly depends on the batch size $b$ and the fan-out size $\beta$, diminishing as either $b$ (under CE) or $\beta$ (under MSE and CE) grows. The magnitude of this impact can be characterized by the absolute slope $|\partial T/\partial \beta|$ of the number of iterations $T$ for convergence with respect to the fan-out size $\beta$, where a steeper slope indicates a stronger impact. Specifically, Theorem 1. gives $|\partial T/\partial \beta| = O \left( \beta^{-3/2} b^{5/2} \right)$ under MSE and Theorem 2. gives $|\partial T/\partial \beta| = O \left( \beta^{-7/2} b^{-1} \right)$ under CE .

**Answering our research question:** Remark 3.1. and Remark 3.2. represent the impact and interplay of the batch size and the fan-out size in the GNN optimization dynamic. Therefore, we conclude that full-graph training does not always provide superior convergence speed than smaller mini-batch settings, especially under MSE.

Furthermore, we present an interesting observation, providing insights into accelerating GNN convergence under memory constraints.

**Obs.1: GNN convergence is more sensitive to batch size than to fan-out size.** Remark 3.1. highlights a stronger dependence of GNN convergence on batch size $b$ than on fan-out size $\beta$, as a larger

batch size $b$ leads to opposite convergence trends under MSE and CE, while increasing the fan-out size $\beta$ exhibits a consistent trend. This observation cannot be fully interpreted by the popular explanation of DNNs, which posits that increasing the batch size reduces gradient variance, resulting in fewer iterations to converge (Cong et al., 2021a; Liu et al., 2024) . We further consider the impact of message passing on the loss and gradient, providing the interpretation of Obs.1. in Appendix F.

**Implication for accelerating convergence.** Under memory constraints, Obs.1. suggests that adjusting the fan-out size $\beta$ offers a more reliable way to accelerate GNN convergence, as the fan-out size $\beta$ keeps the same convergence trends under both MSE and CE. To tune the fan-out size $\beta$, Remark 3.2. highlights that a moderate value of $\beta$ provides a practical balance between convergence and computational efficiency, as the reduction in the number of iterations for convergence becomes smaller when increasing $\beta$ beyond moderate values, particularly with large batches under CE.

## 4  GENERALIZATION OF MINI-BATCH TRAINING

We represent our theoretical study on GNN generalization. First, problem setup is introduced, representing how to isolate the impacts of batch size and fan-out size in generalization by employing Wasserstein distance (Sec. 4.1). Next, we show the generalization result, answering our research question in GNN generalization. We then present an interesting observation, yielding the actionable implication for improving generalization under memory constraints (Sec. 4.2).

### 4.1  PROBLEM SETUP

**Basic setup.** We aim to bound the generalization gap between the expected testing risk and the empirical training risk under the mini-batch training settings, where the expected testing risk is given by $L_{\text{test}}\left(\mathbf{W}^{\text{mini}}, \tilde{\mathbf{A}}_{\text{test}}^{\text{full}}\right) = \mathbb{E}\left[\frac{1}{n_{\text{test}}} \sum_{i \in \text{test set}} l\left(\mathbf{W}^{\text{mini}}, \tilde{\mathbf{a}}_{\text{test},i}^{\text{full}}\right)\right]$, and the empirical training risk is expressed as $\hat{L}_{\text{train}}\left(\mathbf{W}^{\text{mini}}, \tilde{\mathbf{A}}_{\text{train}}^{\text{mini}}\right) = \frac{1}{n_{\text{train}}} \sum_{i \in \text{training set}} l\left(\mathbf{W}^{\text{mini}}, \tilde{\mathbf{a}}_{\text{train},i}^{\text{mini}}\right)$. Note that inference utilizes all testing neighbors across the entire graph, whereas mini-batch training relies on sampled neighbors within limited hops. We then employ the Wasserstein distance (Kantorovich, 1960) to quantify the difference in graph structures between training and testing datasets, as the Wasserstein distance effectively measures differences in non-i.i.d. data, particularly regarding geometric variations.

**Definition 1.** (Distance between Training Set and Testing Set). Define the distance from the training set to the testing set as the Wasserstein distance given by $\Delta\left(\beta, b\right) = \left\{\inf_{\theta \in \Theta[\rho_{\text{train}}, \rho_{\text{test}}]} \sum_{i \in \text{train set}} \sum_{j \in \text{test set}} \theta_{i,j} \delta\left(y_i, y_j, \beta, b\right)\right\}$, where $\rho_{\text{train}}\left(y_i\right)$ and $\rho_{\text{test}}\left(y_i\right)$ denote the probability of $y_i$ appearing in training and testing sets, respectively. $\Theta[\rho_{\text{train}}, \rho_{\text{test}}]$ is the joint probability of $\rho_{\text{train}}$ and $\rho_{\text{test}}$. The infimum in the first equality is conditioned on $\sum_{j \in \text{test set}} \theta_{i,j} = \rho_{\text{train}}\left(y_i\right), \sum_{i \in \text{training set}} \theta_{i,j} = \rho_{\text{test}}\left(y_j\right), \theta_{i,j} \geq 0$. $\delta\left(\mathbf{y}_i, \mathbf{y}_j, \beta, b\right)$ is the distance function of any two points from training and testing sets, respectively.

We set $\delta\left(\mathbf{y}_i, \mathbf{y}_j, \beta, b\right) = \frac{C_\delta h^2}{n_{\min}}\left(\delta_{i,j}^{\text{full}} + \delta_i^{\text{full-mini}}\right)$ with a constant $C_\delta > 0$, $n_{\min} = \min\{n_{\text{train}}, n_{\text{test}}\}$ and $\delta_i^{\text{full}} = \left\|\tilde{\mathbf{a}}_{\text{test},j}^{\text{full}} - \tilde{\mathbf{a}}_{\text{train},i}^{\text{full}}\right\|_F^2 + 2\left\|\tilde{\mathbf{a}}_{\text{test},j}^{\text{full}}\right\|_F^2$, as a constant, mainly capturing the difference of distributions between the training and testing data in full-graph training. $\delta_i^{\text{full-mini}} = \left\|\tilde{\mathbf{a}}_{\text{train},i}^{\text{full}} - \tilde{\mathbf{a}}_{\text{train},i}^{\text{mini}}\right\|_F^2$ reflects the structural difference between full-graph and mini-batch graphs per node during training.

**Isolating the impacts of batch size and fan-out size in generalization.** To isolate these impacts, we focus on the discrepancy $U$ between expected training and testing losses before training, which is the only term for non-i.i.d. graph data in our generalization analysis, with detailed proof in Appendix M. Since the training and testing datasets are split beforehand, $U$ depends on the structural difference between training and testing graphs, which we quantify using the Wasserstein distance $\Delta\left(\beta, b\right)$. We show that greater similarity between training and testing graph structures leads to a smaller $U$.

### 4.2  GENERALIZATION RESULT

Building on the aforementioned setup in Sec 4.1, we use the Wasserstein distance to study the generalization result in PAC-Bayesian framework (McAllester, 2003) under mini-batch GNN training with MSE, given suitable assumptions on the boundedness of the Frobenius norm of the feature matrix and the parameter norm (see Assumptions G.1. and G.2. and the detailed proof in Appendix G).

**Theorem 3.** *Suppose $\mathbf{W}^{\text{mini}}$ are generated by Gaussian initialization. Under Assumptions G.1. and G.2, with high probability, for the posterior distribution $\mathcal{Q}$ over hypothe-*

*sis space in the mini-batch training settings with MSE, we have $L_{test}\left(\mathbf{W}^{mini}, \tilde{\mathbf{A}}_{test}^{full}; \mathcal{Q}\right) - \hat{L}_{train}\left(\mathbf{W}^{mini}, \tilde{\mathbf{A}}_{train}^{mini}; \mathcal{Q}\right) = O\left(\frac{1}{n_{train}} + \Delta\left(\beta, b\right)\right)$, where $\Delta(\beta, b_1) \leq \Delta(\beta, b_2)$ with $b_1 \geq b_2$, $\Delta(\beta, b) \propto \sum_{i \in training\ set} \sum_{j \in testing\ set} \theta_{i,j} \delta_i^{full-mini}, \theta_{i,j} \in \Theta[\rho_{train}, \rho_{test}]$ and $\delta_i^{full-mini}$ has an overall non-increasing trend as the fan-out size $\beta$ grows but small non-monotonic fluctuations exist. The posterior distribution $\mathcal{Q}$ represents the distribution of model parameters after training, and the hypothesis space denotes all possible models.*

*Remark 4.1.* Theorem 3 reveals that increasing either the batch size $b$ or the fan-out size $\beta$ improves the GNN generalization. This is because the role of $b$ and $\beta$ in GNN generalization is captured by the Wasserstein distance $\Delta(\beta, b)$, where larger $\Delta(\beta, b)$ leads to poorer generalization performance. In Definition 1, the Wasserstein distance $\Delta(\beta, b)$ is proportional to the weighted sum of $\delta_i^{full-mini}$ (i.e., the structural difference between full-graph and mini-batch graphs per node during training) over all training nodes , where $\delta_i^{full-mini}$ decreases with either the batch size $b$ or the fan-out size $\beta$, though slightly non-monotonic fluctuations exist when varying $\beta$.

**Answering our research question:** Remark 4.1. represents how the batch size and the fan-out size characterize GNN generalization via the Wasserstein distance $\Delta(\beta, b)$. While full-graph training is expected to outperform smaller mini-batch settings, we remain cautious about the degradation in generalization performance at very large batch sizes or fan-out sizes, as similar issues have been observed in DNNs (You et al., 2019; 2017) . We conduct an empirical study for further investigation.

In addition, we interpret an interesting observation, providing the implication for improving GNN generalization under memory constraints.

**Obs.2: GNN generalization is more sensitive to fan-out size than to batch size.** While increasing the fan-out size $\beta$ and the batch size $b$ both help align the mini-batch with the full graph during training, $\beta$ has a greater impact on the generalization by directly controlling receptive field of each training node. Based on Remark 4.1, this can be interpreted using the Wasserstein distance $\Delta(\beta, b)$, which increases the weighted sum of $\delta_i^{full-mini}$ over all training nodes. A larger $\beta$ can include unsampled but valid edges, turning zero terms $\tilde{\mathbf{a}}_{train,i}^{mini}$ into non-zero values in $\delta_i^{full-mini}$, potentially causing slight non-monotonic fluctuations. In contrast, increasing $b$ does not introduce these edges, as all training nodes are included during summation of $\delta_i^{full-mini}$. With the more complex impact of $\beta$ in $\Delta(\beta, b)$, we conclude that GNN generalization is more sensitive to fan-out size $\beta$ than to batch size $b$ (see Appendix M for the detailed proof).

**Implication for improving generalization.** Under memory constraints, Obs.2. suggests that adjusting the batch size $b$ offers a more stable way to improve GNN generalization, as the batch size $b$ introduces less non-monotonic fluctuations than the fan-out size $\beta$.

## 5 EMPIRICAL STUDY

We first explain the rationale for using the metrics (e.g., iteration-to-accuracy) in Sec. 5.1. We validate Remarks 3.1 - 3.2. and Obs.1. on GNN convergence (Sec. 5.2), and Remark 4.1. and Obs.2. on GNN generalization with the discussion about performance degradation (Sec. 5.3). We compare computational efficiency across varying batch sizes and fan-out sizes, answering our research question in computational efficiency (Sec. 5.4). Finally, we present an overall comparison of generalization performance between full-graph and mini-batch training after tuning batch size and fan-out size, yielding implications for tuning these two hyperparameters (Sec. 5.5).

**Results overview.** Non-isotropic impacts of batch size and fan-out size exist in model performance (i.e., generalization and convergence) and computational efficiency. Full-graph training does not always yield superior model performance or computational efficiency compared to well-tuned smaller mini-batch settings. Carefully tuning the batch size and the fan-out size in mini-batch settings often achieves more favorable trade-offs, such as faster convergence or better generalization.

**Datasets and models:** We conduct experiments on four real-world datasets: reddit (Hamilton et al., 2017), ogbn-arxiv (Hu et al., 2020), ogbn-products (Hu et al., 2020) and ogbn-papers100M (Hu et al., 2020). We train three representative GNN models: GCN (Zhang et al., 2019), GraphSAGE (Hamilton et al., 2017) with mean aggregation, and GAT (Veličković et al., 2017) with 2 heads for ogbn-papers100M and 4 heads for the other datasets. See more training settings in Appendix N.

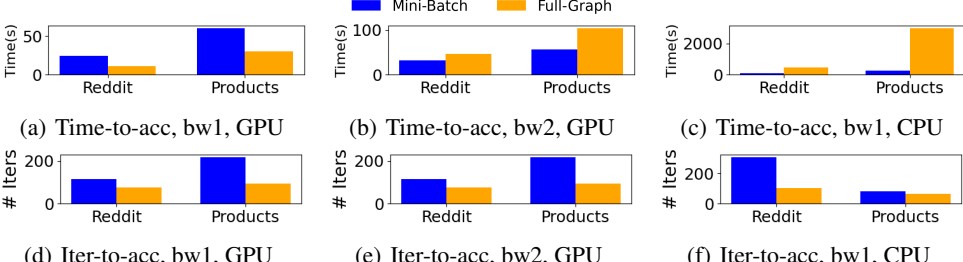

Figure 1: Time-to-acc and iteration-to-acc in mini-batch and full-graph training with varying bandwidths (i.e., two inter-GPU bandwidth values: bw1=infinity > bw2=900GB/s) and computational capacities (i.e., GPU with 40GB of memory and CPU with 512GB of host memory ).

## 5.1  METRIC: ITERATION-TO-ACCURACY

We evaluate convergence performance using three metrics: iteration-to-loss (i.e., the number of iterations to reach a target training loss), iteration-to-accuracy (i.e., the number of iterations to reach a target validation accuracy), and time-to-accuracy (i.e., the time to reach a target validation accuracy). Since iteration-to-loss is from the theoretical analysis in Sec. 3 and time-to-accuracy is commonly used in empirical studies (Bajaj et al., 2024; Hu et al., 2020), we do not provide further explanation.

**Rationale for using iteration-to-accuracy.** However, time-to-accuracy is highly sensitive to hardware differences, entangling model performance improvement per iteration (e.g., accuracy) and computational efficiency (e.g., processed nodes per second). Thus, we additionally introduce *iteration-to-accuracy*, a hardware-agnostic metric, to capture this performance improvement during training.

To illustrate this rationale more clearly, we provide a simple, non-rigorous mathematical derivation, with details in Appendix N. Let $b$ denote the batch size, $\beta$ the fan-out size, and $\nu_l$ the iteration-to-accuracy. Suppose we compare two training setups under the same compute capacity but different bandwidths in distributed systems: a full-graph setting ($b = 1000$, $\beta = 50$, $\nu_l = 10$) and a mini-batch setting ($b = 10$, $\beta = 10$, $\nu_l = 10000$). At high bandwidths (1000 nodes/s), the full-graph setting converges faster, in $5.1 \times 10^5$ seconds, compared with $1.1 \times 10^6$ seconds for mini-batch training. In contrast, at low bandwidths (0.1 nodes/s), mini-batch training converges faster, requiring $2.1 \times 10^6$ seconds, whereas the full-graph setting requires $5.6 \times 10^6$ seconds.

Empirically, Figure 1 illustrates time-to-accuracy and iteration-to-accuracy with two training approaches under different inter-GPU bandwidth levels (i.e., bw1=infinity, simulated by a single GPU with no inter-device communication; bw2=900GB/s, two-GPU NVLink 4.0 setup) and computation capacities (i.e., GPU and CPU). Detailed settings are in Appendix N. For time-to-accuracy, mini-batch training underperforms full-graph training on a single GPU but outperforms it on two GPUs or a single CPU. In contrast, iteration-to-accuracy remains consistent across hardware environments, with a maximum variation of 41.28%, compared to 2787.05% for time-to-accuracy.

Therefore, both mathematical and empirical examples indicate that time-to-accuracy cannot reliably generalize convergence performance across hardware environments, while the iteration-to-accuracy is more reliable to guide early-stage configuration decisions. For example, in a new hardware setup, practitioners can use known iteration-to-accuracy trends to narrow the range of batch and fan-out size, and perform short runs to consider hardware-specific runtime, thereby reducing tuning overhead.

## 5.2  CONVERGENCE

**Empirical Validation of Remarks 3.1, 3.2. and Obs.1.** Remark 3.1. and Obs.1. are empirically validated by Figure 2 and Figures 7- 10 in Appendix N, which illustrate iteration-to-loss for three one-layer GNNs across four real-world datasets under varying fan-out sizes or batch sizes with different learning rates. In addition, Figure 4 in more general settings (e.g., multi-layer GraphSAGE) further confirms Remarks 3.1, 3.2. and Obs.1. using iteration-to-loss (see detailed settings in Appendix N). Due to more complex optimization dynamics in deeper GNNs, Figure 4 shows minor fluctuations across varying batch and fan-out sizes, where the batch size and fan-out size increase until mini-batch training transitions into full-graph training.

**Extended experiments using iteration-to-accuracy and time-to-accuracy.** To study model performance improvement during training, Figure 5 illustrates iteration-to-accuracy and time-to-accuracy

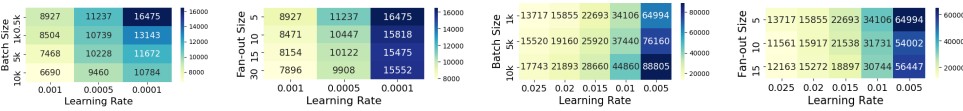

(a) Batch size, CE    (b) Fan-out size, CE    (c) Batch size, b, MSE    (d) Fan-out size, MSE

Figure 2: Iteration-to-loss of one-layer GraphSAGE under *CE* and *MSE* across varying learning rates and batch sizes or fan-out sizes for ogbn-products.

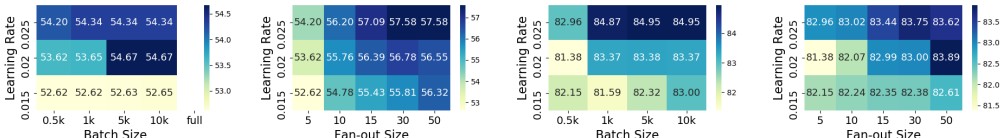

(a) Products, Batch size    (b) Products, Fan-out size    (c) Reddit, Batch size    (d) Reddit, Fan-out size

Figure 3: Test accuracy of one-layer GraphSAGE under *MSE* across varying learning rates and batch sizes or fan-out sizes for ogbn-products and reddit.

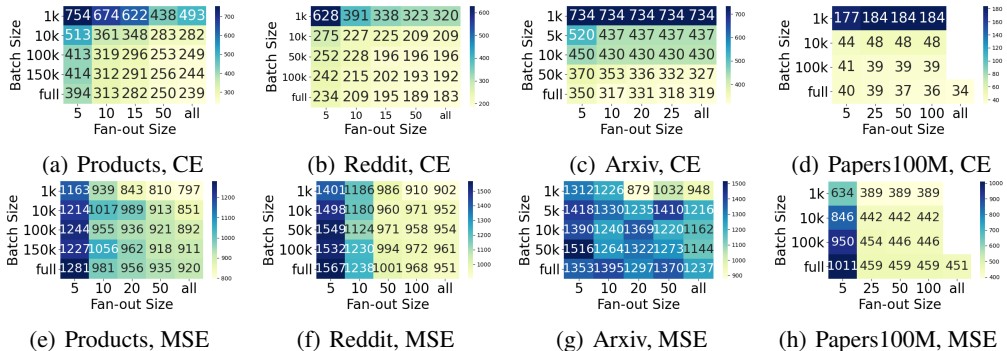

(a) Products, CE    (b) Reddit, CE    (c) Arxiv, CE    (d) Papers100M, CE

(e) Products, MSE    (f) Reddit, MSE    (g) Arxiv, MSE    (h) Papers100M, MSE

Figure 4: Iteration-to-loss of GraphSAGE under *CE* and *MSE* across varying batch and fan-out sizes.

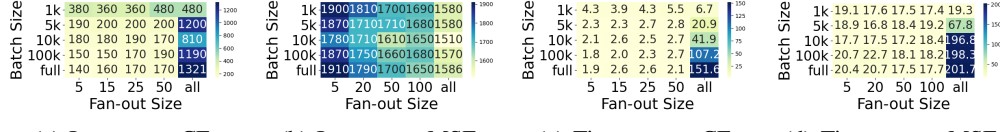

(a) Iter-to-acc, CE    (b) Iter-to-acc, MSE    (c) Time-to-acc, CE    (d) Time-to-acc, MSE

Figure 5: Iteration-to-accuracy and time-to-accuracy of GraphSAGE under *CE* and *MSE* across varying batch sizes and fan-out sizes for reddit.

across varying batch sizes and fan-out sizes for reddit (see more datasets in Appendix N), showing unstable convergence trends with varying batch sizes and very large fan-out sizes (explained further in Sec. 5.3). This is because these two metrics capture both convergence and generalization performance due to the dependency on validation accuracy. Moderate fan-out sizes (e.g., around 15) are shown to balance convergence speed and computational efficiency (shown in time-to-accuracy), supporting the convergence acceleration implications in Sec 3.

## 5.3 GENERALIZATION

**Empirical Validation of Remark 4.1. and Obs.2.** Remark 4.1. and Obs.2. are empirically validated by Figure 3 and Figures 15-16 of one-layer GNNs in Appendix N, which illustrate test accuracies for three one-layer GNNs across four datasets under varying fan-out sizes or batch sizes with different learning rates. In addition, Figures 6(a)-(b) in more general settings for ogbn-products further confirm Obs.2. (see more datasets and details in Appendix N), as the variation of fan-out size induces more frequent and diverse shifts in test accuracies. Regarding Remark 4.1, Figure 6(b) under MSE generally aligns with our theoretical prediction, while Figure 6(a) under CE further shows that performance degradation occurs with very large fan-out sizes (typically more than 15 on these datasets) or batch sizes (exceeding half of the training nodes). This degradation is more severe with fan-out sizes than with batch sizes. We justify *our answer in Sec. 4 to the research question: full-graph training*

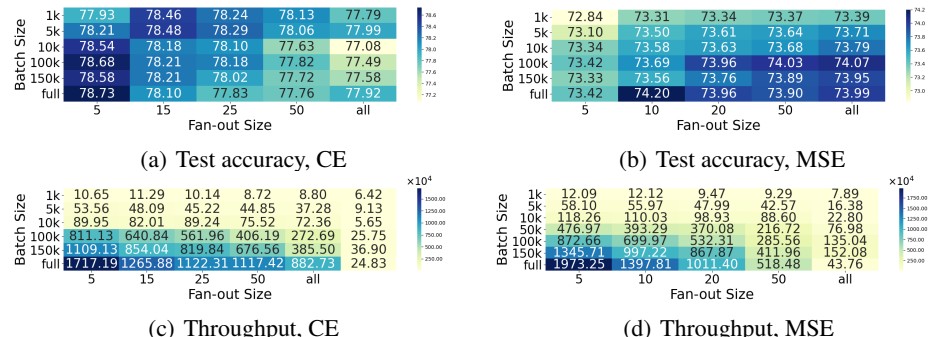

Figure 6: Test accuracies and training throughput (# nodes/s) of GraphSAGE under *CE* and *MSE* across varying batch sizes and fan-out sizes for ogbn-products.

Table 1: Best test accuracies of full-graph and mini-batch training of multi-layer GraphSAGE model without dropout layers after graph-based hyperparameter tuning.

| Datasets | Reddit | Ogbn-arxiv | Ogbn-products | Ogbn-papers100M |
|---|---|---|---|---|
| Full-graph | 96.13 | 70.96 | 77.92 | **59.54** |
| Mini-batch | **96.32** | **71.16** | **78.80** | 58.52 |

*does not always outperform the smaller mini-batch settings in generalization due to degradation in generalization performance.*

**Understanding performance degradation.** This degradation under CE arises as the models tend to converge to sharp minima under large batch sizes (Keskar et al., 2016). Since gradient variance decreases with larger batch and fan-out sizes, similar issues likely occur with large fan-out sizes. This degradation is more severe with fan-out sizes than batch sizes, as aggregating information from too many neighbors causes overfitting and weakens generalization. In contrast, such degradation is not obvious under MSE, which produces flatter minima due to weaker gradients near prediction boundaries (Bosman et al., 2020).

## 5.4 COMPUTATIONAL EFFICIENCY

Figures 6(c)-(d) show the training throughput as the number of target nodes processed per second on a single GPU for ogbn-products (see other datasets in Appendix N).

**Answering our research question:** Computational Efficiency improves with batch size as fixed computations (e.g., parameter updates) are distributed across more data, but becomes worse with larger fan-out sizes due to higher computational demands in message passing. Overall, mini-batch training achieves better computational efficiency than full-graph training.

**Non-isotropic impacts of batch size and fan-out size in convergence, generalization, and computational efficiency.** Based on the observations in Sec. 5.2 - 5.4, the batch size and the fan-out size exhibit distinct, non-uniform effects across different aspects of GNN training. These non-isotropic impacts highlight the need for careful tuning of both hyperparameters to balance computational efficiency, convergence, and generalization.

## 5.5 FULL-GRAPH VS. MINI-BATCH TRAINING AFTER HYPERPARAMETER TUNING

Table 1 compares the generalization performance of full-graph and mini-batch training after tuning batch size and fan-out size via grid search. For the ogbn-papers100M dataset, two hidden layers with a hidden dimension of 128 are used due to resource constraints, limiting representation capacity. The best accuracy from mini-batch training is within 1.74% of full-graph training, suggesting that full-graph training does not consistently outperform well-tuned mini-batch settings.

**Implications for tuning batch size $b$ and fan-out size $\beta$.** Based on both the theoretical and empirical observations above, we recommend keeping the batch size $b$ below half of the training nodes and the fan-out size $\beta$ under 15 for datasets with an average degree less than 50, to avoid generalization degradation and balance the trade-offs in computational efficiency and model performance.

# 6 RELATED WORK

The only existing comparison work (Bajaj et al., 2024) between full-graph and mini-batch GNN training empirically evaluates overall performance but does not investigate the impact of key hyper-parameters (e.g., batch size and fan-out size) on model performance and computational efficiency, thereby overlooking the trade-offs achieved by tuning these hyperparameters. Recent efforts (Yuan et al., 2023; Hu et al., 2021) focus on these hyperparameters but remain limited. For instance, Yuan et al. (Yuan et al., 2023) lack theoretical support, consider only limited batch sizes and fan-out values that are far smaller than those of full-graph training, and overlook the interplay of batch size and fan-out size. Hu et al. (Hu et al., 2021) rely on gradient variance to explain the role of batch size but do not consider fan-out size; thus their explanation conflicts with their empirical observations. Meanwhile, existing theoretical analyses of GNN training (Yang et al., 2023; Tang & Liu, 2023; Xu et al., 2021; Verma & Zhang, 2019; Yadati, 2022; Awasthi et al., 2021) overlook key graph-related factors (e.g., irregular graphs, the difference between training and testing graphs in mini-batch settings) and the impact of non-linear activation on gradients. Furthermore, due to GNN's message-passing process, performance insights from DNNs (You et al., 2019; Smith, 2017; Golmant et al., 2018; Zou et al., 2020a; Bassily et al., 2018; Nabavinejad et al., 2021; Su et al., 2024) cannot directly transfer to GNNs. We provide a more comprehensive related work discussion in Appendix O.

# 7 CONCLUSION

We provide a comprehensive empirical and theoretical study of full-graph vs. mini-batch GNN training from the view of batch size and fan-out size. We provide a novel theoretical GNN generalization analysis employing the Wasserstein distance, to study the impact of batch size and fan-out size. We empirically highlight the importance of iteration-based convergence metrics for hardware-independent evaluation. Our theoretical and empirical findings reveal the non-isotropic impact of batch size and fan-out size in GNN convergence and generalization. Finally, full-graph training does not consistently outperform well-tuned mini-batch settings in model performance or computational efficiency. These insights clarify the trade-offs between full-graph and mini-batch training. We further discuss the extension (e.g., link prediction tasks) and future work (e.g., different activations) in Appendix P.

## ACKNOWLEDGEMENT

This work was supported in part by a collaborative research grant from Ant Group and grants from Hong Kong RGC under the contracts 17203522 (GRF), C7004-22G (CRF), C5032-23G (CRF), and T43-513/23-N (TRS).

## REPRODUCIBILITY STATEMENT

For the theoretical results, all assumptions and complete proofs are provided in Appendices A–E, G, and I–M, with additional important discussions in Appendices F, H, and P. For the empirical study, the code is publicly available via a github link provided in the abstract: `https://github.com/LIUMENGFAN-gif/GNN_fullgraph_minibatch_training`. Detailed experimental configurations and additional experiment results are represented in Appendix N, and all datasets are properly cited in the main text.

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

THE USE OF LARGE LANGUAGE MODELS (LLMS)

We used a large language model (LLM) only as a general-purpose writing assistant to aid in grammar checking and polishing the writing. The LLM did not contribute to research ideas, experiment design, theoretical analysis, or result interpretation.

# A  NOTATIONS

Table 2: Notations

| | |
|---|---|
| $n$ | Number of nodes of the entire graph |
| $n_{\text{train}}$ / $n_{\text{test}}$ | Number of nodes in the training set / the testing set |
| $n_{\min}$ | The minimal value between training and testing sets |
| $\mathbf{X}/\mathbf{x}_i$ | Node feature matrix / $i$-th row of node feature matrix |
| $y_i$ | Ground truth label of node $i$ |
| $\mathbf{y}_i$ / $\hat{\mathbf{y}}_i$ | Ground truth label in one-hot form / estimated outcomes of node $i$ |
| $r$ | feature size |
| $b$ | Batch size |
| $\beta$ | Fan-out size |
| $\mathbf{A}_{\text{train}}^{\text{mini}}$ / $\mathbf{A}_{\text{train}}^{\text{full}}$ | Adjacency matrix in each mini-batch / full-graph training iteration |
| $\mathbf{D}_{\text{train}}^{\text{in,mini}}$ / $\mathbf{D}_{\text{train}}^{\text{in,full}}$ | Diagonal in-degree matrices in each mini-batch / full-graph training iteration |
| $\mathbf{D}_{\text{train}}^{\text{out,mini}}$ / $\mathbf{D}_{\text{train}}^{\text{out,full}}$ | Diagonal out-degree matrices in each mini-batch / full-graph training iteration |
| $\tilde{\mathbf{A}}_{\text{train}}^{\text{mini}}$ / $\tilde{\mathbf{A}}_{\text{train}}^{\text{full}}$ | Normalized adjacency matrix in a mini-batch / full-graph training iteration |
| $\tilde{\mathbf{a}}_{\text{train},i}^{\text{mini}}$ / $\tilde{\mathbf{a}}_{\text{train},i}^{\text{full}}$ | $i$-th row of normalized adjacency matrix in a mini-batch / full-graph training iteration |
| $\hat{\mathbf{A}}_{\text{test}}$ / $\tilde{\mathbf{a}}_{\text{test},i}$ | Normalized adjacency matrix / $i$-th row of Normalized adjacency matrix in testing set |
| $\mathbf{W}^{\text{mini}}$ / $\mathbf{W}^{\text{full}}$ | Learnable parameters of the GNN under mini-batch / full-graph training |
| $\mathbf{w}_i^{\text{mini}}$ / $\mathbf{w}_i^{\text{full}}$ | $i$-th row of parameters of the GNN under mini-batch / full-graph training |
| $\mathbf{W}^{\text{mini}*}$ / $\mathbf{W}^{\text{full}*}$ | Ground truth of learnable parameters $\mathbf{W}^{\text{mini}}$ / $\mathbf{W}^{\text{full}}$ |
| $\mathbf{w}_i^{\text{mini}*}$ / $\mathbf{w}_i^{\text{full}*}$ | $i$-th row of ground truth of learnable parameters $\mathbf{W}^{\text{mini}}$ / $\mathbf{W}^{\text{full}}$ |
| $h$ | Hidden size |
| $K$ | Number of label categories |
| $\sigma(\cdot)$ | ReLU activation function |
| $\hat{\sigma}(\cdot)$ | Dual activation function |
| $L_{\text{train}}(\cdot)$ / $\hat{L}_{\text{train}}(\cdot)$ | Expected / empirical training risk |
| $L_{\text{train}}^{\text{mini}}(\cdot)$ / $\hat{L}_{\text{train}}^{\text{mini}}(\cdot)$ | Expected / empirical training risk in a mini-batch |
| $L_{\text{test}}(\cdot)$ / $\hat{L}_{\text{test}}(\cdot)$ | Expected / empirical testing risk |
| $\hat{\mathbf{G}}$ | Stochastic gradient in mini-batch training |
| $\eta$ | Learning rate |
| $\mathcal{P}/\mathcal{Q}$ | Prior / Posterior distribution of model parameters |
| $U(\cdot)$ | Expected loss discrepancy between training set $\mathcal{C}$ and testing set $\mathcal{Z}$ |
| $\delta(\cdot)$ | Distance function |
| $\theta$ | Covariance |

To easily distinguish the training risk between full-graph and mini-batch training, we rewrite $L_{\text{train}}(\cdot)$ and $\hat{L}_{\text{train}}(\cdot)$ as $L_{\text{train}}^{\text{full}}(\cdot)$ and $\hat{L}_{\text{train}}^{\text{full}}(\cdot)$ under full-graph training. Similarly, we rewrite the gradient $\nabla\hat{L}_{\text{train}}(\cdot)$ as $\nabla\hat{L}_{\text{train}}^{\text{full}}(\cdot)$ during full-graph training, and the stochastic gradient $\hat{\mathbf{G}}$ as $\nabla\hat{L}_{\text{train}}^{\text{mini}}(\cdot)$.

# B  PROOF OF CONVERGENCE THEOREM IN FULL-GRAPH TRAINING WITH MSE

In this section, we provide the proof of the convergence theorem in full-graph training with MSE. We consider multi-class node classification tasks using a one-round GNN trained with the MSE, defined as $l\left(\mathbf{W}, \tilde{\mathbf{a}}_{\text{train},i}^{\text{full}}\right) = \frac{1}{2}\|\hat{\mathbf{y}}_i - \mathbf{y}_i\|_F^2$. The ground truth label $y_i$ is rewritten as $\mathbf{y}_i \in \mathbb{R}^{1\times K}$ in the one-hot

form, where $K \geq 2$ is the number of label categories. The final output of the GNN model is given by $\hat{\mathbf{y}}_i = \mathbf{z}_i = \sigma\left(\tilde{\mathbf{a}}_{\text{train},i}^{\text{full}} \mathbf{X} \mathbf{W}^\top\right)$, where the ReLU function is modified as $\sigma(x) = \sqrt{2}\max(x,0)$. Note that $1/2$ in the MSE function and $\sqrt{2}$ in the ReLU function are introduced to simplify the proof. The hidden dimension $h$ becomes $K$. Note that The rows of $\mathbf{W}$ are initialized independently from a Gaussian distribution $N\left(0, \kappa^2 \mathbf{I}\right)$.

We decompose the analysis of GNN optimization dynamic into three steps.

*Step 1: Reformulating loss and gradient expressions on irregular graphs.* We decouple the activation function from the aggregated node features. For instance, we extract the aggregation from the ReLU function by reformulating squared loss terms.

*Step 2: Bounding the norm of gradient.* Based on the reformulated loss and gradient expressions, we aim to quantify the magnitude of optimization updates by bounding the gradient norm, facilitating convergence analysis. This can be achieved by leveraging the Polyak–Łojasiewicz (PL) inequality (Polyak, 1963), where the squared norm of the gradient is lower bounded by the loss value scaled by a factor.

*Step 3: Bounding the number of iterations to Convergence.* We first leverage the smoothness of the loss function to derive a per-iteration inequality relating loss reduction to the gradient norm, and then accumulate these iteration-wise inequalities over GD updates to obtain an upper bound on the number of iterations required for convergence.

## B.1 ASSUMPTIONS

**Assumption B.1.** The node feature $\mathbf{x}_i$ is drawn i.i.d from $N\left(0, \mathbf{I}_{r \times r}\right)$ for all $i$ in the graph, with $\|\mathbf{X}\|_2^2 \leq C_x$ for a constant $C_x > 0$.

**Assumption B.2.** The rows of ground truth parameters satisfy $\|\mathbf{w}_i^*\|_2 = 1$ for all $i \in \{1, \ldots, h\}$.

Assumption B.1 specifies the distribution of node features and bounds the norm of the feature matrix, and Assumption B.2 limits the norm of ground truth parameters for the GNN model. These assumptions are also adopted in the GNN convergence analysis on regular graphs (Awasthi et al., 2021). We emphasize Assumptions B.1 and B.2 are introduced to simplify the proof. Note that Assumption B.2 can be relaxed to be that $\|\mathbf{w}_i^*\|_2$ is lower and upper bounded by some constants instead of fixing $\|\mathbf{w}_i^*\|_2 = 1$.

**Definition B.3** (Dual activation (Daniely et al., 2016)) The dual activation of $\sigma$ is the function $\hat{\sigma} : [-1, 1] \rightarrow \mathbb{R}$ defined as $\hat{\sigma}(\theta) = \mathbb{E}[\sigma(x)\sigma(y)]$, where $x$ and $y$ are jointly Gaussian random variables with mean zero, variance one, and covariance $\theta$.

Definition B.3 demonstrated that dual activations hold continuity over the interval $[-1, 1]$ and convexity within the range $[0, 1]$.

## B.2 EXPRESSIONS FOR LOSS AND GRADIENTS.

While our ultimate training objective remains empirical risk minimization, we analyze the optimization dynamics of MSE using its expected risk formulation on node feature distribution. This is done to simplify the proof, as expected risk offers a cleaner mathematical structure and does not affect the graph structure. Although this approximation is more accurate in the large-sample regime, we adopt it here as a modeling tool to study the impact of batch size and fan-out size in convergence, even when analyzing small-sample settings.

**Expression for MSE loss:** We first begin by writing an equivalent expression of $L_{\text{train}}^{\text{full}}(\mathbf{w}_j^{\text{full}})$ with $j \in \{1, \ldots, h\}$ as:

$$
\begin{aligned}
L_{\text{train}}^{\text{full}}\left(\mathbf{w}_j^{\text{full}}\right) = \frac{1}{2n_{\text{train}}} \Bigg( & \mathbb{E}\left[\sum_{i=1}^{n_{\text{train}}} \sigma\left(\tilde{\mathbf{a}}_{\text{train},i}^{\text{full}} \mathbf{X}\left(\mathbf{w}_j^{\text{full}}\right)^\top\right)^2\right] + \mathbb{E}\left[\sum_{i=1}^{n_{\text{train}}} \sigma\left(\tilde{\mathbf{a}}_{\text{train},i}^{\text{full}} \mathbf{X}\left(\mathbf{w}_j^{\text{full}*}\right)^\top\right)^2\right] \\
& - 2\mathbb{E}\left[\sum_{i,j=1}^{n_{\text{train}}} \sigma\left(\tilde{\mathbf{a}}_{\text{train},i}^{\text{full}} \mathbf{X}\left(\mathbf{w}_j^{\text{full}}\right)^\top\right) \sigma\left(\tilde{\mathbf{a}}_{\text{train},i}^{\text{full}} \mathbf{X}\left(\mathbf{w}_j^{\text{full}*}\right)^\top\right)\right]\Bigg)
\end{aligned}
\tag{1}
$$

We next compute expressions for each of the three terms above.

$$\frac{1}{n_{\text{train}}}\mathbb{E}\left[\sum_{i=1}^{n_{\text{train}}}\sigma\left(\tilde{\mathbf{a}}_{\text{train},i}^{\text{full}}\mathbf{X}\left(\mathbf{w}_j^{\text{full}}\right)^{\top}\right)^2\right]$$

$$=\frac{1}{n_{\text{train}}}\mathbb{E}\left[\sum_{i,k=1}^{n_{\text{train}}}p_{ij}\sigma\left(\tilde{\mathbf{a}}_{\text{train},i}^{\text{full}}\mathbf{X}\left(\mathbf{w}_j^{\text{full}}\right)^{\top}\right)\sigma\left(\tilde{\mathbf{a}}_{\text{train},k}^{\text{full}}\mathbf{X}\left(\mathbf{w}_j^{\text{full}}\right)^{\top}\right)\right]$$

$$=\frac{1}{n_{\text{train}}}\mathbb{E}[\left\|\mathbf{w}_j^{\text{full}}\right\|^2\sum_{i,k=1}^{n_{\text{train}}}p_{ij}\sqrt{\left(\tilde{\mathbf{A}}_{\text{train}}^{\text{full}}\mathbb{1}\right)_i\left(\tilde{\mathbf{A}}_{\text{train}}^{\text{full}}\mathbb{1}\right)_k}$$

$$\cdot\,\sigma\left(\frac{\tilde{\mathbf{a}}_{\text{train},i}^{\text{full}}\mathbf{X}\left(\mathbf{w}_j^{\text{full}}\right)^{\top}}{\sqrt{\left(\tilde{\mathbf{A}}_{\text{train}}^{\text{full}}\mathbb{1}\right)_i}\left\|\mathbf{w}_j^{\text{full}}\right\|}\right)\sigma\left(\frac{\tilde{\mathbf{a}}_{\text{train},i}^{\text{full}}\mathbf{X}\left(\mathbf{w}_j^{\text{full}}\right)^{\top}}{\sqrt{\left(\tilde{\mathbf{A}}_{\text{train}}^{\text{full}}\mathbb{1}\right)_k}\left\|\mathbf{w}_j^{\text{full}}\right\|}\right)] \tag{2}$$

$$=\frac{\left\|\mathbf{w}_j^{\text{full}}\right\|^2}{n_{\text{train}}}\sum_{i,k=1}^{n_{\text{train}}}p_{ik}\hat{\sigma}\left(\frac{\varrho_{i,k}^{\text{full}}}{\sqrt{\vartheta_{i,k}^{\text{full}}}}\right)\sqrt{\vartheta_{i,k}^{\text{full}}}$$

$$=\left\|\mathbf{w}_j^{\text{full}}\right\|^2\Gamma^{\text{full}},$$

where the penultimate equality follows Definition B.3. We use $p_{ij} = 1$ if $i = j$ and $p_{ij} = 0$ if $i \neq j$, $\varrho_{i,j}^{\text{full}}$ to denote the amount of common messages between node $i$ and node $j$ at a given training iteration, and we define:

$$\Gamma^{\text{full}} = \frac{1}{n_{\text{train}}}\sum_{i,j=1}^{n_{\text{train}}}p_{ij}\hat{\sigma}\left(\frac{\varrho_{i,j}^{\text{full}}}{\sqrt{\vartheta_{i,j}^{\text{full}}}}\right)\sqrt{\vartheta_{i,j}^{\text{full}}}, \tag{3}$$

$$\vartheta_{i,j}^{\text{full}} = \left(\tilde{\mathbf{A}}_{\text{train}}^{\text{full}}\mathbb{1}\right)_i\left(\tilde{\mathbf{A}}_{\text{train}}^{\text{full}}\mathbb{1}\right)_j. \tag{4}$$

Similarly, we get the second term as:

$$\frac{1}{n_{\text{train}}}\mathbb{E}\left[\sum_{i=1}^{n_{\text{train}}}\sigma\left(\tilde{\mathbf{a}}_{\text{train},i}^{\text{full}}\mathbf{X}\left(\mathbf{w}_j^{\text{full}*}\right)^{\top}\right)^2\right]=\left\|\mathbf{w}_j^{\text{full}*}\right\|^2\Gamma^{\text{full}}. \tag{5}$$

We simplify the last term as:

$$\frac{1}{n_{\text{train}}}\mathbb{E}\left[\sum_{i,k=1}^{n_{\text{train}}}p_{ik}\sigma\left(\tilde{\mathbf{a}}_{\text{train},i}^{\text{full}}\mathbf{X}\left(\mathbf{w}_j^{\text{full}}\right)^{\top}\right)\sigma\left(\tilde{\mathbf{a}}_{\text{train},k}^{\text{full}}\mathbf{X}\left(\mathbf{w}_j^{\text{full}*}\right)^{\top}\right)\right]$$

$$=\frac{1}{n_{\text{train}}}\mathbb{E}[\left\|\mathbf{w}_j^{\text{full}}\right\|\left\|\mathbf{w}_j^{\text{full}*}\right\|\sum_{i,k=1}^{n_{\text{train}}}p_{ik}\sqrt{\left(\tilde{\mathbf{A}}_{\text{train}}^{\text{full}}\mathbb{1}\right)_i\left(\tilde{\mathbf{A}}_{\text{train}}^{\text{full}}\mathbb{1}\right)_k}$$

$$\cdot\,\sigma\left(\frac{\tilde{\mathbf{a}}_{\text{train},i}^{\text{full}}\mathbf{X}\left(\mathbf{w}_j^{\text{full}}\right)^{\top}}{\sqrt{\left(\tilde{\mathbf{A}}_{\text{train}}^{\text{full}}\mathbb{1}\right)_i}\left\|\mathbf{w}_j^{\text{full}}\right\|}\right)\sigma\left(\frac{\tilde{\mathbf{a}}_{\text{train},k}^{\text{full}}\mathbf{X}\left(\mathbf{w}_j^{\text{full}*}\right)^{\top}}{\sqrt{\left(\tilde{\mathbf{A}}_{\text{train}}^{\text{full}}\mathbb{1}\right)_k}\left\|\mathbf{w}_j^{\text{full}*}\right\|}\right)] \tag{6}$$

$$=\frac{1}{n_{\text{train}}}\left\|\mathbf{w}_j^{\text{full}}\right\|\left\|\mathbf{w}_j^{\text{full}*}\right\|\sum_{i,k=1}^{n_{\text{train}}}p_{ik}\hat{\sigma}\left(\frac{\varrho_{i,k}^{\text{full}}}{\sqrt{\vartheta_{i,k}^{\text{full}}}}\frac{\left(\mathbf{w}_j^{\text{full}}\right)^{\top}\mathbf{w}_j^{\text{full}*}}{\left\|\mathbf{w}_j^{\text{full}}\right\|\left\|\mathbf{w}_j^{\text{full}*}\right\|}\right)\sqrt{\vartheta_{i,k}^{\text{full}}}.$$

Therefore, we have the expression of $L_{\text{train}}^{\text{full}}(\mathbf{w}_j^{\text{full}})$ as:

$$L_{\text{train}}^{\text{full}}\left(\mathbf{w}_j^{\text{full}}\right)=\frac{1}{2}(\|\mathbf{w}_j^{\text{full}}\|^2\Gamma^{\text{full}}+\|\mathbf{w}_j^{\text{full}*}\|^2\Gamma^{\text{full}}$$

$$-\frac{2}{n_{\text{train}}}\left\|\mathbf{w}_j^{\text{full}}\right\|\left\|\mathbf{w}_j^{\text{full}*}\right\|\sum_{i,k=1}^{n_{\text{train}}}p_{ik}\hat{\sigma}\left(\frac{\varrho_{i,k}^{\text{full}}}{\sqrt{\vartheta_{i,k}^{\text{full}}}}\frac{\left(\mathbf{w}_j^{\text{full}}\right)^{\top}\mathbf{w}_j^{\text{full}*}}{\left\|\mathbf{w}_j^{\text{full}}\right\|\left\|\mathbf{w}_j^{\text{full}*}\right\|}\right)\sqrt{\vartheta_{i,k}^{\text{full}}}). \tag{7}$$

It is easy to see that if $\mathbf{w}_{j,0}^{\text{full}}$ is the initial value of $\mathbf{w}_j^{\text{full}}$ with $j \in \{1, \ldots, h\}$ then each subsequent iteration will be a linear combination of $\mathbf{w}_{j,0}^{\text{full}}$ and $\mathbf{w}_j^{\text{full}*}$. Hence we can assume that $\mathbf{w}_j^{\text{full}} = \phi^{\text{full}} \mathbf{w}_j^{\text{full}*} + \psi^{\text{full}} \mathbf{w}_j^{\text{full}\perp}$, where $\mathbf{w}^\perp$ is a fixed unit vector (depending on the initialization) orthogonal to $\mathbf{w}_j^{\text{full}*}$. Then rewriting the loss in terms of $\phi^{\text{full}}, \psi^{\text{full}}$ and recalling that $\|\mathbf{w}_j^{\text{full}*}\| = 1$ we get the simplified expression of $L_{\text{train}}^{\text{full}}\left(\mathbf{w}_j^{\text{full}}\right)$:

$$L_{\text{train}}^{\text{full}}\left(\phi^{\text{full}}, \psi^{\text{full}}\right) = \frac{1}{2}\left(\left(\phi^{\text{full}}\right)^2 + \left(\psi^{\text{full}}\right)^2 + 1\right)\Gamma^{\text{full}} - \sqrt{\left(\phi^{\text{full}}\right)^2 + \left(\psi^{\text{full}}\right)^2}\Upsilon^{\text{full}}, \tag{8}$$

where we define:

$$\Upsilon^{\text{full}} = \frac{1}{n_{\text{train}}}\sum_{i,j=1}^{n_{\text{train}}} p_{ij}\hat{\sigma}\left(\frac{\phi^{\text{full}}}{\sqrt{\left(\phi^{\text{full}}\right)^2 + \left(\psi^{\text{full}}\right)^2}}\frac{\varrho_{i,j}^{\text{full}}}{\sqrt{\vartheta_{i,j}^{\text{full}}}}\right)\sqrt{\vartheta_{i,j}^{\text{full}}}. \tag{9}$$

**Expression for gradient:** We compute the gradient of the objective with respect to $\mathbf{w}$ or equivalently with respect to $\phi, \psi$.

$$
\begin{aligned}
&\frac{\partial L^{\text{full}}\left(\phi^{\text{full}}, \psi^{\text{full}}\right)}{\partial \phi^{\text{full}}}\\
=&\phi^{\text{full}}\Gamma^{\text{full}} - \frac{\phi^{\text{full}}\Upsilon^{\text{full}}}{\sqrt{(\phi^{\text{full}})^2 + (\psi^{\text{full}})^2}}\\
&+ \frac{1}{n_{\text{train}}^2}\left(\frac{(\psi^{\text{full}})^2}{(\phi^{\text{full}})^2 + (\psi^{\text{full}})^2}\sum_{i,j=1}^{n_{\text{train}}} p_{ij}\varrho_{i,j}^{\text{full}}\hat{\sigma}'\left(\frac{\phi^{\text{full}}}{\sqrt{(\phi^{\text{full}})^2 + (\psi^{\text{full}})^2}}\frac{\varrho_{i,j}^{\text{full}}}{\sqrt{\vartheta_{i,j}^{\text{full}}}}\right)\right)\\
=&\phi^{\text{full}}\Gamma^{\text{full}} - \frac{\phi^{\text{full}}\Upsilon^{\text{full}}}{\sqrt{(\phi^{\text{full}})^2 + (\psi^{\text{full}})^2}}\\
&+ \frac{1}{n_{\text{train}}^2}\left(\frac{(\psi^{\text{full}})^2}{(\phi^{\text{full}})^2 + (\psi^{\text{full}})^2}\sum_{i,j=1}^{n_{\text{train}}} p_{ij}\varrho_{i,j}^{\text{full}}\hat{\sigma}_{\text{step}}\left(\frac{\phi^{\text{full}}}{\sqrt{(\phi^{\text{full}})^2 + (\psi^{\text{full}})^2}}\frac{\varrho_{i,j}^{\text{full}}}{\sqrt{\vartheta_{i,j}^{\text{full}}}}\right)\right),\\
=&\phi^{\text{full}}\Gamma^{\text{full}} - \frac{\phi^{\text{full}}\Upsilon^{\text{full}}}{\sqrt{(\phi^{\text{full}})^2 + (\psi^{\text{full}})^2}} + \frac{(\psi^{\text{full}})^2\,\Xi^{\text{full}}}{(\phi^{\text{full}})^2 + (\psi^{\text{full}})^2},
\end{aligned}
\tag{10}
$$

where in the second equality we use $\hat{\sigma}' = \hat{\sigma'}$ and $\hat{\sigma'} = \sqrt{2}\mathbb{1}(x \geq 0) = \sigma_{\text{step}}(x)$, $\sigma_{\text{step}}$ is the step function, and we define:

$$\Xi^{\text{full}} = \frac{1}{n_{\text{train}}}\sum_{i,j=1}^{n_{\text{train}}} p_{ij}\varrho_{i,j}^{\text{full}}\hat{\sigma}_{\text{step}}\left(\frac{\phi^{\text{full}}}{\sqrt{\left(\phi^{\text{full}}\right)^2 + \left(\psi^{\text{full}}\right)^2}}\frac{\varrho_{i,j}^{\text{full}}}{\sqrt{\vartheta_{i,j}^{\text{full}}}}\right). \tag{11}$$

Similarly, we have:

$$\frac{\partial L^{\text{full}}\left(\phi^{\text{full}}, \psi^{\text{full}}\right)}{\partial \psi^{\text{full}}} = \psi^{\text{full}}\Gamma^{\text{full}} - \frac{\psi^{\text{full}}\Upsilon^{\text{full}}}{\sqrt{\left(\phi^{\text{full}}\right)^2 + \left(\psi^{\text{full}}\right)^2}} + \frac{\phi^{\text{full}}\psi^{\text{full}}\Xi^{\text{full}}}{\left(\phi^{\text{full}}\right)^2 + \left(\psi^{\text{full}}\right)^2}. \tag{12}$$

## B.3 THEOREM B.4

**Theorem B.4.** (Convergence of Full-graph Training with MSE) *Suppose $\mathbf{W}^{full}$ are generated by Gaussian initialization. Under Assumptions B.1 and B.2, if the maximal degree satisfies $C_1^{full} \leq d_{max} \leq C_2^{full} n_{train}^{\frac{3}{4}}$ for some constants $C_1^{full}, C_2^{full} \in (0, 1)$, then with high probability, the training loss satisfies $L_{train}\left(\mathbf{W}_T^{full}, \mathbf{A}_{train}^{full}\right) \leq \epsilon$, provided that the number of iterations $T = O\left(n_{train}^{\frac{7}{2}}h^2 d_{max}^{-\frac{1}{2}}\epsilon^{-1}\log\left(h^2\epsilon^{-1}\right)\right)$ for any $\epsilon \in (0, 1)$ under the full-graph GNN training.*

## B.4 PROOF OF THEOREM B.4

**Lemma B.5** $\frac{1}{\pi n_{\text{train}}}\|\mathbf{A}_{\text{train}}^{\text{full}}\mathbb{1}\|_1 \leq \Gamma^{\text{full}} \leq \frac{1}{n_{\text{train}}}\|\mathbf{A}_{\text{train}}^{\text{full}}\mathbb{1}\|_1$, and $|\Upsilon_t^{\text{full}}| \leq \Gamma^{\text{full}}$, where $n_{\text{train}}^{\frac{1}{2}}d_{\max}^{-\frac{1}{2}} \leq \|\mathbf{A}_{\text{train},t}^{\text{full}}\mathbb{1}\|_1 \leq n_{\text{train}}d_{\max}$.

**Lemma B.6** If $\mathbf{w}_{j,0}^{\text{full}} \sim N(0, \kappa^2\mathbf{I})$ and the learning rate $\eta \in (0, \frac{1}{6\pi\Gamma^{\text{full}}}]$, then with probability at least $1 - e^{-O(1)}$, it holds that for all $t > 0$, $\sqrt{\left(\phi_t^{\text{full}}\right)^2 + \left(\psi_t^{\text{full}}\right)^2} \leq C$, and $\sqrt{\left(\phi_t^{\text{full}}\right)^2 + \left(\psi_t^{\text{full}}\right)^2} > 0$ for all $t \geq 1$, where $C = \frac{\pi}{2} + O\left(\kappa\sqrt{r}\right)$ is a positive constant.

**Lemma B.7** If $\mathbf{w}_{j,0}^{\text{full}} \sim N(0, \kappa^2\mathbf{I})$ and the learning rate $\eta \in (0, \frac{1}{6\pi\Gamma^{\text{full}}}]$, then for all $t \geq 1$ and any $C_1 \in [0, 1]$ such that $\left(\phi^{\text{full}}, \psi^{\text{full}}\right) = (1 - C_1)\left(\phi_t^{\text{full}}, \psi_t^{\text{full}}\right) + C_1\left(\phi_{t+1}^{\text{full}}, \psi_{t+1}^{\text{full}}\right)$, we have that,

$$\lambda_{\max}(\nabla^2 L_{\text{train}}^{\text{full}}(\phi^{\text{full}}, \psi^{\text{full}})) \leq C_2\Gamma^{\text{full}},$$

where $\lambda_{\max}$ is the maximum eigenvalue of the population Hessian denoted by $\nabla^2 L_{\text{train}}^{\text{full}}\left(\phi^{\text{full}}, \psi^{\text{full}}\right)$, and $C_2 = 4\left(1 + \sqrt{\frac{\pi}{2} + O\left(\kappa\sqrt{r}\right)} + o\left(1\right)\right)$ is a positive constant.

**Lemma B.8** If $\mathbf{w}_{j,0}^{\text{full}} \sim N(0, \kappa^2\mathbf{I})$ and the learning rate $\eta \in (0, \frac{1}{6\pi\Gamma^{\text{full}}}]$, then with at least $1 - 1/h^2$, it holds that for all $t \geq C_3 \log\left(\log h\right)$, $\sqrt{\left(\phi_t^{\text{full}}\right)^2 + \left(\psi_t^{\text{full}}\right)^2} \geq 1 - o\left(1\right)$, where $C_3 > 0$ is an absolute constant.

**Lemma B.9** If $\mathbf{w}_{j,0}^{\text{full}} \sim N(0, \kappa^2\mathbf{I})$ and the learning rate $\eta \in (0, \frac{1}{6\pi\Gamma^{\text{full}}}]$, then there is an absolute constant $C_3$, such that for all $t \geq C_3 \log\left(\log h\right)$, either $\left|\psi_t^{\text{full}}\right| \leq \frac{\epsilon^{\frac{1}{2}}}{2h}$ and $\left\|\sqrt{\left(\phi_t^{\text{full}}\right)^2 + \left(\psi_t^{\text{full}}\right)^2} - 1\right\| \leq \frac{\epsilon^{\frac{1}{2}}}{2h}$ or we have that

$$\left\|\nabla L_{\text{train}}^{\text{full}}\left(\phi_t^{\text{full}}, \psi_t^{\text{full}}\right)\right\|^2 \geq \mu^{\text{full}} L_{\text{train}}^{\text{full}}\left(\phi_t^{\text{full}}, \psi_t^{\text{full}}\right),$$

where $\mu^{\text{full}} \geq C_4\epsilon h^{-2}d_{\max}^{-2}\Gamma^{\text{full}}$, and $C_4$ is a positive constant.

**Proof of Theorem B.4:** We analyze an arbitrary $j \in \{1, \ldots, h\}$ and the iterates of the corresponding $\mathbf{w}_j^{\text{full}}$ vector. Setting $\kappa = 1$, we have from Lemma B.7 that the smoothness parameter $C^{\text{full}}$ of the loss function is

$$C^{\text{full}} \leq C_2 = 4\left(1 + \sqrt{2 + \frac{\pi}{2}} + o\left(1\right)\right) \tag{13}$$

Hence, for any $t > 0$,

$$
\begin{aligned}
L_{\text{train}}^{\text{full}}(\mathbf{w}_{j,t+1}^{\text{full}}) \leq & L_{\text{train}}^{\text{full}}\left(\mathbf{w}_{j,t}^{\text{full}}\right) + \nabla L_{\text{train}}^{\text{full}}\left(\mathbf{w}_{j,t}^{\text{full}}\right)(\mathbf{w}_{j,t+1}^{\text{full}} - \mathbf{w}_{j,t}^{\text{full}}) \\
& + \frac{C^{\text{full}}}{2}\left\|\mathbf{w}_{j,t+1}^{\text{full}} - \mathbf{w}_{j,t}^{\text{full}}\right\|^2 \\
\leq & L_{\text{train}}^{\text{full}}\left(\mathbf{w}_{j,t}^{\text{full}}\right) - \eta\left\|\nabla L_{\text{train}}^{\text{full}}\left(\mathbf{w}_{j,t}^{\text{full}}\right)\right\|^2 + \frac{\eta^2 C^{\text{full}}}{2}\left\|\nabla L_{\text{train}}^{\text{full}}\left(\mathbf{w}_{j,t}^{\text{full}}\right)\right\|^2 \\
= & L_{\text{train}}^{\text{full}}\left(\mathbf{w}_{j,t}^{\text{full}}\right) - \eta\left\|\nabla L_{\text{train}}^{\text{full}}\left(\mathbf{w}_{j,t}^{\text{full}}\right)\right\|^2\left(1 - \frac{\eta C^{\text{full}}}{2}\right).
\end{aligned}
\tag{14}
$$

By Lemma B.6, we know that $\eta \in (0, \frac{1}{6\pi\Gamma^{\text{full}}}]$. Using Lemma B.5, we first assume that $\frac{C^{\text{full}}}{6\pi} \leq d_{\max} \leq \left(\frac{1}{6C_6}\right)^{\frac{1}{4}}n_{\text{train}}^{\frac{3}{4}}$ where $C_6 < \frac{1}{6}$ is a positive constant. Then, we set $\eta \in \left[\frac{C_6 d_{\max}^3}{\pi n_{\text{train}}^3}, \frac{1}{6\pi d_{\max}}\right]$. We are going to prove $\eta \in \left[\frac{C_6 d_{\max}^3}{\pi n_{\text{train}}^3}, \frac{1}{6\pi d_{\max}}\right]$ is still within the range $(0, \frac{1}{6\pi\Gamma^{\text{full}}}]$ and $\frac{C_6 d_{\max}^3}{\pi n_{\text{train}}^3} \leq \frac{1}{6\pi d_{\max}}$.

For the right side of the range, we have $\frac{1}{6\pi d_{\max}} \leq \frac{1}{6\pi \Gamma^{\text{full}}}$ due to $\Gamma^{\text{full}} \leq d_{\max}$. For the left side of the range, $\frac{C_6 d_{\max}^3}{\pi n_{\text{train}}^3} > 0$ with the positive constant $C_6$. Moreover, we have:

$$
\begin{aligned}
\frac{1}{6\pi d_{\max}} - \frac{C_6 d_{\max}^3}{\pi n_{\text{train}}^3} &= \frac{1}{\pi}\left(\frac{1}{6 d_{\max}} - \frac{C_6 d_{\max}^3}{n_{\text{train}}^3}\right) \\
&\geq \frac{1}{\pi}\left(\frac{1}{6 d_{\max}} - \frac{1}{6 d_{\max}}\right) = 0.
\end{aligned}
\tag{15}
$$

With $\eta \in \left[\frac{C_6 d_{\max}^3}{\pi n_{\text{train}}^3}, \frac{1}{6\pi d_{\max}}\right]$, we have:

$$
\eta C^{\text{full}} \leq \frac{C^{\text{full}}}{6\pi d_{\max}} \leq 1.
\tag{16}
$$

Furthermore, using Lemma B.9, we have

$$
L_{\text{train}}^{\text{full}}(\mathbf{w}_{j,t+1}^{\text{full}}) < L_{\text{train}}^{\text{full}}\left(\mathbf{w}_{j,t}^{\text{full}}\right)(1 - \eta\mu^{\text{full}}) \leq L_{\text{train},0}^{\text{full}}(\mathbf{w}_{j,0}^{\text{full}})(1 - \eta\mu^{\text{full}})^t.
\tag{17}
$$

Then we have:

$$
T \leq C_7 \log\left(\frac{h^2}{\epsilon}\right)\frac{1}{\eta\mu^{\text{full}}},
\tag{18}
$$

where $C_7$ is a positive constant.

Moreover, we have:

$$
\eta\mu^{\text{full}} \geq \frac{C_4 C_6 d_{\max}\epsilon\Gamma^{\text{full}}}{\pi n_{\text{train}}^3 h^2} \geq \frac{C_4 C_6 d_{\max}^{\frac{1}{2}}\epsilon}{\pi^2 n_{\text{train}}^{\frac{7}{2}} h^2}
\tag{19}
$$

Hence, we have $T = O\left(\frac{n_{\text{train}}^{\frac{7}{2}} h^2}{\epsilon d_{\max}^{\frac{1}{2}}}\log\frac{h^2}{\epsilon}\right)$.

After $T$ time steps, we either have $L_{\text{train}}^{\text{full}}\left(\mathbf{w}_{j,t}^{\text{full}}\right) \leq \frac{\epsilon}{h}$, or that $\psi_t^{\text{full}} \leq \frac{\epsilon^{\frac{1}{2}}}{2h}$ and $\left(\phi_t^{\text{full}}\right)^2 + \left(\psi_t^{\text{full}}\right)^2 - 1 \leq \frac{\epsilon^{\frac{1}{2}}}{2h}$. The latter implies that $\left\|\mathbf{w}_{j,t}^{\text{full}} - \mathbf{w}_{j,t}^{\text{full}*}\right\|^2 \leq \frac{\epsilon}{h}$. In addition, it is easy to see that $L_{\text{train}}^{\text{full}}\left(\mathbf{w}_{j,t}^{\text{full}}\right) \leq \left\|\mathbf{w}_{j,t}^{\text{full}} - \mathbf{w}_{j,t}^{\text{full}*}\right\|^2$. Hence, if the latter happens, then $L_{\text{train}}^{\text{full}}\left(\mathbf{w}_{j,t}^{\text{full}}\right) \leq \frac{\epsilon}{h}$. Hence $L_{\text{train}}^{\text{full}}(\mathbf{W}_T^{\text{full}}) \leq \epsilon$.

This completes the proof.

## C  PROOF OF CONVERGENCE THEOREM IN MINI-BATCH TRAINING WITH MSE

In this section, we provide the proof of the convergence theorem in mini-batch training with MSE of Section 3. We consider multi-class node classification tasks using a one-round GNN trained with the MSE, defined as $l\left(\mathbf{W}, \tilde{\mathbf{a}}_{\text{train},i}^{\text{mini}}\right) = \frac{1}{2}\left\|\hat{\mathbf{y}}_i - \mathbf{y}_i\right\|_F^2$. The ground truth label $y_i$ is rewritten as $\mathbf{y}_i \in \mathbb{R}^{1 \times K}$ in the one-hot form, where $K \geq 2$ is the number of label categories. The final output of the GNN model is given by $\hat{\mathbf{y}}_i = \mathbf{z}_i = \sigma\left(\tilde{\mathbf{a}}_{\text{train},i}^{\text{mini}}\mathbf{X}\mathbf{W}^\top\right)$, where the ReLU function is modified as $\sigma(x) = \sqrt{2}\max(x, 0)$. Note that $1/2$ in the MSE function and $\sqrt{2}$ in the ReLU function are introduced to simplify the proof. The hidden dimension $h$ becomes $K$. The rows of $\mathbf{W}$ are initialized independently from a Gaussian distribution $N\left(0, \kappa^2\mathbf{I}\right)$.

We decompose the analysis of GNN optimization dynamic into three steps, similar to Appendix B.

### C.1  ASSUMPTION

We still use Assumptions B.1 and B.2 in mini-batch settings for training data and the ground truth.

## C.2 EXPRESSIONS FOR LOSS AND GRADIENTS.

While our ultimate training objective remains empirical risk minimization, we analyze the optimization dynamics of MSE using its expected risk formulation on node feature distribution. This is done to simplify the proof, as expected risk offers a cleaner mathematical structure and does not affect the graph structure. Although this approximation is more accurate in the large-sample regime, we adopt it here as a modeling tool to study the impact of batch size and fan-out size in convergence, even when analyzing small-sample settings.

**Expression for MSE loss:**  We first begin by writing an equivalent expression of $L_{\text{train}}^{\text{mini}}(\mathbf{w}_j^{\text{mini}})$ with $j \in \{1, \ldots, h\}$. We can assume that $\mathbf{w}_j^{\text{mini}} = \phi^{\text{mini}}\mathbf{w}_j^{\text{mini}*} + \psi^{\text{mini}}\mathbf{w}_j^{\text{mini}\perp}$, where $\mathbf{w}^{\perp}$ is a fixed unit vector (depending on the initialization) orthogonal to $\mathbf{w}_j^{\text{mini}*}$. Then rewriting the loss in terms of $\phi^{\text{mini}}$, $\psi^{\text{mini}}$ and recalling that $\|\mathbf{w}_j^{\text{mini}*}\| = 1$ we get the simplified expressions of $L_{\text{train}}^{\text{mini}}\left(\mathbf{w}_j^{\text{mini}}\right)$ and $L_{\text{train}}^{\text{full}}\left(\mathbf{w}_j^{\text{mini}}\right)$:

$$L_{\text{train}}^{\text{mini}}\left(\phi^{\text{mini}}, \psi^{\text{mini}}\right) = \frac{1}{2}\left(\left(\phi^{\text{mini}}\right)^2 + \left(\psi^{\text{mini}}\right)^2 + 1\right)\Gamma^{\text{mini}} - \sqrt{\left(\phi^{\text{mini}}\right)^2 + \left(\psi^{\text{mini}}\right)^2}\Upsilon^{\text{mini}}, \quad (20)$$

and

$$\begin{aligned} L_{\text{train}}^{\text{full}}\left(\phi^{\text{mini}}, \psi^{\text{mini}}, \tilde{\mathbf{A}}_{\text{train}}^{\text{mini}}\right) = &\frac{1}{2}\left(\left(\phi^{\text{mini}}\right)^2 + \left(\psi^{\text{mini}}\right)^2 + 1\right)\Gamma^{\text{full}}\left(\phi^{\text{mini}}, \psi^{\text{mini}}, \tilde{\mathbf{A}}_{\text{train}}^{\text{mini}}\right) \\ &- \sqrt{\left(\phi^{\text{mini}}\right)^2 + \left(\psi^{\text{mini}}\right)^2}\Upsilon^{\text{full}}\left(\phi^{\text{mini}}, \psi^{\text{mini}}, \tilde{\mathbf{A}}_{\text{train}}^{\text{mini}}\right), \end{aligned} \quad (21)$$

where we simplify $\Gamma^{\text{full}}\left(\phi^{\text{mini}}, \psi^{\text{mini}}, \tilde{\mathbf{A}}_{\text{train}}^{\text{mini}}\right)$ and $\Upsilon^{\text{full}}\left(\phi^{\text{mini}}, \psi^{\text{mini}}, \tilde{\mathbf{A}}_{\text{train}}^{\text{mini}}\right)$ as $\Gamma^{\text{full-mini}}$ and $\Upsilon^{\text{full-mini}}$, respectively, and we define:

$$\Gamma^{\text{mini}} = \frac{1}{b}\sum_{i,j=1}^{b} p_{ij}\hat{\sigma}\left(\frac{\varrho_{i,j}^{\text{mini}}}{\sqrt{\vartheta_{i,j}^{\text{mini}}}}\right)\sqrt{\vartheta_{i,j}^{\text{mini}}}, \quad (22)$$

$$\Gamma^{\text{full-mini}} = \Gamma^{\text{full}}\left(\phi^{\text{mini}}, \psi^{\text{mini}}, \tilde{\mathbf{A}}_{\text{train}}^{\text{mini}}\right) = \frac{1}{n_{\text{train}}}\sum_{i,j=1}^{n_{\text{train}}} p_{ij}\hat{\sigma}\left(\frac{\varrho_{i,j}^{\text{mini}}}{\sqrt{\vartheta_{i,j}^{\text{mini}}}}\right)\sqrt{\vartheta_{i,j}^{\text{mini}}}, \quad (23)$$

$$\Upsilon^{\text{mini}} = \frac{1}{b}\sum_{i,j=1}^{b} p_{ij}\hat{\sigma}\left(\frac{\phi^{\text{mini}}}{\sqrt{\left(\phi^{\text{mini}}\right)^2 + \left(\psi^{\text{mini}}\right)^2}}\frac{\varrho_{i,j}^{\text{mini}}}{\sqrt{\vartheta_{i,j}^{\text{mini}}}}\right)\sqrt{\vartheta_{i,j}^{\text{mini}}}, \quad (24)$$

$$\begin{aligned} \Upsilon^{\text{full-mini}} &= \Upsilon^{\text{full}}\left(\phi^{\text{mini}}, \psi^{\text{mini}}, \tilde{\mathbf{A}}_{\text{train}}^{\text{mini}}\right) \\ &= \frac{1}{n_{\text{train}}}\sum_{i,j=1}^{n_{\text{train}}} p_{ij}\hat{\sigma}\left(\frac{\phi^{\text{mini}}}{\sqrt{\left(\phi^{\text{mini}}\right)^2 + \left(\psi^{\text{mini}}\right)^2}}\frac{\varrho_{i,j}^{\text{mini}}}{\sqrt{\vartheta_{i,j}^{\text{mini}}}}\right)\sqrt{\vartheta_{i,j}^{\text{mini}}}, \end{aligned} \quad (25)$$

$$\vartheta_{i,j}^{\text{mini}} = \left(\tilde{\mathbf{A}}_{\text{train}}^{\text{mini}}\mathbb{1}\right)_i\left(\tilde{\mathbf{A}}_{\text{train}}^{\text{mini}}\mathbb{1}\right)_j, \quad (26)$$

where we use $p_{ij} = 1$ if $i = j$ and $p_{ij} = 0$ if $i \neq j$, $\varrho_{i,j}^{\text{mini}}$ to denote the amount of common messages between node $i$ and node $j$ at a given training iteration

**Expression for gradient:**  We compute the gradient of the objective with respect to $\mathbf{w}$ or equivalently with respect to $\phi, \psi$.

$$\frac{\partial L^{\text{mini}}\left(\phi^{\text{mini}}, \psi^{\text{mini}}\right)}{\partial \phi^{\text{mini}}} = \phi^{\text{mini}}\Gamma^{\text{mini}} - \frac{\phi^{\text{mini}}\Upsilon^{\text{mini}}}{\sqrt{\left(\phi^{\text{mini}}\right)^2 + \left(\psi^{\text{mini}}\right)^2}} + \frac{\left(\psi^{\text{mini}}\right)^2\Xi^{\text{mini}}}{\left(\phi^{\text{mini}}\right)^2 + \left(\psi^{\text{mini}}\right)^2}, \quad (27)$$

and

$$\frac{\partial L^{\text{full}}\left(\phi^{\text{mini}}, \psi^{\text{mini}}, \tilde{\mathbf{A}}_{\text{train}}^{\text{mini}}\right)}{\partial \phi^{\text{mini}}} = \phi^{\text{mini}}\Gamma^{\text{full-mini}} - \frac{\phi^{\text{mini}}\Upsilon^{\text{full-mini}}}{\sqrt{\left(\phi^{\text{mini}}\right)^2 + \left(\psi^{\text{mini}}\right)^2}} + \frac{\left(\psi^{\text{mini}}\right)^2\Xi^{\text{full-mini}}}{\left(\phi^{\text{mini}}\right)^2 + \left(\psi^{\text{mini}}\right)^2}, \quad (28)$$

where we define:

$$\Xi^{\text{mini}} = \frac{1}{b} \sum_{i,j=1}^{b} p_{ij} \varrho_{i,j}^{\text{mini}} \hat{\sigma}_{\text{step}} \left( \frac{\phi^{\text{mini}}}{\sqrt{\left(\phi^{\text{mini}}\right)^2 + \left(\psi^{\text{mini}}\right)^2}} \frac{\varrho_{i,j}^{\text{mini}}}{\sqrt{\vartheta_{i,j}^{\text{mini}}}} \right), \tag{29}$$

$$\begin{aligned} \Xi^{\text{full-mini}} &= \Xi^{\text{full}} \left( \phi^{\text{mini}}, \psi^{\text{mini}}, \tilde{\mathbf{A}}_{\text{train}}^{\text{mini}} \right) \\ &= \frac{1}{n_{\text{train}}} \sum_{i,j=1}^{n_{\text{train}}} p_{ij} \varrho_{i,j}^{\text{mini}} \hat{\sigma}_{\text{step}} \left( \frac{\phi^{\text{mini}}}{\sqrt{\left(\phi^{\text{mini}}\right)^2 + \left(\psi^{\text{mini}}\right)^2}} \frac{\varrho_{i,j}^{\text{mini}}}{\sqrt{\vartheta_{i,j}^{\text{mini}}}} \right). \end{aligned} \tag{30}$$

Similarly, we have:

$$\frac{\partial L^{\text{mini}} \left( \phi^{\text{mini}}, \psi^{\text{mini}} \right)}{\partial \psi^{\text{mini}}} = \psi^{\text{mini}} \Gamma^{\text{mini}} - \frac{\psi^{\text{mini}} \Upsilon^{\text{mini}}}{\sqrt{\left(\phi^{\text{mini}}\right)^2 + \left(\psi^{\text{mini}}\right)^2}} + \frac{\phi^{\text{mini}} \psi^{\text{mini}} \Xi^{\text{mini}}}{\left(\phi^{\text{mini}}\right)^2 + \left(\psi^{\text{mini}}\right)^2}, \tag{31}$$

and

$$\frac{\partial L^{\text{full}} \left( \phi^{\text{mini}}, \psi^{\text{mini}}, \tilde{\mathbf{A}}_{\text{train}}^{\text{mini}} \right)}{\partial \psi^{\text{mini}}} = \psi^{\text{mini}} \Gamma^{\text{full-mini}} - \frac{\psi^{\text{mini}} \Upsilon^{\text{full-mini}}}{\sqrt{\left(\phi^{\text{mini}}\right)^2 + \left(\psi^{\text{mini}}\right)^2}} + \frac{\phi^{\text{mini}} \psi^{\text{mini}} \Xi^{\text{full-mini}}}{\left(\phi^{\text{mini}}\right)^2 + \left(\psi^{\text{mini}}\right)^2}. \tag{32}$$

## C.3 PROOF OF THEOREM 1

**Lemma C.1** $\frac{1}{\pi b} \|\mathbf{A}_{\text{train},t}^{\text{mini}} \mathbb{1}\|_1 \leq \Gamma_t^{\text{mini}}, \Gamma_t^{\text{full-mini}} \leq \frac{1}{b} \left\| \tilde{\mathbf{A}}_{\text{train}}^{\text{mini}} \mathbb{1} \right\|_1$, $|\Upsilon_t^{\text{mini}}| \leq \Gamma_t^{\text{mini}}$ and $|\Upsilon_t^{\text{full-mini}}| \leq \Gamma_t^{\text{full-mini}}$, where $b^{\frac{1}{2}} \beta^{-\frac{1}{2}} \leq \|\mathbf{A}_{\text{train},t}^{\text{mini}} \mathbb{1}\|_1 \leq b\beta$.

**Lemma C.2** If $\mathbf{w}_{j,0}^{\text{mini}} \sim N(0, \kappa^2 \mathbf{I})$ and the learning rate $\eta_t \in (0, \frac{1}{6\pi \Gamma_t^{\text{mini}}}]$, then with probability at least $1 - e^{-O(1)}$, it holds that for all $t > 0$, $\sqrt{\left(\phi_t^{\text{mini}}\right)^2 + \left(\psi_t^{\text{mini}}\right)^2} \leq C$, and $\sqrt{\left(\phi_t^{\text{mini}}\right)^2 + \left(\psi_t^{\text{mini}}\right)^2} > 0$ for all $t \geq 1$, where $C = \frac{\pi}{2} + O\left(\kappa\sqrt{r}\right)$ is a positive constant.

**Lemma C.3** If $\mathbf{w}_{j,0}^{\text{mini}} \sim N(0, \kappa^2 \mathbf{I})$ and the learning rate $\eta_t \in (0, \frac{1}{6\pi \Gamma_t^{\text{mini}}}]$, then for all $t \geq 1$ and any $C_1 \in [0,1]$ such that $\left(\phi^{\text{mini}}, \psi^{\text{mini}}\right) = (1 - C_1)\left(\phi_t^{\text{mini}}, \psi_t^{\text{mini}}\right) + C_1\left(\phi_{t+1}^{\text{mini}}, \psi_{t+1}^{\text{mini}}\right)$, we have that,

$$\lambda_{\max}(\nabla^2 L_{\text{train}}^{\text{full}}(\phi^{\text{mini}}, \psi^{\text{mini}}, \tilde{\mathbf{A}}_{\text{train}}^{\text{mini}})) \leq C_2 \Gamma_t^{\text{full-mini}},$$

where $\lambda_{\max}$ is the maximum eigenvalue of the population Hessian denoted by $\nabla^2 L_{\text{train}}^{\text{full}} \left( \phi^{\text{mini}}, \psi^{\text{mini}}, \tilde{\mathbf{A}}_{\text{train}}^{\text{mini}} \right)$, and $C_2 = 4\left(1 + \sqrt{\frac{\pi}{2} + O\left(\kappa\sqrt{r}\right)} + o\left(1\right)\right)$ is a positive constant.

**Lemma C.4** If $\mathbf{w}_{j,0}^{\text{mini}} \sim N(0, \kappa^2 \mathbf{I})$ and the learning rate $\eta_t \in (0, \frac{1}{6\pi \Gamma_t^{\text{mini}}}]$, then with at least $1 - 1/h^2$, it holds that for all $t \geq C_3 \log\left(\log h\right)$, $\sqrt{\left(\phi_t^{\text{mini}}\right)^2 + \left(\psi_t^{\text{mini}}\right)^2} \geq 1 - o\left(1\right)$, where $C_3 > 0$ is an absolute constant.

**Lemma C.5** If $\mathbf{w}_{j,0}^{\text{mini}} \sim N(0, \kappa^2 \mathbf{I})$ and the learning rate $\eta \in (0, \frac{1}{6\pi \Gamma^{\text{mini}}}]$, then there is an absolute constant $C_3$, such that for all $t \geq C_3 \log\left(\log h\right)$, either $\left|\psi_t^{\text{mini}}\right| \leq \frac{\epsilon^{\frac{1}{2}}}{2h}$ and $\left\| \sqrt{\left(\phi_t^{\text{mini}}\right)^2 + \left(\psi_t^{\text{mini}}\right)^2} - 1 \right\| \leq \frac{\epsilon^{\frac{1}{2}}}{2h}$ or we have that

$$\left\| \nabla L_{\text{train}}^{\text{full}} \left( \phi_t^{\text{mini}}, \psi_t^{\text{mini}}, \tilde{\mathbf{A}}_{\text{train}}^{\text{mini}} \right) \right\|^2 \geq \mu_t^{\text{mini}} L_{\text{train}}^{\text{full}} \left( \phi_t^{\text{mini}}, \psi_t^{\text{mini}}, \tilde{\mathbf{A}}_{\text{train}}^{\text{mini}} \right),$$

where $\mu_t^{\text{mini}} \geq C_4 \epsilon h^{-2} \beta^{-2} \Gamma^{\text{full-mini}}$, and $C_4$ is a positive constant.

**Lemma C.6** [Lemma G.2 in (Du et al., 2018)] Regarding $n$ random variables $u_1, \ldots, u_n$ satisfying $\sum_{i=1}^{n} u_i = 0$. Let $\mathcal{B} \in [n]$ denote a subset of $[n]$ and $|\mathcal{B}| = b \leq n$, the following holds,

$$\mathbb{E}\left[\left(\frac{1}{b}\sum_{i\in\mathcal{B}}u_i\right)^2\right] \leq \frac{1}{b}\mathbb{E}\left[u_i^2\right].$$

**Proof of Theorem 1:**  For any $t > 0$, taking expectation conditioning on $\mathbf{w}_{j,t+1}^{\text{mini}}$ gives:

$$
\begin{aligned}
&\mathbb{E}\left[L_{\text{train}}^{\text{full}}(\mathbf{w}_{j,t+1}^{\text{mini}}, \tilde{\mathbf{A}}_{\text{train}}^{\text{mini}})|\mathbf{w}_{j,t}^{\text{mini}}\right]\\
\leq& L_{\text{train}}^{\text{full}}\left(\mathbf{w}_{j,t}^{\text{mini}}, \tilde{\mathbf{A}}_{\text{train}}^{\text{mini}}\right) + \nabla L_{\text{train}}^{\text{full}}\left(\mathbf{w}_{j,t}^{\text{mini}}, \tilde{\mathbf{A}}_{\text{train}}^{\text{mini}}\right)\mathbb{E}\left[\left(\mathbf{w}_{j,t+1}^{\text{mini}} - \mathbf{w}_{j,t}^{\text{mini}}\right)|\mathbf{w}_{j,t}^{\text{mini}}\right]\\
&+ \frac{C^{\text{mini}}}{2}\mathbb{E}\left[\left\|\mathbf{w}_{j,t+1}^{\text{mini}} - \mathbf{w}_{j,t}^{\text{mini}}\right\|^2|\mathbf{w}_{j,t}^{\text{mini}}\right]
\end{aligned}
\tag{33}
$$

Furthermore, using Lemma C.6, we have:

$$
\begin{aligned}
&\mathbb{E}\left[\left\|\mathbf{w}_{j,t+1}^{\text{mini}} - \mathbf{w}_{j,t}^{\text{mini}}\right\|^2|\mathbf{w}_{j,t}^{\text{mini}}\right]\\
=& \eta_t^2\mathbb{E}\left[\left\|\nabla L_{\text{train}}^{\text{mini}}\left(\mathbf{w}_{j,t}^{\text{mini}}, \tilde{\mathbf{A}}_{\text{train}}^{\text{mini}}\right)\right\|_F^2|\mathbf{w}_{j,t}^{\text{mini}}\right]\\
\leq& \eta_t^2(\mathbb{E}\left[\left\|\nabla L_{\text{train}}^{\text{mini}}\left(\mathbf{w}_{j,t}^{\text{mini}}, \tilde{\mathbf{A}}_{\text{train}}^{\text{mini}}\right) - \nabla L_{\text{train}}^{\text{full}}\left(\mathbf{w}_{j,t}^{\text{mini}}, \tilde{\mathbf{A}}_{\text{train}}^{\text{mini}}\right)\right\|^2|\mathbf{w}_{j,t}^{\text{mini}}\right]\\
&+ \left\|\nabla L_{\text{train}}^{\text{full}}\left(\mathbf{w}_{j,t}^{\text{mini}}, \tilde{\mathbf{A}}_{\text{train}}^{\text{mini}}\right)\right\|^2)\\
\leq& \eta_t^2\left(\frac{n_{\text{train}}^2}{n_{\text{train}}b}\left\|\nabla L_{\text{train}}^{\text{full}}\left(\mathbf{w}_{j,t}^{\text{mini}}, \tilde{\mathbf{A}}_{\text{train}}^{\text{mini}}\right)\right\|^2 + \left\|\nabla L_{\text{train}}^{\text{full}}\left(\mathbf{w}_{j,t}^{\text{mini}}, \tilde{\mathbf{A}}_{\text{train}}^{\text{mini}}\right)\right\|^2\right)\\
\leq& \eta_t^2\left(\frac{2n_{\text{train}}}{b}\left\|\nabla L_{\text{train}}^{\text{full}}\left(\mathbf{w}_{j,t}^{\text{mini}}, \tilde{\mathbf{A}}_{\text{train}}^{\text{mini}}\right)\right\|^2\right).
\end{aligned}
\tag{34}
$$

Moreover, we have:

$$
\begin{aligned}
&\nabla L_{\text{train}}^{\text{full}}\left(\mathbf{w}_{j,t}^{\text{mini}}, \tilde{\mathbf{A}}_{\text{train}}^{\text{mini}}\right)\mathbb{E}\left[\left(\mathbf{w}_{j,t+1}^{\text{mini}} - \mathbf{w}_{j,t}^{\text{mini}}\right)|\mathbf{w}_{j,t+1}^{\text{mini}}\right]\\
=& -\eta_t\nabla L_{\text{train}}^{\text{full}}\left(\mathbf{w}_{j,t}^{\text{mini}}, \tilde{\mathbf{A}}_{\text{train}}^{\text{mini}}\right)\mathbb{E}\left[\nabla L_{\text{train},t}^{\text{mini}}\left(\mathbf{w}_{j,t}^{\text{mini}}\right)\mathbb{E}|\mathbf{w}_{j,t+1}^{\text{mini}}\right]\\
=& -\eta_t\left\|\nabla L_{\text{train}}^{\text{full}}\left(\mathbf{w}_{j,t}^{\text{mini}}, \tilde{\mathbf{A}}_{\text{train}}^{\text{mini}}\right)\right\|^2
\end{aligned}
\tag{35}
$$

Hence, we have:

$$
\begin{aligned}
&\mathbb{E}\left[L_{\text{train}}^{\text{full}}(\mathbf{w}_{j,t+1}^{\text{mini}}, \tilde{\mathbf{A}}_{\text{train}}^{\text{mini}})|\mathbf{w}_{j,t}^{\text{mini}}\right]\\
\leq& L_{\text{train}}^{\text{full}}\left(\mathbf{w}_{j,t}^{\text{mini}}, \tilde{\mathbf{A}}_{\text{train}}^{\text{mini}}\right) - \eta_t\left\|\nabla L_{\text{train}}^{\text{full}}\left(\mathbf{w}_{j,t}^{\text{mini}}, \tilde{\mathbf{A}}_{\text{train}}^{\text{mini}}\right)\right\|^2\\
&+ \frac{C^{\text{mini}}n_{\text{train}}}{b}\eta_t^2\left\|\nabla L_{\text{train}}^{\text{full}}\left(\mathbf{w}_{j,t}^{\text{mini}}, \tilde{\mathbf{A}}_{\text{train}}^{\text{mini}}\right)\right\|^2\\
\leq& L_{\text{train}}^{\text{full}}\left(\mathbf{w}_{j,t}^{\text{mini}}, \tilde{\mathbf{A}}_{\text{train}}^{\text{mini}}\right) - \eta_t\left\|\nabla L_{\text{train}}^{\text{full}}\left(\mathbf{w}_{j,t}^{\text{mini}}, \tilde{\mathbf{A}}_{\text{train}}^{\text{mini}}\right)\right\|^2\left(1 - \frac{C^{\text{mini}}n_{\text{train}}}{b}\eta_t\right)
\end{aligned}
\tag{36}
$$

By Lemma C.2, we know that $\eta_t \in (0, \frac{1}{6\pi\Gamma_t^{\text{mini}}}]$. Using Lemma C.1, we first assume that $\frac{C^{\text{mini}}}{6\pi} \leq \beta \leq \left(\frac{1}{6C_6}\right)^{\frac{1}{4}}b^{\frac{3}{4}}$ where $C_6 < \frac{1}{6}$ is a positive constant. Then, we set $\eta_t \in \left[\frac{C_6\beta^3}{\pi n_{\text{train}}b^2}, \frac{b}{6\pi\beta n_{\text{train}}}\right]$. We are going to prove $\eta_t \in \left[\frac{C_6\beta^3}{\pi n_{\text{train}}b^2}, \frac{b}{6\pi\beta n_{\text{train}}}\right]$ is still within the range $(0, \frac{1}{6\pi\Gamma^{\text{mini}}}]$ and $\frac{C_6\beta^3}{\pi n_{\text{train}}b^2} \leq \frac{b}{6\pi\beta n_{\text{train}}}$.

For the right side of the range, we have $\frac{b}{6\pi\beta n_{\text{train}}} \leq \frac{b}{6\pi\Gamma_t^{\text{mini}} n_{\text{train}}} \leq \frac{1}{6\pi\Gamma_t^{\text{mini}}}$ due to $\Gamma_t^{\text{mini}} \leq \beta$ and $b \leq n_{\text{train}}$. For the left side of the range, $\frac{C_6\beta^3}{\pi n_{\text{train}} b^2}$ with the positive constant $C_6$. Moreover, we have:

$$
\begin{aligned}
\frac{b}{6\pi\beta n_{\text{train}}} - \frac{C_6\beta^3}{\pi n_{\text{train}} b^2} &= \frac{b}{\pi n_{\text{train}}}\left(\frac{1}{6\beta} - \frac{C_6\beta^3}{b^3}\right) \\
&\geq \frac{1}{\pi}\left(\frac{1}{6\beta} - \frac{1}{6\beta}\right) = 0.
\end{aligned}
\tag{37}
$$

With $\eta_t \in \left[\frac{C_6\beta^3}{\pi n_{\text{train}} b^2}, \frac{b}{6\pi\beta n_{\text{train}}}\right]$, we have:

$$
\frac{C^{\text{mini}} n_{\text{train}}}{b}\eta_t \leq \frac{C^{\text{mini}}}{6\pi\beta} \leq 1.
\tag{38}
$$

Furthermore, using Lemma C.5, we have

$$
\begin{aligned}
L_{\text{train}}^{\text{full}}(\mathbf{w}_{j,t+1}^{\text{mini}}, \tilde{\mathbf{A}}_{\text{train}}^{\text{mini}}) &\leq L_{\text{train}}^{\text{full}}\left(\mathbf{w}_{j,t}^{\text{mini}}, \tilde{\mathbf{A}}_{\text{train}}^{\text{mini}}\right)(1 - \eta_t\mu_t^{\text{mini}}) \\
&\leq L_{\text{train}}^{\text{mini}}(\mathbf{w}_{j,0}^{\text{mini}}, \tilde{\mathbf{A}}_{\text{train}}^{\text{mini}})\prod_{\tau=1}^{t}(1 - \eta_\tau\mu_\tau^{\text{mini}}) \\
&\leq L_{\text{train}}^{\text{mini}}(\mathbf{w}_{j,0}^{\text{mini}}, \tilde{\mathbf{A}}_{\text{train}}^{\text{mini}})(1 - \frac{1}{t}\sum_{\tau=1}^{t}\eta_\tau\mu_\tau^{\text{mini}})^t,
\end{aligned}
\tag{39}
$$

where the last inequality can be proved: $f(x) = \log(1 - x)$ is a concave function on $0 < x < 1$, then, for $0 < x_i < 1$ with $i = \{1, \ldots, n\}$, we have $f(\frac{1}{n}\sum_{i=1}^{n} x_i) \geq \frac{1}{n}\sum_{i=1}^{n} f(x_i)$, which can be written as $\log(1 - \frac{1}{n}\sum_{i=1}^{n} x_i) \geq \frac{1}{n}\sum_{i=1}^{n}\log(1 - x_i)$. Therefore, we have $(1 - \frac{1}{n}\sum_{i=1}^{n} x_i)^n \geq \prod_{i=1}^{n}(1 - x_i)$.

Then we have:

$$
T \leq C_7 \log\left(\frac{h^2}{\epsilon}\right)\frac{1}{\frac{1}{T}\sum_{\tau=1}^{T}\eta_\tau\mu_\tau^{\text{mini}}},
\tag{40}
$$

where $C_7$ is a positive constant.

Moreover, we have:

$$
\begin{aligned}
\frac{1}{T}\sum_{\tau=1}^{T}\eta_\tau\mu_\tau^{\text{mini}} &\geq \frac{1}{T}\sum_{\tau=1}^{T}\frac{C_4 C_6\beta\epsilon\Gamma_\tau^{\text{full-mini}}}{\pi n_{\text{train}} b^2 h^2} \\
&\geq \frac{1}{T}\sum_{\tau=1}^{T}\frac{C_4 C_6\beta^{\frac{1}{2}}\epsilon}{\pi^2 n_{\text{train}} b^{\frac{5}{2}} h^2} \\
&= \frac{C_4 C_6\beta^{\frac{1}{2}}\epsilon}{\pi^2 n_{\text{train}} b^{\frac{5}{2}} h^2}
\end{aligned}
\tag{41}
$$

Hence, we have $T = O\left(\frac{n_{\text{train}} b^{\frac{5}{2}} h^2}{\epsilon\beta^{\frac{1}{2}}}\log\frac{h^2}{\epsilon}\right)$.

After $T$ time steps, we either have $L_{\text{train}}^{\text{full}}\left(\mathbf{w}_{j,t}^{\text{mini}}, \tilde{\mathbf{A}}_{\text{train}}^{\text{mini}}\right) \leq \frac{\epsilon}{h}$, or that $\psi_t^{\text{mini}} \leq \frac{\epsilon^{\frac{1}{2}}}{2h}$ and $(\phi_t^{\text{mini}})^2 + (\psi_t^{\text{mini}})^2 - 1 \leq \frac{\epsilon^{\frac{1}{2}}}{2h}$. The latter implies that $\left\|\mathbf{w}_{j,t}^{\text{mini}} - \mathbf{w}_{j,t}^{\text{mini}*}\right\|^2 \leq \frac{\epsilon}{h}$. In addition, it is easy to see that $L_{\text{train}}^{\text{full}}\left(\mathbf{w}_{j,t}^{\text{mini}}, \tilde{\mathbf{A}}_{\text{train}}^{\text{mini}}\right) \leq \|\mathbf{w}_{j,t}^{\text{mini}} - \mathbf{w}_{j,t}^{\text{mini}*}\|^2$. Hence, if the latter happens, then $L_{\text{train}}^{\text{full}}\left(\mathbf{w}_{j,t}^{\text{mini}}, \tilde{\mathbf{A}}_{\text{train}}^{\text{mini}}\right) \leq \frac{\epsilon}{h}$. Hence $L_{\text{train}}^{\text{full}}(\mathbf{W}_T^{\text{mini}}) \leq \epsilon$.

This completes the proof.

## D    PROOF OF CONVERGENCE THEOREM IN FULL-GRAPH TRAINING WITH CE

In this section, we provide the proof of the convergence theorem in full-graph training with CE. To simplify the analysis, we focus on binary node classification using a one-round GNN trained with the

CE, defined as $l\left(\mathbf{W}, \tilde{\mathbf{a}}_{\text{train},i}^{\text{full}}\right) = \log\left(1 + \exp\left(-y_i \hat{y}_i\right)\right)$. The final output of the GNN model is given by $\hat{y}_i = \mathbf{z}_i \mathbf{v}^\top = \sigma\left(\tilde{\mathbf{a}}_{\text{train},i}^{\text{full}} \mathbf{X} \mathbf{W}^\top\right) \mathbf{v}^\top, \forall i \in$ training set, where $\mathbf{v} \in \{-1, +1\} \in \mathbb{R}^{1 \times h}$ is the fixed output layer vector with half $1$ and half $-1$. The rows of $\mathbf{W}$ are initialized independently from a Gaussian distribution $N\left(0, \kappa^2 \mathbf{I}\right)$.

We decompose the analysis of GNN optimization dynamic into three steps.

*Step 1: Reformulating loss and gradient expressions on irregular graphs.* We represent the ReLU function implicitly using a position-wise 0/1 indicator matrix that can directly multiply the aggregated node features.

*Step 2: Bounding the norm of gradient.* Based on the reformulated loss and gradient expressions, we aim to quantify the magnitude of optimization updates by bounding the gradient norm, facilitating convergence analysis. We can bound the Frobenius norm of the gradient by the average of individual node-level gradients.

*Step 3: Bounding the number of iterations to Convergence.* We first leverage the smoothness of the loss function to derive a per-iteration inequality relating loss reduction to the gradient norm, and then accumulate these iteration-wise inequalities over GD updates to obtain an upper bound on the number of iterations required for convergence.

## D.1 ASSUMPTION

We still use Assumptions D.3 on the training data.

**Assumption D.1.** $\forall i, i' \in$ training set, if $y_i \neq y_{i'}$, then $\|\tilde{\mathbf{a}}_{\text{train},i}^{\text{full}} \mathbf{X} - \tilde{\mathbf{a}}_{\text{train},i'}^{\text{full}} \mathbf{X}\|_2 \geq \alpha$ for some $\alpha > 0$.

Assumption D.1 requires that aggregated node features with different labels in the training data are separated by at least a constant, which is often satisfied in practice and can be easily verified based on the training data. A similar assumption on the non-aggregated features $\|\mathbf{x}_i - \mathbf{x}_{i'}\|_2$ has been adopted in prior analyses of the DNN optimization dynamics without message passing (Zou et al., 2020a; 2018).

## D.2 EXPRESSIONS FOR GRADIENTS FOR CE LOSS.

We first provide some basic expressions regarding the gradients for the CE loss in the GNN under our setting. Note that the node classification task in this case is binary, denoted as $K = 2$.

**Output after the 1-st layer:** Given an input $\mathbf{X}$, the $i$-th column of output after the first layer of the GNN under the full-graph training is

$$\mathbf{z}_i^{\text{full}} = \sigma\left(\tilde{\mathbf{a}}_{\text{train},i}^{\text{full}} \mathbf{X}\left(\mathbf{W}^{\text{full}}\right)^\top\right) = \tilde{\mathbf{a}}_{\text{train},i}^{\text{full}} \mathbf{X}(\mathbf{\Sigma}_i^{\text{full}} \mathbf{W}^{\text{full}})^\top, \tag{42}$$

where $\mathbf{\Sigma}_i^{\text{full}} = Diag\left(\mathbb{1}\left\{\tilde{\mathbf{a}}_{\text{train},i}^{\text{full}} \mathbf{X}\left(\mathbf{W}^{\text{full}}\right)^\top > 0\right\}\right) \in \mathbb{R}^{h \times h}$ represents whether the $j$-th element $\left\{\tilde{\mathbf{a}}_{\text{train},i} \mathbf{X}\left(\mathbf{W}^{\text{full}}\right)^\top\right\}_j$ is more than zero (1) or is zeroed out (0). Here we slightly abuse the notation and denote $\mathbb{1}\{\mathbf{x} > 0\} = (\mathbb{1}\{\mathbf{x}_1 > 0\}, \ldots, \mathbb{1}\{\mathbf{x}_m > 0\})^\top$ for a vector $\mathbf{x} \in \mathbb{R}^m$.

**Output of one-round GNN for the CE loss:** The output of the one-round GNN for the CE loss with input $\mathbf{X}$ under the full-graph training can be expressed as:

$$\hat{y}_i^{\text{full}} = \sigma\left(\tilde{\mathbf{a}}_{\text{train},i}^{\text{full}} \mathbf{X}\left(\mathbf{W}^{\text{full}}\right)^\top\right) \mathbf{v}^\top = \tilde{\mathbf{a}}_{\text{train},i}^{\text{full}} \mathbf{X}(\mathbf{\Sigma}_i^{\text{full}} \mathbf{W}^{\text{full}})^\top \mathbf{v}^\top, \tag{43}$$

where $\mathbf{v} \in \{-1, +1\} \in \mathbb{R}^{1 \times h}$ is the fixed output layer weight vector with half $1$ and half $-1$, corresponding to the binary classification task setting in this case.

**Gradient for CE loss in GNN:** The partial gradient of the training loss $\hat{L}_{\text{train}}^{\text{full}}\left(\mathbf{W}^{\text{full}}\right)$ with respect to $\mathbf{W}^{\text{full}}$ under full-graph training can be expressed as:

$$\nabla \hat{L}_{\text{train}}^{\text{full}}\left(\mathbf{W}^{\text{full}}\right) = \frac{1}{n_{\text{train}}} \sum_{i=0}^{n_{\text{train}}} l'\left(y_i \hat{y}_i^{\text{full}}\right) \cdot y_i \cdot \nabla_{\mathbf{W}^{\text{full}}}\left[\hat{y}_i^{\text{full}}\right], \tag{44}$$

where the gradient of the GNN is defined as $\nabla_{\mathbf{W}^{\text{full}}}\left[\hat{y}_i^{\text{full}}\right] = \left(\mathbf{v}\boldsymbol{\Sigma}_i^{\text{full}}\right)^\top \tilde{\mathbf{a}}_{\text{train},i}^{\text{full}}\mathbf{X}.$

### D.3 THEOREM D.2.

**Theorem D.2.** (Convergence of Full-graph Training with CE) *Suppose $\mathbf{W}^{full}$ are generated by Gaussian initialization. Under Assumptions D.3 and D.1, if the hidden dimension of a one-round GNN satisfies $h = \Omega\left(\log\left(n_{train}\right) d_{max}^{-1}\left(n_{train}^2 + \epsilon^{-1}\right)\right)$, then with high probability, the training loss satisfies $\hat{L}_{train}\left(\mathbf{W}_T^{full}, \mathbf{A}_{train}^{full}\right) \leq \epsilon$, provided that the number of iterations $T = O\left(n_{train}\left(\log\left(n_{train}\right)\right)^{\frac{1}{2}}\alpha^{-2}d_{max}^{-\frac{5}{2}}\left(n_{train}^2 + \epsilon^{-1}\right)\right)$ for any $\epsilon \geq 0$ under the full-graph training.*

### D.4 PROOF OF THEOREM D.2.

We first provide the following lemmas.

**Lemma D.3** (Bounded initial training loss) Under Assumptions D.3 and D.1, with the probability at least $1 - \delta$, at the initialization the training loss satisfies $\hat{L}_{\text{train}}^{\text{full}}\left(\mathbf{W}_0^{\text{full}}\right) \leq C\sqrt{d_{\max}\log(n_{\text{train}}/\delta)}$, where $C$ is an absolute constant.

**Lemma D.4** (Gradient lower and upper bound) Under Assumptions D.3 and D.1, with the probability at least $1 - \exp\left(-C_1 h\alpha^2/n_{\text{train}}^2\right)$, there exist positive constants $C_1$, $C_2$ and $C_3$, such that

$$\left\|\nabla_{\mathbf{W}^{\text{full}}}\hat{L}_{\text{train}}^{\text{full}}\left(\mathbf{W}^{\text{full}}\right)\right\|_F^2 \geq \frac{C_2 h\alpha^2 d_{\max}^3}{n_{\text{train}}^3}\left(\sum_{i=1}^{n_{\text{train}}} l'\left(y_i\hat{y}_i^{\text{full}}\right)\right)^2,$$

$$\left\|\nabla_{\mathbf{W}^{\text{full}}}\hat{L}_{\text{train}}^{\text{full}}\left(\mathbf{W}^{\text{full}}\right)\right\|_F \leq -\frac{C_3 h^{\frac{1}{2}} d_{\max}^{\frac{1}{2}}}{n_{\text{train}}}\sum_{i=1}^{n_{\text{train}}} l'\left(y_i\hat{y}_i^{\text{full}}\right).$$

**Lemma D.4** (Sufficient descent) Let $\mathbf{W}_0^{\text{full}}$ be generated via Gaussian random initialization. Let $\mathbf{W}_t^{\text{full}}$ be the $t$-th iterate in the gradient descent. If $\mathbf{W}_t^{\text{full}}, \mathbf{W}_{t+1}^{\text{full}} \in \mathcal{B}\left(\mathbf{W}_0^{\text{full}}, \tau\right)$ and $\tau \leq C_6/(\alpha d_{\max}^{\frac{1}{2}})$, then there exist constants $C_4$, $C_5$ and $C_6$ such that, with probability at least $1 - \exp(-O(1))$, the following holds:

$$\hat{L}_{\text{train}}^{\text{full}}\left(\mathbf{W}_{t+1}^{\text{full}}\right) - \hat{L}_{\text{train}}^{\text{full}}\left(\mathbf{W}_t^{\text{full}}\right) \leq -(\eta - C_4 d_{\max} h\eta^2)\left\|\nabla_{\mathbf{W}^{\text{full}}}\hat{L}_{\text{train}}^{\text{full}}\left(\mathbf{W}_t^{\text{full}}\right)\right\|_F^2$$
$$-\frac{C_5\eta d_{\max}^{\frac{1}{2}} h^{\frac{1}{2}}\left\|\nabla_{\mathbf{W}^{\text{full}}}\hat{L}_{\text{train}}^{\text{full}}\left(\mathbf{W}_t^{\text{full}}\right)\right\|_F}{n_{\text{train}}\tau}\sum_{i=1}^{n_{\text{train}}} l'\left(y_i\hat{y}_{i,t}^{\text{full}}\right).$$

**Proof of Theorem D.2:** We first prove that gradient descent can achieve the training loss at the value of $\epsilon$ under the condition that all iterates are staying inside the perturbation region $\mathcal{B}\left(\mathbf{W}_0^{\text{full}}, \tau\right) = \left\{\mathbf{W} : \left\|\mathbf{W} - \mathbf{W}_0^{\text{full}}\right\|_2 \leq \tau\right\}$.

Using Lemma D.4, there exists a constant $C_2$ such that

$$\left\|\nabla_{\mathbf{W}^{\text{full}}}\hat{L}_{\text{train}}^{\text{full}}\left(\mathbf{W}_t^{\text{full}}\right)\right\|_F^2 \geq \frac{C_2 h\alpha^2 d_{\max}^3}{n_{\text{train}}^3}\left(\sum_{i=1}^{n_{\text{train}}} l'\left(y_i\hat{y}_{i,t}^{\text{full}}\right)\right)^2 \tag{45}$$

We then set the step size $\eta$ and the radius $\tau$ as follows:

$$\eta = \frac{1}{4C_4 d_{\max} h} = O\left(d_{\max}^{-1} h^{-1}\right), \tag{46}$$

$$\tau = \frac{4C_5 n_{\text{train}}^{\frac{1}{2}}}{C_2 \alpha d_{\max}} = O\left(n_{\text{train}}^{\frac{1}{2}} d_{\max}^{-1} \alpha^{-1}\right). \tag{47}$$

Then we have

$$
\begin{aligned}
&\hat{L}_{\text{train}}^{\text{full}}\left(\mathbf{W}_{t+1}^{\text{full}}\right) - \hat{L}_{\text{train}}^{\text{full}}\left(\mathbf{W}_{t}^{\text{full}}\right) \\
&\leq -\frac{3}{4}\eta \left\|\nabla_{\mathbf{W}^{\text{full}}}\hat{L}_{\text{train}}^{\text{full}}\left(\mathbf{W}_{t}^{\text{full}}\right)\right\|_F^2 - \frac{C_2 \eta h^{\frac{1}{2}} \alpha d_{\max}^{\frac{3}{2}}}{4 n_{\text{train}}^{\frac{3}{2}}} \left\|\nabla_{\mathbf{W}^{\text{full}}}\hat{L}_{\text{train}}^{\text{full}}\left(\mathbf{W}_{t}^{\text{full}}\right)\right\|_F \sum_{i=1}^{n_{\text{train}}} l'\left(y_i \hat{y}_{i,t}^{\text{full}}\right) \\
&\leq -\frac{3}{4}\eta \left\|\nabla_{\mathbf{W}^{\text{full}}}\hat{L}_{\text{train}}^{\text{full}}\left(\mathbf{W}_{t}^{\text{full}}\right)\right\|_F^2 + \frac{\eta}{4}\left\|\nabla_{\mathbf{W}^{\text{full}}}\hat{L}_{\text{train}}^{\text{full}}\left(\mathbf{W}_{t}^{\text{full}}\right)\right\|_F^2 \\
&= -\frac{1}{2}\eta \left\|\nabla_{\mathbf{W}^{\text{full}}}\hat{L}_{\text{train}}^{\text{full}}\left(\mathbf{W}_{t}^{\text{full}}\right)\right\|_F^2 \\
&\leq -\eta \frac{C_2 h \alpha^2 d_{\max}^3}{2 n_{\text{train}}^3}\left(\sum_{i=0}^{n_{\text{train}}} l'\left(y_i \hat{y}_{i,t}^{\text{full}}\right)\right)^2,
\end{aligned}
\tag{48}
$$

where the first inequality is derived from Lemma D.4 and the settings of $\eta$ and $\tau$, the second inequality is derived from Lemma D.4, as well as the last inequality follows the gradient lower bound in Lemma D.4.

We note that $l(x) = \log(1 + \exp(-x))$ satisfies $-l'(x) = 1/(1 + \exp(x)) \geq \min\{u_0, u_1 l(x)\}$, where $u_0 = 1/2$, $u_1 = 1/(2\log(2))$. This implies that:

$$
\begin{aligned}
-\sum_{i=0}^{n_{\text{train}}} l'\left(y_i \hat{y}_{i,t}^{\text{full}}\right) &\geq \min\left\{u_0, \sum_{i=0}^{n_{\text{train}}} u_1 l'\left(y_i \hat{y}_{i,t}^{\text{full}}\right)\right\} \\
&\geq \min\left\{u_0, n_{\text{train}} u_0 \hat{L}_{\text{train}}^{\text{full}}\left(\mathbf{W}_{t}^{\text{full}}\right)\right\}.
\end{aligned}
\tag{49}
$$

Since $\min\{a, b\} \geq 1/(1/a + 1/b)$, we have:

$$
\begin{aligned}
&\hat{L}_{\text{train}}^{\text{full}}\left(\mathbf{W}_{t+1}^{\text{full}}\right) - \hat{L}_{\text{train}}^{\text{full}}\left(\mathbf{W}_{t}^{\text{full}}\right) \\
&\leq -\eta \min\left\{\frac{C_2 h \alpha^2 d_{\max}^3}{2 n_{\text{train}}^3} u_0^2, \frac{C_2 h \alpha^2 d_{\max}^3}{2 n_{\text{train}}} u_1^2 \left(\hat{L}_{\text{train}}^{\text{full}}\left(\mathbf{W}_{t}^{\text{full}}\right)\right)^2\right\} \\
&\leq -\eta \left(\frac{2 n_{\text{train}}^3}{C_2 h \alpha^2 d_{\max}^3 u_0^2} + \frac{2 n_{\text{train}}}{C_2 h \alpha^2 d_{\max}^3 u_1^2 \left(\hat{L}_{\text{train}}^{\text{full}}\left(\mathbf{W}_{t}^{\text{full}}\right)\right)^2}\right)^{-1}.
\end{aligned}
\tag{50}
$$

Rearranging terms, we have:

$$\frac{2 n_{\text{train}}^2}{C_2 h \alpha^2 d_{\max}^2 u_0^2}\left(\hat{L}_{\text{train}}^{\text{full}}\left(\mathbf{W}_{t+1}^{\text{full}}\right) - \hat{L}_{\text{train}}^{\text{full}}\left(\mathbf{W}_{t}^{\text{full}}\right)\right) + \frac{2\left(\hat{L}_{\text{train}}^{\text{full}}\left(\mathbf{W}_{t+1}^{\text{full}}\right) - \hat{L}_{\text{train}}^{\text{full}}\left(\mathbf{W}_{t}^{\text{full}}\right)\right)}{C_2 h \alpha^2 d_{\max}^2 u_1^2 \left(\hat{L}_{\text{train}}^{\text{full}}\left(\mathbf{W}_{t}^{\text{full}}\right)\right)^2} \leq -\eta. \tag{51}$$

Using $(x - y)/y^2 \geq y^{-1} - x^{-1}$ and taking telescope sum over $t$, we have:

$$
\begin{aligned}
t\eta &\leq \frac{2 n_{\text{train}}^3}{C_2 h \alpha^2 d_{\max}^3 u_0^2}\left(\hat{L}_{\text{train}}^{\text{full}}\left(\mathbf{W}_{0}^{\text{full}}\right) - \hat{L}_{\text{train}}^{\text{full}}\left(\mathbf{W}_{t}^{\text{full}}\right)\right) + \frac{2 n_{\text{train}}\left(\left(\hat{L}_{\text{train}}^{\text{full}}\left(\mathbf{W}_{t}^{\text{full}}\right)\right)^{-1} - \left(\hat{L}_{\text{train}}^{\text{full}}\left(\mathbf{W}_{0}^{\text{full}}\right)\right)^{-1}\right)}{C_2 h \alpha^2 d_{\max}^3 u_1^2} \\
&\leq \frac{2 n_{\text{train}}^3}{C_2 h \alpha^2 d_{\max}^3 u_0^2}\hat{L}_{\text{train}}^{\text{full}}\left(\mathbf{W}_{0}^{\text{full}}\right) + \frac{2 n_{\text{train}}\left(\left(\hat{L}_{\text{train}}^{\text{full}}\left(\mathbf{W}_{t}^{\text{full}}\right)\right)^{-1} - \left(\hat{L}_{\text{train}}^{\text{full}}\left(\mathbf{W}_{0}^{\text{full}}\right)\right)^{-1}\right)}{C_2 h \alpha^2 d_{\max}^3 u_1^2}.
\end{aligned}
\tag{52}
$$

Next, we guarantee that, after $T$ gradient steps, the loss function $\hat{L}_{\text{train}}^{\text{full}}\left(\mathbf{W}_{T}^{\text{full}}\right)$ is smaller than $\epsilon$.

Using Lemma D.3, we have $\hat{L}_{\text{train}}^{\text{full}}\left(\mathbf{W}_0^{\text{full}}\right) = O\left(d_{\max}^{\frac{1}{2}}\left(\log\left(n_{\text{train}}\right)\right)^{\frac{1}{2}}\right)$.

Therefore, $T$ satisfies:

$$T = O\left(n_{\text{train}}^3 \left(\log\left(n_{\text{train}}\right)\right)^{\frac{1}{2}} / \left(\alpha^2 d_{\max}^{\frac{5}{2}}\right) + n_{\text{train}}\left(\log\left(n_{\text{train}}\right)\right)^{\frac{1}{2}} / \left(\epsilon\alpha^2 d_{\max}^{\frac{5}{2}}\right)\right). \tag{53}$$

Then we are going to verify the condition that all iterates stay inside the perturbation region $\mathcal{B}\left(\mathbf{W}_0^{\text{full}}, \tau\right)$. Obviously, we have $\mathbf{W}_0^{\text{full}} \in \mathcal{B}\left(\mathbf{W}_0^{\text{full}}, \tau\right)$. Hence, we need to prove $\mathbf{W}_{t+1}^{\text{full}} \in \mathcal{B}\left(\mathbf{W}_0^{\text{full}}, \tau\right)$ under the induction hypothesis that $\mathbf{W}_t^{\text{full}} \in \mathcal{B}\left(\mathbf{W}_0^{\text{full}}, \tau\right)$ holds for all $t \leq T$.

Since we have $\hat{L}_{\text{train}}^{\text{full}}\left(\mathbf{W}_{t+1}^{\text{full}}\right) - \hat{L}_{\text{train}}^{\text{full}}\left(\mathbf{W}_t^{\text{full}}\right) \leq -\frac{1}{2}\eta\left\|\nabla_{\mathbf{W}^{\text{full}}}\hat{L}_{\text{train}}^{\text{full}}\left(\mathbf{W}_t^{\text{full}}\right)\right\|_F^2$ for any $t \leq T$, using triangle inequality, we have:

$$\begin{aligned}
\left\|\mathbf{W}_t^{\text{full}} - \mathbf{W}_0^{\text{full}}\right\|_2 &\leq \eta\sum_{k=0}^{t-1}\left\|\nabla_{\mathbf{W}_k^{\text{full}}}\hat{L}_{\text{train}}^{\text{full}}\left(\mathbf{W}_k^{\text{full}}\right)\right\|_F^2 \\
&\leq \eta\sqrt{t\sum_{k=0}^{t-1}\left\|\nabla_{\mathbf{W}_k^{\text{full}}}\hat{L}_{\text{train}}^{\text{full}}\left(\mathbf{W}_k^{\text{full}}\right)\right\|_F^2} \\
&\leq \sqrt{2t\eta\sum_{k=0}^{t-1}\left[\hat{L}_{\text{train}}^{\text{full}}\left(\mathbf{W}_k^{\text{full}}\right) - \hat{L}_{\text{train}}^{\text{full}}\left(\mathbf{W}_{k+1}^{\text{full}}\right)\right]} \\
&\leq \sqrt{2t\eta\hat{L}_{\text{train}}^{\text{full}}\left(\mathbf{W}_0^{\text{full}}\right)}.
\end{aligned} \tag{54}$$

Using $\hat{L}_{\text{train}}^{\text{full}}\left(\mathbf{W}_0^{\text{full}}\right) = O\left(d_{\max}^{\frac{1}{2}}\left(\log\left(n_{\text{train}}\right)\right)^{\frac{1}{2}}\right)$ in Lemma D.3 and our settings of $\eta$, we have:

$$\begin{aligned}
\left\|\mathbf{W}_t^{\text{full}} - \mathbf{W}_0^{\text{full}}\right\|_2 &\leq \sqrt{2t\eta\hat{L}_{\text{train}}^{\text{full}}\left(\mathbf{W}_0^{\text{full}}\right)} \\
&= O\left(n_{\text{train}}^{\frac{3}{2}}\left(\log\left(n_{\text{train}}\right)\right)^{\frac{1}{2}}\alpha^{-1}d_{\max}^{-\frac{3}{2}}h^{-\frac{1}{2}} + n_{\text{train}}^{\frac{1}{2}}\left(\log\left(n_{\text{train}}\right)\right)^{\frac{1}{2}}\epsilon^{-\frac{1}{2}}\alpha^{-1}d_{\max}^{-\frac{3}{2}}h^{-\frac{1}{2}}\right).
\end{aligned} \tag{55}$$

In addition, by Lemma D.4 and our choice of $\eta$, we have

$$\begin{aligned}
\eta\left\|\nabla_{\mathbf{W}^{\text{full}}}\hat{L}_{\text{train}}^{\text{full}}\left(\mathbf{W}^{\text{full}}\right)\right\|_2 &\leq -\frac{\eta C_3 h^{\frac{1}{2}}d_{\max}^{\frac{1}{2}}}{n_{\text{train}}}\sum_{i=0}^{n_{\text{train}}}l'\left(y_i\hat{y}_i^{\text{full}}\right) \\
&\leq O\left(\left(\log\left(n_{\text{train}}\right)\right)^{\frac{1}{2}}h^{-\frac{1}{2}}d_{\max}^{-\frac{1}{2}}\right),
\end{aligned} \tag{56}$$

where the second inequality is derived from the fact that $-1 \leq l'(\cdot) \leq 0$.

Therefore, by triangle inequality, we assume that $h = \Omega\left(n_{\text{train}}^2\log\left(n_{\text{train}}\right)d_{\max}^{-1} + \log\left(n_{\text{train}}\right)\epsilon^{-1}d_{\max}^{-1}\right)$ and we have:

$$\begin{aligned}
\left\|\mathbf{W}_{t+1}^{\text{full}} - \mathbf{W}_t^{\text{full}}\right\|_2 &\leq \eta\left\|\nabla_{\mathbf{W}^{\text{full}}}\hat{L}_{\text{train}}^{\text{full}}\left(\mathbf{W}_t^{\text{full}}\right)\right\|_2 + \left\|\mathbf{W}_t^{\text{full}} - \mathbf{W}_0^{\text{full}}\right\|_2 \\
&= O\left(n_{\text{train}}^{\frac{3}{2}}\left(\log\left(n_{\text{train}}\right)\right)^{\frac{1}{2}}\alpha^{-1}d_{\max}^{-\frac{3}{2}}h^{-\frac{1}{2}} + n_{\text{train}}^{\frac{1}{2}}\left(\log\left(n_{\text{train}}\right)\right)^{\frac{1}{2}}\epsilon^{-\frac{1}{2}}\alpha^{-1}d_{\max}^{-\frac{3}{2}}h^{-\frac{1}{2}}\right) \\
&= O\left(n_{\text{train}}^{\frac{1}{2}}d_{\max}^{-1}\alpha^{-1}\right),
\end{aligned} \tag{57}$$

which is exactly the same order of $\tau$ in our settings.

This verifies $\mathbf{W}_{t+1}^{\text{full}} \in \mathcal{B}\left(\mathbf{W}_0^{\text{full}}, \tau\right)$.

Proved.

# E PROOF OF CONVERGENCE THEOREM IN MINI-BATCH TRAINING WITH CE AND INTERPRETATION OF THE OBS.1

In this section, we provide the proof of the convergence theorem in mini-batch training with CE. To simplify the analysis, we focus on binary node classification using a one-round GNN trained with the CE, defined as $l\left(\mathbf{W}, \tilde{\mathbf{a}}_{\text{train},i}^{\text{mini}}\right) = \log\left(1 + \exp\left(-y_i \hat{y}_i\right)\right)$. The final output of the GNN model is given by $\hat{y}_i = \mathbf{z}_i \mathbf{v}^\top = \sigma\left(\tilde{\mathbf{a}}_{\text{train},i}^{\text{mini}} \mathbf{X} \mathbf{W}^\top\right) \mathbf{v}^\top, \forall i \in$ training set, where $\mathbf{v} \in \{-1, +1\} \in \mathbb{R}^{1 \times h}$ is the fixed output layer vector with half $1$ and half $-1$. The rows of $\mathbf{W}$ are initialized independently from a Gaussian distribution $N\left(0, \kappa^2 \mathbf{I}\right)$.

We decompose the analysis of GNN optimization dynamic into three steps, similar to Appendix D.

## E.1 ASSUMPTION

We still use Assumptions B.1 on the training data.

**Assumption E.1.** $\forall i, i' \in$ training set, if $y_i \neq y_{i'}$, then $\|\tilde{\mathbf{a}}_{\text{train},i}^{\text{mini}} \mathbf{X} - \tilde{\mathbf{a}}_{\text{train},i'}^{\text{mini}} \mathbf{X}\|_2 \geq \alpha$ for some $\alpha > 0$.

Assumption E.1 requires that aggregated node features with different labels in the training data are separated by at least a constant.

## E.2 EXPRESSIONS FOR GRADIENTS FOR CE LOSS.

We first provide some basic expressions regarding the gradients for the CE loss in the GNN under our setting. Note that the node classification task in this case is binary, denoted as $K = 2$.

The $i$-th column of the output $\mathbf{z}_i^{\text{mini}}$ after the first layer, as well as the output $\hat{y}_i^{\text{mini}}$ of the one-round GNN for the CE loss under mini-batch training, are similar to those in full-graph training in Sec. D, with $\mathbf{W}^{\text{full}}$ and $\tilde{\mathbf{a}}_{\text{train},i}^{\text{full}}$ replaced by $\mathbf{W}^{\text{mini}}$ and $\tilde{\mathbf{a}}_{\text{train},i}^{\text{mini}}$, respectively.

**Gradient for CE loss in GNN:** The partial gradients of the training losses $\hat{L}_{\text{train}}^{\text{mini}}\left(\mathbf{W}^{\text{mini}}, \tilde{\mathbf{A}}_{\text{train}}^{\text{mini}}\right)$ and $\hat{L}_{\text{train}}^{\text{full}}\left(\mathbf{W}^{\text{mini}}, \tilde{\mathbf{A}}_{\text{train}}^{\text{mini}}\right)$ with respect to $\mathbf{W}^{\text{mini}}$ under full-graph training can be expressed as:

$$\nabla \hat{L}_{\text{train}}^{\text{mini}}\left(\mathbf{W}^{\text{mini}}, \tilde{\mathbf{A}}_{\text{train}}^{\text{mini}}\right) = \frac{1}{b} \sum_{i=0}^{b} l'\left(y_i \hat{y}_i^{\text{mini}}\right) \cdot y_i \cdot \nabla_{\mathbf{W}^{\text{mini}}}\left[\hat{y}_i^{\text{mini}}\right], \tag{58}$$

$$\nabla \hat{L}_{\text{train}}^{\text{full}}\left(\mathbf{W}^{\text{mini}}, \tilde{\mathbf{A}}_{\text{train}}^{\text{mini}}\right) = \frac{1}{n_{\text{train}}} \sum_{i=0}^{n_{\text{train}}} l'\left(y_i \hat{y}_i^{\text{mini}}\right) \cdot y_i \cdot \nabla_{\mathbf{W}^{\text{mini}}}\left[\hat{y}_i^{\text{mini}}\right], \tag{59}$$

where the gradient of the GNN is defined as $\nabla_{\mathbf{W}^{\text{mini}}}\left[\hat{y}_i^{\text{mini}}\right] = \left(\mathbf{v}\boldsymbol{\Sigma}_i^{\text{mini}}\right)^\top \tilde{\mathbf{a}}_{\text{train},i}^{\text{mini}} \mathbf{X}$.

## E.3 THEOREM E.2.

**Theorem E.2.** (Convergence of Mini-batch Training with CE) *Suppose $\mathbf{W}^{mini}$ are generated by Gaussian initialization. Under Assumptions B.1 and E.1, if the hidden dimension of a one-round GNN satisfies $h = \Omega\left(n_{train}^2 \log\left(n_{train}\right)\beta^{-1} + \log\left(n_{train}\right)\beta^{-1}\epsilon^{-1}\right)$, then with high probability, the training loss satisfies $\hat{L}_{train}\left(\mathbf{W}_T^{mini}, \mathbf{A}_{train}^{mini}\right) \leq \epsilon$, provided that the number of iterations $T = \tilde{O}\left(n_{train}^4\left(\log\left(n_{train}\right)\right)^{\frac{1}{2}}\alpha^{-2}\beta^{-\frac{5}{2}}b^{-1} + n_{train}^2\left(\log\left(n_{train}\right)\right)^{\frac{1}{2}}\alpha^{-2}\beta^{-\frac{5}{2}}b^{-1}\epsilon^{-1}\right)$ for any $\epsilon \geq 0$ under the mini-batch GNN training.*

Our bound on the hidden dimension $h$ reveals an over-parameterization setting in this case, where the number of trainable parameters exceeds the number of training samples. Since the hidden dimension $h$ remains finite, our analysis is conducted in the finite-width setting, in contrast to the infinite-width Neural Tangent Kernel (NTK) framework (Yang et al., 2023; Lin et al., 2023).

### E.4 PROOF OF E.2.

We first provide the following lemmas.

**Lemma E.3** (Bounded initial training loss) Under Assumptions B.1-E.1, with the probability at least $1 - \delta$, at the initialization the training loss satisfies $\hat{L}_{\text{train}}^{\text{full}} \left( \mathbf{W}_0^{\text{mini}} \right), \hat{L}_{\text{train}}^{\text{mini}} \left( \mathbf{W}_0^{\text{mini}} \right) \leq C \sqrt{\beta \log(n_{\text{train}}/\delta)}$, where $C$ is an absolute constant.

**Lemma E.4** (Gradient lower and upper bound) Under Assumptions B.1-E.1, with the probability at least $1 - \exp\left( -C_1 h \alpha^2 / (n\beta) \right)$, there exist positive constants $C_1$, $C_2$ and $C_3$, such that

$$\left\| \nabla_{\mathbf{W}^{\text{mini}}} \hat{L}_{\text{train}}^{\text{full}} \left( \mathbf{W}^{\text{mini}}, \tilde{\mathbf{A}}_{\text{train}}^{\text{mini}} \right) \right\|_F^2 \geq \frac{C_2 h \alpha^2 \beta^3}{n_{\text{train}}^3} \left( \sum_{i=1}^{n_{\text{train}}} l' \left( y_i \hat{y}_i^{\text{mini}} \right) \right)^2,$$

$$\left\| \nabla_{\mathbf{W}^{\text{mini}}} \hat{L}_{\text{train}}^{\text{mini}} \left( \mathbf{W}^{\text{mini}}, \tilde{\mathbf{A}}_{\text{train}}^{\text{mini}} \right) \right\|_F \leq -\frac{C_3 h^{\frac{1}{2}} \beta^{\frac{1}{2}}}{b} \sum_{i=1}^{b} l' \left( y_i \hat{y}_i^{\text{mini}} \right).$$

**Lemma E.5** (Sufficient descent) Let $\mathbf{W}_0^{\text{mini}}$ be generated via Gaussian random initialization. Let $\mathbf{W}_t^{\text{mini}}$ be the $t$-th iterate in the stochastic gradient descent. If $\mathbf{W}_t^{\text{mini}}, \mathbf{W}_{t+1}^{\text{mini}} \in \mathcal{B} \left( \mathbf{W}_0^{\text{mini}}, \tau \right)$ and $\tau \leq C_6 n^{\frac{1}{2}} / (\alpha\beta)$, then there exist constants $C_4$, $C_5$ and $C_6$ such that, with probability at least $1 - \exp(-O(1))$, the following holds:

$$\mathbb{E} \left[ \hat{L}_{\text{train}}^{\text{full}} \left( \mathbf{W}_{t+1}^{\text{mini}}, \tilde{\mathbf{A}}_{\text{train}}^{\text{mini}} \right) | \mathbf{W}_t^{\text{mini}} \right] - \hat{L}_{\text{train}}^{\text{full}} \left( \mathbf{W}_t^{\text{mini}}, \tilde{\mathbf{A}}_{\text{train}}^{\text{mini}} \right)$$

$$\leq - \left( \eta - \frac{C_4 \beta h \eta^2 n_{\text{train}}}{b} \right) \left\| \nabla_{\mathbf{W}_t^{\text{mini}}} \hat{L}_{\text{train}}^{\text{full}} \left( \mathbf{W}_t^{\text{mini}}, \tilde{\mathbf{A}}_{\text{train}}^{\text{mini}} \right) \right\|_F^2$$

$$- \frac{C_5 \eta \beta^{\frac{1}{2}} h^{\frac{1}{2}} \left\| \nabla_{\mathbf{W}_t^{\text{mini}}} \hat{L}_{\text{train}}^{\text{full}} \left( \mathbf{W}_t^{\text{mini}}, \tilde{\mathbf{A}}_{\text{train}}^{\text{mini}} \right) \right\|_F}{n_{\text{train}} \tau} \sum_{i=1}^{n_{\text{train}}} l' \left( y_i \hat{y}_{i,t+1}^{\text{mini}} \right).$$

**Proof of E.2:** We first prove that stochastic gradient descent can achieve the training loss at the value of $\epsilon$ under the condition that all iterates are staying inside the perturbation region $\mathcal{B} \left( \mathbf{W}_0^{\text{mini}}, \tau \right) = \left\{ \mathbf{W} : \left\| \mathbf{W} - \mathbf{W}_0^{\text{mini}} \right\|_2 \leq \tau \right\}$.

Using Lemma E.4, there exists a constant $C_2$ such that

$$\left\| \nabla_{\mathbf{W}^{\text{full}}} \hat{L}_{\text{train}}^{\text{mini}} \left( \mathbf{W}_t^{\text{mini}}, \tilde{\mathbf{A}}_{\text{train}}^{\text{mini}} \right) \right\|_F^2 \geq \frac{C_2 h \alpha^2 \beta^3}{n_{\text{train}}^3} \left( \sum_{i=1}^{n_{\text{train}}} l' \left( y_i \hat{y}_{i,t}^{\text{mini}} \right) \right)^2 \tag{60}$$

We then set the step size $\eta$ and the radius $\tau$ as follows:

$$\eta = \frac{b}{4 C_4 \beta h n_{\text{train}}} = O \left( b \beta^{-1} h^{-1} n_{\text{train}}^{-1} \right), \tag{61}$$

$$\tau = \frac{4 C_5 n_{\text{train}}^{\frac{1}{2}}}{C_2 \alpha \beta} = O \left( n_{\text{train}}^{\frac{1}{2}} \alpha^{-1} \beta^{-1} \right). \tag{62}$$

Then we have:

$$\mathbb{E} \left[ \hat{L}_{\text{train}}^{\text{full}} \left( \mathbf{W}_{t+1}^{\text{mini}}, \tilde{\mathbf{A}}_{\text{train}}^{\text{mini}} \right) | \mathbf{W}_t^{\text{mini}} \right] - \hat{L}_{\text{train}}^{\text{full}} \left( \mathbf{W}_t^{\text{mini}}, \tilde{\mathbf{A}}_{\text{train}}^{\text{mini}} \right)$$

$$\leq - \eta \left( \frac{2 n_{\text{train}}^3}{C_2 h \alpha^2 \beta^3 u_0^2} + \frac{2 n_{\text{train}}}{C_2 h \alpha^2 \beta^3 u_1^2 \left( \hat{L}_{\text{train}}^{\text{full}} \left( \mathbf{W}_t^{\text{mini}}, \tilde{\mathbf{A}}_{\text{train}}^{\text{mini}} \right) \right)^2} \right)^{-1}, \tag{63}$$

where $u_0 = 1/2$, $u_1 = 1/(2 \log(2))$.

Rearranging terms, we have:

$$\frac{2n_{\text{train}}^3}{C_2 h\alpha^2 \beta^3 u_0^2} \left( \mathbb{E}\left[ \hat{L}_{\text{train}}^{\text{full}}\left(\mathbf{W}_{t+1}^{\text{mini}}, \tilde{\mathbf{A}}_{\text{train}}^{\text{mini}}\right) \mid \mathbf{W}_t^{\text{mini}}\right] - \hat{L}_{\text{train}}^{\text{full}}\left(\mathbf{W}_t^{\text{mini}}, \tilde{\mathbf{A}}_{\text{train}}^{\text{mini}}\right)\right)$$

$$+ \frac{2n_{\text{train}}\left( \mathbb{E}\left[ \hat{L}_{\text{train}}^{\text{full}}\left(\mathbf{W}_{t+1}^{\text{mini}}, \tilde{\mathbf{A}}_{\text{train}}^{\text{mini}}\right) \mid \mathbf{W}_t^{\text{mini}}\right] - \hat{L}_{\text{train}}^{\text{full}}\left(\mathbf{W}_t^{\text{mini}}, \tilde{\mathbf{A}}_{\text{train}}^{\text{mini}}\right)\right)}{C_2 h\alpha^2 \beta^3 u_1^2 \left( \hat{L}_{\text{train}}^{\text{full}}\left(\mathbf{W}_t^{\text{mini}}, \tilde{\mathbf{A}}_{\text{train}}^{\text{mini}}\right)\right)^2} \le -\eta. \tag{64}$$

Using $(x-y)/y^2 \ge y^{-1} - x^{-1}$ and taking telescope sum over $t$, we have:

$$t\eta \le \frac{2n_{\text{train}}^3}{C_2 h\alpha^2 \beta^3 u_0^2} \hat{L}_{\text{train}}^{\text{full}}\left(\mathbf{W}_0^{\text{mini}}, \tilde{\mathbf{A}}_{\text{train}}^{\text{mini}}\right)$$

$$+ \frac{2n_{\text{train}}\left( \left( \mathbb{E}\left[ \hat{L}_{\text{train}}^{\text{full}}\left(\mathbf{W}_t^{\text{mini}}, \tilde{\mathbf{A}}_{\text{train}}^{\text{mini}}\right)\right]^{-1}\right) - \left( \hat{L}_{\text{train}}^{\text{full}}\left(\mathbf{W}_0^{\text{mini}}, \tilde{\mathbf{A}}_{\text{train}}^{\text{mini}}\right)\right)^{-1}\right)}{C_2 h\alpha^2 \beta^3 u_1^2}. \tag{65}$$

Next, we guarantee that, after $T$ gradient steps, the loss function $\hat{L}_{\text{train}}^{\text{full}}\left(\mathbf{W}_T^{\text{mini}}, \tilde{\mathbf{A}}_{\text{train}}^{\text{mini}}\right)$ is smaller than $\epsilon$.

Using Lemma E.3, we have $\hat{L}_{\text{train}}^{\text{full}}\left(\mathbf{W}_0^{\text{mini}}, \tilde{\mathbf{A}}_{\text{train}}^{\text{mini}}\right) = O\left(\beta^{\frac{1}{2}}\left(\log\left(n_{\text{train}}\right)\right)^{\frac{1}{2}}\right)$.

Therefore, $T$ satisfies:

$$T = \tilde{O}\left( n_{\text{train}}^4 \left(\log\left(n_{\text{train}}\right)\right)^{\frac{1}{2}} / \left(\alpha^2 \beta^{\frac{5}{2}} b\right) + n_{\text{train}}^2 \left(\log\left(n_{\text{train}}\right)\right)^{\frac{1}{2}} / \left(\epsilon \alpha^2 \beta^{\frac{5}{2}} b\right)\right). \tag{66}$$

Then we are going to verify the condition that all iterates stay inside the perturbation region $\mathcal{B}\left(\mathbf{W}_0^{\text{mini}}, \tau\right)$. Obviously, we have $\mathbf{W}_0^{\text{mini}} \in \mathcal{B}\left(\mathbf{W}_0^{\text{mini}}, \tau\right)$. Hence, we need to prove $\mathbf{W}_{t+1} \in \mathcal{B}\left(\mathbf{W}_0^{\text{mini}}, \tau\right)$ under the induction hypothesis that $\mathbf{W}_t \in \mathcal{B}\left(\mathbf{W}_0^{\text{mini}}, \tau\right)$ holds for all $t \le T$.

Since we have $\hat{L}_{\text{train}}^{\text{mini}}\left(\mathbf{W}_{t+1}^{\text{mini}}\right) - \hat{L}_{\text{train}}^{\text{mini}}\left(\mathbf{W}_t^{\text{mini}}\right) \le -\frac{1}{2}\eta \left\|\nabla_{\mathbf{W}^{\text{mini}}} \hat{L}_{\text{train}}^{\text{mini}}\left(\mathbf{W}_t^{\text{mini}}, \tilde{\mathbf{A}}_{\text{train}}^{\text{mini}}\right)\right\|_F^2$ for any $t \le T$, using triangle inequality, we have:

$$\left\|\mathbf{W}_t^{\text{mini}} - \mathbf{W}_0^{\text{mini}}\right\|_2 \le \sqrt{2t\eta \hat{L}_{\text{train}}^{\text{mini}}\left(\mathbf{W}_0^{\text{mini}}, \tilde{\mathbf{A}}_{\text{train}}^{\text{mini}}\right)}. \tag{67}$$

Using $\hat{L}_{\text{train}}^{\text{mini}}\left(\mathbf{W}_0^{\text{mini}}, \tilde{\mathbf{A}}_{\text{train}}^{\text{mini}}\right) = O\left(\beta^{\frac{1}{2}}\left(\log\left(n_{\text{train}}\right)\right)^{\frac{1}{2}}\right)$ in Lemma E.3 and our settings of $\eta$, we have:

$$\left\|\mathbf{W}_t^{\text{mini}} - \mathbf{W}_0^{\text{mini}}\right\|_2$$

$$\le \sqrt{2t\eta \hat{L}_{\text{train}}^{\text{mini}}\left(\mathbf{W}_0^{\text{mini}}, \tilde{\mathbf{A}}_{\text{train}}^{\text{mini}}\right)} \tag{68}$$

$$= O\left( n_{\text{train}}^{\frac{3}{2}} \left(\log\left(n_{\text{train}}\right)\right)^{\frac{1}{2}} b^{\frac{1}{2}} \alpha^{-1} \beta^{-\frac{3}{2}} h^{-\frac{1}{2}} + n_{\text{train}}^{\frac{1}{2}} \left(\log\left(n_{\text{train}}\right)\right)^{\frac{1}{2}} b^{\frac{1}{2}} \epsilon^{-\frac{1}{2}} \alpha^{-1} \beta^{-\frac{3}{2}} h^{-\frac{1}{2}}\right).$$

In addition, by Lemma E.4 and our choice of $\eta$, we have

$$\eta \left\|\nabla_{\mathbf{W}^{\text{mini}}} \hat{L}_{\text{train}}^{\text{mini}}\left(\mathbf{W}^{\text{mini}}, \tilde{\mathbf{A}}_{\text{train}}^{\text{mini}}\right)\right\|_2 \le -\frac{\eta C_3 h^{\frac{1}{2}} \beta^{\frac{1}{2}}}{b} \sum_{i=0}^b l'\left(y_i \hat{y}_i^{\text{mini}}\right) \tag{69}$$

$$\le O\left(\left(\log\left(n_{\text{train}}\right)\right)^{\frac{1}{2}} h^{-\frac{1}{2}} b^{-\frac{1}{2}}\right),$$

where the second inequality is derived from the fact that $-1 \le l'(\cdot) \le 0$.

Therefore, by triangle inequality, we assume that $h = \Omega\left(n_{\text{train}}^2 \log\left(n_{\text{train}}\right)\beta^{-1} + \log\left(n_{\text{train}}\right)\epsilon^{-1}\beta^{-1}\right)$ and we have:

$$\left\|\mathbf{W}_{t+1}^{\text{mini}} - \mathbf{W}_t^{\text{mini}}\right\|_2 \le \eta \left\|\nabla_{\mathbf{W}^{\text{mini}}} \hat{L}_{\text{train}}^{\text{mini}}\left(\mathbf{W}_t^{\text{mini}}\right)\right\|_2 + \left\|\mathbf{W}_t^{\text{mini}} - \mathbf{W}_0^{\text{mini}}\right\|_2$$

$$= O\left( n_{\text{train}}^{\frac{1}{2}} \alpha^{-1} \beta^{-1}\right), \tag{70}$$

which is exactly the same order of $\tau$ in our settings.

This verifies $\mathbf{W}_{t+1}^{\text{mini}} \in \mathcal{B}\left(\mathbf{W}_0^{\text{mini}}, \tau\right)$.

Proved.

## F    Interpretation of the Obs.1 from Convergence Theorems

**Understanding the impact of batch size on GNN convergence.** The popular explanation posits that increasing batch size reduces gradient variance, resulting in fewer iterations to converge (Cong et al., 2021a; Zou et al., 2020b; Liu et al., 2024; Li & Liang, 2018; Hu et al., 2021). This explanation does not fully account for the impact of batch size on GNN convergence, necessitating additional consideration of the impact of message passing on the loss and gradient.

*MSE:* Taking the MSE as an example, the impact of batch size on GNN convergence is explained in three steps: (1). Activation similarity: Larger batch place more sampled nodes and their neighbors into the same graph subset in a single iteration, where message passing enables direct or indirect information exchange, resulting in similar activations processed by the same GNN parameters. In contrast, smaller batches spread nodes across iterations with varying graph subsets and updated parameters, reducing such similarity. (2). Gradient similarity: As MSE penalizes the numerical difference between predicted and target activations, the nodes with similar activations produce similar gradients. The GNN with larger batch sizes yields more coherent update directions after gradient averaging, capturing dominant structural patterns among nodes. (3). Bias: These updates may reduce node representational distinctiveness and overlook graph structural diversity, introducing bias and steering optimization toward suboptimal local minima. As batch size grows, convergence requires more iterations to escape these biased regions. DNNs typically assume i.i.d. training samples, enabling large batches to retain diversity and reduce gradient bias. This explains why GNN findings on MSE differ from expectations based on gradient variance alone, highlighting how the interplay between message passing and the loss function affects the impact of batch sizes on the GNN optimization dynamic, diverging from DNN behavior.

*CE:*CE focuses on optimizing the predicted probability of the true class, rather than minimizing the numerical differences between activations. Thus, the activation similarity does not necessarily lead to similar gradient directions under CE. This allows larger batch sizes to benefit from reduced gradient variance without introducing significant bias under CE, leading to fewer iterations to converge.

**Understanding the impact of fan-out size on convergence.** A larger fan-out size allows each node to aggregate more neighbors, enriching the node's embedding and enhancing the effective gradient even when using MSE. This leads to the reduced gradient variance, thereby more stable updates and fewer iterations for GNN convergence.

## G    Proof of Generalization Theorem in Mini-batch Training

In this section, we provide the proof of Theorem 3 in Section 4.

We can characterize the GNN generalization under mini-batch training via the PAC-Bayesian framework (McAllester, 2003). This framework decomposes the generalization gap into two components: (1) the divergence between the prior distribution $\mathcal{P}$ and the posterior distribution $\mathcal{Q}$ over the hypothesis space that includes all possible models that a learning algorithm can select, and (2) the discrepancy between expected training and testing losses over $\mathcal{P}$. The first component is easily re-derived following the PAC-Bayesian framework. We mainly focus on bounding the second component, namely the discrepancy $U$ between expected training and testing losses over $\mathcal{P}$.

As the training and testing datasets are split before training, analyzing this loss discrepancy $U$ reflects the structural difference between training and testing graphs. To isolate the impact of this structural difference on generalization, we demonstrate that the discrepancy $U$ is bounded by the Wasserstein distance $\Delta\left(\beta, b\right)$ from the training graph to the testing graph, such that $U \leq C_u \Delta\left(\beta, b\right)$ for a constant $C_u > 0$. This bound suggests that the more similar the training and testing graph structures are, the smaller the expected loss discrepancy is.

## G.1 Assumptions

We introduce assumptions on graph data and model parameters.

**Assumption G.1.** There exists a constant $C_F > 0$ such that $\|\mathbf{X}\|_F^2 \leq C_F$.

**Assumption G.2.** There exists a constant $C_w > 0$ such that $\|\mathbf{w}_i\|_2^2 \leq C_w$ for all $i$.

Assumption G.1 bounds the Frobenius norm of the feature matrix, and Assumption G.2 requires the norm of parameters to be upper-bounded during mini-batch training. These assumptions are also employed in the analyses of GNN generalization (Tang & Liu, 2023; Garg et al., 2020; Liao et al., 2020), which are introduced to simplify the proof.

The rows of $\mathbf{W}$ are initialized independently from a Gaussian distribution $N\left(0, \kappa^2 \mathbf{I}\right)$.

## G.2 Proof of Theorem 3

**Definition G.3.** (Expected Loss Discrepancy (Ma et al., 2021)). For a constant $C_u > 0$, define the expected loss discrepancy between training and testing datasets before GNN training as:

$$U = ln\mathbb{E}_{\mathbf{W}^{\mathrm{mini}} \sim \mathcal{P}} \left[ e^{C_u \left( L_{\mathrm{test}} \left( \mathbf{W}^{\mathrm{mini}}, \mathbf{A}_{\mathrm{test}}^{\mathrm{full}} \right) - L_{\mathrm{train}} \left( \mathbf{W}^{\mathrm{mini}}, \mathbf{A}_{\mathrm{train}}^{\mathrm{mini}} \right) \right)} \right],$$

where $\mathcal{P}$ represents the prior distribution over hypothesis space that includes all possible models that a learning algorithm can select.

Definition G.3 captures the difference between training and testing datasets.

**Definition G.4.** (Distance between Training Set and Testing Set). Define the distance from the training set to the testing set as the Wasserstein distance given by:

$$
\Delta\left(\beta, b\right) = \left\{ \inf_{\theta \in \Theta[\rho_{\mathrm{train}}, \rho_{\mathrm{test}}]} \sum_{i \in \mathrm{train\ set}} \sum_{j \in \mathrm{test\ set}} \theta_{i,j} \delta\left(y_i, y_j, \beta, b\right) \right\}
$$
$$
= \left\{ \sup_{f(\cdot), g(\cdot)} \sum_{i \in \mathrm{train\ set}} f\left(y_i\right) \rho_{\mathrm{train}}\left(y_i\right) + \sum_{j \in \mathrm{test\ set}} g\left(y_j\right) \rho_{\mathrm{test}}\left(y_j\right) \right\}, \tag{71}
$$

where $\rho_{\mathrm{train}}\left(y_i\right)$ and $\rho_{\mathrm{test}}\left(y_i\right)$ denote the probability of $y_i$ appearing in training and testing sets, respectively. $\Theta[\rho_{\mathrm{train}}, \rho_{\mathrm{test}}]$ is the joint probability of $\rho_{\mathrm{train}}$ and $\rho_{\mathrm{test}}$, and $f\left(\mathbf{y}_i\right)$ and $g\left(\mathbf{y}_i\right)$ are functions of $\mathbf{y}_i$ with $i \in \mathcal{V}$. The infimum in the first equality is conditioned on $\sum_{j \in \mathrm{test\ set}} \theta_{i,j} = \rho_{\mathrm{train}}\left(y_i\right), \sum_{i \in \mathrm{training\ set}} \theta_{i,j} = \rho_{\mathrm{test}}\left(y_j\right), \theta_{i,j} \geq 0$, and the supremum in the second equality is conditioned on $f\left(y_i\right) + g\left(y_j\right) \leq \delta\left(y_i, y_j, \beta, b\right)$. $\delta\left(\mathbf{y}_i, \mathbf{y}_j, \beta, b\right)$ is the distance function of any two points from training and testing sets, respectively.

The Wasserstein distance effectively measures differences in non-i.i.d. data, particularly regarding geometric variations. A dual representation is provided in Eq (71).

**Theorem G.5.** (PAC-Bayesian Generalization Theorem). For any $C_u > 0$, for any "prior" distribution $\mathcal{P}$ of the output hypothesis function of a GNN that is independent of node labels from training dataset, with probability at least $1 - C_G$, for the distribution $\mathcal{Q}$ of the output hypothesis function of a GNN, we have:

$$L_{\mathrm{test}}(\mathbf{W}^{\mathrm{mini}}, \tilde{\mathbf{A}}_{\mathrm{test}}^{\mathrm{full}}; \mathcal{Q}) \leq \hat{L}_{\mathrm{train}}^{\mathrm{full}}(\mathbf{W}^{\mathrm{mini}}, \tilde{\mathbf{A}}_{\mathrm{train}}^{\mathrm{mini}}; \mathcal{Q}) + \frac{1}{C_u}(D_{KL}(\mathcal{Q}\|\mathcal{P}) + ln\frac{1}{C_G} + \frac{C_u^2}{4n_{\mathrm{train}}} + U).$$

**Lemma G.6** For any $C_u > 0$, assume the "prior" $\mathcal{P}$ on hypothesis space is defined by sampling the model parameters. If the in-degree of each node is $O(\beta)$ and the out-degree of each node is $O(b)$, we have:

$$U \leq C_u \Delta(\beta, b),$$

*and,*

$$
\Delta(\beta, b) \propto \sum_{i \in \mathrm{train\ set}} \sum_{j \in \mathrm{test\ set}} \theta_{i,j} \delta_i^{\mathrm{full\text{-}mini}} = \sum_{i \in \mathrm{train\ set}} \sum_{j \in \mathrm{test\ set}} \theta_{i,j} \left\| \tilde{\mathbf{a}}_{\mathrm{train},i}^{\mathrm{full}} - \tilde{\mathbf{a}}_{\mathrm{train},i}^{\mathrm{mini}} \right\|_F^2. \tag{72}
$$
$$
\Delta(\beta, b_1) \leq \Delta(\beta, b_2) \text{ with } b_1 \geq b_2
$$

where $\delta_i^{\text{full-mini}}$ has a overall non-increasing trend when the fan-out size $\beta$ increases but small non-monotonic fluctuations can exist. Note that fan-out size $\beta$ plays a more dominant role than batch size $b$ in influencing generalization.

**Proof of Theorem 3:**   Using Theorem G.5, we have

$$
\begin{aligned}
L_{\text{test}}\left(\mathbf{W}^{\text{mini}}, \tilde{\mathbf{A}}_{\text{train}}^{\text{full}}; \mathcal{Q}\right) \leq & \hat{L}_{\text{train}}^{\text{full}}\left(\mathbf{W}^{\text{mini}}, \tilde{\mathbf{A}}_{\text{train}}^{\text{mini}}; \mathcal{Q}\right) \\
& + \frac{1}{C_u}(D_{KL}(\mathcal{Q}\|\mathcal{P}) + ln\frac{1}{C_G} + \frac{C_u^2}{4n_{\text{train}}} + U.
\end{aligned}
\tag{73}
$$

Since both $\mathcal{P}$ and $\mathcal{Q}$ are normal distributions (Ma et al., 2021), assuming that $\left\|\mathbf{w}_{j,T}^{\text{mini}}\right\|_F^2 \leq C_w$, we know that

$$
D_{KL}(\mathcal{Q}\|\mathcal{P}) \leq \frac{\left\|\mathbf{W}_T^{\text{mini}}\right\|_F^2}{2h\kappa^2} = \frac{\sum_j^h \left\|\mathbf{w}_{j,T}^{\text{mini}}\right\|_F^2}{2h\kappa^2} \leq \frac{C_w}{2\kappa^2},
\tag{74}
$$

where $C_T$ is a positive constant.

Hence,

$$
\begin{aligned}
& L_{\text{test}}\left(\mathbf{W}^{\text{mini}}, \tilde{\mathbf{A}}_{\text{train}}^{\text{full}}; \mathcal{Q}\right) \\
\leq & \hat{L}_{\text{train}}^{\text{full}}\left(\mathbf{W}^{\text{mini}}, \tilde{\mathbf{A}}_{\text{train}}^{\text{mini}}; \mathcal{Q}\right) + \frac{1}{C_u}(D_{KL}(\mathcal{Q}\|\mathcal{P}) + ln\frac{1}{C_G} + \frac{C_u^2}{4n_{\text{train}}} + U \\
\leq & \hat{L}_{\text{train}}^{\text{full}}\left(\mathbf{W}^{\text{mini}}, \tilde{\mathbf{A}}_{\text{train}}^{\text{mini}}; \mathcal{Q}\right) + \frac{1}{C_u}\left(\frac{C_w}{2\kappa^2} + ln\frac{1}{C_G} + \frac{C_u^2}{4n_{\text{train}}} + C_u\Delta\left(\beta, b\right)\right).
\end{aligned}
\tag{75}
$$

## H   EXTENSION TO MULTI-LAYER GNNS

Our theoretical analysis readily extends to multi-layer GNNs, as long as each layer introduces only one non-linearity (e.g., ReLU activation). In such settings, the key difference is that the output of each layer is recursively defined based on the previous layer.

This recursive definition preserves the same message-passing structure at each layer. In convergence analysis, we bound the gradient norms layer by layer; in generalization analysis, the pre-training loss discrepancy propagates across layers. These recursive structures allow our convergence and generalization bounds to translate naturally to multi-layer GNNs.

Our key theoretical insights (from the view of batch size and fan-out size) are generalizable to multi-layer GNNs. This is because adding more layers simply nests the same operations, without changing the qualitative roles of batch size and fan-out size. Hence, the analytical trends observed in the one-layer case remain consistent.

Therefore, our theoretical analyses support the multi-layer GNN settings.

## I   PROOF OF THE MAIN LEMMAS OF CONVERGENCE THEOREMS WITH MSE

### I.1   PROOF OF LEMMA B.5 AND C.1

We first focus on the mini-batch training. Note that $\hat{\sigma}(x) \geq \frac{1}{\pi}$ whenever $x \geq 0$ (Daniely et al., 2016). Then, the bound on $\Gamma^{\text{mini}}$ follows as:

$$
\begin{aligned}
\Gamma^{\text{mini}} = & \frac{1}{b}\sum_{i,j=1}^b p_{ij}\hat{\sigma}\left(\frac{\varrho_{i,j}^{\text{mini}}}{\sqrt{\vartheta_{i,j}^{\text{mini}}}}\right)\sqrt{\vartheta_{i,j}^{\text{mini}}} \\
\geq & \frac{1}{\pi b}\sum_{i,j=1}^b p_{ij}\sqrt{\vartheta_{i,j}^{\text{mini}}} = \frac{1}{\pi b}\sum_{i,j=1}^b p_{ij}\sqrt{\left(\tilde{\mathbf{A}}_{\text{train}}^{\text{mini}}\mathbb{1}\right)_i \left(\tilde{\mathbf{A}}_{\text{train}}^{\text{mini}}\mathbb{1}\right)_j} \\
= & \frac{1}{\pi b}\sum_{i=1}^b \left(\tilde{\mathbf{A}}_{\text{train}}^{\text{mini}}\mathbb{1}\right)_i, \\
= & \frac{1}{\pi b}\left\|\tilde{\mathbf{A}}_{\text{train}}^{\text{mini}}\mathbb{1}\right\|_1.
\end{aligned}
\tag{76}
$$

To bound $\Upsilon_t^{\text{mini}}$, we notice that $|\hat{\sigma}(x)| \leq \hat{\sigma}(|x|)$, and $\hat{\sigma}(\cdot)$ is a non-decreasing function in $[0,1]$. Hence, we get $|\Upsilon^{\text{mini}}| \leq \Gamma^{\text{mini}}$.

we have the normalized adjacency matrix of a graph with $b$ nodes as:

$$
\tilde{\mathbf{A}}_{\text{train}}^{\text{mini}} = \begin{bmatrix} \frac{1}{\sqrt{d_1^{\text{in}}}} & & \\ & \ddots & \\ & & \frac{1}{\sqrt{d_b^{\text{in}}}} \end{bmatrix} \begin{bmatrix} a_{11} & \cdots & a_{1n} \\ \vdots & \ddots & \vdots \\ a_{b1} & \cdots & a_{bn} \end{bmatrix} \begin{bmatrix} \frac{1}{\sqrt{d_1^{\text{out}}}} & & \\ & \ddots & \\ & & \frac{1}{\sqrt{d_n^{\text{out}}}} \end{bmatrix}
$$
$$
= \begin{bmatrix} \frac{1}{\sqrt{d_1^{\text{in}}}}\frac{1}{\sqrt{d_1^{\text{out}}}}a_{11} & \cdots & \frac{1}{\sqrt{d_1^{\text{in}}}}\frac{1}{\sqrt{d_n^{\text{out}}}}a_{1n} \\ \vdots & \ddots & \vdots \\ \frac{1}{\sqrt{d_b^{\text{in}}}}\frac{1}{\sqrt{d_1^{\text{out}}}}a_{b1} & \cdots & \frac{1}{\sqrt{d_b^{\text{in}}}}\frac{1}{\sqrt{d_n^{\text{out}}}}a_{bn} \end{bmatrix},
\tag{77}
$$

where $a_{ij} \in \{0,1\}$ represents whether node $i$ connects with node $j$ (1) or not (0).

Since $d_i^{\text{in}} \leq \beta$ and $d^{\text{out}} \leq b$, we have:

$$
\begin{aligned}
\|\tilde{\mathbf{A}}_{\text{train}}^{\text{mini}}\mathbb{1}\|_1 &= \sum_{i=1}^{b} \frac{1}{\sqrt{d_i^{\text{in}}}\sqrt{d_1^{\text{out}}}}a_{i1} + \cdots + \frac{1}{\sqrt{d_i^{\text{in}}}\sqrt{d_n^{\text{out}}}}a_{in} \\
&\geq \sum_{i=1}^{b} \frac{1}{\beta^{\frac{1}{2}}b^{\frac{1}{2}}}(a_{i1} + \cdots + a_{in}) \\
&\geq \frac{b^{\frac{1}{2}}}{\beta^{\frac{1}{2}}}\min_i (a_{i1} + \cdots + a_{in}) \\
&\geq \frac{b^{\frac{1}{2}}}{\beta^{\frac{1}{2}}}.
\end{aligned}
\tag{78}
$$

Moreover, since $\varrho_{i,j}^{\text{mini}}$ denotes the amount of common messages between node $i$ and node $j$ at a given training iteration, we know that $\frac{\varrho_{i,j}^{\text{mini}}}{\sqrt{\vartheta_{i,j}^{\text{mini}}}} \leq 1$. Then we have:

$$
\begin{aligned}
\Gamma^{\text{mini}} &= \frac{1}{b}\sum_{i,j=1}^{b} p_{ij}\hat{\sigma}\left(\frac{\varrho_{i,j}^{\text{mini}}}{\sqrt{\vartheta_{mini}^{\text{mini}}}}\right)\sqrt{\vartheta_{i,j}^{\text{full}}} \\
&\leq \frac{1}{b}\left\|\tilde{\mathbf{A}}_{\text{train}}^{\text{mini}}\mathbb{1}\right\|_1
\end{aligned}
\tag{79}
$$

Moreover, since $1/\left(\sqrt{d_i^{\text{in}}}\sqrt{d_1^{\text{out}}}\right) \leq 1$, we have:

$$
\begin{aligned}
\|\tilde{\mathbf{A}}_{\text{train}}^{\text{mini}}\mathbb{1}\|_1 &= \sum_{i=1}^{b} \frac{1}{\sqrt{d_i^{\text{in}}}\sqrt{d_1^{\text{out}}}}a_{i1} + \cdots + \frac{1}{\sqrt{d_i^{\text{in}}}\sqrt{d_n^{\text{out}}}}a_{in} \\
&\leq \sum_{i=1}^{b} (a_{i1} + \cdots + a_{in}) \\
&\leq \beta b,
\end{aligned}
\tag{80}
$$

where the last inequality holds because there exist at most $\beta$ terms that are not equal to $0$.

Similarly, for $\Gamma^{\text{full-mini}}$, we have:

$$
\begin{aligned}
\Gamma^{\text{full-mini}} &\geq \frac{1}{\pi n_{\text{train}}}\sum_{i=1}^{n_{\text{train}}}\left(\tilde{\mathbf{A}}_{\text{train}}^{\text{mini}}\mathbb{1}\right)_i, \\
&= \frac{1}{\pi n_{\text{train}}}\frac{n_{\text{train}}}{b}\sum_{i=1}^{b}\left(\tilde{\mathbf{A}}_{\text{train}}^{\text{mini}}\mathbb{1}\right)_i, \\
&= \frac{1}{\pi b}\left\|\tilde{\mathbf{A}}_{\text{train}}^{\text{mini}}\mathbb{1}\right\|_1,
\end{aligned}
\tag{81}
$$

and

$$\Gamma^{\text{full-mini}} \leq \frac{1}{b} \left\| \tilde{\mathbf{A}}_{\text{train}}^{\text{mini}} \mathbb{1} \right\|_1. \tag{82}$$

Moreover, $|\Upsilon^{\text{full-mini}}| \leq \Gamma^{\text{full-mini}}$ holds.

Similarly, in the full-graph training, we can replace $b$ and $\beta$ by $n_{\text{train}}$ and $d_{\max}$, respectively. Therefore, we have:

$$\frac{1}{\pi n_{\text{train}}} \left\| \tilde{\mathbf{A}}_{\text{train}}^{\text{full}} \mathbb{1} \right\|_1 \leq \Gamma^{\text{full}} \leq \frac{1}{n_{\text{train}}} \left\| \tilde{\mathbf{A}}_{\text{train}}^{\text{full}} \mathbb{1} \right\|_1. \tag{83}$$

$$\frac{n_{\text{train}}^{\frac{1}{2}}}{d_{\max}^{\frac{1}{2}}} \leq \|\tilde{\mathbf{A}}_{\text{train}}^{\text{full}} \mathbb{1}\|_1 \leq n_{\text{train}} d_{\max}. \tag{84}$$

$$|\Upsilon^{\text{full}}| \leq \Gamma^{\text{full}} \tag{85}$$

This complete the proof.

### I.2  PROOF OF LEMMA B.6 AND C.2

**Lemma I.1**   In the mini-batch training, $|\Xi_t^{\text{mini}}| = o(\Gamma_t^{\text{mini}})$, $|\Xi_t^{\text{full-mini}}| = o(\Gamma_t^{\text{full-mini}})$, and, when $\phi_t^{\text{mini}} \geq -\frac{1}{100}$ and $\sqrt{\left(\phi_t^{\text{mini}}\right)^2 + \left(\psi_t^{\text{mini}}\right)^2} \geq 1 - o(1)$, then $\Xi_t^{\text{mini}} \geq \frac{\Gamma_t^{\text{mini}}}{2\beta}$ and $\Xi_t^{\text{full-mini}} \geq \frac{\Gamma_t^{\text{full-mini}}}{2\beta}$. In the full-graph training, $|\Xi_t^{\text{full}}| = o(\Gamma^{\text{full}})$, and, when $\phi_t^{\text{full}} \geq -\frac{1}{100}$ and $\sqrt{\left(\phi_t^{\text{full}}\right)^2 + \left(\psi_t^{\text{full}}\right)^2} \geq 1 - o(1)$, then $\Xi_t^{\text{full}} \geq \frac{\Gamma_t^{\text{full}}}{2d_{\max}}$.

**Proof of Lemma B.6 and C.2:**   We first focus on the mini-batch training. Considering the gradient, we are going to analyze $\left(\phi_{t+1}^{\text{mini}}\right)^2 + \left(\psi_{t+1}^{\text{mini}}\right)^2$:

$$
\begin{aligned}
&\left(\phi_{t+1}^{\text{mini}}\right)^2 + \left(\psi_{t+1}^{\text{mini}}\right)^2 \\
&= \left(\phi_t^{\text{mini}} - \eta_t \frac{\partial L_{\text{train},t}^{\text{mini}}\left(\phi_t^{\text{mini}}, \psi_t^{\text{mini}}\right)}{\partial \phi_t^{\text{mini}}}\right)^2 + \left(\psi_t^{\text{mini}} - \eta_t \frac{\partial L_{\text{train},t}^{\text{mini}}\left(\phi_t^{\text{mini}}, \psi_t^{\text{mini}}\right)}{\partial \psi_t^{\text{mini}}}\right)^2 \\
&= \left(\phi_t^{\text{mini}} - \eta_t \phi_t^{\text{mini}} \Gamma_t^{\text{mini}} + \eta_t \frac{\phi_t^{\text{mini}}}{\sqrt{\left(\phi_t^{\text{mini}}\right)^2 + \left(\psi_t^{\text{mini}}\right)^2}} \Upsilon_t^{\text{mini}} + \eta_t \frac{\left(\psi_t^{\text{mini}}\right)^2}{\left(\phi_t^{\text{mini}}\right)^2 + \left(\psi_t^{\text{mini}}\right)^2} \Xi_t^{\text{mini}}\right)^2 \\
&\quad + \left(\psi_t^{\text{mini}} - \eta_t \psi_t^{\text{mini}} \Gamma_t^{\text{mini}} + \eta_t \frac{\psi_t^{\text{mini}}}{\sqrt{\left(\phi_t^{\text{mini}}\right)^2 + \left(\psi_t^{\text{mini}}\right)^2}} \Upsilon_t^{\text{mini}} + \eta_t \frac{\phi_t^{\text{mini}} \psi_t^{\text{mini}}}{\left(\phi_t^{\text{mini}}\right)^2 + \left(\psi_t^{\text{mini}}\right)^2} \Xi_t^{\text{mini}}\right)^2 \\
&= \left(\phi_t^{\text{mini}}\right)^2 + \left(\psi_t^{\text{mini}}\right)^2 + \eta_t^2 \Gamma_t^{\text{mini}\,2} \left(\left(\phi_t^{\text{mini}}\right)^2 + \left(\psi_t^{\text{mini}}\right)^2\right) + \eta_t^2 \Upsilon_t^{\text{mini}\,2} \\
&\quad + \eta_t^2 \frac{\left(\psi_t^{\text{mini}}\right)^2}{\left(\phi_t^{\text{mini}}\right)^2 + \left(\psi_t^{\text{mini}}\right)^2} \left(\Xi_t^{\text{mini}}\right)^2 - 2\eta_t \Gamma_t^{\text{mini}} \left(\left(\phi_t^{\text{mini}}\right)^2 + \left(\psi_t^{\text{mini}}\right)^2\right) \\
&\quad + 2\eta_t \sqrt{\left(\phi_t^{\text{mini}}\right)^2 + \left(\psi_t^{\text{mini}}\right)^2} \Upsilon_t^{\text{mini}} - 2\eta_t^2 \Gamma_t^{\text{mini}} \Upsilon_t^{\text{mini}} \sqrt{\left(\phi_t^{\text{mini}}\right)^2 + \left(\psi_t^{\text{mini}}\right)^2}
\end{aligned}
\tag{86}
$$

Hence, By Lemma C.1 and Lemma I.1, we have:

$$\left(\phi_{t+1}^{\text{mini}}\right)^2 + \left(\psi_{t+1}^{\text{mini}}\right)^2 \leq \left(\sqrt{\left(\phi_t^{\text{mini}}\right)^2 + \left(\psi_t^{\text{mini}}\right)^2} \left(1 - \frac{C}{\pi}\right)\right)^2 + C, \tag{87}$$

when the learning rate $\eta_t \in \left[\frac{C}{\pi \Gamma_t^{\text{mini}}}, \frac{1}{6\pi \Gamma_t^{\text{mini}}}\right]$ (Awasthi et al., 2021), where $C \leq \frac{1}{16}$ is a small enough and positive constant. Hence, we can rewrite the range of $\eta$ as $\eta_t \in (0, \frac{1}{6\pi \Gamma_t^{\text{mini}}}]$

Then, for all $t \geq 1$, we have:

$$\sqrt{(\phi_t^{\text{mini}})^2 + (\psi_t^{\text{mini}})^2} \leq \left(1 - \frac{C}{\pi}\right)^t \sqrt{(\phi_0^{\text{mini}})^2 + (\psi_0^{\text{mini}})^2} + \frac{C}{1 - \left(1 - \frac{C}{\pi}\right)^2}$$

$$< \sqrt{(\phi_0^{\text{mini}})^2 + (\psi_0^{\text{mini}})^2} + \frac{\pi}{2 - C} \tag{88}$$

$$< \sqrt{(\phi_0^{\text{mini}})^2 + (\psi_0^{\text{mini}})^2} + \frac{\pi}{2}$$

Moreover, with probability at least $1 - e^{-O(r)}$, we will have $\sqrt{\left(\phi_0^{\text{mini}}\right)^2 + \left(\psi_0^{\text{mini}}\right)^2} = O(\kappa\sqrt{r})$.

Hence, we have $\sqrt{\left(\phi_t^{\text{mini}}\right)^2 + \left(\psi_t^{\text{mini}}\right)^2} \leq C_1$.

We also have that if $\Upsilon_t^{\text{mini}} > 0$, then

$$\left(\phi_{t+1}^{\text{mini}}\right)^2 + \left(\psi_{t+1}^{\text{mini}}\right)^2 \geq \left(\left(\phi_t^{\text{mini}}\right)^2 + \left(\psi_t^{\text{mini}}\right)^2\right)(1 - \eta_t \Gamma_t^{\text{mini}})^2$$

$$+ 2\eta_t \Upsilon_t^{\text{mini}} \sqrt{(\phi_t^{\text{mini}})^2 + (\psi_t^{\text{mini}})^2}(1 - \eta_t \Gamma_t^{\text{mini}}) + \eta_t^2 \Upsilon_t^{\text{mini}\,2} \tag{89}$$

$$> \eta_t^2 \Upsilon_t^{\text{mini}\,2} > 0$$

Similarly, in the full-graph training, we can replace $b$ and $\beta$ by $n_{\text{train}}$ and $d_{\max}$, respectively.

This completes the proof.

### I.3 PROOF OF LEMMA B.7 AND C.3

We first focus on the mini-batch training. From Lemma C.2 , we immediately have $\sqrt{(\phi^{\text{mini}})^2 + (\psi^{\text{mini}})^2} \leq C$, where $C$ is a positive constant.

Next, we analyze the upper bound of $\lambda_{\max}(\nabla^2 L_{\text{train}}^{\text{full}}(\phi^{\text{mini}}, \psi^{\text{mini}}, \tilde{\mathbf{A}}_{\text{train}}^{\text{mini}}))$. We have:

$$\lambda_{\max}(\nabla^2 L_{\text{train}}^{\text{full}}(\phi^{\text{mini}}, \psi^{\text{mini}}, \tilde{\mathbf{A}}_{\text{train}}^{\text{mini}}))$$

$$\leq \left| \frac{\partial^2 L_{\text{train}}^{\text{full}}(\phi_t^{\text{mini}}, \psi_t^{\text{mini}}, \tilde{\mathbf{A}}_{\text{train}}^{\text{mini}})}{\partial \left(\phi_t^{\text{mini}}\right)^2} \right| + \left| \frac{\partial^2 L_{\text{train}}^{\text{full}}(\phi_t^{\text{mini}}, \psi_t^{\text{mini}}, \tilde{\mathbf{A}}_{\text{train}}^{\text{mini}})}{\partial \left(\psi_t^{\text{mini}}\right)^2} \right| \tag{90}$$

$$+ \left| \frac{\partial^2 L_{\text{train}}^{\text{full}}(\phi_t^{\text{mini}}, \psi_t^{\text{mini}}, \tilde{\mathbf{A}}_{\text{train}}^{\text{mini}})}{\partial \phi_t^{\text{mini}} \partial \psi_t^{\text{mini}}} \right| + \left| \frac{\partial^2 L_{\text{train}}^{\text{full}}(\phi_t^{\text{mini}}, \psi_t^{\text{mini}}, \tilde{\mathbf{A}}_{\text{train}}^{\text{mini}})}{\partial \psi_t^{\text{mini}} \partial \phi_t^{\text{mini}}} \right|.$$

Taking the second derivatives, we get:

$$\left| \frac{\partial^2 L_{\text{train}}^{\text{full}}(\phi_t^{\text{mini}}, \psi_t^{\text{mini}}, \tilde{\mathbf{A}}_{\text{train}}^{\text{mini}})}{\partial \left(\phi_t^{\text{mini}}\right)^2} \right| = \Gamma_t^{\text{full-mini}} - \frac{\phi_t^{\text{mini}}}{\left\| \mathbf{w}_{j,t}^{\text{mini}} \right\|} \frac{\partial \Upsilon_t^{\text{full-mini}}}{\partial \phi_t^{\text{mini}}} - \frac{\left(\psi_t^{\text{mini}}\right)^2}{\left\| \mathbf{w}_{j,t}^{\text{mini}} \right\|^{\frac{3}{2}}} \Upsilon_t^{\text{full-mini}}$$

$$- \frac{\left(\psi_t^{\text{mini}}\right)^2}{\left\| \mathbf{w}_{j,t}^{\text{mini}} \right\|^2} \frac{\partial \Xi_t^{\text{full-mini}}}{\partial \phi_t^{\text{mini}}} + \frac{2\phi_t^{\text{mini}} \left(\psi_t^{\text{mini}}\right)^2}{\left\| \mathbf{w}_{j,t}^{\text{mini}} \right\|^4} \Xi_t^{\text{full-mini}}, \tag{91}$$

$$\left| \frac{\partial^2 L_{\text{train}}^{\text{full}}(\phi_t^{\text{mini}}, \psi_t^{\text{mini}}, \tilde{\mathbf{A}}_{\text{train}}^{\text{mini}})}{\partial \left(\psi_t^{\text{mini}}\right)^2} \right|$$

$$= \Gamma_t^{\text{full-mini}} - \frac{\psi_t^{\text{mini}}}{\left\| \mathbf{w}_{j,t}^{\text{mini}} \right\|} \frac{\partial \Upsilon_t^{\text{full-mini}}}{\partial \psi_t^{\text{mini}}} + \frac{\phi_t^{\text{mini}} \psi_t^{\text{mini}}}{\left\| \mathbf{w}_{j,t}^{\text{mini}} \right\|^{\frac{3}{2}}} \Upsilon_t^{\text{full-mini}}$$

$$+ \frac{\phi_t^{\text{mini}} \psi_t^{\text{mini}}}{\left\| \mathbf{w}_{j,t}^{\text{mini}} \right\|^2} \frac{\partial \Xi_t^{\text{full-mini}}}{\partial \phi_t^{\text{mini}}} + \frac{\phi_t^{\text{mini}}((\phi_t^{\text{mini}})^2 - (\psi_t^{\text{mini}})^2)}{\left\| \mathbf{w}_{j,t}^{\text{mini}} \right\|^4} \Xi_t^{\text{full-mini}}, \tag{92}$$

$$\left| \frac{\partial^2 L_{\text{train}}^{\text{full}}(\phi_t^{\text{mini}}, \psi_t^{\text{mini}}, \tilde{\mathbf{A}}_{\text{train}}^{\text{mini}})}{\partial \phi_t^{\text{mini}} \partial \psi_t^{\text{mini}}} \right| = - \frac{\psi_t^{\text{mini}}}{\left\| \mathbf{w}_{j,t}^{\text{mini}} \right\|} \frac{\partial \Upsilon_t^{\text{full-mini}}}{\partial \phi_t^{\text{mini}}} + \frac{\phi_t^{\text{mini}} \psi_t^{\text{mini}}}{\left\| \mathbf{w}_{j,t}^{\text{mini}} \right\|^{\frac{3}{2}}} \Upsilon_t^{\text{full-mini}}$$

$$+ \frac{\phi_t^{\text{mini}} \psi_t^{\text{mini}}}{\left\| \mathbf{w}_{j,t}^{\text{mini}} \right\|^2} \frac{\partial \Xi_t^{\text{full-mini}}}{\partial \phi_t^{\text{mini}}} + \frac{\phi_t^{\text{mini}} \psi_t^{\text{mini}}}{\left\| \mathbf{w}_{j,t}^{\text{mini}} \right\|^2} \frac{\partial \Xi_t^{\text{full-mini}}}{\partial \phi_t^{\text{mini}}} \tag{93}$$

$$+ \frac{\psi_t^{\text{mini}} \left( \left( \psi_t^{\text{mini}} \right)^2 - \left( \phi_t^{\text{mini}} \right)^2 \right)}{\left\| \mathbf{w}_{j,t}^{\text{mini}} \right\|^4} \Xi_t^{\text{full-mini}},$$

$$\left| \frac{\partial^2 L_{\text{train}}^{\text{full}}(\phi_t^{\text{mini}}, \psi_t^{\text{mini}}, \tilde{\mathbf{A}}_{\text{train}}^{\text{mini}})}{\partial \psi_t^{\text{mini}} \partial \phi_t^{\text{mini}}} \right| = - \frac{\phi_t^{\text{mini}}}{\left\| \mathbf{w}_{j,t}^{\text{mini}} \right\|} \frac{\partial \Upsilon_t^{\text{full-mini}}}{\partial \psi_t^{\text{mini}}} + \frac{\phi_t^{\text{mini}} \psi_t^{\text{mini}}}{\left\| \mathbf{w}_{j,t}^{\text{mini}} \right\|^{\frac{3}{2}}} \Upsilon_t^{\text{full-mini}}$$

$$+ \frac{\phi_t^{\text{mini}} \psi_t^{\text{mini}}}{\left\| \mathbf{w}_{j,t}^{\text{mini}} \right\|^2} \frac{\partial \Xi_t^{\text{full-mini}}}{\partial \phi_t^{\text{mini}}} - 2 \frac{\phi_t^{\text{mini}} \left( \psi_t^{\text{mini}} \right)^2}{\left\| \mathbf{w}_{j,t}^{\text{mini}} \right\|^4} \Xi_t^{\text{full-mini}}. \tag{94}$$

Next we have

$$\left| \frac{\partial \Upsilon_t^{\text{full-mini}}}{\partial \phi_t^{\text{mini}}} \right| = \left| \frac{1}{n_{\text{train}}} \sum_{i,k=1}^{n_{\text{train}}} p_{ik} \varrho_{i,k}^{\text{mini}} \hat{\sigma}_{\text{step}}\left( \frac{\varrho_{i,k}^{\text{mini}}}{\sqrt{\vartheta_{i,k}^{\text{mini}}}} \frac{\phi_t^{\text{mini}}}{\sqrt{\left( \phi_t^{\text{mini}} \right)^2 + \left( \psi_t^{\text{mini}} \right)^2}} \right) \cdot \frac{\left( \psi_t^{\text{mini}} \right)^2}{\left\| \mathbf{w}_{j,t}^{\text{mini}} \right\|^{\frac{3}{2}}} \right|$$

$$\leq \left( \left( \phi_t^{\text{mini}} \right)^2 + \left( \psi_t^{\text{mini}} \right)^2 \right)^{\frac{1}{4}} \left| \Xi_t^{\text{full-mini}} \right| = o \left( \Gamma_t^{\text{full-mini}} \right) \tag{95}$$

$$\left| \frac{\partial \Upsilon_t^{\text{full-mini}}}{\partial \psi_t^{\text{mini}}} \right| = \left| \frac{1}{n_{\text{train}}} \sum_{i,k=1}^{n_{\text{train}}} p_{ik} \varrho_{i,k}^{\text{mini}} \hat{\sigma}_{\text{step}}\left( \frac{\varrho_{i,k}^{\text{mini}}}{\sqrt{\vartheta_{i,k}^{\text{mini}}}} \frac{\phi_t^{\text{mini}}}{\sqrt{\left( \phi_t^{\text{mini}} \right)^2 + \left( \psi_t^{\text{mini}} \right)^2}} \right) \cdot \frac{\phi_t^{\text{mini}} \psi_t^{\text{mini}}}{\left\| \mathbf{w}_{j,t}^{\text{mini}} \right\|^{\frac{3}{2}}} \right|$$

$$\leq \left( \left( \phi_t^{\text{mini}} \right)^2 + \left( \psi_t^{\text{mini}} \right)^2 \right)^{\frac{1}{4}} \left| \Xi_t^{\text{full-mini}} \right| = o \left( \Gamma_t^{\text{full-mini}} \right), \tag{96}$$

where we use $|\Xi_t^{\text{full-mini}}| = o(\Gamma_t^{\text{full-mini}})$ in the Lemma I.1.

To differentiate $\Xi_t^{\text{full-mini}}$, we employ $\hat{\sigma}(\theta) = 1 - \frac{arccos(\theta)}{\pi}$ (Daniely et al., 2016) and $arccos'(\theta) = -\frac{1}{\sqrt{1-\theta^2}}$ to get:

$$\left| \frac{\partial \Xi_t^{\text{full-mini}}}{\partial \phi_t^{\text{mini}}} \right| = \left| \frac{1}{n_{\text{train}}} \sum_{i,k=1}^{n_{\text{train}}} p_{ik} \frac{\left( \varrho_{i,k}^{\text{mini}} \right)^2}{\vartheta_{i,k}^{\text{mini}}} \frac{\left\| \mathbf{w}_{j,t}^{\text{mini}} \right\|}{\psi_t^{\text{mini}}} \frac{\left( \psi_t^{\text{mini}} \right)^2}{\left\| \mathbf{w}_{j,t}^{\text{mini}} \right\|^{\frac{3}{2}}} \right|$$

$$\leq \frac{1}{n_{\text{train}}} \sum_{i,k=1}^{n_{\text{train}}} \left( \left( \phi_t^{\text{mini}} \right)^2 + \left( \psi_t^{\text{mini}} \right)^2 \right)^{\frac{1}{4}} \frac{\left( \varrho_{i,k}^{\text{mini}} \right)^2}{\vartheta_{i,k}^{\text{mini}}} = o \left( \Gamma_t^{\text{full-mini}} \right), \tag{97}$$

$$\left| \frac{\partial \Xi_t^{\text{full-mini}}}{\partial \psi_t^{\text{mini}}} \right| = \left| \frac{1}{n_{\text{train}}} \sum_{i,k=1}^{n_{\text{train}}} p_{ik} \frac{\left( \varrho_{i,k}^{\text{mini}} \right)^2}{\vartheta_{i,k}^{\text{mini}}} \frac{\left\| \mathbf{w}_{j,t}^{\text{mini}} \right\|}{\psi_t^{\text{mini}}} \frac{\phi_t^{\text{mini}} \psi_t^{\text{mini}}}{\left\| \mathbf{w}_{j,t}^{\text{mini}} \right\|^{\frac{3}{2}}} \right|$$

$$\leq \frac{1}{n_{\text{train}}} \sum_{i,k=1}^{n_{\text{train}}} \left( \left( \phi_t^{\text{mini}} \right)^2 + \left( \psi_t^{\text{mini}} \right)^2 \right)^{\frac{1}{4}} \frac{\left( \varrho_{i,k}^{\text{mini}} \right)^2}{\vartheta_{i,k}^{\text{mini}}} = o \left( \Gamma_t^{\text{full-mini}} \right). \tag{98}$$

Therefore, we have:

$$\left| \frac{\partial^2 L_{\text{train}}^{\text{full}}(\phi_t^{\text{mini}}, \psi_t^{\text{mini}}, \tilde{\mathbf{A}}_{\text{train}}^{\text{mini}})}{\partial \left( \phi_t^{\text{mini}} \right)^2} \right| \leq \Gamma_t^{\text{full-mini}} + \left( \left( \phi_t^{\text{mini}} \right)^2 + \left( \psi_t^{\text{mini}} \right)^2 \right)^{\frac{1}{4}} \Gamma_t^{\text{full-mini}} + o \left( \Gamma_t^{\text{full-mini}} \right)$$

$$\leq \Gamma_t^{\text{full-mini}} (1 + \sqrt{C} + o(1)) = C_1 \Gamma_t^{\text{full-mini}} \tag{99}$$

$$\left| \frac{\partial^2 L_{\text{train}}^{\text{full}}(\phi_t^{\text{mini}}, \psi_t^{\text{mini}}, \tilde{\mathbf{A}}_{\text{train}}^{\text{mini}})}{\partial \left( \psi_t^{\text{mini}} \right)^2} \right| \leq \Gamma_t^{\text{full-mini}} + \left( \left( \phi_t^{\text{mini}} \right)^2 + \left( \psi_t^{\text{mini}} \right)^2 \right)^{\frac{1}{4}} \Gamma_t^{\text{full-mini}} + o \left( \Gamma_t^{\text{full-mini}} \right)$$

$$\leq \Gamma_t^{\text{full-mini}} (1 + \sqrt{C} + o(1)) = C_1 \Gamma_t^{\text{full-mini}} \tag{100}$$

$$\left| \frac{\partial^2 L_{\text{train}}^{\text{full}}(\phi_t^{\text{mini}}, \psi_t^{\text{mini}}, \tilde{\mathbf{A}}_{\text{train}}^{\text{mini}})}{\partial \phi_t^{\text{mini}} \partial \psi_t^{\text{mini}}} \right| \leq \left( \left( \phi_t^{\text{mini}} \right)^2 + \left( \psi_t^{\text{mini}} \right)^2 \right)^{\frac{1}{4}} \Gamma_t^{\text{full-mini}} + o \left( \Gamma_t^{\text{full-mini}} \right)$$

$$\leq \Gamma_t^{\text{full-mini}}(\sqrt{C} + o(1)) = C_2 \Gamma_t^{\text{full-mini}} \tag{101}$$

$$\left| \frac{\partial^2 L_{\text{train}}^{\text{full}}(\phi_t^{\text{mini}}, \psi_t^{\text{mini}}, \tilde{\mathbf{A}}_{\text{train}}^{\text{mini}})}{\partial \psi_t^{\text{mini}} \partial \phi_t^{\text{mini}}} \right| \leq \left( \left( \phi_t^{\text{mini}} \right)^2 + \left( \psi_t^{\text{mini}} \right)^2 \right)^{\frac{1}{4}} \Gamma_t^{\text{full-mini}} + o \left( \Gamma_t^{\text{full-mini}} \right)$$

$$\leq \Gamma_t^{\text{full-mini}}(\sqrt{C} + o(1)) = C_2 \Gamma_t^{\text{full-mini}}, \tag{102}$$

where $C_1$ and $C_2$ are absolute constants.

Hence, we have:

$$\lambda_{\max}(\nabla^2 L_{\text{train}}^{\text{full}}(\phi^{\text{mini}}, \psi^{\text{mini}}, \tilde{\mathbf{A}}_{\text{train}}^{\text{mini}})) \leq C_3 \Gamma_t^{\text{full-mini}}, \tag{103}$$

where $C_3 = 4 \left( 1 + \sqrt{\frac{\pi}{2} + O\left( \kappa \sqrt{r} \right) + o(1)} \right)$ is an absolute constant.

Similarly, in the full-graph training, we have:

$$\lambda_{\max}(\nabla^2 L_{\text{train}}^{\text{full}}(\phi^{\text{full}}, \psi^{\text{full}})) \leq C_4 \Gamma^{\text{full}}, \tag{104}$$

where $C_4 = 4 \left( 1 + \sqrt{\frac{\pi}{2} + O\left( \kappa \sqrt{r} \right) + o(1)} \right)$ is an absolute constant.

### I.4 PROOF OF LEMMA B.8 AND C.4

We first focus on the mini-batch training. Due to random initialization, with probability at least $1 - \frac{1}{h^2}$, we have that $\sqrt{\left( \phi_0^{\text{mini}} \right)^2 + \left( \psi_0^{\text{mini}} \right)^2} = o\left( \kappa \sqrt{r} \right)$ and $\phi_0^{\text{mini}} \geq -C\kappa\sqrt{\log h}$ with a constant $C > 0$. Furthermore, we have the following updates:

$$\left( \phi_{t+1}^{\text{mini}} \right)^2 + \left( \psi_{t+1}^{\text{mini}} \right)^2$$

$$= \left( \sqrt{(\phi_t^{\text{mini}})^2 + (\psi_t^{\text{mini}})^2} \left( 1 - \eta_t \Gamma_t^{\text{mini}} + \eta_t \Upsilon_t^{\text{mini}} \right) \right)^2 + \eta_t^2 \frac{\left( \psi_t^{\text{mini}} \right)^2}{(\phi_t^{\text{mini}})^2 + (\psi_t^{\text{mini}})^2} \left( \Xi_t^{\text{mini}} \right)^2, \tag{105}$$

$$\phi_{t+1}^{\text{mini}} = \phi_t^{\text{mini}} \left( 1 - \eta_t \Gamma_t^{\text{mini}} \right) + \eta_t \frac{\phi_t^{\text{mini}}}{\sqrt{(\phi_t^{\text{mini}})^2 + (\psi_t^{\text{mini}})^2}} \Upsilon_t^{\text{mini}} + \eta_t \frac{\left( \psi_t^{\text{mini}} \right)^2}{(\phi_t^{\text{mini}})^2 + (\psi_t^{\text{mini}})^2} \Xi_t^{\text{mini}}. \tag{106}$$

Since $\Upsilon_t^{\text{mini}} > 0$ and is bounded by $\Gamma_t^{\text{mini}}$ and $\Xi_t^{\text{mini}} = o(\Gamma_t^{\text{mini}})$, if $\phi_t^{\text{mini}} < 0$ and $\sqrt{\left( \phi_t^{\text{mini}} \right)^2 + \left( \psi_t^{\text{mini}} \right)^2} \geq 2$, we have:

$$\sqrt{(\phi_{t+1}^{\text{mini}})^2 + (\psi_{t+1}^{\text{mini}})^2} \geq \sqrt{(\phi_t^{\text{mini}})^2 + (\psi_t^{\text{mini}})^2} \left( 1 - \eta_t \Gamma_t^{\text{mini}} \right), \tag{107}$$

$$\phi_{t+1}^{\text{mini}} \geq \phi_t^{\text{mini}} \left( 1 - \frac{\eta_t}{2} \Gamma_t^{\text{mini}} - \eta_t o \left( \Gamma_t^{\text{mini}} \right) \right). \tag{108}$$

Hence, after $t \geq C_1 \log(\kappa \log h)$ steps, we have that $\phi_t^{\text{mini}} \geq -\frac{1}{100}$ and $\sqrt{\left( \phi_t^{\text{mini}} \right)^2 + \left( \psi_t^{\text{mini}} \right)^2} \geq 2$.

Next we show that from this point on $\phi_t^{\text{mini}}$ and $\sqrt{\left( \phi_t^{\text{mini}} \right)^2 + \left( \psi_t^{\text{mini}} \right)^2}$, the conditions in this Lemma continue to be satisfied. We have:

$$\sqrt{(\phi_{t+1}^{\text{mini}})^2 + (\psi_{t+1}^{\text{mini}})^2}$$

$$\geq \sqrt{(\phi_t^{\text{mini}})^2 + (\psi_t^{\text{mini}})^2} - \eta_t \Gamma_t^{\text{mini}} \left( \sqrt{(\phi_t^{\text{mini}})^2 + (\psi_t^{\text{mini}})^2} - 1 \right) + \eta_t \left( \Upsilon_t^{\text{mini}} - \Gamma_t^{\text{mini}} \right), \tag{109}$$

where

$$\Upsilon_t^{\text{mini}} - \Gamma_t^{\text{mini}}$$

$$= \frac{1}{b} \sum_{i,j : \varrho_{i,j}^{\text{mini}} \neq 0}^{b} p_{ij} \sqrt{\vartheta_{i,j}^{\text{mini}}} \left( \hat{\sigma} \left( \frac{\varrho_{i,j}^{\text{mini}}}{\sqrt{\vartheta_{i,j}^{\text{mini}}}} \frac{\phi_t^{\text{mini}}}{\sqrt{(\phi_t^{\text{mini}})^2 + (\psi_t^{\text{mini}})^2}} \right) - \hat{\sigma} \left( \frac{\varrho_{i,j}^{\text{mini}}}{\sqrt{\vartheta_{i,j}^{\text{mini}}}} \right) \right). \tag{110}$$

Once $\phi_t^{\text{mini}} \geq -\frac{1}{100}$, we have:

$$\left| \hat{\sigma} \left( \frac{\varrho_{i,j}^{\text{mini}}}{\sqrt{\vartheta_{i,j}^{\text{mini}}}} \frac{\phi_t^{\text{mini}}}{\sqrt{\left(\phi_t^{\text{mini}}\right)^2 + \left(\psi_t^{\text{mini}}\right)^2}} \right) - \hat{\sigma} \left( \frac{\varrho_{i,j}^{\text{mini}}}{\sqrt{\vartheta_{i,j}^{\text{mini}}}} \right) \right| \leq 2 \frac{\varrho_{i,j}^{\text{mini}}}{\sqrt{\vartheta_{i,j}^{\text{mini}}}}. \tag{111}$$

Hence, we have that if $\phi_t^{\text{mini}} \geq -\frac{1}{100}$, then $\sqrt{\left(\phi_t^{\text{mini}}\right)^2 + \left(\psi_t^{\text{mini}}\right)^2} \geq 1 - o(1)$.

Next we discuss that $\phi_t^{\text{mini}}$ continues to be larger than $-\frac{1}{100}$. First, if $\phi_t^{\text{mini}} > 0$, then $\sqrt{\left(\phi_t^{\text{mini}}\right)^2 + \left(\psi_t^{\text{mini}}\right)^2} \geq 1 - o(1)$ remains. Furthermore, if $\phi_t^{\text{mini}} \in [-\frac{1}{100}, 0)$, then $\Xi_t^{\text{mini}}$ is non negative and is at least $\frac{1}{4b} \sum_{i,j,\varrho_{i,j}^{\text{mini}} \neq 0}^{b} p_{ij} \frac{\varrho_{i,j}^{\text{mini}}}{\sqrt{\vartheta_{i,j}^{\text{mini}}}}$. Hence, we have:

$$\begin{aligned}
\phi_{t+1}^{\text{mini}} \geq & \phi_t^{\text{mini}} - \eta_t \Gamma_t^{\text{mini}} \phi_t^{\text{mini}} \left( 1 - \frac{1}{\sqrt{\left(\phi_t^{\text{mini}}\right)^2 + \left(\psi_t^{\text{mini}}\right)^2}} \right) \\
& + \eta_t \frac{\phi_t^{\text{mini}}}{\sqrt{\left(\phi_t^{\text{mini}}\right)^2 + \left(\psi_t^{\text{mini}}\right)^2}} \left( \Upsilon_t^{\text{mini}} - \Gamma_t^{\text{mini}} \right) \\
& + \eta_t \frac{\left(\psi_t^{\text{mini}}\right)^2}{\left(\phi_t^{\text{mini}}\right)^2 + \left(\psi_t^{\text{mini}}\right)^2} \frac{\sum_{i,j,\varrho_{i,j}^{\text{mini}} \neq 0}^{b} p_{ij} \frac{\varrho_{i,j}^{\text{mini}}}{\sqrt{\vartheta_{i,j}^{\text{mini}}}}}{4b}.
\end{aligned} \tag{112}$$

Using $\left| \sqrt{\left(\phi_t^{\text{mini}}\right)^2 + \left(\psi_t^{\text{mini}}\right)^2} - 1 \right| = O(1)$ and the fact that if $\phi_t^{\text{mini}} \in [-\frac{1}{100}, 0)$ and $\sqrt{\left(\phi_t^{\text{mini}}\right)^2 + \left(\psi_t^{\text{mini}}\right)^2} \geq 1 - o(1)$, then $\left| \frac{\psi_t^{\text{mini}}}{\sqrt{\left(\phi_t^{\text{mini}}\right)^2 + \left(\psi_t^{\text{mini}}\right)^2}} \right| \geq \frac{1}{2}$, we have that $\phi_{t+1}^{\text{mini}} \geq \phi_t^{\text{mini}}$.

Similarly, under the full-graph training, we can replace $b$ by $n_{\text{train}}$.

## I.5 PROOF OF LEMMA B.9 AND E.5

We first focus on the full-graph training. We have

$$\begin{aligned}
& \left\| \nabla L_{\text{train},t}^{\text{full}} \left( \phi_t^{\text{full}}, \psi_t^{\text{full}} \right) \right\|^2 \\
& = \left( \phi_t^{\text{full}} \Gamma^{\text{full}} - \frac{\phi_t^{\text{full}}}{\sqrt{\left(\phi_t^{\text{full}}\right)^2 + \left(\psi_t^{\text{full}}\right)^2}} \Upsilon_t^{\text{full}} - \frac{\left(\psi_t^{\text{full}}\right)^2}{\left(\phi_t^{\text{full}}\right)^2 + \left(\psi_t^{\text{full}}\right)^2} \Xi_t^{\text{full}} \right)^2 \\
& \quad + \left( \psi_t^{\text{full}} \Gamma^{\text{full}} - \frac{\psi_t^{\text{full}}}{\sqrt{\left(\phi_t^{\text{full}}\right)^2 + \left(\psi_t^{\text{full}}\right)^2}} \Upsilon_t^{\text{full}} - \frac{\phi_t^{\text{full}} \psi_t^{\text{full}}}{\left(\phi_t^{\text{full}}\right)^2 + \left(\psi_t^{\text{full}}\right)^2} \Xi_t^{\text{full}} \right)^2 \\
& = \left( \sqrt{\left(\phi_t^{\text{full}}\right)^2 + \left(\psi_t^{\text{full}}\right)^2} \Gamma^{\text{full}} - \Upsilon_t^{\text{full}} \right)^2 + \frac{\left(\psi_t^{\text{full}}\right)^2}{\left(\phi_t^{\text{full}}\right)^2 + \left(\psi_t^{\text{full}}\right)^2} \left( \Xi_t^{\text{full}} \right)^2.
\end{aligned} \tag{113}$$

On the other hand, the loss $L_{\text{train},t}^{\text{full}} \left( \phi_t^{\text{full}}, \psi_t^{\text{full}} \right)$ can be written as :

$$\begin{aligned}
& L_{\text{train},t}^{\text{full}} \left( \phi_t^{\text{full}}, \psi_t^{\text{full}} \right) \\
& = \frac{1}{2} \left( \left(\phi_t^{\text{full}}\right)^2 + \left(\psi_t^{\text{full}}\right)^2 + 1 \right) \Gamma^{\text{full}} - \sqrt{\left(\phi_t^{\text{full}}\right)^2 + \left(\psi_t^{\text{full}}\right)^2} \Upsilon_t^{\text{full}} \\
& \leq \frac{1}{2} \left( \left(\phi_t^{\text{full}}\right)^2 + \left(\psi_t^{\text{full}}\right)^2 + 1 \right) \Gamma^{\text{full}}.
\end{aligned} \tag{114}$$

Hence, we have:

$$\frac{\left\|\nabla L_{\text{train},t}^{\text{full}}\left(\phi_t^{\text{full}},\psi_t^{\text{full}}\right)\right\|^2}{L_{\text{train},t}^{\text{full}}\left(\phi_t^{\text{full}},\psi_t^{\text{full}}\right)} \geq \frac{\left(\sqrt{\left(\phi_t^{\text{full}}\right)^2+\left(\psi_t^{\text{full}}\right)^2}\Gamma^{\text{full}}-\Upsilon_t^{\text{full}}\right)^2}{\Gamma^{\text{full}}}$$
$$+2\frac{\left(\psi_t^{\text{full}}\right)^2\left(\Xi_t^{\text{full}}\right)^2}{\left(\left(\phi_t^{\text{full}}\right)^2+\left(\psi_t^{\text{full}}\right)^2\right)\left(\left(\phi_t^{\text{full}}\right)^2+\left(\psi_t^{\text{full}}\right)^2+1\right)\Gamma^{\text{full}}}. \tag{115}$$

If $\sqrt{\left(\phi_t^{\text{full}}\right)^2+\left(\psi_t^{\text{full}}\right)^2}-1 > \frac{\epsilon^{\frac{1}{2}}}{2h}$, then the first term above combined with $\Upsilon_t^{\text{full}} \leq \Gamma^{\text{full}}$ leads to

$$\frac{\left\|\nabla L_{\text{train},t}^{\text{full}}\left(\phi_t^{\text{full}},\psi_t^{\text{full}}\right)\right\|^2}{L_{\text{train},t}^{\text{full}}\left(\phi_t^{\text{full}},\psi_t^{\text{full}}\right)} \geq \frac{\epsilon\Gamma^{\text{full}2}}{4h^2\Gamma^{\text{full}}} = \frac{\epsilon\Gamma^{\text{full}}}{4h^2}. \tag{116}$$

If $\left|\sqrt{\left(\phi_t^{\text{full}}\right)^2+\left(\psi_t^{\text{full}}\right)^2}-1\right| \leq \frac{\epsilon^{\frac{1}{2}}}{2h} \leq 2$ and $\left|\psi_t^{\text{full}}\right| > \frac{\epsilon^{\frac{1}{2}}}{2h}$, then the second term leads to

$$\frac{\left\|\nabla L_{\text{train},t}^{\text{full}}\left(\phi_t^{\text{full}},\psi_t^{\text{full}}\right)\right\|^2}{L_{\text{train},t}^{\text{full}}\left(\phi_t^{\text{full}},\psi_t^{\text{full}}\right)} \geq 2\frac{\left(\psi_t^{\text{full}}\right)^2\left(\Xi_t^{\text{full}}\right)^2}{\left(\left(\phi_t^{\text{full}}\right)^2+\left(\psi_t^{\text{full}}\right)^2\right)\left(\left(\phi_t^{\text{full}}\right)^2+\left(\psi_t^{\text{full}}\right)^2+1\right)\Gamma^{\text{full}}}$$
$$\geq 2\frac{\left(\psi_t^{\text{full}}\right)^2\left(\Xi_t^{\text{full}}\right)^2}{9(9+1)\Gamma^{\text{full}}}$$
$$\geq 2\frac{\left(\Xi_t^{\text{full}}\right)^2}{90\Gamma^{\text{full}}}\frac{\epsilon}{4h^2}$$
$$\geq 2\frac{\Gamma^{\text{full}}}{360d_{\max}^2}\frac{\epsilon}{4h^2} \tag{117}$$

Hence, we have:

$$\left\|\nabla L_{\text{train},t}^{\text{full}}\left(\phi_t^{\text{full}},\psi_t^{\text{full}}\right)\right\|^2 \geq \mu^{\text{full}}L_{\text{train},t}^{\text{full}}\left(\phi_t^{\text{full}},\psi_t^{\text{full}}\right), \tag{118}$$

where $\mu^{\text{full}} \geq C_1\epsilon h^{-2}d_{\max}^{-2}\Gamma^{\text{full}}$, and $C_1$ is a positive constant.

Similarly, in the mini-batch training, we can replace $d_{\max}$ by $\beta$, we have:

$$\left\|\nabla L_{\text{train},t}^{\text{full}}\left(\phi_t^{\text{mini}},\psi_t^{\text{mini}},\tilde{\mathbf{A}}_{\text{train}}^{\text{mini}}\right)\right\|^2 \geq \mu_t^{\text{mini}}L_{\text{train},t}^{\text{full}}\left(\phi_t^{\text{mini}},\psi_t^{\text{mini}},\tilde{\mathbf{A}}_{\text{train}}^{\text{mini}}\right), \tag{119}$$

where $\mu_t^{\text{mini}} \geq C_2\epsilon h^{-2}\beta^{-2}\Gamma^{\text{full-mini}}$, and $C_2$ is a positive constant.

Finally, we are going to consider the case in the full-graph training when $\sqrt{\left(\phi_t^{\text{full}}\right)^2+\left(\psi_t^{\text{full}}\right)^2} \leq 1-\frac{\epsilon^{\frac{1}{2}}}{2h}$. We can assume that $\left|\Upsilon_t^{\text{full}}-\sqrt{\left(\phi_t^{\text{full}}\right)^2+\left(\psi_t^{\text{full}}\right)^2}\Gamma^{\text{full}}\right| \leq \frac{\epsilon^{\frac{1}{2}}}{2h}\Gamma^{\text{full}}$ since otherwise we get the same bound as in (116). In this case, we show that $\left\|\psi_t^{\text{full}}\right\|$ must be at least $\frac{\epsilon^{\frac{1}{2}}}{2h}$ and hence the bound of (117) can be applicable. Using $\hat{\sigma}(\cdot)$ is convex in $[0,1]$, we can get

$$n_{\text{train}}p_{ij}\left(\sqrt{\vartheta_{i,j}^{\text{full}}}\left(\hat{\sigma}\left(\frac{\varrho_{i,j}^{\text{full}}}{\sqrt{\vartheta_{i,j}^{\text{full}}}}\frac{\phi_t^{\text{full}}}{\sqrt{\left(\phi_t^{\text{full}}\right)^2+\left(\psi_t^{\text{full}}\right)^2}}\right)-\hat{\sigma}\left(\frac{\varrho_{i,j}^{\text{full}}}{\sqrt{\vartheta_{i,j}^{\text{full}}}}\right)\right)\right)$$
$$\geq p_{ij}\varrho_{i,j}^{\text{full}}\frac{\phi_t^{\text{full}}-\sqrt{\left(\phi_t^{\text{full}}\right)^2+\left(\psi_t^{\text{full}}\right)^2}}{\sqrt{\left(\phi_t^{\text{full}}\right)^2+\left(\psi_t^{\text{full}}\right)^2}}\hat{\sigma}_{\text{step}}\left(\frac{\varrho_{i,j}^{\text{full}}}{\sqrt{\vartheta_{i,j}^{\text{full}}}}\right). \tag{120}$$

Summing over $i,j$, we have

$$n_{\text{train}}\left(\Upsilon_t^{\text{full}}-\Gamma^{\text{full}}\right) \geq \frac{\phi_t^{\text{full}}-\sqrt{\left(\phi_t^{\text{full}}\right)^2+\left(\psi_t^{\text{full}}\right)^2}}{\sqrt{\left(\phi_t^{\text{full}}\right)^2+\left(\psi_t^{\text{full}}\right)^2}}\sum_{i,j}p_{ij}\varrho_{i,j}^{\text{full}}\hat{\sigma}_{\text{step}}(\frac{\varrho_{i,j}^{\text{full}}}{\sqrt{\vartheta_{i,j}^{\text{full}}}}). \tag{121}$$

Substituting $\Upsilon_t^{\text{full}} = \sqrt{\left(\phi_t^{\text{full}}\right)^2 + \left(\psi_t^{\text{full}}\right)^2}\Gamma^{\text{full}} \pm \frac{\epsilon^{\frac{1}{2}}\Gamma^{\text{full}}}{2h}$, we have

$$
\begin{aligned}
n_{\text{train}} &\left( \left( \sqrt{\left(\phi_t^{\text{full}}\right)^2 + \left(\psi_t^{\text{full}}\right)^2} - 1 \right) \Gamma^{\text{full}} \pm \frac{\epsilon^{\frac{1}{2}}\Gamma^{\text{full}}}{2h} \right) \\
&\geq \frac{\phi_t^{\text{full}} - \sqrt{\left(\phi_t^{\text{full}}\right)^2 + \left(\psi_t^{\text{full}}\right)^2}}{\sqrt{\left(\phi_t^{\text{full}}\right)^2 + \left(\psi_t^{\text{full}}\right)^2}} \sum_{i,j} p_{ij} \varrho_{i,j}^{\text{full}} \hat{\sigma}_{\text{step}}(\frac{\varrho_{i,j}^{\text{full}}}{\sqrt{\vartheta_{i,j}^{\text{full}}}}).
\end{aligned}
\tag{122}
$$

Using the bound on $\sqrt{\left(\phi_t^{\text{full}}\right)^2 + \left(\psi_t^{\text{full}}\right)^2}$, the above implies that

$$
\frac{\epsilon^{\frac{1}{2}} n_{\text{train}}\Gamma^{\text{full}}}{2h} \leq \frac{\sqrt{\left(\phi_t^{\text{full}}\right)^2 + \left(\psi_t^{\text{full}}\right)^2} - \phi_t^{\text{full}}}{\sqrt{\left(\phi_t^{\text{full}}\right)^2 + \left(\psi_t^{\text{full}}\right)^2}} \sum_{i,j} p_{ij} \varrho_{i,j}^{\text{full}} \hat{\sigma}_{\text{step}}(\frac{\varrho_{i,j}^{\text{full}}}{\sqrt{\vartheta_{i,j}^{\text{full}}}})
\tag{123}
$$

Noticing that $\Gamma^{\text{full}} \geq \frac{1}{\pi n_{\text{train}}}\|\tilde{\mathbf{A}}_{\text{train},t}^{\text{full}}\mathbb{1}\|_1$ by Lemma C.1, we have

$$
\frac{\sqrt{\left(\phi_t^{\text{full}}\right)^2 + \left(\psi_t^{\text{full}}\right)^2} - \phi_t^{\text{full}}}{\sqrt{\left(\phi_t^{\text{full}}\right)^2 + \left(\psi_t^{\text{full}}\right)^2}} \geq \frac{\epsilon^{\frac{1}{2}}\|\tilde{\mathbf{A}}_{\text{train},t}^{\text{full}}\mathbb{1}\|_1}{2\pi h \sum_{i,j} p_{ij} \varrho_{i,j}^{\text{full}} \hat{\sigma}_{\text{step}}(\frac{\varrho_{i,j}^{\text{full}}}{\sqrt{\vartheta_{i,j}^{\text{full}}}})}
\tag{124}
$$

Therefore, we have

$$
\begin{aligned}
\psi_t^{\text{full}} &\geq \sqrt{\left(\phi_t^{\text{full}}\right)^2 + \left(\psi_t^{\text{full}}\right)^2} - \phi_t^{\text{full}} \\
&\geq \frac{\epsilon^{\frac{1}{2}}\|\tilde{\mathbf{A}}_{\text{train},t}^{\text{full}}\mathbb{1}\|_1}{2\pi h \sum_{i,j} p_{ij} \varrho_{i,j}^{\text{full}} \hat{\sigma}_{\text{step}}(\frac{\varrho_{i,j}^{\text{full}}}{\sqrt{\vartheta_{i,j}^{\text{full}}}})} \sqrt{\left(\phi_t^{\text{full}}\right)^2 + \left(\psi_t^{\text{full}}\right)^2} \\
&\geq \frac{\epsilon^{\frac{1}{2}}\|\tilde{\mathbf{A}}_{\text{train},t}^{\text{full}}\mathbb{1}\|_1}{2\pi h \sum_{i,j} p_{ij} \varrho_{i,j}^{\text{full}} \hat{\sigma}_{\text{step}}(\frac{\varrho_{i,j}^{\text{full}}}{\sqrt{\vartheta_{i,j}^{\text{full}}}})} \\
&> \frac{\epsilon^{\frac{1}{2}}\|\tilde{\mathbf{A}}_{\text{train},t}^{\text{full}}\mathbb{1}\|_1}{2h n_{\text{train}}} \\
&\geq \frac{\epsilon^{\frac{1}{2}}}{2h n_{\text{train}}^{\frac{1}{2}} d_{\max}^{\frac{1}{2}}} > \frac{\epsilon^{\frac{1}{2}}}{2h}
\end{aligned}
\tag{125}
$$

where the second last inequality uses $\sum_{i,j} p_{ij} \varrho_{i,j}^{\text{full}} \hat{\sigma}_{\text{step}}(\frac{\varrho_{i,j}^{\text{full}}}{\sqrt{\vartheta_{i,j}^{\text{full}}}}) \leq n_{\text{train}}$ because there exist $n_{\text{train}}$ nodes that have the common messages.

Similarly, in the mini-batch training, we can replace $n_{\text{train}}$ and $d_{\max}$ by $b$ and $\beta$, respectively.

# J  PROOF OF AUXILIARY LEMMAS OF CONVERGENCE THEOREMS WITH MSE

## J.1  PROOF OF LEMMA I.1:

We first focus on the mini-batch training. We are going to analyze the upper bound of $\Xi^{\text{mini}}$:

$$
\Xi^{\text{mini}} = \frac{1}{b} \sum_{i,j=1}^{b} p_{ij} \varrho_{i,j}^{\text{mini}} \hat{\sigma}_{\text{step}} \left( \frac{\phi^{\text{mini}}}{\sqrt{\left(\phi^{\text{mini}}\right)^2 + \left(\psi^{\text{mini}}\right)^2}} \frac{\varrho_{i,j}^{\text{mini}}}{\sqrt{\vartheta_{i,j}^{\text{mini}}}} \right),
\tag{126}
$$

where each term in summation is non-zero only when $\varrho_{i,j}^{\text{mini}} \neq 0$ if $i = j$.

Hence, there are at most $b$ non-zero terms in the summation, and $\Xi_t^{\text{mini}}$ is upper bounded by $\Gamma_t^{\text{mini}}$, namely $\Xi_t^{\text{mini}} = o(\Gamma_t^{\text{mini}})$.

Note that $\hat{\sigma}_{\text{step}}(x) \geq \frac{1}{2}$ whenever $|x| \leq \frac{1}{50}$ (Daniely et al., 2016), which is ensured by $\phi_t \geq -\frac{1}{100}$ and $\sqrt{\left(\phi_t^{\text{mini}}\right)^2 + \left(\psi_t^{\text{mini}}\right)^2} \geq 1 - o(1)$. Hence, in this case, each term in the summation in the expression of $\Xi^{\text{mini}}$ will be non-negative. Then, for $i = j$, each of the $b$ terms in the summation will contributed at least $\frac{1}{200}$ (Awasthi et al., 2021). Therefore, in this case $\Xi_t^{\text{mini}} \geq \frac{1}{2} \geq \frac{\Gamma_t^{\text{mini}}}{2\beta}$ with $\Gamma_t^{\text{mini}} \leq \beta$.

Similarly, we have $\Xi_t^{\text{full-mini}} = o(\Gamma_t^{\text{full-mini}})$, and, when $\phi_t^{\text{mini}} \geq -\frac{1}{100}$ and $\sqrt{\left(\phi_t^{\text{mini}}\right)^2 + \left(\psi_t^{\text{mini}}\right)^2} \geq 1 - o(1)$, $\Xi_t^{\text{full-mini}} \geq \frac{\Gamma_t^{\text{full-mini}}}{2\beta}$.

Similarly, under the full-graph training, we place $b$ and $\beta$ by $n_{\text{train}}$ and $d_{\text{max}}$, respectively.

## K PROOF OF THE MAIN LEMMAS OF CONVERGENCE THEOREMS WITH CE

### K.1 PROOF OF LEMMA D.3 AND E.3

**Lemma K.1** Let $\tilde{\mathbf{A}}$ be the normalized adjacency matrix with self-loops. Given a mini-batch of size $b$ and fan-out size $\beta$, the following inequalities hold:

$$\left\|\tilde{\mathbf{a}}_{\text{train},i}^{\text{mini}}\right\|_2^2 \leq \beta,$$

and

$$\left\|\tilde{\mathbf{a}}_{\text{train},i}^{\text{full}}\right\|_2^2 \leq d_{\text{max}},$$

for any $i$ in the training set.

**Lemma K.2** With Gaussian random initialization, for any $\delta \in (0,1)$, if $h \geq C \log(n/\delta)$ for some large enough constant $C$, then with probability at least $1 - \delta$, the following inequalities hold:

$$\left|\left\|\mathbf{z}_i^{\text{mini}}\right\|_2 - C_x^{\frac{1}{2}} \beta^{\frac{1}{2}}\right| \leq C_1 \sqrt{\beta \frac{\log(n_{\text{train}}/\delta)}{h}},$$

and

$$\left|\left\|\mathbf{z}_i^{\text{full}}\right\|_2 - C_x^{\frac{1}{2}} d_{\text{max}}^{\frac{1}{2}}\right| \leq C_2 \sqrt{d_{\text{max}} \frac{\log(n_{\text{train}}/\delta)}{h}},$$

for any $i$ in the training set, where $C_1$ and $C_2$ are absolute constants.

**Proof of Lemma D.3 and E.3:** Since half of the elements of $\mathbf{v}$ are 1's and the other half of the elements are $-1$'s, without loss of generality, we can assume that $\mathbf{v}_1 = \cdots = \mathbf{v}_{h/2} = 1$ and $\mathbf{v}_{h/2+1} = \cdots = \mathbf{v}_h = -1$.

Obviously, we have $\mathbb{E}\left(\hat{y}_i^{\text{mini}}\right) = \mathbb{E}\left(\hat{y}_i^{\text{full}}\right) = 0$ for any $i$ in the training set.

We first focus on the mini-batch training. Using the value of $\mathbf{v}$, we have:

$$\hat{y}_i^{\text{mini}} = \sum_{i=1}^{h/2} \left[\sigma\left(\tilde{\mathbf{a}}_{\text{train},i}^{\text{mini}}\mathbf{X}\left(\mathbf{w}_j^{\text{mini}}\right)^\top\right) - \sigma\left(\tilde{\mathbf{a}}_{\text{train},i}^{\text{mini}}\mathbf{X}\left(\mathbf{w}_{j+h/2}^{\text{mini}}\right)^\top\right)\right] \tag{127}$$

With the Lipschitz property of ReLU function, we have:

$$\begin{aligned}
&\left\|\sigma\left(\tilde{\mathbf{a}}_{\text{train},i}^{\text{mini}}\mathbf{X}\left(\mathbf{w}_j^{\text{mini}}\right)^\top\right) - \sigma\left(\tilde{\mathbf{a}}_{\text{train},i}^{\text{mini}}\mathbf{X}\left(\mathbf{w}_{j+h/2}^{\text{mini}}\right)^\top\right)\right\|_2 \\
&\leq \left\|\tilde{\mathbf{a}}_{\text{train},i}^{\text{mini}}\mathbf{X}\left(\mathbf{w}_j^{\text{mini}}\right)^\top - \tilde{\mathbf{a}}_{\text{train},i}^{\text{mini}}\mathbf{X}\left(\mathbf{w}_{j+h/2}^{\text{mini}}\right)^\top\right\|_2 \\
&= \left\|\tilde{\mathbf{a}}_{\text{train},i}^{\text{mini}}\mathbf{X}\left(\mathbf{w}_j^{\text{mini}} - \mathbf{w}_{j+h/2}^{\text{mini}}\right)^\top\right\|_2 \\
&\leq \left\|\tilde{\mathbf{a}}_{\text{train},i}^{\text{mini}}\right\|_2 \|\mathbf{X}\|_2 \left\|\mathbf{w}_j^{\text{mini}} - \mathbf{w}_{j+h/2}^{\text{mini}}\right\|_2 \\
&\leq C_3 h^{-\frac{1}{2}} \beta^{\frac{1}{2}},
\end{aligned} \tag{128}$$

for some absolute constant $C_3$. Here the last inequality follows Lemma K.1.

Therefore, by Hoeffding's inequality and Lemma K.2, we have:

$$\mathbb{P}\left(\left|\hat{y}_i^{\text{mini}}\right| > u\right) \leq 2\exp\left(-\frac{u^2}{\sum_{j=0}^{h/2}\left(C_3 h^{-\frac{1}{2}}\beta^{\frac{1}{2}}\right)^2}\right)$$

$$\mathbb{P}\left(\left|\hat{y}_i^{\text{mini}}\right| > u\right) \leq 2\exp\left(-\frac{2u^2}{C_3^2\beta}\right)$$

(129)

Taking union bound over $i$, we have

$$\mathbb{P}\left(\left|\hat{y}_i^{\text{mini}}\right| > u, i = 1, \ldots, n_{\text{train}}\right) \leq 2n_{\text{train}}\exp\left(-\frac{2u^2}{C_3^2\beta}\right).$$

(130)

Let $2n\exp\left(-\frac{2u^2}{C_3^2\beta}\right) = \delta$, we have:

$$exp\left(-\frac{2u^2}{C_3^2\beta}\right) = \frac{\delta}{2n_{\text{train}}},$$

$$-\frac{2u^2}{C_3^2\beta} = \log\left(\frac{\delta}{2n_{\text{train}}}\right),$$

$$u^2 = \frac{C_3^2\beta}{2}\log\left(\frac{\delta}{2n_{\text{train}}}\right) > 0,$$

$$u = C_4\sqrt{\beta\log\left(\frac{\delta}{n_{\text{train}}}\right)}.$$

(131)

Then $\mathbb{P}\left(\left|\hat{y}_i^{\text{mini}}\right| > C_4\sqrt{\beta\log\left(\frac{\delta}{n_{\text{train}}}\right)}, i = 1, \ldots, n_{\text{train}}\right) \leq \delta$.

Therefore, with the probability at least $1 - \delta$, it holds that

$$\left|\hat{y}_i^{\text{mini}}\right| \leq C_4\sqrt{\beta\log\left(\frac{\delta}{n_{\text{train}}}\right)},$$

(132)

for any $i$ in the training set.

Then substituting the above bound into the formula of loss function $l(y_i\hat{y}_i^{\text{mini}})$, we complete the proof of Lemma E.3. Further, substituting the $\beta$ with $d_{\max}$, we complete the proof of Lemma D.3 for the full-graph training.

## K.2 PROOF OF LEMMA D.4 AND E.4

**Lemma K.3** There exist absolute constants $C, C_1, C_2 > 0$ such that, with the probability at least $1 - \exp\left(-Ch\alpha^2/(n_{\text{train}}d_{\max})\right)$, for any $\mathbf{m} = (\mathbf{m}_1, \ldots, \mathbf{m}_{n_{\text{train}}}) \in \mathbb{R}_+^{n_{\text{train}}}$, there exist at least $C_1 h\alpha^2/(n_{\text{train}}d_{\max})$ GNN nodes in $\{1, \ldots, j, \ldots, h\}$ that satisfy:

$$\left\|\frac{1}{n_{\text{train}}}\sum_{i=1}^{n_{\text{train}}} m_i y_i \sigma'\left(\tilde{\mathbf{a}}_{\text{train},i}^{\text{full}}\mathbf{X}\left(\mathbf{w}_j^{\text{mini}}\right)^\top\right)\tilde{\mathbf{a}}_{\text{train},i}^{\text{full}}\mathbf{X}\right\|_2 \geq C_2\|\mathbf{m}\|_\infty d_{\max}^2.$$

**Lemma K.4** There exist absolute constants $C_3, C_4, C_5 > 0$ such that, with the probability at least $1 - \exp\left(-C_3 h\alpha^2/(n_{\text{train}}\beta)\right)$, for any $\mathbf{m} = (\mathbf{m}_1, \ldots, \mathbf{m}_{n_{\text{train}}}) \in \mathbb{R}_+^{n_{\text{train}}}$, there exist at least $C_4 h\alpha^2/(n_{\text{train}}\beta)$ GNN nodes in $\{1, \ldots, j, \ldots, h\}$ that satisfy:

$$\left\|\frac{1}{n_{\text{train}}}\sum_{i=1}^{n_{\text{train}}} m_i y_i \sigma'\left(\tilde{\mathbf{a}}_{\text{train},i}^{\text{mini}}\mathbf{X}\left(\mathbf{w}_j^{\text{mini}}\right)^\top\right)\tilde{\mathbf{a}}_{\text{train},i}^{\text{mini}}\mathbf{X}\right\|_2 \geq C_5\|\mathbf{m}\|_\infty \beta^2.$$

**Proof of Lemma D.4 and E.4:**   We first focus on the mini-batch training. We are going to prove the gradient upper bound. The gradient $\nabla_{\mathbf{W}^{\text{mini}}} l\left(y_i \hat{y}_i^{\text{mini}}\right)$ can be written as:

$$
\begin{aligned}
\nabla_{\mathbf{W}^{\text{mini}}} l\left(y_i \hat{y}_i^{\text{mini}}\right) &= l'\left(y_i \hat{y}_i^{\text{mini}}\right) \cdot y_i \cdot \nabla_{\mathbf{W}^{\text{mini}}} \hat{y}_i^{\text{mini}} \\
&= l'\left(y_i \hat{y}_i^{\text{mini}}\right) \cdot y_i \cdot \left(\mathbf{v}\boldsymbol{\Sigma}_i^{\text{mini}}\right)^\top \tilde{\mathbf{a}}_{\text{train},i}^{\text{mini}} \mathbf{X}.
\end{aligned}
\tag{133}
$$

Since $\boldsymbol{\Sigma}_i^{\text{mini}}$ is a diagonal matrix with $\left(\boldsymbol{\Sigma}_i^{\text{mini}}\right)_{jj} \in \{0,1\}$ for any $j \in \{1, \ldots, h\}$, we have $\left\|\boldsymbol{\Sigma}_i^{\text{mini}}\right\|_2 = 1$ for any $i$ in the training set.

Hence, we have the following upper bound on $\left\|\nabla_{\mathbf{W}^{\text{mini}}} l\left(y_i \hat{y}_i^{\text{mini}}\right)\right\|_F$:

$$
\begin{aligned}
\left\|\nabla_{\mathbf{W}^{\text{mini}}} l\left(y_i \hat{y}_i^{\text{mini}}\right)\right\|_F &= \left\|\nabla_{\mathbf{W}^{\text{mini}}} l\left(y_i \hat{y}_i^{\text{mini}}\right)\right\|_2 \\
&\leq -l'\left(y_i \hat{y}_i^{\text{mini}}\right) \left\|\boldsymbol{\Sigma}_i^{\text{mini}}\right\|_2 \|\mathbf{v}\|_2 \left\|\tilde{\mathbf{a}}_{\text{train},i}^{\text{mini}}\right\|_2 \|\mathbf{X}\|_2 \\
&\leq -l'\left(y_i \hat{y}_i^{\text{mini}}\right) C_x^{\frac{1}{2}} h^{\frac{1}{2}} \beta^{\frac{1}{2}},
\end{aligned}
\tag{134}
$$

where the first equality holds due to the fact that $\nabla_{\mathbf{W}^{\text{mini}}} l\left(y_i \hat{y}_i^{\text{mini}}\right)$ is a rank-one matrix, and the last ineuqality follows Lemma K.1 and $\|\mathbf{v}\|_2 = h^{\frac{1}{2}}$.

Further, we have the following for $\nabla_{\mathbf{W}^{\text{mini}}} \hat{L}_{\text{train}}^{\text{mini}}\left(\mathbf{W}^{\text{mini}}, \tilde{\mathbf{A}}_{\text{train}}^{\text{mini}}\right)$:

$$
\begin{aligned}
\left\|\nabla_{\mathbf{W}^{\text{mini}}} \hat{L}_{\text{train}}^{\text{mini}}\left(\mathbf{W}^{\text{mini}}, \tilde{\mathbf{A}}_{\text{train}}^{\text{mini}}\right)\right\|_F &= \left\|\frac{1}{b} \sum_{i=0}^{b} \nabla_{\mathbf{W}^{\text{mini}}} l\left(y_i \hat{y}_i^{\text{mini}}\right)\right\|_F \\
&\leq \frac{1}{b} \sum_{i=0}^{b} \left\|\nabla_{\mathbf{W}^{\text{mini}}} l\left(y_i \hat{y}_i^{\text{mini}}\right)\right\|_F \\
&\leq -\frac{C_6 h^{\frac{1}{2}} \beta^{\frac{1}{2}}}{b} \sum_{i=0}^{b} l'\left(y_i \hat{y}_i^{\text{mini}}\right),
\end{aligned}
\tag{135}
$$

where $C_6$ is a positive constant.

Then, replacing $b$ and $\beta$ by $n_{\text{train}}$ and $d_{\text{max}}$ respectively, we have:

$$
\left\|\nabla_{\mathbf{W}^{\text{full}}} \hat{L}_{\text{train}}^{\text{full}}\left(\mathbf{W}^{\text{full}}\right)\right\|_F \leq -\frac{C_6 h^{\frac{1}{2}} d_{\text{max}}^{\frac{1}{2}}}{n_{\text{train}}} \sum_{i=1}^{n_{\text{train}}} l'\left(y_i \hat{y}_i^{\text{full}}\right),
\tag{136}
$$

for the full-graph training.

Next, we still focus on the mini-batch training. We are going to prove the gradient lower bound.

Given the initilization $\mathbf{W}_0^{\text{mini}}$ and any $\tilde{\mathbf{W}}^{\text{mini}} \in \mathcal{B}\left(\mathbf{W}_0^{\text{mini}}, \tau\right)$, where $\mathcal{B}\left(\mathbf{W}_0^{\text{mini}}, \tau\right) = \left\{\mathbf{W} : \left\|\mathbf{W} - \mathbf{W}_0^{\text{mini}}\right\|_2 \leq \tau\right\}$.

We define:

$$
\mathbf{g}_j = \frac{1}{n_{\text{train}}} \sum_{i=0}^{n_{\text{train}}} l'\left(y_i \hat{y}_i^{\text{mini}}\right) y_i \mathbf{v}_j \sigma'\left(\tilde{\mathbf{a}}_{\text{train},i}^{\text{mini}} \mathbf{X} \left(\mathbf{W}_{i,0}^{\text{mini}}\right)^\top\right) \tilde{\mathbf{a}}_{\text{train},i}^{\text{mini}} \mathbf{X}.
\tag{137}
$$

Then, since $\mathbf{W}_0$ is generated via Gaussian random initialization, by Lemma K.4, we have the following inequality holds for at least $C_4 h \alpha^2 / \left(n_{\text{train}} \beta\right)$ GNN nodes:

$$
\|\mathbf{g}_j\|_2 \geq C_5 \max_i \left| l'\left(y_i \hat{y}_i^{\text{mini}}\right)\right| \beta^2,
\tag{138}
$$

where $C_4$ and $C_5$ are positive absolute constants.

Further, we can rewrite $\nabla_{\mathbf{w}_j^{\text{mini}}} \hat{L}_{\text{train}}^{\text{full}}\left(\mathbf{w}_j^{\text{mini}}, \tilde{\mathbf{A}}_{\text{train}}^{\text{mini}}\right)$ as follows:

$$
\nabla_{\tilde{\mathbf{w}}_j^{\text{mini}}} \hat{L}_{\text{train}}^{\text{full}}\left(\tilde{\mathbf{w}}_j^{\text{mini}}, \tilde{\mathbf{A}}_{\text{train}}^{\text{mini}}\right) = \frac{1}{n_{\text{train}}} \sum_{i=1}^{n_{\text{train}}} l'\left(y_i \hat{y}_i^{\text{mini}}\right) y_i \mathbf{v}_j \sigma'\left(\tilde{\mathbf{a}}_{\text{train},i}^{\text{mini}} \mathbf{X} \left(\tilde{\mathbf{w}}_j^{\text{mini}}\right)^\top\right) \tilde{\mathbf{a}}_{\text{train},i}^{\text{mini}} \mathbf{X}.
\tag{139}
$$

Let $\mathbf{z}_{i,j} = l'\left(y_i \hat{y}_i^{\text{mini}}\right) y_i \mathbf{v}_j$, we have:

$$
\begin{aligned}
& \|\mathbf{g}_j\|_2 - \left\| \nabla_{\tilde{\mathbf{w}}_j^{\text{mini}}} \hat{L}_{\text{train}}^{\text{full}} \left( \tilde{\mathbf{w}}_j^{\text{mini}}, \tilde{\mathbf{A}}_{\text{train}}^{\text{mini}} \right) \right\|_2 \\
& = \left\| \frac{1}{n_{\text{train}}} \sum_{i=1}^{n_{\text{train}}} \mathbf{z}_{i,j} \sigma' \left( \tilde{\mathbf{a}}_{\text{train},i}^{\text{mini}} \mathbf{X} \left( \mathbf{w}_{j,0}^{\text{mini}} \right)^\top \right) \tilde{\mathbf{a}}_{\text{train},i}^{\text{mini}} \mathbf{X} \right\|_2 \\
& \quad - \left\| \frac{1}{n_{\text{train}}} \sum_{i=1}^{n_{\text{train}}} \mathbf{z}_{i,j} \sigma' \left( \tilde{\mathbf{a}}_{\text{train},i}^{\text{mini}} \mathbf{X} \left( \tilde{\mathbf{w}}_j^{\text{mini}} \right)^\top \right) \tilde{\mathbf{a}}_{\text{train},i}^{\text{mini}} \mathbf{X} \right\|_2 \\
& \leq \left\| \frac{1}{n_{\text{train}}} \sum_{i=1}^{n_{\text{train}}} \mathbf{z}_{i,j} \left( \sigma' \left( \tilde{\mathbf{a}}_{\text{train},i}^{\text{mini}} \mathbf{X} \left( \mathbf{w}_{j,0}^{\text{mini}} \right)^\top \right) - \sigma' \left( \tilde{\mathbf{a}}_{\text{train},i}^{\text{mini}} \mathbf{X} \left( \tilde{\mathbf{w}}_j^{\text{mini}} \right)^\top \right) \right) \tilde{\mathbf{a}}_{\text{train},i}^{\text{mini}} \mathbf{X} \right\|_2 \\
& \leq \frac{1}{n_{\text{train}}} \sum_{i=1}^{n_{\text{train}}} C_x^{\frac{1}{2}} \beta^{\frac{1}{2}} \max_i \left| l'\left( y_i \hat{y}_i^{\text{mini}} \right) \right| \\
& = C_7 \beta^{\frac{1}{2}} \max_i \left| l'\left( y_i \hat{y}_i^{\text{mini}} \right) \right|,
\end{aligned}
\tag{140}
$$

where $C_7$ is an absolute constant.

Therefore, there are at least $C_4 h \alpha^2 / (n_{\text{train}} \beta)$ GNN nodes, satisfying:

$$
\begin{aligned}
& \left\| \nabla_{\tilde{\mathbf{w}}_j^{\text{mini}}} \hat{L}_{\text{train}}^{\text{full}} \left( \tilde{\mathbf{w}}_j^{\text{mini}}, \tilde{\mathbf{A}}_{\text{train}}^{\text{mini}} \right) \right\|_2 \\
& \geq C_5 \max_i \left| l'\left( y_i \hat{y}_i^{\text{mini}} \right) \right| \beta^2 - C_7 \beta^{\frac{1}{2}} \max_i \left| l'\left( y_i \hat{y}_i^{\text{mini}} \right) \right| \\
& \geq C_8 \max_i \left| l'\left( y_i \hat{y}_i^{\text{mini}} \right) \right| \beta^2.
\end{aligned}
\tag{141}
$$

Therefore, we have:

$$
\begin{aligned}
& \left\| \nabla_{\tilde{\mathbf{W}}^{\text{mini}}} \hat{L}_{\text{train}}^{\text{full}} \left( \tilde{\mathbf{W}}^{\text{mini}}, \tilde{\mathbf{A}}_{\text{train}}^{\text{mini}} \right) \right\|_2 \\
& = \sum_{j=1}^h \left\| \nabla_{\tilde{\mathbf{w}}_j^{\text{mini}}} \hat{L}_{\text{train}}^{\text{full}} \left( \tilde{\mathbf{w}}_j^{\text{mini}}, \tilde{\mathbf{A}}_{\text{train}}^{\text{mini}} \right) \right\|_2 \\
& \geq \frac{C_4 h \alpha^2}{n_{\text{train}} \beta} \left( C_8 \max_i \left| l'\left( y_i \hat{y}_i^{\text{mini}} \right) \right| \beta^2 \right)^2 \\
& \geq \frac{C_9 h \alpha^2 \beta^3}{n_{\text{train}}^3} \left( \sum_{i=1}^{n_{\text{train}}} l'\left( y_i \hat{y}_i^{\text{mini}} \right) \right)^2.
\end{aligned}
\tag{142}
$$

Then, replacing $\beta$ by $d_{\max}$, we have:

$$
\left\| \nabla_{\tilde{\mathbf{W}}^{\text{full}}} \hat{L}_{\text{train}}^{\text{full}} \left( \tilde{\mathbf{W}}^{\text{full}} \right) \right\|_2 \geq \frac{C_9 h \alpha^2 d_{\max}^3}{n_{\text{train}}^3} \left( \sum_{i=1}^{n_{\text{train}}} l'\left( y_i \hat{y}_i^{\text{full}} \right) \right)^2,
\tag{143}
$$

for the full-graph training.

Proved.

### K.3 PROOF OF LEMMA D.5

**Lemma K.5** For any $\delta > 0$, with probability at least $1 - e^{-O(1)}$, if $\mathbf{W}_t^{\text{full}} \in \mathcal{B}\left( \mathbf{W}_0^{\text{full}}, \tau \right)$, it holds that:

$$
\left\| \mathbf{w}_{j,t}^{\text{full}} \right\|_2 \leq C + \tau,
$$

and

$$
\left\| \mathbf{w}_{j,0}^{\text{full}} \right\|_2 \leq C,
$$

for $j \in \{1, \ldots, h\}$, where $C = \kappa\left( \sqrt{r} + \delta \right)$ is positive constant.

**Proof of Lemma D.5:** Since $l(x)$ is $1/4$-smooth, the following holds for any $\Delta$ and $x$:

$$l(x+\Delta) \leq l(x) + l'(x)\Delta + \frac{1}{8}\Delta^2. \tag{144}$$

Then we have the following upper bound on $\hat{L}_{\text{train}}^{\text{full}}\left(\mathbf{W}_{t+1}^{\text{full}}\right) - \hat{L}_{\text{train}}^{\text{full}}\left(\mathbf{W}_t^{\text{full}}\right)$:

$$
\begin{aligned}
\hat{L}_{\text{train}}^{\text{full}}\left(\mathbf{W}_{t+1}^{\text{full}}\right) - \hat{L}_{\text{train}}^{\text{full}}\left(\mathbf{W}_t^{\text{full}}\right) &= \frac{1}{n_{\text{train}}}\sum_{i=1}^{n_{\text{train}}}\left[l\left(y_i\hat{y}_{i,t+1}^{\text{full}}\right) - l\left(y_i\hat{y}_{i,t}^{\text{full}}\right)\right]\\
&= \frac{1}{n_{\text{train}}}\sum_{i=1}^{n_{\text{train}}}\left[l'\left(y_i\hat{y}_{i,t+1}^{\text{full}}\right)\Delta_{i,t+1}^{\text{full}} + \frac{1}{8}\left(\Delta_{i,t+1}^{\text{full}}\right)^2\right],
\end{aligned}
\tag{145}
$$

where $\Delta_{i,t+1}^{\text{full}} = y_i\left(\hat{y}_{i,t+1}^{\text{full}} - \hat{y}_{i,t}^{\text{full}}\right)$.

Therefore, we are going to bound $\Delta_{i,t}^{\text{full}}$. The upper bound of $\left|\Delta_{i,t}^{\text{full}}\right|$ can be derived as:

$$
\begin{aligned}
\left|\Delta_{i,t}^{\text{full}}\right| &= \left|y_i\tilde{\mathbf{a}}_{\text{train},i}^{\text{full}}\mathbf{X}\left(\mathbf{\Sigma}_{i,t+1}^{\text{full}}\mathbf{W}_{t+1}^{\text{full}}\right)^\top\mathbf{v}^\top - y_i\tilde{\mathbf{a}}_{\text{train},i}^{\text{full}}\mathbf{X}\left(\mathbf{\Sigma}_{i,t}^{\text{full}}\mathbf{W}_t^{\text{full}}\right)^\top\mathbf{v}^\top\right|\\
&= \left|y_i\tilde{\mathbf{a}}_{\text{train},i}^{\text{full}}\mathbf{X}\left(\mathbf{\Sigma}_{i,t+1}^{\text{full}}\mathbf{W}_{t+1}^{\text{full}} - \mathbf{\Sigma}_{i,t}^{\text{full}}\mathbf{W}_t^{\text{full}}\right)^\top\mathbf{v}^\top\right|\\
&\leq C_x^{\frac{1}{2}}d_{\max}^{\frac{1}{2}}h^{\frac{1}{2}}\left\|\mathbf{\Sigma}_{i,t+1}^{\text{full}}\mathbf{W}_{t+1}^{\text{full}} - \mathbf{\Sigma}_{i,t}^{\text{full}}\mathbf{W}_t^{\text{full}}\right\|_2,
\end{aligned}
\tag{146}
$$

where the last inequality follows Lemma K.1.

Hence, we have:

$$
\begin{aligned}
\left|\Delta_{i,t}^{\text{full}}\right| &\leq C_x^{\frac{1}{2}}d_{\max}^{\frac{1}{2}}h^{\frac{1}{2}}\left\|\left(\mathbf{W}_{t+1}^{\text{full}} - \mathbf{W}_t^{\text{full}}\right)\mathbf{\Sigma}_{i,t+1}^{\text{full}}\right\|_2 + \left\|\mathbf{W}_t^{\text{full}}\left(\mathbf{\Sigma}_{i,t+1}^{\text{full}} - \mathbf{\Sigma}_{i,t}^{\text{full}}\right)\right\|_2\\
&\leq 2C_x^{\frac{1}{2}}d_{\max}^{\frac{1}{2}}h^{\frac{1}{2}}\left(\left\|\mathbf{W}_{t+1}^{\text{full}} - \mathbf{W}_t^{\text{full}}\right\|_2 + \left\|\mathbf{W}_t^{\text{full}}\right\|_2\right)\\
&\leq C_1 d_{\max}^{\frac{1}{2}}h^{\frac{1}{2}}\eta\left(\left\|\nabla_{\mathbf{W}_t^{\text{full}}}\hat{L}_{\text{train}}^{\text{full}}\left(\mathbf{W}_t^{\text{full}}\right)\right\|_2 + C + \tau\right)\\
&= C_1 d_{\max}^{\frac{1}{2}}h^{\frac{1}{2}}\eta\left(\left\|\nabla_{\mathbf{W}_t^{\text{full}}}\hat{L}_{\text{train}}^{\text{full}}\left(\mathbf{W}_t^{\text{full}}\right)\right\|_F + C + \tau\right).
\end{aligned}
\tag{147}
$$

Note that $\tau$ has an upper bound, the third term in the brackets on the right-hand side of the above inequality is dominated by the first one. Then we have:

$$\left|\Delta_{i,t}^{\text{full}}\right| \leq C_1 d_{\max}^{\frac{1}{2}}h^{\frac{1}{2}}\eta\left\|\nabla_{\mathbf{W}_t^{\text{full}}}\hat{L}_{\text{train}}^{\text{full}}\left(\mathbf{W}_t^{\text{full}}\right)\right\|_F. \tag{148}$$

Then we are going to prove the lower bound of $\Delta_{i,t}^{\text{full}}$.

Since $\Delta_{i,t}^{\text{full}} = y_i\tilde{\mathbf{a}}_{\text{train},i}^{\text{full}}\mathbf{X}\left(\mathbf{\Sigma}_{i,t+1}^{\text{full}}\mathbf{W}_{t+1}^{\text{full}} - \mathbf{\Sigma}_{i,t}^{\text{full}}\mathbf{W}_t^{\text{full}}\right)^\top\mathbf{v}^\top = y_i\left(\mathbf{z}_{i,t+1}^{\text{full}} - \mathbf{z}_{i,t}^{\text{full}}\right)\mathbf{v}^\top$, thus we mainly focus on bounding the term $\mathbf{z}_{i,t+1}^{\text{full}} - \mathbf{z}_{i,t}^{\text{full}}$.

We define the diagonal matrix $\tilde{\mathbf{\Sigma}}_{i,t}^{\text{full}}$ as:

$$\left(\tilde{\mathbf{\Sigma}}_{i,t}^{\text{full}}\right)_{jj} = \left(\mathbf{\Sigma}_{i,t+1}^{\text{full}} - \mathbf{\Sigma}_{i,t}^{\text{full}}\right)_{jj}\frac{\left(\mathbf{w}_{j,t+1}^{\text{full}}\right)^\top}{\left(\mathbf{w}_{j,t+1}^{\text{full}} - \mathbf{w}_{j,t}^{\text{full}}\right)^\top}, \tag{149}$$

for any $j \in \{1, \ldots, h\}$.

Then we have:

$$
\begin{aligned}
&\mathbf{z}_{i,t+1}^{\text{full}} - \mathbf{z}_{i,t}^{\text{full}}\\
=&\tilde{\mathbf{a}}_{\text{train},i}^{\text{full}}\mathbf{X}\left(\mathbf{W}_{t+1}^{\text{full}} - \mathbf{W}_t^{\text{full}}\right)^\top\left(\mathbf{\Sigma}_{i,t}^{\text{full}} + \tilde{\mathbf{\Sigma}}_{i,t}^{\text{full}}\right)^\top\\
=&-\eta\tilde{\mathbf{a}}_{\text{train},i}^{\text{full}}\mathbf{X}\nabla_{\mathbf{W}_t^{\text{full}}}\hat{L}_{\text{train}}^{\text{full}}\left(\mathbf{W}_t^{\text{full}}\right)\left(\mathbf{\Sigma}_{i,t}^{\text{full}} + \tilde{\mathbf{\Sigma}}_{i,t}^{\text{full}}\right)^\top.
\end{aligned}
\tag{150}
$$

Thus, the following holds:

$$
\begin{aligned}
\Delta_{i,t}^{\text{full}} =& y_i \left( \mathbf{z}_{i,t+1}^{\text{full}} - \mathbf{z}_{i,t}^{\text{full}} \right) \mathbf{v}^\top \\
=& - \eta y_i \tilde{\mathbf{a}}_{\text{train},i}^{\text{full}} \mathbf{X} \nabla_{\mathbf{W}_t^{\text{full}}} \hat{L}_{\text{train}}^{\text{full}} \left( \mathbf{W}_t^{\text{full}} \right) \left( \mathbf{v} \left( \boldsymbol{\Sigma}_{i,t}^{\text{full}} + \tilde{\boldsymbol{\Sigma}}_{i,t}^{\text{full}} \right) \right)^\top \\
=& - \eta y_i \tilde{\mathbf{a}}_{\text{train},i}^{\text{full}} \mathbf{X} \nabla_{\mathbf{W}_t^{\text{full}}} \hat{L}_{\text{train}}^{\text{full}} \left( \mathbf{W}_t^{\text{full}} \right) \left( \mathbf{v} \tilde{\boldsymbol{\Sigma}}_{i,t}^{\text{full}} \right)^\top \\
& - \eta y_i \tilde{\mathbf{a}}_{\text{train},i}^{\text{full}} \mathbf{X} \nabla_{\mathbf{W}_t^{\text{full}}} \hat{L}_{\text{train}}^{\text{full}} \left( \mathbf{W}_t^{\text{full}} \right) \left( \mathbf{v} \boldsymbol{\Sigma}_{i,t}^{\text{full}} \right)^\top \\
=& \mathbf{U}_{i,t}^{(1)} + \mathbf{U}_{i,t}^{(2)},
\end{aligned}
\tag{151}
$$

where we define:

$$
\mathbf{U}_{i,t}^{(1)} = - \eta y_i \tilde{\mathbf{a}}_{\text{train},i}^{\text{full}} \mathbf{X} \nabla_{\mathbf{W}_t^{\text{full}}} \hat{L}_{\text{train}}^{\text{full}} \left( \mathbf{W}_t^{\text{full}} \right) \left( \mathbf{v} \tilde{\boldsymbol{\Sigma}}_{i,t}^{\text{full}} \right)^\top,
\tag{152}
$$

and

$$
\mathbf{U}_{i,t}^{(2)} = - \eta y_i \tilde{\mathbf{a}}_{\text{train},i}^{\text{full}} \mathbf{X} \nabla_{\mathbf{W}_t^{\text{full}}} \hat{L}_{\text{train}}^{\text{full}} \left( \mathbf{W}_t^{\text{full}} \right) \left( \mathbf{v} \boldsymbol{\Sigma}_{i,t}^{\text{full}} \right)^\top.
\tag{153}
$$

For $\mathbf{U}_{i,t}^{(1)}$, we notice that:

$$
\begin{aligned}
\left\| \mathbf{v} \tilde{\boldsymbol{\Sigma}}_{i,t}^{\text{full}} \right\|_2 \leq& \| \mathbf{v} \|_2 \left\| \tilde{\boldsymbol{\Sigma}}_{i,t}^{\text{full}} \right\|_2 \\
\leq& h^{\frac{1}{2}} \max_j \left| \left( \boldsymbol{\Sigma}_{i,t+1}^{\text{full}} - \boldsymbol{\Sigma}_{i,t}^{\text{full}} \right)_{jj} \frac{\left( \mathbf{w}_{j,t+1}^{\text{full}} \right)^\top}{\left( \mathbf{w}_{j,t+1}^{\text{full}} - \mathbf{w}_{j,t}^{\text{full}} \right)^\top} \right| \\
\leq& h^{\frac{1}{2}} \max_j \left| \frac{\left( \mathbf{w}_{j,t+1}^{\text{full}} \right)^\top}{\left( \mathbf{w}_{j,t+1}^{\text{full}} - \mathbf{w}_{j,t}^{\text{full}} \right)^\top} \right|.
\end{aligned}
\tag{154}
$$

Using Lemma K.5 and noticing that $\tau$ has a upper bound, we have:

$$
\left| \frac{\left( \mathbf{w}_{j,t+1}^{\text{full}} \right)^\top}{\left( \mathbf{w}_{j,t+1}^{\text{full}} - \mathbf{w}_{j,t}^{\text{full}} \right)^\top} \right| \leq \frac{\left\| \mathbf{w}_{j,t+1}^{\text{full}} \right\|_2}{\left\| \mathbf{w}_{j,t+1}^{\text{full}} \right\|_2 - \left\| \mathbf{w}_{j,t}^{\text{full}} \right\|_2} \leq \frac{\left\| \mathbf{w}_{j,0}^{\text{full}} \right\|_2 + \tau}{\varepsilon \tau} \leq C_2 \tau^{-1},
\tag{155}
$$

where $\varepsilon$ represents a positive small enough constant and $C_2$ is a positive constant.

Then we have $\left\| \mathbf{v} \tilde{\boldsymbol{\Sigma}}_{i,t}^{\text{full}} \right\|_2 \leq C_2 h^{\frac{1}{2}} \tau^{-1}$, thereby we know that $\left| \mathbf{U}_{i,t}^{(1)} \right| \leq C_3 \eta d_{\max}^{\frac{1}{2}} h^{\frac{1}{2}} \tau^{-1} \left\| \nabla_{\mathbf{W}_t^{\text{full}}} \hat{L}_{\text{train}}^{\text{full}} \left( \mathbf{W}_t^{\text{full}} \right) \right\|_F$.

Moreover, we have:

$$
\begin{aligned}
& \frac{1}{n_{\text{train}}} \sum_{}^{n_{\text{train}}} l' \left( y_i \hat{y}_{i,t}^{\text{full}} \right) \mathbf{U}_{i,t}^{(2)} \\
=& - \frac{\eta}{n_{\text{train}}} \sum_{}^{n_{\text{train}}} l' \left( y_i \hat{y}_{i,t}^{\text{full}} \right) y_i \tilde{\mathbf{a}}_{\text{train},i}^{\text{full}} \mathbf{X} \nabla_{\mathbf{W}_t^{\text{full}}} \hat{L}_{\text{train}}^{\text{full}} \left( \mathbf{W}_t^{\text{full}} \right) \left( \mathbf{v} \boldsymbol{\Sigma}_{i,t}^{\text{full}} \right)^\top \\
=& - \eta \left\| \nabla_{\mathbf{W}_t^{\text{full}}} \hat{L}_{\text{train}}^{\text{full}} \left( \mathbf{W}_t^{\text{full}} \right) \right\|_F^2.
\end{aligned}
\tag{156}
$$

Therefore, putting everything together, we have:

$$
\begin{aligned}
& \hat{L}_{\text{train}}^{\text{full}} \left( \mathbf{W}_{t+1}^{\text{full}} \right) - \hat{L}_{\text{train}}^{\text{full}} \left( \mathbf{W}_t^{\text{full}} \right) \\
=& \frac{1}{n_{\text{train}}} \sum_{i=1}^{n_{\text{train}}} \left[ l' \left( y_i \hat{y}_{i,t+1}^{\text{full}} \right) \Delta_{i,t+1}^{\text{full}} + \frac{1}{8} \left( \Delta_{i,t+1}^{\text{full}} \right)^2 \right] \\
\leq& \frac{1}{n_{\text{train}}} l' \left( y_i \hat{y}_{i,t+1}^{\text{full}} \right) \left( \mathbf{U}_{i,t}^{(1)} + \mathbf{U}_{i,t}^{(2)} \right) + C_4 d_{\max} h \eta^2 \left\| \nabla_{\mathbf{W}_t^{\text{full}}} \hat{L}_{\text{train}}^{\text{full}} \left( \mathbf{W}_t^{\text{full}} \right) \right\|_F^2 \\
\leq& - \left( \eta - C_4 d_{\max} h \eta^2 \right) \left\| \nabla_{\mathbf{W}_t^{\text{full}}} \hat{L}_{\text{train}}^{\text{full}} \left( \mathbf{W}_t^{\text{full}} \right) \right\|_F^2 \\
& - \frac{C_3 \eta d_{\max}^{\frac{1}{2}} h^{\frac{1}{2}} \left\| \nabla_{\mathbf{W}_t^{\text{full}}} \hat{L}_{\text{train}}^{\text{full}} \left( \mathbf{W}_t^{\text{full}} \right) \right\|_F}{n_{\text{train}} \tau} \sum_{i=1}^{n_{\text{train}}} l' \left( y_i \hat{y}_{i,t+1}^{\text{full}} \right).
\end{aligned}
\tag{157}
$$

Since we both have the condition "with the probability at least $1 - \exp\left(-O\left(h\alpha^2 / \left(n_{\text{train}} d_{\max}\right)\right)\right)$" and "with the probability at least $1 - \exp\left(-O\left(1\right)\right)$", we can write the condition as "with the probability at least $1 - \exp\left(-O\left(1\right)\right)$".

Proved.

### K.4 PROOF OF LEMMA E.5

**Lemma K.6** For any $\delta > 0$, with probability at least $1 - e^{-O(1)}$, if $\mathbf{W}_t^{\text{mini}} \in \mathcal{B}\left(\mathbf{W}_0^{\text{mini}}, \tau\right)$, it holds that:

$$\left\|\mathbf{w}_{j,t}^{\text{mini}}\right\|_2 \leq C + \tau,$$

and

$$\left\|\mathbf{w}_{j,0}^{\text{mini}}\right\|_2 \leq C,$$

for $j \in \{1, \ldots, h\}$, where $C = \kappa\left(\sqrt{r} + \delta\right)$ is positive constant.

**Proof of Lemma E.5:** Since $l(x)$ is $1/4$-smooth, the following holds for any $\Delta$ and $x$:

$$l(x + \Delta) \leq l(x) + l'(x)\Delta + \frac{1}{8}\Delta^2. \tag{158}$$

Then we have the following upper bound on $\hat{L}_{\text{train}}^{\text{full}}\left(\mathbf{W}_{t+1}^{\text{mini}}\right) - \hat{L}_{\text{train}}^{\text{full}}\left(\mathbf{W}_t^{\text{mini}}\right)$:

$$
\begin{aligned}
&\hat{L}_{\text{train}}^{\text{full}}\left(\mathbf{W}_{t+1}^{\text{mini}}, \tilde{\mathbf{A}}_{\text{train}}^{\text{mini}}\right) - \hat{L}_{\text{train}}^{\text{full}}\left(\mathbf{W}_t^{\text{mini}}, \tilde{\mathbf{A}}_{\text{train}}^{\text{mini}}\right) \\
&= \frac{1}{n_{\text{train}}} \sum_{i=1}^{n_{\text{train}}} \left[l\left(y_i \hat{y}_{i,t+1}^{\text{mini}}\right) - l\left(y_i \hat{y}_{i,t}^{\text{mini}}\right)\right] \\
&= \frac{1}{n_{\text{train}}} \sum_{i=1}^{n_{\text{train}}} \left[l'\left(y_i \hat{y}_{i,t+1}^{\text{mini}}\right)\Delta_{i,t+1}^{\text{mini}} + \frac{1}{8}\left(\Delta_{i,t+1}^{\text{mini}}\right)^2\right],
\end{aligned} \tag{159}
$$

where $\Delta_{i,t+1}^{\text{mini}} = y_i\left(\hat{y}_{i,t+1}^{\text{mini}} - \hat{y}_{i,t}^{\text{mini}}\right)$.

Then, taking expectation conditioning $\mathbf{W}_t^{\text{mini}}$ gives:

$$
\begin{aligned}
&\mathbb{E}\left[\hat{L}_{\text{train}}^{\text{full}}\left(\mathbf{W}_{t+1}^{\text{mini}}, \tilde{\mathbf{A}}_{\text{train}}^{\text{mini}}\right) | \mathbf{W}_t^{\text{mini}}\right] - \hat{L}_{\text{train}}^{\text{full}}\left(\mathbf{W}_t^{\text{mini}}, \tilde{\mathbf{A}}_{\text{train}}^{\text{mini}}\right) \\
&= \frac{1}{n_{\text{train}}} \sum_{i=1}^{n_{\text{train}}} \left[l'\left(y_i \hat{y}_{i,t+1}^{\text{mini}}\right)\mathbb{E}\left[\Delta_{i,t+1}^{\text{mini}} | \mathbf{W}_t^{\text{mini}}\right] + \frac{1}{8}\mathbb{E}\left[\left(\Delta_{i,t+1}^{\text{mini}}\right)^2 | \mathbf{W}_t^{\text{mini}}\right]\right].
\end{aligned} \tag{160}
$$

Similar to the proof of Lemma D.5, we have:

$$
\begin{aligned}
&\frac{1}{n_{\text{train}}} \sum_{i=1}^{n_{\text{train}}} l'\left(y_i \hat{y}_{i,t+1}^{\text{mini}}\right)\mathbb{E}\left[\Delta_{i,t+1}^{\text{mini}} | \mathbf{W}_t^{\text{mini}}\right] \\
&\leq -\eta \left\|\nabla_{\mathbf{W}_t^{\text{mini}}} \hat{L}_{\text{train}}^{\text{full}}\left(\mathbf{W}_t^{\text{mini}}, \tilde{\mathbf{A}}_{\text{train}}^{\text{mini}}\right)\right\|_F^2 \\
&\quad - \frac{C_1 \eta \beta^{\frac{1}{2}} h^{\frac{1}{2}} \left\|\nabla_{\mathbf{W}_t^{\text{mini}}} \hat{L}_{\text{train}}^{\text{full}}\left(\mathbf{W}_t^{\text{mini}}, \tilde{\mathbf{A}}_{\text{train}}^{\text{mini}}\right)\right\|_F}{n_{\text{train}}\tau} \sum_{i=1}^{n_{\text{train}}} l'\left(y_i \hat{y}_{i,t+1}^{\text{mini}}\right).
\end{aligned} \tag{161}
$$

In terms of $\mathbb{E}\left[\left(\Delta_{i,t+1}^{\text{mini}}\right)^2 | \mathbf{W}_t^{\text{mini}}\right]$, similar to the proof of Lemma D.5, we have:

$$\mathbb{E}\left[\left(\Delta_{i,t+1}^{\text{mini}}\right)^2 | \mathbf{W}_t^{\text{mini}}\right] \leq C_2 \beta h \eta^2 \mathbb{E}\left[\left\|\nabla_{\mathbf{W}_t^{\text{mini}}} \hat{L}_{\text{train}}^{\text{mini}}\left(\mathbf{W}_t^{\text{mini}}, \tilde{\mathbf{A}}_{\text{train}}^{\text{mini}}\right)\right\|_F^2 | \mathbf{W}_t^{\text{mini}}\right]. \tag{162}$$

Furthermore, using Lemma E.6, we have:

$$
\mathbb{E}\left[\left\|\nabla_{\mathbf{W}_t^{\text{mini}}}\hat{L}_{\text{train}}^{\text{mini}}\left(\mathbf{W}_t^{\text{mini}},\tilde{\mathbf{A}}_{\text{train}}^{\text{mini}}\right)\right\|_F^2|\mathbf{W}_t^{\text{mini}}\right]
$$

$$
\leq \mathbb{E}\left[\left\|\nabla_{\mathbf{W}_t^{\text{mini}}}\hat{L}_{\text{train}}^{\text{mini}}\left(\mathbf{W}_t^{\text{mini}},\tilde{\mathbf{A}}_{\text{train}}^{\text{mini}}\right)-\nabla_{\mathbf{W}_t^{\text{mini}}}\hat{L}_{\text{train}}^{\text{full}}\left(\mathbf{W}_t^{\text{mini}},\tilde{\mathbf{A}}_{\text{train}}^{\text{mini}}\right)\right\|_F^2|\mathbf{W}_t^{\text{mini}}\right]
$$

$$
+\left\|\nabla_{\mathbf{W}_t^{\text{mini}}}\hat{L}_{\text{train}}^{\text{full}}\left(\mathbf{W}_t^{\text{mini}},\tilde{\mathbf{A}}_{\text{train}}^{\text{mini}}\right)\right\|_F^2 \tag{163}
$$

$$
\leq \frac{n_{\text{train}}^2}{n_{\text{train}}b}\left\|\nabla_{\mathbf{W}_t^{\text{mini}}}\hat{L}_{\text{train}}^{\text{full}}\left(\mathbf{W}_t^{\text{mini}},\tilde{\mathbf{A}}_{\text{train}}^{\text{mini}}\right)\right\|_F^2+\left\|\nabla_{\mathbf{W}_t^{\text{mini}}}\hat{L}_{\text{train}}^{\text{full}}\left(\mathbf{W}_t^{\text{mini}},\tilde{\mathbf{A}}_{\text{train}}^{\text{mini}}\right)\right\|_F^2
$$

$$
\leq \frac{2n_{\text{train}}}{b}\left\|\nabla_{\mathbf{W}_t^{\text{mini}}}\hat{L}_{\text{train}}^{\text{full}}\left(\mathbf{W}_t^{\text{mini}},\tilde{\mathbf{A}}_{\text{train}}^{\text{mini}}\right)\right\|_F^2.
$$

Hence, the following holds:

$$
\mathbb{E}\left[\left(\Delta_{i,t+1}^{\text{mini}}\right)^2|\mathbf{W}_t^{\text{mini}}\right]\leq \frac{C_3\beta hn_{\text{train}}\eta^2}{b}\left\|\nabla_{\mathbf{W}_t^{\text{mini}}}\hat{L}_{\text{train}}^{\text{full}}\left(\mathbf{W}_t^{\text{mini}},\tilde{\mathbf{A}}_{\text{train}}^{\text{mini}}\right)\right\|_F^2 \tag{164}
$$

Therefore, we have:

$$
\mathbb{E}\left[\hat{L}_{\text{train}}^{\text{full}}\left(\mathbf{W}_{t+1}^{\text{mini}},\tilde{\mathbf{A}}_{\text{train}}^{\text{mini}}\right)|\mathbf{W}_t^{\text{mini}}\right]-\hat{L}_{\text{train}}^{\text{full}}\left(\mathbf{W}_t^{\text{mini}},\tilde{\mathbf{A}}_{\text{train}}^{\text{mini}}\right)
$$

$$
\leq -\left(\eta-\frac{C_3n_{\text{train}}\beta h\eta^2}{b}\right)\left\|\nabla_{\mathbf{W}_t^{\text{mini}}}\hat{L}_{\text{train}}^{\text{full}}\left(\mathbf{W}_t^{\text{mini}},\tilde{\mathbf{A}}_{\text{train}}^{\text{mini}}\right)\right\|_F^2 \tag{165}
$$

$$
-\frac{C_1\eta\beta^{\frac{1}{2}}h^{\frac{1}{2}}\left\|\nabla_{\mathbf{W}_t^{\text{mini}}}\hat{L}_{\text{train}}^{\text{full}}\left(\mathbf{W}_t^{\text{mini}},\tilde{\mathbf{A}}_{\text{train}}^{\text{mini}}\right)\right\|_F}{n_{\text{train}}\tau}\sum_{i=1}^{n_{\text{train}}}l'\left(y_i\hat{y}_{i,t+1}^{\text{mini}}\right)
$$

Since we both have the condition "with the probability at least $1-\exp\left(-O\left(h\alpha^2/\left(n_{\text{train}}\beta\right)\right)\right)$" and "with the probability at least $1-\exp\left(-O\left(1\right)\right)$", we can write the condition as "with the probability at least $1-\exp\left(-O\left(1\right)\right)$".

Proved.

## L PROOF OF AUXILIARY LEMMAS OF CONVERGENCE THEOREMS WITH CE

### L.1 PROOF OF LEMMA K.1:

We first focus on the mini-batch training. The normalized adjacency matrix can be expressed as:

$$
\tilde{\mathbf{A}}_{\text{train}}^{\text{mini}}=\begin{bmatrix}\frac{1}{\sqrt{d_1^{\text{in}}}} & & \\ & \ddots & \\ & & \frac{1}{\sqrt{d_b^{\text{in}}}}\end{bmatrix}\begin{bmatrix}a_{11} & \cdots & a_{1n} \\ \vdots & \ddots & \vdots \\ a_{b1} & \cdots & a_{bn}\end{bmatrix}\begin{bmatrix}\frac{1}{\sqrt{d_1^{\text{out}}}} & & \\ & \ddots & \\ & & \frac{1}{\sqrt{d_n^{\text{out}}}}\end{bmatrix}
$$

$$
=\begin{bmatrix}\frac{1}{\sqrt{d_1^{\text{in}}}}\frac{1}{\sqrt{d_1^{\text{out}}}}a_{11} & \cdots & \frac{1}{\sqrt{d_1^{\text{in}}}}\frac{1}{\sqrt{d_n^{\text{out}}}}a_{1n} \\ \vdots & \ddots & \vdots \\ \frac{1}{\sqrt{d_b^{\text{in}}}}\frac{1}{\sqrt{d_1^{\text{out}}}}a_{b1} & \cdots & \frac{1}{\sqrt{d_b^{\text{in}}}}\frac{1}{\sqrt{d_n^{\text{out}}}}a_{bn}\end{bmatrix}, \tag{166}
$$

where $a_{ij}\in\{0,1\}$ for any node $i$ in the mini batch and $j\in\{1,\ldots,n\}$.

Then, the following inequality holds on the $l_2$-norm of $\tilde{\mathbf{a}}_{\text{train},i}^{\text{mini}}$:

$$
\left\|\tilde{\mathbf{a}}_{\text{train},i}^{\text{mini}}\right\|_2^2\leq \frac{1}{d_i^{\text{in}}d_1^{\text{out}}}+\cdots+\frac{1}{d_i^{\text{in}}d_n^{\text{out}}}\leq \beta, \tag{167}
$$

where the first inequality has at most $\beta$ terms because there exist at most $\beta$ terms with $a_{ij}=1$, and the last inequality follows $\frac{1}{d_i^{\text{in}}d_j^{\text{out}}}\leq 1$.

Similarly, under the full-graph training, we have:

$$
\left\|\tilde{\mathbf{a}}_{\text{train},i}^{\text{full}}\right\|_2^2\leq \frac{1}{d_i^{\text{in}}d_1^{\text{out}}}+\cdots+\frac{1}{d_i^{\text{in}}d_n^{\text{out}}}\leq d_{\max}. \tag{168}
$$

This completes the proof.

### L.2 PROOF OF LEMMA K.2:

We first focus on the mini-batch training.

For any fixed $i \in \{1, \ldots, n_{\text{train}}\}$ and $j \in \{1, \ldots, h\}$, conditioned on $\tilde{\mathbf{a}}_{\text{train},i}^{\text{mini}}\mathbf{X}$, we have:

$$\mathbb{E}\left[\sigma^2\left(\tilde{\mathbf{a}}_{\text{train},i}^{\text{mini}}\mathbf{X}\left(\mathbf{w}_j^{\text{mini}}\right)^\top\right)|\tilde{\mathbf{a}}_{\text{train},i}^{\text{mini}}\mathbf{X}\right] = \frac{1}{2}\mathbb{E}\left[\left(\tilde{\mathbf{a}}_{\text{train},i}^{\text{mini}}\mathbf{X}\left(\mathbf{w}_j^{\text{mini}}\right)^\top\right)^2|\tilde{\mathbf{a}}_{\text{train},i}^{\text{mini}}\mathbf{X}\right]$$
$$= \left\|\tilde{\mathbf{a}}_{\text{train},i}^{\text{mini}}\mathbf{X}\right\|_2^2 \kappa^2, \tag{169}$$

where the last inequality is due to $\tilde{\mathbf{a}}_{\text{train},i}^{\text{mini}}\mathbf{X}\left(\mathbf{w}_j^{\text{mini}}\right)^\top \sim \mathcal{N}\left(0, \left\|\tilde{\mathbf{a}}_{\text{train},i}^{\text{mini}}\mathbf{X}\right\|_2^2\kappa^2\mathbf{I}\right)$ conditioned on $\tilde{\mathbf{a}}_{\text{train},i}^{\text{mini}}\mathbf{X}$.

Then, since $\mathbf{z}_i^{\text{mini}} = \tilde{\mathbf{a}}_{\text{train},i}^{\text{mini}}\mathbf{X}(\boldsymbol{\Sigma}_i^{\text{mini}}\mathbf{W}^{\text{mini}})^\top$, by Bernstein inequality, for any $\xi \geq 0$, we have:

$$\mathbb{P}\left(\left|\left\|\mathbf{z}_i^{\text{mini}}\right\|_2^2 - \left\|\tilde{\mathbf{a}}_{\text{train},i}^{\text{mini}}\mathbf{X}\right\|_2^2\right| \geq \left\|\tilde{\mathbf{a}}_{\text{train},i}^{\text{mini}}\mathbf{X}\right\|_2^2\xi|\tilde{\mathbf{a}}_{\text{train},i}^{\text{mini}}\mathbf{X}\right)$$
$$\leq 2\exp\left(-Ch\min\left\{\xi^2, \xi\right\}\right). \tag{170}$$

Taking union bound over $i$, we have:

$$\mathbb{P}\left(\left|\left\|\mathbf{z}_i^{\text{mini}}\right\|_2^2 - \left\|\tilde{\mathbf{a}}_{\text{train},i}^{\text{mini}}\mathbf{X}\right\|_2^2\right| \leq \left\|\tilde{\mathbf{a}}_{\text{train},i}^{\text{mini}}\mathbf{X}\right\|_2^2\xi, i = 1, \ldots, n_{\text{train}}\right)$$
$$\leq 1 - 2n_{\text{train}}\exp\left(-Ch\min\left\{\xi^2, \xi\right\}\right), \tag{171}$$

which further implies that, if $h \geq C_1^2 \log\left(n_{\text{train}}/\delta\right)$, then with probability at least $1 - \delta$, we have:

$$\left|\left\|\mathbf{z}_i^{\text{mini}}\right\|_2^2 - \left\|\tilde{\mathbf{a}}_{\text{train},i}^{\text{mini}}\mathbf{X}\right\|_2^2\right| \leq C_1\left\|\tilde{\mathbf{a}}_{\text{train},i}^{\text{mini}}\mathbf{X}\right\|_2^2\sqrt{\frac{\log\left(n_{\text{train}}/\delta\right)}{h}}, \tag{172}$$

for any $i = 1, \ldots, n_{\text{train}}$, where $C_1$ is an absolute constant.

This inequality implies that:

$$\left\|\mathbf{z}_i^{\text{mini}}\right\|_2^2 \leq \left[1 + C_1\sqrt{\frac{\log\left(n_{\text{train}}/\delta\right)}{h}}\right]^{\frac{1}{2}}\left\|\tilde{\mathbf{a}}_{\text{train},i}^{\text{mini}}\mathbf{X}\right\|_2^2$$
$$\leq \left[1 + C_1\sqrt{\frac{\log\left(n_{\text{train}}/\delta\right)}{h}}\right]^{\frac{1}{2}}\left\|\tilde{\mathbf{a}}_{\text{train},i}^{\text{mini}}\right\|_2^2\|\mathbf{X}\|_2^2 \tag{173}$$
$$\leq C_x^{\frac{1}{2}}\beta^{\frac{1}{2}}\left(1 + C_1\sqrt{\frac{\log\left(n_{\text{train}}/\delta\right)}{h}}\right),$$

where the last inequality fowllows by the fact that $(1 + x)^{\frac{1}{2}} \leq 1 + x$ for $x > 0$, which is applicable here.

Similarly, we can also prove that:

$$\left\|\mathbf{z}_i^{\text{mini}}\right\|_2^2 \geq C_x^{\frac{1}{2}}\beta^{\frac{1}{2}}\left(1 - C_2\sqrt{\frac{\log\left(n_{\text{train}}/\delta\right)}{h}}\right), \tag{174}$$

for some absolute constant $C_2$. Hence, we have:

$$\left|\left\|\mathbf{z}_i^{\text{mini}}\right\|_2^2 - C_x^{\frac{1}{2}}\beta^{\frac{1}{2}}\right| \leq C_3\sqrt{\beta\frac{\log\left(n_{\text{train}}/\delta\right)}{h}}, \tag{175}$$

where $C_3$ is an absolute constant.

For the full-graph training, we can replace $\beta$ by $d_{\max}$ as:

$$\left|\left\|\mathbf{z}_i^{\text{mini}}\right\|_2^2 - C_x^{\frac{1}{2}}d_{\max}^{\frac{1}{2}}\right| \leq C_4\sqrt{d_{\max}\frac{\log\left(n_{\text{train}}/\delta\right)}{h}}, \tag{176}$$

where $C_4$ is an absolute constant.

This completes the proof.

L.3  PROOF OF LEMMA K.3 AND K.4

**Lemma L.1**  Assume that for $i \neq j$ such that $y_i \neq y_j$, $\left\| \tilde{\mathbf{a}}_{\text{train},i}^{\text{full}} \mathbf{X} - \tilde{\mathbf{a}}_{\text{train},j}^{\text{full}} \mathbf{X} \right\|_2 \geq \alpha$ and $\left\| \tilde{\mathbf{a}}_{\text{train},i}^{\text{mini}} \mathbf{X} - \tilde{\mathbf{a}}_{\text{train},j}^{\text{mini}} \mathbf{X} \right\|_2 \geq \alpha$. For any $\mathbf{m} = (\mathbf{m}_1, \ldots, \mathbf{m}_{n_{\text{train}}}) \in \mathbb{R}_+^{n_{\text{train}}}$, we define $h\left( \mathbf{w}_j^{\text{full}} \right) = \sum_{i=1}^{n_{\text{train}}} \mathbf{m}_i y_i \sigma'\left( \tilde{\mathbf{a}}_{\text{train},i}^{\text{full}} \mathbf{X} \left( \mathbf{w}_j^{\text{full}} \right)^\top \right) \tilde{\mathbf{a}}_{\text{train},i}^{\text{full}} \mathbf{X}$ and $h\left( \mathbf{w}_j^{\text{mini}} \right) = \sum_{i=1}^{n_{\text{train}}} \mathbf{m}_i y_i \sigma'\left( \tilde{\mathbf{a}}_{\text{train},i}^{\text{mini}} \mathbf{X} \left( \mathbf{w}_j^{\text{mini}} \right)^\top \right) \tilde{\mathbf{a}}_{\text{train},i}^{\text{mini}} \mathbf{X}$, where $\mathbf{w}_j$ is a Gaussian random vector for any $j = 1, \ldots, h$. There exist absolute constant $C, C_1, C_2, C_3 > 0$ such that:

$$\mathbb{P}\left[ \left\| h\left( \mathbf{w}_j^{\text{full}} \right) \right\|_2 \geq C \left\| \mathbf{m} \right\|_\infty \right] \geq C_1 \frac{\alpha^2}{n_{\text{train}} d_{\max}}.$$

and

$$\mathbb{P}\left[ \left\| h\left( \mathbf{w}_j^{\text{mini}} \right) \right\|_2 \geq C_2 \left\| \mathbf{m} \right\|_\infty \right] \geq C_3 \frac{\alpha^2}{n_{\text{train}} \beta}.$$

**Proof of Lemma K.3 and K.4:**  We first focus on the mini-batch training. Under the assumption, we know that for $i \neq j$ and $y_i \neq y_j$, $\left\| \tilde{\mathbf{a}}_{\text{train},i}^{\text{mini}} \mathbf{X} - \tilde{\mathbf{a}}_{\text{train},j}^{\text{mini}} \mathbf{X} \right\|_2 \geq \alpha$. For any given $j = \{1, \ldots, h\}$ and $\hat{\mathbf{m}}$ with $\|\hat{\mathbf{m}}\|_\infty = 1$, by Lemma L.1, we have:

$$\mathbb{P}\left[ \left\| \frac{1}{n_{\text{train}}} \sum_{i=1}^{n_{\text{train}}} \hat{\mathbf{m}}_i y_i \sigma'\left( \tilde{\mathbf{a}}_{\text{train},i}^{\text{mini}} \mathbf{X} \left( \mathbf{w}_j^{\text{mini}} \right)^\top \right) \tilde{\mathbf{a}}_{\text{train},i}^{\text{mini}} \mathbf{X} \right\|_2 \geq \frac{C_2}{n_{\text{train}}} \right] \geq \frac{C_3 \alpha^2}{n_{\text{train}} \beta}. \tag{177}$$

Let $\mathcal{S}_{\infty,+}^{n_{\text{train}}-1} = \left\{ \mathbf{m} \in \mathbb{R}_+^{n_{\text{train}}} : \|\mathbf{m}\|_\infty = 1 \right\}$, and $\mathcal{U} = \mathcal{U}\left[ \mathcal{S}_{\infty,+}^{n_{\text{train}}-1}, C_2 / (4 n_{\text{train}}) \right]$ be a $C_2 / (4 n_{\text{train}})$-net covering $\mathcal{S}_{\infty,+}^{n_{\text{train}}-1}$ in $l_\infty$-norm. Then we have:

$$|\mathcal{U}| \leq (4 n_{\text{train}} / C_2)^{n_{\text{train}}}. \tag{178}$$

For $j = 1, \ldots, h$, we define:

$$\mathbf{Z}_{ij} = \mathbb{1}\left[ \left\| \frac{1}{n_{\text{train}}} \sum_{i=1}^{n_{\text{train}}} \hat{\mathbf{m}}_i y_i \sigma'\left( \tilde{\mathbf{a}}_{\text{train},i}^{\text{mini}} \mathbf{X} \left( \mathbf{w}_j^{\text{mini}} \right)^\top \right) \tilde{\mathbf{a}}_{\text{train},i}^{\text{mini}} \mathbf{X} \right\|_2 \geq \frac{C_2}{n_{\text{train}}} \right]. \tag{179}$$

Let $p_\alpha = \frac{C_3 \alpha^2}{n_{\text{train}} \beta}$, by Bernstein ineuqality and union bound, with probability at least $1 - \exp\left( -C_4 h p_\alpha + n_{\text{train}} \log\left( 4 n_{\text{train}} / C_2 \right) \right) \geq 1 - \exp\left( -C_5 h \alpha^2 / (n_{\text{train}} \beta) \right)$, we have:

$$\frac{1}{h} \sum_{j=1}^h \mathbf{Z}_j \geq \frac{p_\alpha}{2}, \tag{180}$$

where $C_4$ and $C_5$ are absolute constants.

For any $\mathbf{m} \in \mathcal{S}_{\infty,+}^{n_{\text{train}}-1}$, there exists $\hat{\mathbf{m}} \in \mathcal{U}$ such that:

$$\|\mathbf{m} - \hat{\mathbf{m}}\|_\infty \leq C_2 / (4 n_{\text{train}}). \tag{181}$$

Therefore, we have:

$$\begin{aligned}
&\left| \left\| \frac{1}{n_{\text{train}}} \sum_{i=1}^{n_{\text{train}}} \mathbf{m}_i y_i \sigma'\left( \tilde{\mathbf{a}}_{\text{train},i}^{\text{mini}} \mathbf{X} \left( \mathbf{w}_j^{\text{mini}} \right)^\top \right) \tilde{\mathbf{a}}_{\text{train},i}^{\text{mini}} \mathbf{X} \right\|_2 \right. \\
&\left. - \left\| \frac{1}{n_{\text{train}}} \sum_{i=1}^{n_{\text{train}}} \hat{\mathbf{m}}_i y_i \sigma'\left( \tilde{\mathbf{a}}_{\text{train},i}^{\text{mini}} \mathbf{X} \left( \mathbf{w}_j^{\text{mini}} \right)^\top \right) \tilde{\mathbf{a}}_{\text{train},i}^{\text{mini}} \mathbf{X} \right\|_2 \right| \leq C_6 \beta^2.
\end{aligned} \tag{182}$$

where $C_6$ is an absolute constant.

It is clear that with probability $1 - \exp\left( -C_5 h \alpha^2 / (n_{\text{train}} \beta) \right)$, for any $\mathbf{m} \in \mathcal{S}_{\infty,+}^{n_{\text{train}}-1}$, there exist at least $C_3 h \alpha^2 / (n_{\text{train}} \beta)$ GNN nodes that satisfy:

$$\left\| \frac{1}{n_{\text{train}}} \sum_{i=1}^{n_{\text{train}}} \hat{\mathbf{m}}_i y_i \sigma'\left( \tilde{\mathbf{a}}_{\text{train},i}^{\text{mini}} \mathbf{X} \left( \mathbf{w}_j^{\text{mini}} \right)^\top \right) \tilde{\mathbf{a}}_{\text{train},i}^{\text{mini}} \mathbf{X} \right\|_2 \geq C_6 \beta^2 = C_6 \beta^2 \|\mathbf{m}\|_\infty. \tag{183}$$

Similarly, for the full-graph training, we replace $\beta$ by $d_{\max}$. It is clear that with probability $1 - \exp\left(-C_7 h\alpha^2 / \left(n_{\text{train}} d_{\max}\right)\right)$, for any $\mathbf{m} \in \mathcal{S}_{\infty,+}^{n_{\text{train}}-1}$, there exist at least $C_8 h\alpha^2 / \left(n_{\text{train}} d_{\max}\right)$ GNN nodes that satisfy:

$$\left\| \frac{1}{n_{\text{train}}} \sum_{i=1}^{n_{\text{train}}} \hat{\mathbf{m}}_i y_i \sigma'\left( \tilde{\mathbf{a}}_{\text{train},i}^{\text{full}} \mathbf{X} \left(\mathbf{w}_j^{\text{full}}\right)^{\top} \right) \tilde{\mathbf{a}}_{\text{train},i}^{\text{full}} \mathbf{X} \right\|_2 \geq C_9 d_{\max}^2 = C_9 d_{\max}^2 \left\| \mathbf{m} \right\|_{\infty}. \tag{184}$$

where $C_7$, $C_8$ and $C_9$ are absolute constants.

This completes the proof.

### L.4   PROOF OF LEMMA K.5 AND K.6:

We first focus on the mini-batch training. Under the assumption, we know that each row of $\mathbf{W}_0^{\text{mini}}$ follows $\mathcal{N}\left(0, \kappa^2 \mathbf{I}\right)$. We define:

$$\mathbf{W}_0^{\text{mini}} = \kappa \mathbf{Z}, \tag{185}$$

where $\mathbf{Z} \in \mathbb{R}^{h \times r}$ with $\mathbf{Z}_j \in \mathbb{R}^{1 \times r} \sim \mathcal{N}(0, \mathbf{I})$.

Using Vershynin's result, we have:

$$\mathbb{P}\left( \left\| \mathbf{Z} \right\|_2 \leq \sqrt{r} + \sqrt{h} + \delta \right) \geq 1 - e^{-\frac{\delta^2}{2}}, \tag{186}$$

and

$$\mathbb{P}\left( \left\| \mathbf{Z}_j \right\|_2 \leq \sqrt{r} + \delta \right) \geq 1 - e^{-\frac{\delta^2}{2}}. \tag{187}$$

Therefore, with probability at least $1 - e^{-\frac{\delta^2}{2}}$, we have:

$$\left\| \mathbf{w}_{j,0}^{\text{mini}} \right\|_2 \leq \kappa \left( \sqrt{r} + \delta \right) \tag{188}$$

Since $\mathbf{W}_t^{\text{mini}} \in \mathcal{B}\left( \mathbf{W}_0^{\text{mini}}, \tau \right)$, we have:

$$\left\| \mathbf{w}_{j,t}^{\text{mini}} \right\|_2 \leq \kappa \left( \sqrt{r} + \delta \right) + \tau \tag{189}$$

Similarly, under the full-graph training, we have:

$$\left\| \mathbf{w}_{j,0}^{\text{full}} \right\|_2 \leq \kappa \left( \sqrt{r} + \delta \right) \tag{190}$$

and

$$\left\| \mathbf{w}_{j,t}^{\text{full}} \right\|_2 \leq \kappa \left( \sqrt{r} + \delta \right) + \tau \tag{191}$$

### L.5   PROOF OF LEMMA L.1:

We first focus on the mini-batch training. Without loss of generality, we assume that $\mathbf{m}_1 = \left\| \mathbf{m} \right\|_{\infty}$. Let $\tilde{\mathbf{z}}_1 = \tilde{\mathbf{a}}_{\text{train},1}^{\text{mini}} \mathbf{X} / \left\| \tilde{\mathbf{a}}_{\text{train},1}^{\text{mini}} \mathbf{X} \right\|_2$, we can construct an orthonormal matrix $\mathbf{Q} = [\tilde{\mathbf{z}}_1, \mathbf{Q}'] \in \mathbb{R}^{r \times r}$.

Let $\mathbf{u} = \mathbf{Q}^{\top} \mathbf{w}_j^{\text{mini}} \sim \mathcal{N}\left( 0, \kappa^2 \mathbf{I} \right)$ be a Gaussian random vector with $0 < \kappa \leq 1$. Then we have:

$$\mathbf{w}_j^{\text{mini}} = \left\| \tilde{\mathbf{a}}_{\text{train},1}^{\text{mini}} \mathbf{X} \right\|_2 \mathbf{Q}\mathbf{u} = \mathbf{u}_1 \tilde{\mathbf{a}}_{\text{train},1}^{\text{mini}} \mathbf{X} + \left\| \tilde{\mathbf{a}}_{\text{train},1}^{\text{mini}} \mathbf{X} \right\|_2 \mathbf{Q}' \mathbf{u}', \tag{192}$$

where $\mathbf{u}' = \left( \mathbf{u}_2, \ldots, \mathbf{u}_r \right)^{\top}$.

We define the following two events based on a parameter $\gamma \in (0, 1]$:

$$\mathcal{E}_1(\gamma) = \left\{ C_x \beta \left| \mathbf{u}_1 \right| \leq \gamma \right\}, \tag{193}$$

and

$$\begin{aligned} \mathcal{E}_2(\gamma) = \{ &\left| < \left\| \tilde{\mathbf{a}}_{\text{train},1}^{\text{mini}} \mathbf{X} \right\|_2 \mathbf{Q}' \mathbf{u}', \tilde{\mathbf{a}}_{\text{train},i}^{\text{mini}} \mathbf{X} > \right| \leq \gamma \\ &\text{for all } \tilde{\mathbf{a}}_{\text{train},i}^{\text{mini}} \mathbf{X} \text{ such that } \left\| \tilde{\mathbf{a}}_{\text{train},i}^{\text{mini}} \mathbf{X} - \tilde{\mathbf{a}}_{\text{train},1}^{\text{mini}} \mathbf{X} \right\|_2 \geq \alpha \}. \end{aligned} \tag{194}$$

Let $\mathcal{E}(\gamma) = \mathcal{E}_1(\gamma) \cap \mathcal{E}_2(\gamma)$. We first give lower bound for $\mathbb{P}(\mathcal{E}) = \mathbb{P}(\mathcal{E}_1)\mathbb{P}(\mathcal{E}_2)$.

Since $\mathbf{u}_1 \sim \mathcal{N}(0, \kappa^2)$, we have:

$$
\begin{aligned}
\mathbb{P}(\mathcal{E}_1) \geq& \mathbb{P}\left(\left\{C_x\beta\,|\mathbf{u}_1| \leq \kappa^2\gamma\right\}\right) \\
=& \frac{1}{\sqrt{2\pi}} \int_{-\frac{\gamma\kappa^2}{C_x\beta}}^{\frac{\gamma\kappa^2}{C_x\beta}} \exp\left(-\frac{1}{2}x^2\right) dx \\
\geq& \sqrt{\frac{2}{\pi e}}\frac{\gamma\kappa^2}{C_x\beta}.
\end{aligned}
\tag{195}
$$

Moreover, by definition, for any $i = 1, \ldots, n_{\text{train}}$, we have:

$$
\begin{aligned}
&< \left\|\tilde{\mathbf{a}}_{\text{train},1}^{\text{mini}}\mathbf{X}\right\|_2 \mathbf{Q}'\mathbf{u}', \tilde{\mathbf{a}}_{\text{train},i}^{\text{mini}}\mathbf{X} > \\
&\sim \mathcal{N}\left[0, \left\|\tilde{\mathbf{a}}_{\text{train},1}^{\text{mini}}\mathbf{X}\right\|_2^2 \left\|\tilde{\mathbf{a}}_{\text{train},i}^{\text{mini}}\mathbf{X}\right\|_2^2 - \left(\left(\tilde{\mathbf{a}}_{\text{train},1}^{\text{mini}}\mathbf{X}\right)^\top \tilde{\mathbf{a}}_{\text{train},i}^{\text{mini}}\mathbf{X}\right)^2\right].
\end{aligned}
\tag{196}
$$

Let $\mathcal{I} = \left\{i : \left\|\tilde{\mathbf{a}}_{\text{train},i}^{\text{mini}}\mathbf{X} - \tilde{\mathbf{a}}_{\text{train},1}^{\text{mini}}\mathbf{X}\right\|_2 \geq \alpha\right\}$. For any $i \in \mathcal{I}$, we have:

$$
\begin{aligned}
&\left\|\tilde{\mathbf{a}}_{\text{train},i}^{\text{mini}}\mathbf{X} - \tilde{\mathbf{a}}_{\text{train},1}^{\text{mini}}\mathbf{X}\right\|_2 \\
=& \left\|\tilde{\mathbf{a}}_{\text{train},i}^{\text{mini}}\mathbf{X}\right\|_2 + \left\|\tilde{\mathbf{a}}_{\text{train},1}^{\text{mini}}\mathbf{X}\right\|_2 - 2 < \tilde{\mathbf{a}}_{\text{train},i}^{\text{mini}}\mathbf{X}, \tilde{\mathbf{a}}_{\text{train},1}^{\text{mini}}\mathbf{X} > .
\end{aligned}
\tag{197}
$$

Then we have:

$$
-C_x\beta + \frac{\alpha^2}{2} \leq < \tilde{\mathbf{a}}_{\text{train},i}^{\text{mini}}\mathbf{X}, \tilde{\mathbf{a}}_{\text{train},1}^{\text{mini}}\mathbf{X} > \leq C_x\beta - \frac{\alpha^2}{2},
\tag{198}
$$

and if $\alpha^2 \leq 2C_x\beta$, then:

$$
\begin{aligned}
&C_x^2\beta^2 - \left(\left(\tilde{\mathbf{a}}_{\text{train},1}^{\text{mini}}\mathbf{X}\right)^\top \tilde{\mathbf{a}}_{\text{train},i}^{\text{mini}}\mathbf{X}\right)^2 \\
\geq& C_x^2\beta^2 - \left(C_x^2\beta^2 - \frac{\alpha^2}{2}\right)^2 \\
\geq& C_x^2\beta^2 \geq \frac{\alpha^2}{4}
\end{aligned}
\tag{199}
$$

Therefore, for any $i \in \mathcal{I}$, we have:

$$
\begin{aligned}
&\mathbb{P}\left[\left|< \left\|\tilde{\mathbf{a}}_{\text{train},1}^{\text{mini}}\mathbf{X}\right\|_2 \mathbf{Q}'\mathbf{u}', \tilde{\mathbf{a}}_{\text{train},i}^{\text{mini}}\mathbf{X} >\right| \leq \gamma\right] \\
=& \frac{1}{\sqrt{2\pi}} \int_{-\left[\|\tilde{\mathbf{a}}_{\text{train},1}^{\text{mini}}\mathbf{X}\|_2^2\|\tilde{\mathbf{a}}_{\text{train},i}^{\text{mini}}\mathbf{X}\|_2^2 - \left(\left(\tilde{\mathbf{a}}_{\text{train},1}^{\text{mini}}\mathbf{X}\right)^\top \tilde{\mathbf{a}}_{\text{train},i}^{\text{mini}}\mathbf{X}\right)^2\right]^{\frac{1}{2}}\gamma}^{\left[\|\tilde{\mathbf{a}}_{\text{train},1}^{\text{mini}}\mathbf{X}\|_2^2\|\tilde{\mathbf{a}}_{\text{train},i}^{\text{mini}}\mathbf{X}\|_2^2 - \left(\left(\tilde{\mathbf{a}}_{\text{train},1}^{\text{mini}}\mathbf{X}\right)^\top \tilde{\mathbf{a}}_{\text{train},i}^{\text{mini}}\mathbf{X}\right)^2\right]^{\frac{1}{2}}\gamma} \exp\left(-\frac{1}{2}x^2\right) dx \\
\leq& \frac{1}{\sqrt{2\pi}} \int_{-\left[C_x^2\beta^2 - \left(\left(\tilde{\mathbf{a}}_{\text{train},1}^{\text{mini}}\mathbf{X}\right)^\top \tilde{\mathbf{a}}_{\text{train},i}^{\text{mini}}\mathbf{X}\right)^2\right]^{\frac{1}{2}}\gamma}^{\left[C_x^2\beta^2 - \left(\left(\tilde{\mathbf{a}}_{\text{train},1}^{\text{mini}}\mathbf{X}\right)^\top \tilde{\mathbf{a}}_{\text{train},i}^{\text{mini}}\mathbf{X}\right)^2\right]^{\frac{1}{2}}\gamma} \exp\left(-\frac{1}{2}x^2\right) dx \\
\leq& \sqrt{\frac{2}{\pi}} \frac{\gamma}{\left[C_x^2\beta^2 - \left(\left(\tilde{\mathbf{a}}_{\text{train},1}^{\text{mini}}\mathbf{X}\right)^\top \tilde{\mathbf{a}}_{\text{train},i}^{\text{mini}}\mathbf{X}\right)^2\right]^{\frac{1}{2}}} \\
\leq& \sqrt{\frac{2}{\pi}} \frac{\gamma}{\alpha^2/2}
\end{aligned}
\tag{200}
$$

Taking union bound over $\mathcal{I}$, we have:

$$
\mathbb{P}(\mathcal{E}_2) \geq 1 - \frac{2\sqrt{2}}{\sqrt{\pi}}n_{\text{train}}\gamma\alpha^{-2}.
\tag{201}
$$

Therefore, we have:

$$\mathbb{P}\left(\mathcal{E}\right) \geq \sqrt{\frac{2}{\pi e}} \frac{\gamma \kappa^2}{C_x \beta} \left(1 - \frac{2\sqrt{2}}{\sqrt{\pi}} n_{\text{train}} \gamma \alpha^{-2}\right). \tag{202}$$

Setting $\gamma = \frac{\sqrt{\pi}\alpha^2}{4\sqrt{2}n_{\text{train}}}$, we obtain:

$$\begin{aligned}
\mathbb{P}\left(\mathcal{E}\right) &\geq \sqrt{\frac{2}{\pi e}} \frac{\kappa^2}{C_x \beta} \frac{\sqrt{\pi}\alpha^2}{4\sqrt{2}n_{\text{train}}} \left(1 - \frac{2\sqrt{2}}{\sqrt{\pi}} n_{\text{train}} \frac{\sqrt{\pi}\alpha^2}{4\sqrt{2}n_{\text{train}}} \alpha^{-2}\right) \\
&= \frac{\kappa^2 \alpha^2}{8\sqrt{e}C_x n_{\text{train}} \beta}.
\end{aligned} \tag{203}$$

Let $\mathcal{I}' = [n_{\text{train}}] \setminus (\mathcal{I} \cup \{1\})$. Conditioning on event $\mathcal{E}$, we have:

$$\begin{aligned}
&h\left(\mathbf{w}_j^{\text{mini}}\right) \\
&= \sum_{i=1}^{n_{\text{train}}} \mathbf{m}_i y_i \sigma'\left(\tilde{\mathbf{a}}_{\text{train},i}^{\text{mini}} \mathbf{X} \left(\mathbf{w}_j^{\text{mini}}\right)^\top\right) \tilde{\mathbf{a}}_{\text{train},i}^{\text{mini}} \mathbf{X} \\
&= \mathbf{m}_1 y_1 \sigma'\left(\mathbf{u}_1\right) \tilde{\mathbf{a}}_{\text{train},1}^{\text{mini}} \mathbf{X} \\
&\quad + \sum_{i \in \mathcal{I}} \mathbf{m}_i y_i \sigma'\left(\mathbf{u}_1 < \tilde{\mathbf{a}}_{\text{train},1}^{\text{mini}} \mathbf{X}, \tilde{\mathbf{a}}_{\text{train},i}^{\text{mini}} \mathbf{X} >, < \left\|\tilde{\mathbf{a}}_{\text{train},1}^{\text{mini}} \mathbf{X}\right\|_2 \mathbf{Q}'\mathbf{u}', \tilde{\mathbf{a}}_{\text{train},i}^{\text{mini}} \mathbf{X} >\right) \tilde{\mathbf{a}}_{\text{train},i}^{\text{mini}} \mathbf{X} \\
&\quad + \sum_{i \in \mathcal{I}'} \mathbf{m}_i y_i \sigma'\left(\mathbf{u}_1 < \tilde{\mathbf{a}}_{\text{train},1}^{\text{mini}} \mathbf{X}, \tilde{\mathbf{a}}_{\text{train},i}^{\text{mini}} \mathbf{X} >, < \left\|\tilde{\mathbf{a}}_{\text{train},1}^{\text{mini}} \mathbf{X}\right\|_2 \mathbf{Q}'\mathbf{u}', \tilde{\mathbf{a}}_{\text{train},i}^{\text{mini}} \mathbf{X} >\right) \tilde{\mathbf{a}}_{\text{train},i}^{\text{mini}} \mathbf{X} \\
&= \mathbf{m}_1 y_1 \sigma'\left(\mathbf{u}_1\right) \tilde{\mathbf{a}}_{\text{train},1}^{\text{mini}} \mathbf{X} + \sum_{i \in \mathcal{I}} \mathbf{m}_i y_i \sigma'\left(< \left\|\tilde{\mathbf{a}}_{\text{train},1}^{\text{mini}} \mathbf{X}\right\|_2 \mathbf{Q}'\mathbf{u}', \tilde{\mathbf{a}}_{\text{train},i}^{\text{mini}} \mathbf{X} >\right) \tilde{\mathbf{a}}_{\text{train},i}^{\text{mini}} \mathbf{X} \\
&\quad + \sum_{i \in \mathcal{I}'} \mathbf{m}_i y_i \sigma'\left(\mathbf{u}_1 < \tilde{\mathbf{a}}_{\text{train},1}^{\text{mini}} \mathbf{X}, \tilde{\mathbf{a}}_{\text{train},i}^{\text{mini}} \mathbf{X} >, < \left\|\tilde{\mathbf{a}}_{\text{train},1}^{\text{mini}} \mathbf{X}\right\|_2 \mathbf{Q}'\mathbf{u}', \tilde{\mathbf{a}}_{\text{train},i}^{\text{mini}} \mathbf{X} >\right) \tilde{\mathbf{a}}_{\text{train},i}^{\text{mini}} \mathbf{X},
\end{aligned} \tag{204}$$

where the last equality follows from the fact that conditioning on event $\mathcal{E}$, for all $i \in \mathcal{I}$, it hold that:

$$\left|< \left\|\tilde{\mathbf{a}}_{\text{train},1}^{\text{mini}} \mathbf{X}\right\|_2 \mathbf{Q}'\mathbf{u}', \tilde{\mathbf{a}}_{\text{train},i}^{\text{mini}} \mathbf{X} >\right| \geq |\mathbf{u}_1 C_x \beta| \geq \left|\mathbf{u}_1 < \tilde{\mathbf{a}}_{\text{train},1}^{\text{mini}} \mathbf{X}, \tilde{\mathbf{a}}_{\text{train},i}^{\text{mini}} \mathbf{X} >\right|. \tag{205}$$

We then consider two cases: $\mathbf{u}_1 > 0$ and $\mathbf{u}_1 < 0$, which occur equally likely conditioning on $\mathcal{E}$.

Therefore, we have:

$$\mathbb{P}\left[\left\|h\left(\mathbf{w}_j^{\text{mini}}\right)\right\|_2 \geq \inf_{\mathbf{u}_1^{(1)}>0, \mathbf{u}_1^{(2)}<0} \max\left\{\left\|h\left(\mathbf{w}_j^{\text{mini},(1)}\right)\right\|_2, \left\|h\left(\mathbf{w}_j^{\text{mini},(2)}\right)\right\|_2\right\} \Big| \mathcal{E}\right] \geq \frac{1}{2}, \tag{206}$$

where we define $\mathbf{w}_j^{\text{mini},(1)} = \mathbf{u}_1^{(1)} \tilde{\mathbf{a}}_{\text{train},1}^{\text{mini}} \mathbf{X} + \left\|\tilde{\mathbf{a}}_{\text{train},1}^{\text{mini}} \mathbf{X}\right\|_2 \mathbf{Q}'\mathbf{u}'$ and $\mathbf{w}_j^{\text{mini},(2)} = \mathbf{u}_1^{(2)} \tilde{\mathbf{a}}_{\text{train},1}^{\text{mini}} \mathbf{X} + \left\|\tilde{\mathbf{a}}_{\text{train},1}^{\text{mini}} \mathbf{X}\right\|_2 \mathbf{Q}'\mathbf{u}'$.

By the inequality $\max\left\{\|\mathbf{a}\|_2, \|\mathbf{b}\|_2\right\} \geq \|\mathbf{a} - \mathbf{b}\|_2 / 2$, we have:

$$\mathbb{P}\left[\left\|h\left(\mathbf{w}_j^{\text{mini}}\right)\right\|_2 \geq \inf_{\mathbf{u}_1^{(1)}>0, \mathbf{u}_1^{(2)}<0} \left\|h\left(\mathbf{w}_j^{\text{mini},(1)}\right) - h\left(\mathbf{w}_j^{\text{mini},(2)}\right)\right\|_2 / 2 \Big| \mathcal{E}\right] \geq \frac{1}{2}, \tag{207}$$

By Eq 204, we have:

$$h\left(\mathbf{w}_j^{\text{mini},(1)}\right) - h\left(\mathbf{w}_j^{\text{mini},(2)}\right) = \mathbf{m}_1 y_1 \tilde{\mathbf{a}}_{\text{train},1}^{\text{mini}} \mathbf{X} + \sum_{i \in \mathcal{I}'} \mathbf{m}_i' y_i \tilde{\mathbf{a}}_{\text{train},i}^{\text{mini}} \mathbf{X}, \tag{208}$$

where we define:

$$\begin{aligned}
\mathbf{m}_i' = \mathbf{m}_i [ &\sigma'\left(\mathbf{u}_1^{(1)} < \tilde{\mathbf{a}}_{\text{train},1}^{\text{mini}} \mathbf{X}, \tilde{\mathbf{a}}_{\text{train},i}^{\text{mini}} \mathbf{X} >, < \left\|\tilde{\mathbf{a}}_{\text{train},1}^{\text{mini}} \mathbf{X}\right\|_2 \mathbf{Q}'\mathbf{u}', \tilde{\mathbf{a}}_{\text{train},i}^{\text{mini}} \mathbf{X} >\right) \\
&- \sigma'\left(\mathbf{u}_1^{(2)} < \tilde{\mathbf{a}}_{\text{train},1}^{\text{mini}} \mathbf{X}, \tilde{\mathbf{a}}_{\text{train},i}^{\text{mini}} \mathbf{X} >, < \left\|\tilde{\mathbf{a}}_{\text{train},1}^{\text{mini}} \mathbf{X}\right\|_2 \mathbf{Q}'\mathbf{u}', \tilde{\mathbf{a}}_{\text{train},i}^{\text{mini}} \mathbf{X} >\right) ].
\end{aligned} \tag{209}$$

Note that for all $i \in \mathcal{I}'$, we have $y_i = y_1$ and $< \tilde{\mathbf{a}}_{\text{train},1}^{\text{mini}} \mathbf{X}, \tilde{\mathbf{a}}_{\text{train},i}^{\text{mini}} \mathbf{X} > \geq C_x \beta - \alpha^2 / 2 \geq 0$. Therefore, since $\mathbf{u}_1^{(1)} > 0 > \mathbf{u}_1^{(2)}$, we have:

$$
\begin{aligned}
&\sigma' \left( \mathbf{u}_1^{(1)} < \tilde{\mathbf{a}}_{\text{train},1}^{\text{mini}} \mathbf{X}, \tilde{\mathbf{a}}_{\text{train},i}^{\text{mini}} \mathbf{X} >, < \left\| \tilde{\mathbf{a}}_{\text{train},1}^{\text{mini}} \mathbf{X} \right\|_2 \mathbf{Q}' \mathbf{u}', \tilde{\mathbf{a}}_{\text{train},i}^{\text{mini}} \mathbf{X} > \right) \\
&- \sigma' \left( \mathbf{u}_1^{(2)} < \tilde{\mathbf{a}}_{\text{train},1}^{\text{mini}} \mathbf{X}, \tilde{\mathbf{a}}_{\text{train},i}^{\text{mini}} \mathbf{X} >, < \left\| \tilde{\mathbf{a}}_{\text{train},1}^{\text{mini}} \mathbf{X} \right\|_2 \mathbf{Q}' \mathbf{u}', \tilde{\mathbf{a}}_{\text{train},i}^{\text{mini}} \mathbf{X} > \right) \geq 0.
\end{aligned}
\tag{210}
$$

Therefore, $\mathbf{m}_i' \geq 0$ for all $i \in \mathcal{I}'$ and

$$
h \left( \mathbf{w}_j^{\text{mini},(1)} \right) - h \left( \mathbf{w}_j^{\text{mini},(2)} \right) = y_1 \left( \mathbf{m}_1 \tilde{\mathbf{a}}_{\text{train},1}^{\text{mini}} \mathbf{X} + \sum_{i \in \mathcal{I}'} \mathbf{m}_i' \tilde{\mathbf{a}}_{\text{train},i}^{\text{mini}} \mathbf{X} \right).
\tag{211}
$$

Then we have:

$$
\begin{aligned}
& \left\| h \left( \mathbf{w}_j^{\text{mini},(1)} \right) - h \left( \mathbf{w}_j^{\text{mini},(2)} \right) \right\|_2 \\
\geq & \left\| y_1 \left( \mathbf{m}_1 \tilde{\mathbf{a}}_{\text{train},1}^{\text{mini}} \mathbf{X} + \sum_{i \in \mathcal{I}'} \mathbf{m}_i' \tilde{\mathbf{a}}_{\text{train},i}^{\text{mini}} \mathbf{X} \right) \right\|_2 \\
\geq & < \mathbf{m}_1 \tilde{\mathbf{a}}_{\text{train},1}^{\text{mini}} \mathbf{X} + \sum_{i \in \mathcal{I}'} \mathbf{m}_i' \tilde{\mathbf{a}}_{\text{train},i}^{\text{mini}} \mathbf{X}, \tilde{\mathbf{a}}_{\text{train},1}^{\text{mini}} \mathbf{X} > / \left\| \tilde{\mathbf{a}}_{\text{train},1}^{\text{mini}} \mathbf{X} \right\|_2 \\
\geq & \mathbf{m}_1.
\end{aligned}
\tag{212}
$$

Since the inequality above holds for any $\mathbf{u}_1^{(1)} > 0$ and $\mathbf{u}_1^{(2)} < 0$, taking infimum, we have:

$$
\inf_{\mathbf{u}_1^{(1)} > 0, \mathbf{u}_1^{(2)} < 0} \left\| h \left( \mathbf{w}_j^{\text{mini},(1)} \right) - h \left( \mathbf{w}_j^{\text{mini},(2)} \right) \right\|_2 \geq \mathbf{m}_1.
\tag{213}
$$

Therefore, we have:

$$
\mathbb{P} \left[ \left\| h \left( \mathbf{w}_j^{\text{mini}} \right) \right\|_2 \geq \mathbf{m}_1 / 2 | \mathcal{E} \right] \geq \frac{1}{2}.
\tag{214}
$$

Since $\mathbf{m}_1 = \|\mathbf{m}\|_\infty$ and $\mathbb{P}(\mathcal{E}) \geq \frac{\kappa^2 \alpha^2}{8\sqrt{e} C_x n_{\text{train}} \beta}$, we have:

$$
\mathbb{P} \left[ \left\| h \left( \mathbf{w}_j^{\text{mini}} \right) \right\|_2 \geq C \|\mathbf{m}\|_\infty \right] \geq \frac{C_1 \alpha^2}{n_{\text{train}} \beta},
\tag{215}
$$

where $C$ and $C_1$ are absolute constants.

Similarly, for the full-graph training, we can replace $\beta$ by $d_{\max}$ as:

$$
\mathbb{P} \left[ \left\| h \left( \mathbf{w}_j^{\text{full}} \right) \right\|_2 \geq C_2 \|\mathbf{m}\|_\infty \right] \geq \frac{C_3 \alpha^2}{n_{\text{train}} d_{\max}},
\tag{216}
$$

where $C_2$ and $C_3$ are absolute constants.

Proved.

# M   PROOFS OF THE MAIN THEOREM AND LEMMA OF THEOREM 5

## M.1   PROOF OF THEOREM G.5

**Lemma M.1.** (Lemma 4 in (Ma et al., 2021)) For any two distributions $\mathcal{P}$ and $\mathcal{Q}$ defined on the hypothesis space, and any function $f(\cdot) \in \mathbb{R}$ with $\mathbf{dom} f$ in this hypothesis space, we have:

$$
\mathbb{E}_{x \sim \mathcal{Q}} \leq D_{KL} \left( \mathcal{Q} \| \mathcal{P} \right) + \mathbb{E}_{x \sim \mathcal{P}} e^{f(x)}.
$$

**Lemma M.2.** (Lemma 2 in (Ma et al., 2021)) Suppose $x_1, x_2, \ldots, x_n$ are independent random variables with $a_i \leq x_i \leq b_i, \forall i = 1, 2, \ldots, n$. Let $\overline{x} = \frac{1}{n} \sum_{i=1}^n x_i$. Then, for any $C > 0$,

$$
\mathbb{P} \left( |\overline{x} - \mathbb{E}(\overline{x})| > C \right) \leq 2 e^{- \frac{n^2 C^2}{\sum_{i=1}^n (b_i - a_i)^2}}.
$$

**Lemma M.3.** (Lemma 3 in (Ma et al., 2021)) If $x$ is a centered random variable, i.e., $\mathbb{E}(x) = 0$, and if $\exists C_1 > 0$, for any $C_2 > 0$,

$$\mathbb{P}(|x| > C_2) \leq 2e^{-C_1 C_2^2}.$$

Then, for any $C_u > 0$,

$$\mathbb{E}\left(e^{C_u x}\right) \leq e^{\frac{C_u^2}{2C_1}}.$$

**Proof of Theorem G.5:** We are going to prove the result by upper-bounding the quantity $C_u(L_{\text{test}}\left(\mathbf{W}^{\text{mini}}, \tilde{\mathbf{A}}_{\text{test}}^{\text{full}}; \mathcal{Q}\right) - \hat{L}_{\text{train}}^{\text{full}}\left(\mathbf{W}^{\text{mini}}, \tilde{\mathbf{A}}_{\text{train}}^{\text{mini}}; \mathcal{Q}\right)$. First, we have:

$$C_u(L_{\text{test}}\left(\mathbf{W}^{\text{mini}}, \tilde{\mathbf{A}}_{\text{test}}^{\text{full}}; \mathcal{Q}\right) - \hat{L}_{\text{train}}^{\text{full}}\left(\mathbf{W}^{\text{mini}}, \tilde{\mathbf{A}}_{\text{train}}^{\text{mini}}; \mathcal{Q}\right)$$

$$\leq \mathbb{E}_{\mathbf{W}^{\text{mini}} \sim \mathcal{Q}}\left[C_u\left(L_{\text{test}}\left(\mathbf{W}^{\text{mini}}, \tilde{\mathbf{A}}_{\text{test}}^{\text{full}}\right) - \hat{L}_{\text{train}}^{\text{full}}\left(\mathbf{W}^{\text{mini}}, \tilde{\mathbf{A}}_{\text{train}}^{\text{mini}}\right)\right)\right] \tag{217}$$

$$\leq D_{KL}(\mathcal{Q}\|\mathcal{P}) + \ln \mathbb{E}_{\mathbf{W}^{\text{mini}} \sim \mathcal{P}}\left[e^{\left(L_{\text{test}}\left(\mathbf{W}^{\text{mini}}, \tilde{\mathbf{A}}_{\text{test}}^{\text{full}}\right) - \hat{L}_{\text{train}}^{\text{full}}\left(\mathbf{W}^{\text{mini}}, \tilde{\mathbf{A}}_{\text{train}}^{\text{mini}}\right)\right)}\right],$$

where the last inequality uses Lemma M.1.

Next, we upper-bound the second term in the RHS of (217). Here the term $\Lambda = \mathbb{E}_{\mathbf{W}^{\text{mini}} \sim \mathcal{P}}\left[e^{\left(L_{\text{test}}\left(\mathbf{W}^{\text{mini}}, \tilde{\mathbf{A}}_{\text{test}}^{\text{full}}\right) - \hat{L}_{\text{train}}^{\text{full}}\left(\mathbf{W}^{\text{mini}}, \tilde{\mathbf{A}}_{\text{train}}^{\text{mini}}\right)\right)}\right]$ is a random variable with the randomness coming from the sample of node labels in training dataset, and $\mathcal{P}$ is independent of node labels $\mathbf{y}$ from training dataset. Applying Markov's inequality to the term $\Lambda$, we have for any $C_G > 0$, with probability at least $1 - C_G$ over $\mathbf{y}$ from training set,

$$\Lambda \leq \frac{1}{C_G}\mathbb{E}_{\mathbf{y} \sim \text{training set}}[\Lambda], \tag{218}$$

and hence,

$$\ln \Lambda \leq \ln \frac{1}{C_G}\mathbb{E}_{\mathbf{y} \sim \text{training set}}[\Lambda] = \ln \frac{1}{C_G} + \ln \mathbb{E}_{\mathbf{y} \sim \text{training set}}[\Lambda]. \tag{219}$$

Then we need to upper-bound $\ln \mathbb{E}_{\mathbf{y} \sim \text{training set}}[\Lambda]$. We can rewrite it as:

$$\ln \mathbb{E}_{\mathbf{y} \sim \text{training set}}[\Lambda]$$

$$= \ln \mathbb{E}_{\mathbf{y} \sim \text{training set}}\mathbb{E}_{\mathbf{W}^{\text{mini}} \sim \mathcal{P}}\left[e^{\left(L_{\text{test}}\left(\mathbf{W}^{\text{mini}}, \tilde{\mathbf{A}}_{\text{test}}^{\text{full}}\right) - \hat{L}_{\text{train}}^{\text{full}}\left(\mathbf{W}^{\text{mini}}, \tilde{\mathbf{A}}_{\text{train}}^{\text{mini}}\right)\right)}\right] \tag{220}$$

$$= \ln \mathbb{E}_{\mathbf{W}^{\text{mini}} \sim \mathcal{P}}\mathbb{E}_{\mathbf{y} \sim \text{training set}}\left[e^{\left(L_{\text{test}}\left(\mathbf{W}^{\text{mini}}, \tilde{\mathbf{A}}_{\text{test}}^{\text{full}}\right) - \hat{L}_{\text{train}}^{\text{full}}\left(\mathbf{W}^{\text{mini}}, \tilde{\mathbf{A}}_{\text{train}}^{\text{mini}}\right)\right)}\right].$$

For a fixed model with model parameters $\mathbf{W}^{\text{mini}}$, we have

$$\mathbb{E}_{\mathbf{y} \sim \text{training set}}\left[e^{\left(L_{\text{test}}\left(\mathbf{W}^{\text{mini}}, \tilde{\mathbf{A}}_{\text{test}}^{\text{full}}\right) - \hat{L}_{\text{train}}^{\text{full}}\left(\mathbf{W}^{\text{mini}}, \tilde{\mathbf{A}}_{\text{train}}^{\text{mini}}\right)\right)}\right]$$

$$= \mathbb{E}_{\mathbf{y} \sim \text{training set}}\left[e^{\left(L_{\text{test}}\left(\mathbf{W}^{\text{mini}}, \tilde{\mathbf{A}}_{\text{test}}^{\text{full}}\right) - L_{\text{train}}^{\text{full}}\left(\mathbf{W}^{\text{mini}}, \tilde{\mathbf{A}}_{\text{train}}^{\text{mini}}\right) + L_{\text{train}}^{\text{full}}\left(\mathbf{W}^{\text{mini}}, \tilde{\mathbf{A}}_{\text{train}}^{\text{mini}}\right) - \hat{L}_{\text{train}}^{\text{full}}\left(\mathbf{W}^{\text{mini}}, \tilde{\mathbf{A}}_{\text{train}}^{\text{mini}}\right)\right)}\right]$$

$$= \mathbb{E}_{\mathbf{y} \sim \text{training set}}\left[e^{\left(L_{\text{test}}\left(\mathbf{W}^{\text{mini}}, \tilde{\mathbf{A}}_{\text{test}}^{\text{full}}\right) - L_{\text{train}}^{\text{full}}\left(\mathbf{W}^{\text{mini}}, \tilde{\mathbf{A}}_{\text{train}}^{\text{mini}}\right)\right)}e^{\left(L_{\text{train}}^{\text{full}}\left(\mathbf{W}^{\text{mini}}, \tilde{\mathbf{A}}_{\text{train}}^{\text{mini}}\right) - \hat{L}_{\text{train}}^{\text{full}}\left(\mathbf{W}^{\text{mini}}, \tilde{\mathbf{A}}_{\text{train}}^{\text{mini}}\right)\right)}\right] \tag{221}$$

$$= e^{\left(L_{\text{test}}\left(\mathbf{W}^{\text{mini}}, \tilde{\mathbf{A}}_{\text{test}}^{\text{full}}\right) - L_{\text{train}}^{\text{full}}\left(\mathbf{W}^{\text{mini}}, \tilde{\mathbf{A}}_{\text{train}}^{\text{mini}}\right)\right)}\mathbb{E}_{\mathbf{y} \sim \text{training set}}\left[e^{\left(L_{\text{train}}^{\text{full}}\left(\mathbf{W}^{\text{mini}}, \tilde{\mathbf{A}}_{\text{train}}^{\text{mini}}\right) - \hat{L}_{\text{train}}^{\text{full}}\left(\mathbf{W}^{\text{mini}}, \tilde{\mathbf{A}}_{\text{train}}^{\text{mini}}\right)\right)}\right].$$

In the following, we wil give an upper bound on $\mathbb{E}_{\mathbf{y} \sim \text{training set}}\left[e^{\left(L_{\text{train}}^{\text{full}}\left(\mathbf{W}^{\text{mini}}, \tilde{\mathbf{A}}_{\text{train}}^{\text{mini}}\right) - \hat{L}_{\text{train}}^{\text{full}}\left(\mathbf{W}^{\text{mini}}, \tilde{\mathbf{A}}_{\text{train}}^{\text{mini}}\right)\right)}\right]$ that is independent of $\mathbf{W}^{\text{mini}}$. For the entire training dataset, $\hat{L}_{\text{train}}^{\text{full}}\left(\mathbf{W}^{\text{mini}}, \tilde{\mathbf{A}}_{\text{train}}^{\text{mini}}\right)$ can be written as:

$$\hat{L}_{\text{train}}^{\text{full}}\left(\mathbf{W}^{\text{mini}}, \tilde{\mathbf{A}}_{\text{train}}^{\text{mini}}\right) = \frac{1}{n_{\text{train}}}\sum_{i \in \text{training set}}\|\hat{\mathbf{y}}_i^{\text{mini}} - \mathbf{y}_i\|_F^2, \tag{222}$$

where the node labels are independently sampled. Hence, $\hat{L}_{\text{train}}^{\text{full}}\left(\mathbf{W}^{\text{mini}}, \tilde{\mathbf{A}}_{\text{train}}^{\text{mini}}\right)$ is the empirical mean of $n_{\text{train}}$ independent Bernoulli random variables and $L_{\text{train}}^{\text{full}}\left(\mathbf{W}^{\text{mini}}, \tilde{\mathbf{A}}_{\text{train}}^{\text{mini}}\right)$ is the expectation of $\hat{L}_{\text{train}}^{\text{full}}\left(\mathbf{W}^{\text{mini}}, \tilde{\mathbf{A}}_{\text{train}}^{\text{mini}}\right)$. By Lemma M.2, for any $C_1 > 0$,

$$\mathbb{P}\left(\left|L_{\text{train}}^{\text{full}}\left(\mathbf{W}^{\text{mini}}, \tilde{\mathbf{A}}_{\text{train}}^{\text{mini}}\right) - \hat{L}_{\text{train}}^{\text{full}}\left(\mathbf{W}^{\text{mini}}, \tilde{\mathbf{A}}_{\text{train}}^{\text{mini}}\right)\right| \geq C_1\right) \leq 2e^{-2n_{\text{train}}C_1^2}, \tag{223}$$

and hence, by Lemma M.1, we have

$$\mathbb{E}_{\mathbf{y}\sim\text{training set}}\left[e^{C_u\left(L_{\text{train}}^{\text{full}}\left(\mathbf{W}^{\text{mini}},\tilde{\mathbf{A}}_{\text{train}}^{\text{mini}}\right)-\hat{L}_{\text{train}}^{\text{full}}\left(\mathbf{W}^{\text{mini}},\tilde{\mathbf{A}}_{\text{train}}^{\text{mini}}\right)\right)}\right]\leq e^{\frac{C_u^2}{4n_{\text{train}}}}. \tag{224}$$

Therefore, we have

$$\ln\Lambda\leq\ln\mathbb{E}_{\mathbf{W}^{\text{mini}}\sim\mathcal{P}}\left[e^{\left(L_{\text{test}}\left(\mathbf{W}^{\text{mini}},\tilde{\mathbf{A}}_{\text{test}}^{\text{full}}\right)-L_{\text{train}}^{\text{full}}\left(\mathbf{W}^{\text{mini}},\tilde{\mathbf{A}}_{\text{train}}^{\text{mini}}\right)\right)}e^{\frac{C_u^2}{4n_{\text{train}}}}\right]$$
$$=U+\frac{C_u^2}{4n_{\text{train}}}. \tag{225}$$

Finally, we get

$$C_u\left(L_{\text{test}}\left(\mathbf{W}^{\text{mini}},\tilde{\mathbf{A}}_{\text{test}}^{\text{full}};\mathcal{Q}\right)-\hat{L}_{\text{train}}^{\text{full}}\left(\mathbf{W}^{\text{mini}},\tilde{\mathbf{A}}_{\text{train}}^{\text{mini}};\mathcal{Q}\right)\right)$$
$$\leq D_{KL}(\mathcal{Q}\|\mathcal{P})+\ln\mathbb{E}_{\mathbf{W}^{\text{mini}}\sim\mathcal{P}}\left[e^{\left(L_{\text{test}}\left(\mathbf{W}^{\text{mini}},\tilde{\mathbf{A}}_{\text{test}}^{\text{full}}\right)-\hat{L}_{\text{train}}^{\text{full}}\left(\mathbf{W}^{\text{mini}},\tilde{\mathbf{A}}_{\text{train}}^{\text{mini}}\right)\right)}\right]$$
$$\leq D_{KL}(\mathcal{Q}\|\mathcal{P})+\ln\frac{1}{C_G}+\frac{C_u^2}{4n_{\text{train}}}+U. \tag{226}$$

Hence, we have the final result

$$L_{\text{test}}\left(\mathbf{W}^{\text{mini}},\tilde{\mathbf{A}}_{\text{test}}^{\text{full}};\mathcal{Q}\right)$$
$$\leq\hat{L}_{\text{train}}^{\text{full}}\left(\mathbf{W}^{\text{mini}},\tilde{\mathbf{A}}_{\text{train}}^{\text{mini}};\mathcal{Q}\right)+\frac{1}{C_u}\left(D_{KL}(\mathcal{Q}\|\mathcal{P})+\ln\frac{1}{C_G}+\frac{C_u^2}{4n_{\text{train}}}+U\right). \tag{227}$$

## M.2 PROOF OF LEMMA G.6

Recall that

$$U=\ln\mathbb{E}_{\mathbf{W}^{\text{mini}}\sim\mathcal{P}}\left[e^{C_u\left(L_{\text{test}}\left(\mathbf{W}^{\text{mini}},\tilde{\mathbf{A}}_{\text{test}}^{\text{full}}\right)-L_{\text{train}}^{\text{full}}\left(\mathbf{W}^{\text{mini}},\tilde{\mathbf{A}}_{\text{train}}^{\text{mini}}\right)\right)}\right]. \tag{228}$$

First, we focus on the term $L_{\text{test}}\left(\mathbf{W}^{\text{mini}},\tilde{\mathbf{A}}_{\text{test}}^{\text{full}}\right)-L_{\text{train}}^{\text{full}}\left(\mathbf{W}^{\text{mini}},\tilde{\mathbf{A}}_{\text{train}}^{\text{mini}}\right)$. Set $l_{\text{train}}^{\text{mini}}(\mathbf{y}_i)=\|\hat{\mathbf{y}}_i^{\text{mini}}-\mathbf{y}_i\|_F^2,\forall i\in$ train set and $l_{\text{test}}^{\text{mini}}(\mathbf{y}_j)=\left\|\hat{\mathbf{y}}_i^{\text{mini}}-\mathbf{y}_j\right\|_F^2,\forall i\in$ test set. Then we have

$$L_{\text{test}}\left(\mathbf{W}^{\text{mini}},\tilde{\mathbf{A}}_{\text{test}}^{\text{full}}\right)-L_{\text{train}}^{\text{full}}\left(\mathbf{W}^{\text{mini}},\tilde{\mathbf{A}}_{\text{train}}^{\text{mini}}\right)$$
$$=\mathbb{E}_{\mathbf{y}\sim\text{test set}}\left[\frac{1}{n_{\text{test}}}\sum_{j\in\text{test set}}l_{\text{test}}^{\text{mini}}(\mathbf{y}_j)\right]-\mathbb{E}_{\mathbf{y}\sim\text{train set}}\left[\frac{1}{n_{\text{train}}}\sum_{i\in\text{train set}}l_{\text{train}}^{\text{mini}}(\mathbf{y}_i)\right] \tag{229}$$
$$=\frac{1}{n_{\text{test}}}\sum_{j\in\text{test set}}l_{\text{test}}^{\text{mini}}(\mathbf{y}_j)\rho_{\text{test}}(\mathbf{y}_j)-\frac{1}{n_{\text{train}}}\sum_{i\in\text{train set}}l_{\text{train}}^{\text{mini}}(\mathbf{y}_i)\rho_{\text{train}}(\mathbf{y}_i)$$

Furthermore, we define $\boldsymbol{\Sigma}_{\text{train},i}=Diag\left(\mathbb{1}\left\{\tilde{\mathbf{a}}_{\text{train},i}^{\text{mini}}\mathbf{X}\left(\mathbf{W}^{\text{mini}}\right)^\top>0\right\}\right)\in\mathbb{R}^{h\times h}$ to represent whether the $j$-th element $\left\{\tilde{\mathbf{a}}_{\text{train},i}\mathbf{X}\left(\mathbf{W}^{\text{full}}\right)^\top\right\}_j$ is more than zero (1) or is zeroed out (0). Then we have:

$$\sigma\left(\tilde{\mathbf{a}}_{\text{train},i}^{\text{mini}}\mathbf{X}\left(\mathbf{W}^{\text{mini}}\right)^\top\right)=\tilde{\mathbf{a}}_{\text{train},i}^{\text{mini}}\mathbf{X}(\boldsymbol{\Sigma}_{\text{train},i}\mathbf{W}^{\text{mini}})^\top. \tag{230}$$

Similarly, we have $\sigma\left(\tilde{\mathbf{a}}_{\text{train},i}^{\text{mini}}\mathbf{X}\left(\mathbf{W}^{\text{mini}*}\right)^\top\right)=\tilde{\mathbf{a}}_{\text{train},i}^{\text{mini}}\mathbf{X}(\boldsymbol{\Sigma}_{\text{train},i}^*\mathbf{W}^{\text{mini}*})^\top$, $\sigma\left(\tilde{\mathbf{a}}_{\text{test},i}^{\text{full}}\mathbf{X}\left(\mathbf{W}^{\text{mini}}\right)^\top\right)=\tilde{\mathbf{a}}_{\text{test},i}^{\text{full}}\mathbf{X}(\boldsymbol{\Sigma}_{\text{test},i}\mathbf{W}^{\text{mini}})^\top$, and $\sigma\left(\tilde{\mathbf{a}}_{\text{test},i}^{\text{full}}\mathbf{X}\left(\mathbf{W}^{\text{mini}*}\right)^\top\right)=\tilde{\mathbf{a}}_{\text{test},i}^{\text{full}}\mathbf{X}(\boldsymbol{\Sigma}_{\text{test},i}^*\mathbf{W}^{\text{mini}*})^\top$.

we set $f(\mathbf{y}_i)=-\frac{1}{n_{\text{train}}}l_{\text{train}}^{\text{mini}}(\mathbf{y}_i)$ with $\forall i\in$ train set and $g(\mathbf{y}_j)=\frac{1}{n_{\text{test}}}l_{\text{test}}^{\text{mini}}(\mathbf{y}_j)$ with $\forall j\in$ test set and $n_{\min}=\min\{n_{\text{train}},n_{\text{test}}\}$.

Hence, we have:

$$
\begin{aligned}
&f(\mathbf{y}_i) + g(\mathbf{y}_j) \\
&= \frac{1}{n_{\text{test}}} l_{\text{test}}^{\text{mini}}(\mathbf{y}_j) - \frac{1}{n_{\text{train}}} l_{\text{train}}^{\text{mini}}(\mathbf{y}_i) \\
&\leq \frac{1}{n_{\min}} \left( l_{\text{test}}^{\text{mini}}(\mathbf{y}_j) - l_{\text{train}}^{\text{mini}}(\mathbf{y}_i) \right) \\
&= \frac{1}{n_{\min}} \left\| \sigma\left( \tilde{\mathbf{a}}_{\text{test},j}^{\text{full}} \mathbf{X} \left( \mathbf{W}^{\text{mini}} \right)^\top \right) - \sigma\left( \tilde{\mathbf{a}}_{\text{test},j}^{\text{full}} \mathbf{X} \left( \mathbf{W}^{\text{mini}*} \right)^\top \right) \right\|_F^2 \\
&\quad - \frac{1}{n_{\min}} \left\| \sigma\left( \tilde{\mathbf{a}}_{\text{train},i}^{\text{mini}} \mathbf{X} \left( \mathbf{W}^{\text{mini}} \right)^\top \right) - \sigma\left( \tilde{\mathbf{a}}_{\text{train},i}^{\text{mini}} \mathbf{X} \left( \mathbf{W}^{\text{mini}*} \right)^\top \right) \right\|_F^2 \\
&\leq \frac{1}{n_{\min}} \| \sigma\left( \tilde{\mathbf{a}}_{\text{test},j}^{\text{full}} \mathbf{X} \left( \mathbf{W}^{\text{mini}} \right)^\top \right) - \sigma\left( \tilde{\mathbf{a}}_{\text{test},j}^{\text{full}} \mathbf{X} \left( \mathbf{W}^{\text{mini}*} \right)^\top \right) \\
&\quad - \left( \sigma\left( \tilde{\mathbf{a}}_{\text{train},i}^{\text{mini}} \mathbf{X} \left( \mathbf{W}^{\text{mini}} \right)^\top \right) - \sigma\left( \tilde{\mathbf{a}}_{\text{train},i}^{\text{mini}} \mathbf{X} \left( \mathbf{W}^{\text{mini}*} \right)^\top \right) \right) \|_F^2 \\
&= \frac{1}{n_{\min}} \| \tilde{\mathbf{a}}_{\text{test},j}^{\text{full}} \mathbf{X} \left( \mathbf{W}^{\text{mini}} \right)^\top \boldsymbol{\Sigma}_{\text{test},j}^\top - \tilde{\mathbf{a}}_{\text{test},j}^{\text{full}} \mathbf{X} \left( \mathbf{W}^{\text{mini}*} \right)^\top \boldsymbol{\Sigma}_{\text{test},j}^{*\top} \\
&\quad - \left( \tilde{\mathbf{a}}_{\text{train},i}^{\text{mini}} \mathbf{X} \left( \mathbf{W}^{\text{mini}} \right)^\top \boldsymbol{\Sigma}_{\text{train},i}^\top - \tilde{\mathbf{a}}_{\text{train},i}^{\text{mini}} \mathbf{X} \left( \mathbf{W}^{\text{mini}*} \right)^\top \boldsymbol{\Sigma}_{\text{train},i}^{*\top} \right) \|_F^2 \\
&\leq \frac{\|\mathbf{X}\|_F^2}{n_{\min}} \Big( \left( \left\| \mathbf{W}^{\text{mini}*} \right\|_F^2 \left\| \boldsymbol{\Sigma}_{\text{train},i}^* \right\|_F^2 + \left\| \mathbf{W}^{\text{mini}} \right\|_F^2 \left\| \boldsymbol{\Sigma}_{\text{train},i} \right\|_F^2 \right) \left\| \tilde{\mathbf{a}}_{\text{test},j}^{\text{full}} - \tilde{\mathbf{a}}_{\text{train},i}^{\text{mini}} \right\|_F^2 \\
&\quad + \left\| \mathbf{W}^{\text{mini}*} \right\|_F^2 \left\| \boldsymbol{\Sigma}_{\text{train},i}^* - \boldsymbol{\Sigma}_{\text{test},j}^* \right\|_F^2 \left\| \tilde{\mathbf{a}}_{\text{test},j}^{\text{full}} \right\|_F^2 \\
&\quad + \left\| \mathbf{W}^{\text{mini}} \right\|_F^2 \left\| \boldsymbol{\Sigma}_{\text{train},i} - \boldsymbol{\Sigma}_{\text{test},j} \right\|_F^2 \left\| \tilde{\mathbf{a}}_{\text{test},j}^{\text{full}} \right\|_F^2 \Big) \\
&\leq \frac{C_F \left( C_w + 1 \right) h^2}{n_{\min}} \left( \left\| \tilde{\mathbf{a}}_{\text{test},j}^{\text{full}} - \tilde{\mathbf{a}}_{\text{train},i}^{\text{mini}} \right\|_F^2 + 2 \left\| \tilde{\mathbf{a}}_{\text{test},j}^{\text{full}} \right\|_F^2 \right) \\
&\leq \frac{C_F \left( C_w + 1 \right) h^2}{n_{\min}} \left( \left\| \tilde{\mathbf{a}}_{\text{test},j}^{\text{full}} - \tilde{\mathbf{a}}_{\text{train},i}^{\text{full}} \right\|_F^2 + 2 \left\| \tilde{\mathbf{a}}_{\text{test},j}^{\text{full}} \right\|_F^2 + \left\| \tilde{\mathbf{a}}_{\text{train},i}^{\text{full}} - \tilde{\mathbf{a}}_{\text{train},i}^{\text{mini}} \right\|_F^2 \right), \\
&= \frac{C_F \left( C_w + 1 \right) h^2}{n_{\min}} \left( \delta_{i,j}^{\text{full}} + \delta_i^{\text{full-mini}} \right) \\
&= \delta(\mathbf{y}_i, \mathbf{y}_j, \beta, b)
\end{aligned}
\tag{231}
$$

where the penultimate inequality follows $\left\| \boldsymbol{\Sigma}_{\text{train},i} \right\|_F^2, \left\| \boldsymbol{\Sigma}_{\text{train},i}^* \right\|_F^2 \leq h$, $\left\| \boldsymbol{\Sigma}_{\text{train},i}^* - \boldsymbol{\Sigma}_{\text{test},i}^* \right\|_F^2, \left\| \boldsymbol{\Sigma}_{\text{train},i} - \boldsymbol{\Sigma}_{\text{test},i} \right\|_F^2 \leq 2h$ because $\boldsymbol{\Sigma}_i$ is a diagonal matrix with $(\boldsymbol{\Sigma}_i)_{jj} \in \{0, 1\}$ for any $j \in \{1, \ldots, h\}$. The penultimate expression is exactly the distance function defined in Definition 1., $\delta_{i,j}^{\text{full-mini}} = \left\| \tilde{\mathbf{a}}_{\text{train},i}^{\text{full}} - \tilde{\mathbf{a}}_{\text{train},i}^{\text{mini}} \right\|_F^2$, and $\delta_{i,j}^{\text{full}} = \left\| \tilde{\mathbf{a}}_{\text{test},j}^{\text{full}} - \tilde{\mathbf{a}}_{\text{train},i}^{\text{full}} \right\|_F^2 + 2 \left\| \tilde{\mathbf{a}}_{\text{test},j}^{\text{full}} \right\|_F^2$ is a constant based on the split of training and testing.

Hence, we have

$$
\begin{aligned}
&L_{\text{test}} \left( \mathbf{W}^{\text{mini}}, \tilde{\mathbf{A}}_{\text{test}}^{\text{full}} \right) - L_{\text{train}}^{\text{full}} \left( \mathbf{W}^{\text{mini}}, \tilde{\mathbf{A}}_{\text{train}}^{\text{mini}} \right) \\
&= \frac{1}{2n_{\text{test}}} \sum_{j \in \text{test set}} l_{\text{test}}^{\text{mini}}(\mathbf{y}_j) \rho_{\text{test}}(\mathbf{y}_j) - \frac{1}{2n_{\text{train}}} \sum_{i \in \text{train set}} l_{\text{train}}^{\text{mini}}(\mathbf{y}_i) \rho_{\text{train}}(\mathbf{y}_i) \\
&\leq \Delta_{\text{train,test}}(\beta, b) = \min \sum_{i \in \text{train set}} \sum_{j \in \text{test set}} \theta_{i,j} \delta(\mathbf{y}_i, \mathbf{y}_j, \beta, b) \\
&= \min \sum_{i \in \text{train set}} \sum_{j \in \text{test set}} \theta_{i,j} \frac{C_F \left( C_w + 1 \right) h^2}{n_{\min}} \left( \delta_{i,j}^{\text{full}} + \delta_i^{\text{full-mini}} \right)
\end{aligned}
\tag{232}
$$

Then we mainly focus on the elements of $\delta_i^{\text{full-mini}}$.

Recall that $\tilde{\mathbf{a}}_{\text{train},i,j}^{\text{full}} = \frac{1}{\sqrt{d_i^{\text{in,full}}} \sqrt{d_j^{\text{out,full}}}} a_{ij}^{\text{full}}$ and $\tilde{\mathbf{a}}_{\text{train},i,j}^{\text{mini}} = \frac{1}{\sqrt{d_i^{\text{in,mini}}} \sqrt{d_j^{\text{out,mini}}}} a_{ij}^{\text{mini}}$, where $a_{ij}^{\text{full}}, a_{ij}^{\text{mini}} \in \{0, 1\}$ represents whether node $i$ receives a message from node $j$ (1) or not (0).

Hence, we have:

$$\left\| \tilde{\mathbf{a}}_{\text{train},i}^{\text{full}} - \tilde{\mathbf{a}}_{\text{train},i}^{\text{mini}} \right\|_F^2 = \sum_{j=1}^n \left| \frac{1}{\sqrt{d_i^{\text{in,full}}} \sqrt{d_j^{\text{out,full}}}} a_{ij}^{\text{full}} - \frac{1}{\sqrt{d_i^{\text{in,mini}}} \sqrt{d_j^{\text{out,mini}}}} a_{ij}^{\text{mini}} \right|^2, \tag{233}$$

where $\frac{1}{\sqrt{d_i^{\text{in,full}}} \sqrt{d_j^{\text{out,full}}}} \leq \frac{1}{\sqrt{d_i^{\text{in,mini}}} \sqrt{d_j^{\text{out,mini}}}}$.

We fix the batch size $b$. Notice that when the fan-out size $\beta$ increases, $d_j^{\text{out,mini}}$ may increase and we have four cases: (1). $a_{ij}^{\text{mini}}$ keeps as 0 given $a_{ij}^{\text{full}} = 0$, (2). $a_{ij}^{\text{mini}}$ keeps as 0 given $a_{ij}^{\text{full}} = 1$, (3). $a_{ij}^{\text{mini}}$ keeps as 1 given $a_{ij}^{\text{full}} = 1$, (4). $a_{ij}^{\text{mini}}$ becomes 1 from 0 given $a_{ij}^{\text{full}} = 1$. Then we have $\sum_{j=1}^n \left| \frac{1}{\sqrt{d_i^{\text{in,full}}} \sqrt{d_j^{\text{out,full}}}} a_{ij}^{\text{full}} - \frac{1}{\sqrt{d_i^{\text{in,mini}}} \sqrt{d_j^{\text{out,mini}}}} a_{ij}^{\text{mini}} \right|^2$ are non-increasing when $\beta$ increases at the first three cases. However, at the fourth case, we may both have $\left| \frac{1}{\sqrt{d_i^{\text{in,full}}} \sqrt{d_j^{\text{out,full}}}} \right|^2 \leq$
$\left| \frac{1}{\sqrt{d_i^{\text{in,full}}} \sqrt{d_j^{\text{out,full}}}} - \frac{1}{\sqrt{d_i^{\text{in,mini}}} \sqrt{d_j^{\text{out,mini}}}} \right|^2$ and $\left| \frac{1}{\sqrt{d_i^{\text{in,full}}} \sqrt{d_j^{\text{out,full}}}} \right|^2 \geq \left| \frac{1}{\sqrt{d_i^{\text{in,full}}} \sqrt{d_j^{\text{out,full}}}} - \frac{1}{\sqrt{d_i^{\text{in,mini}}} \sqrt{d_j^{\text{out,mini}}}} \right|^2$.
Hence, $\delta_i^{\text{full-mini}}$ has a overall non-increasing trend when $\beta$ increases but small non-monotonic fluctuations can exist.

We fix the fan-out size $\beta$. Notice that when the batch size $b$ increases, $d_j^{\text{in,mini}}$ may increase and we have three situations: (1). $a_{ij}^{\text{mini}}$ keeps as 0 given $a_{ij}^{\text{full}} = 0$, (2). $a_{ij}^{\text{mini}}$ keeps as 0 given $a_{ij}^{\text{full}} = 1$, (3). $a_{ij}^{\text{mini}}$ keeps as 1 given $a_{ij}^{\text{full}} = 1$. Then we have $\delta_i^{\text{full-mini}}$ is non-increasing when $b$ increases.

Note that fan-out size $\beta$ plays a more dominant role than batch size $b$ in influencing generalization. This is because the variation in fan-out size $\beta$ not only increases the number of sampled neighbors but also potentially alters the structure of the adjacency matrix of node $i$ — by introducing new connections during mini-batch sampling (i.e., the fourth case). This structural change can lead to more significant variations in generalization performance. In contrast, changes in batch size b primarily supplement the number of sampled nodes without modifying the adjacency structure of the node $i$.

Since $\delta(\mathbf{y}_i, \mathbf{y}_j, \beta, b)$ is proporional to $\delta_{i,j}^{\text{full-mini}} = \left\| \tilde{\mathbf{a}}_{\text{train},i}^{\text{full}} - \tilde{\mathbf{a}}_{\text{train},i}^{\text{mini}} \right\|_F^2$ and $\Delta(\beta, b)$ is proporional to $\delta(\mathbf{y}_i, \mathbf{y}_j, \beta, b)$, we have the upper bound $\Delta(\beta, b)$ of $L_{\text{test}}\left( \mathbf{W}^{\text{mini}}, \tilde{\mathbf{A}}_{\text{test}}^{\text{full}} \right) - L_{\text{train}}^{\text{full}}\left( \mathbf{W}^{\text{mini}}, \tilde{\mathbf{A}}_{\text{train}}^{\text{mini}} \right)$ keeps non-increasing when $b$ increases, and overall have the non-increasing trend when $\beta$ increases but small non-monotonic fluctuations exist.

Finally, we have

$$\begin{aligned} U &= \ln \mathbb{E}_{\mathbf{W}^{\text{mini}} \sim \mathcal{P}} \left[ e^{C_u \left( L_{\text{test}}\left( \mathbf{W}^{\text{mini}}, \tilde{\mathbf{A}}_{\text{test}}^{\text{full}} \right) - L_{\text{train}}^{\text{full}}\left( \mathbf{W}^{\text{mini}}, \tilde{\mathbf{A}}_{\text{train}}^{\text{mini}} \right) \right)} \right] \\ &\leq \ln \mathbb{E}_{\mathbf{W}^{\text{mini}} \sim \mathcal{P}} \left[ e^{C_u \Delta(\beta, b)} \right] \\ &= \ln(e^{C_u \Delta(\beta, b)}) \\ &= C_u \Delta(\beta, b). \end{aligned} \tag{234}$$

This completes the proof.

# N  EXPERIMENTS

## N.1  TRAINING SETTINGS

**Testbed:** The experiments, except those on the ogbn-papers100M, are conducted on a machine with 512GB of host memory and four NVIDIA A100 GPUs, each with 40GB of memory, inter-connected via 900GB/s NVLink 4.0. The experiments on ogbn-papers100M are run on two machines without GPUs, each equipped with 1024GB of host memory and an interconnect bandwidth of 50 Gbps.

Table 3: Datasets Info1.

| Datasets | #Nodes | #Edges | Avg. Degree | #Classes | #Features |
|---|---|---|---|---|---|
| Reddit | 232,965 | 11,606,919 | 50 | 41 | 602 |
| Ogbn-arxiv | 169,343 | 1,166,243 | 7 | 40 | 128 |
| Ogbn-products | 2,449,029 | 61,859,140 | 25 | 47 | 100 |
| Ogbn-papers100M | 111,059,956 | 1,615,685,872 | 15 | 172 | 128 |

Table 4: Datasets Info2.

| Datasets | Train/Val/Test |
|---|---|
| Reddit | 152,410/23,699/55,334 |
| Ogbn-arxiv | 90,941/29,799/48,603 |
| Ogbn-products | 195,922/48,980/2,204,127 |
| Ogbn-papers100M | 1,207,179 / 125,265/214,338 |

**Metrics:** 1). We evaluate convergence performance using three metrics: iteration-to-loss, iteration-to-accuracy, and time-to-accuracy. These metrics measure training progress towards a target convergence point in terms of training loss or validation accuracy. For all GNN models and datasets except ogbn-papers100M, the target training loss is defined as the maximum loss observed over 100 consecutive iterations at the smallest batch size, provided that the variance of these loss values is below $5 \times 10^{-4}$. Similarly, the target validation accuracy is defined as the minimum accuracy over 100 consecutive iterations at the smallest batch size, provided that the variance of these accuracies is below $4 \times 10^{-4}$. Note that the defined target training loss and the defined target validation accuracy are applied across all hyperparameter settings for the specific model and dataset. For ogbn-papers100M, training is limited to 200 iterations due to the extremely large graph size and training time constraints. Note that using the smallest batch size as the reference is common in prior works (Bajaj et al., 2024), and serves as a conservative criterion: because fluctuations are most pronounced under the smallest batch size, requiring stability in this setting to prevent mistaking transient variations for convergence and to provide a uniform benchmark across batch sizes. Moreover, by enforcing a variance threshold, this definition remains unbiased toward larger or smaller batch sizes and offers a fair basis for comparing convergence across settings. 2). For generalization, test accuracy is used as the metric in the training iteration. 3).For system efficiency, we measure the training throughput in terms of the number of target nodes processed per second (number of nodes/s). This metric ensures that throughput reflects the rate of training examples processed.

We run all implementations using Python 3.8.10 and dgl>=1.0.0. The uniform neighbor sampling is used for mini-batch training. Due to the massive comparisons, adding error bars to every figure would make them overly cluttered and difficult to interpret. We have repeated all experiments at least three times using different seeds and observed low variance. For example, in Figure 6, the standard deviation of the final accuracy is less than 3.17%. This small variance does not affect the observed convergence trends, which remain consistent across runs.

## N.2 METRICS: ITERATION-TO-LOSS

**Simple mathematical derivation.** In distributed systems with two devices, assuming:

- Per-iteration calculation time $t_{cal}$: $t_{cal} = (b * \beta + b)/C$, where $b$ is batch size, $\beta$ is fan-out size, and $C$ is compute capacity (nodes/s);

- Per-iteration communication time $t_{comm}$ : $t_{comm} = b/H$ for mini-batch training and $t_{comm} = (b * \beta + b)/H$ for full-graph training, where $H$ is the bandwidth.

- time-to-accuracy $t$: $t = n \times (t_{cal} + t_{comm})$, where $n$ is iteration-to-accuracy.

Consider two training setups:

- Full-graph training: $b_l = 1000$, $\beta_l = 50$, $n_l = 10$ iterations to converge

- Mini-batch training: $b_s = 10$, $\beta_s = 10$, $n_s = 10000$ iterations to converge

Under the same compute power $C = 1$ node/s, but different bandwidths:

- High bandwidth: $H_h = 1000$ nodes/s
- Low bandwidth: $H_s = 0.1$ nodes/s

Plugging into the formulas:

- High bandwidth: 1).Full-graph: $t = 10 \times \left( \frac{1000 \cdot 50 + 1000}{1} + \frac{1000 \cdot 50 + 1000}{1000} \right) = 5.1051 \times 10^5$ s
  2). Mini-batch: $t = 10000 \times \left( \frac{10 \cdot 10 + 10}{1} + \frac{10}{1000} \right) = 1.1001 \times 10^6$ s
  Therefore, Full-graph training is faster.
- Low bandwidth: 1). Full-graph: $t = 10 \times \left( \frac{1000 \cdot 50 + 1000}{1} + \frac{1000 \cdot 50 + 1000}{0.1} \right) = 5.61 \times 10^6$ s
  2). Mini-batch: $t = 10000 \times \left( \frac{10 \cdot 10 + 10}{1} + \frac{10}{0.1} \right) = 2.1 \times 10^6$ s
  Therefore, Mini-batch training is faster.

This example shows that time-to-accuracy may flip conclusions depending on hardware, while iteration-to-accuracy remains stable.

**Experiments.** The vanilla distributed system (i.e., the standard implementation without any optimizations) is used for full-graph training, and the Distributed Data Parallel (DDP) technique (Li et al., 2020) is applied for mini-batch training. We examine a three-layer GraphSAGE model on Reddit and a three-layer GCN model on ogbn-products. These models include normalization layers and are trained using a cross-entropy loss function and Adam optimizer with a learning rate of 0.01. The target validation accuracy is set at 0.9 for ogbn-products and 0.95 for Reddit. The total batch size is 2000 and the fan-out size is [5,10,15]. To simulate infinite bandwidth (i.e., bw1), we use a single GPU or CPU. For limited bandwidth (i.e., bw2), we use two GPUs interconnected via 900GB/s NVLink.

### N.3 CONVERGENCE

For experiments on one-layer GNN models, the basic setups are without drop-out or normalization layers and with ReLU activation, and SGD optimizer for both full-graph and mini-batch training. For experiments in more general settings, multiple-layer GNNs are adopted without dropout layers and with ReLU activation and Adam optimizer for both full-graph and mini-batch training. The SAR system (Mostafa, 2022) is used for full-graph and mini-batch training on ogbn-papers100M via the *gloo* backend, while other datasets are mainly trained on a single GPU.

**Convergence of one-round GNN trained with MSE.** To align with theoretical analysis, we use iteration-to-loss here. The details are as follows: 1). The target training losses are 0.0226 for ogbn-arxiv, 0.0225 for reddit and ogbn-products, and [0.005, 0.0054, 0.0065] for ogbn-papers100M on GCN, GraphSAGE, GAT, respectively. 2). When varying the batch sizes, the fan-out size is 5. 3). When varying the fan-out sizes, the batch size is 500 for ogbn-arxiv, ogbn-products and reddit, as well as is 10000 for for ogbn-papers100M.

Figure 7-8 shows the iteration-to-loss for four datasets under GAT, GCN, and GraphSAGE trained with MSE across different learning rates and either varying batch sizes or varying fan-out sizes.

**Convergence of one-round GNN trained with CE.** To align with theoretical analysis, we use iteration-to-loss here. We set the original multi-class node classification task as the binary node classification task. The details are as follows: 1). The target training losses are 0.51 for ogbn-arxiv, [0.325,0.325,0.2] for reddit on GCN, GraphSAGE, GAT, respectively, [0.08,0.051,0.051] for ogbn-products on GCN, GraphSAGE, GAT, respectively, and [0.009, 0.0087, 0.0087] for ogbn-papers100M on GCN, GraphSAGE, GAT, respectively. 2). When varying the batch sizes, the fan-out size is 5. 3). When varying the fan-out sizes, the batch size is 500 for ogbn-arxiv, ogbn-products and reddit, as well as is 10000 for for ogbn-papers100M.

Figure 9-10 shows the iteration-to-loss for four datasets under GAT, GCN, and GraphSAGE trained with MSE across different learning rates and either varying batch sizes or varying fan-out sizes.

**Convergence in more general settings.** For the comparison at the dimension of batch size and fan-out size, we use 3-layer GraphSAGE models with hidden dimension of 256 for reddit, ogbn-products,

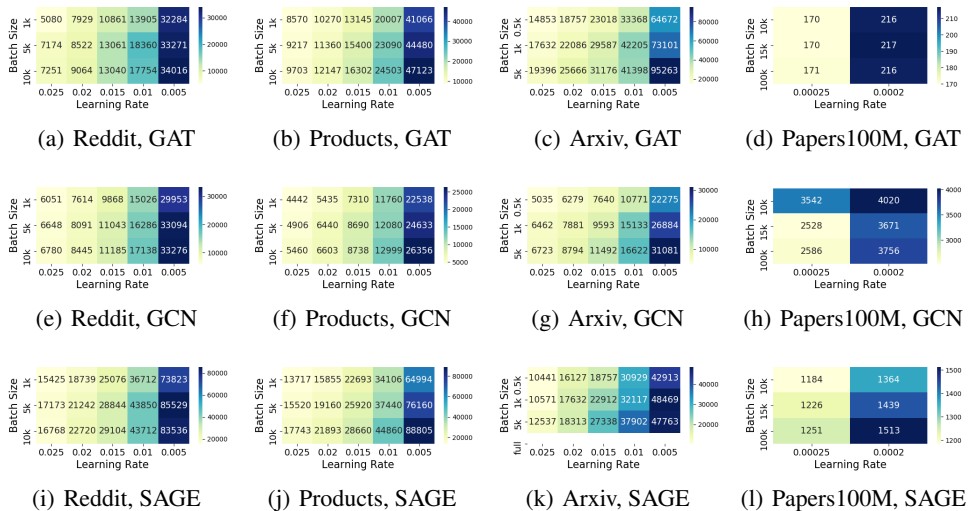

Figure 7: Iteration-to-loss for real-world datasets for one-round GAT, GCN, GraphSAGE across different batch sizes and learning rates under *MSE*.

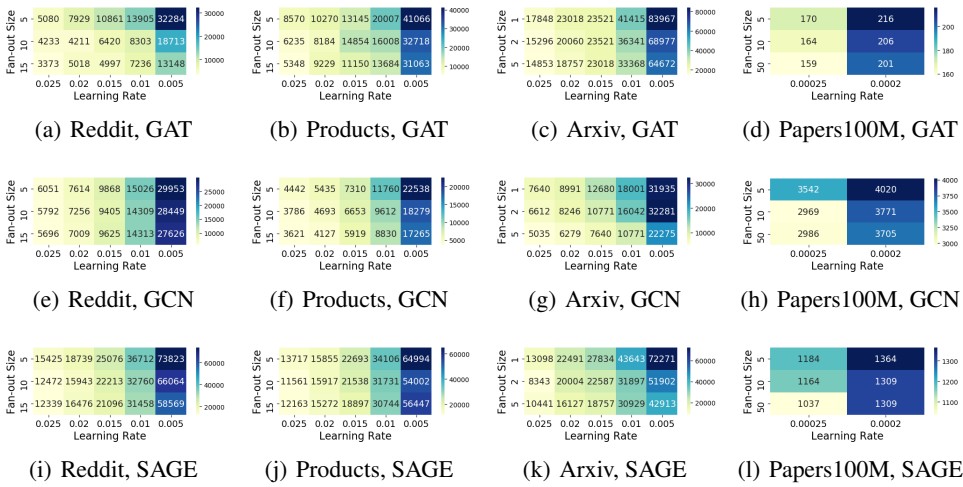

Figure 8: Iteration-to-loss for real-world datasets for one-round GAT, GCN, GraphSAGE across different fan-out sizes and learning rates under *MSE*.

and ogbn-arxiv, and 2-layer GraphSAGE models with hidden dimension of 128 for ogbn-papers100M. The activation function is ReLU function. The optimizer is Adam with a learning rate of 0.001 and a weight decay of 0. Due to the extremely large graph size of the ogbn-papers100M dataset and limited computational resources, we use separate machines for full-graph and mini-batch training on this dataset, making it infeasible to compare system efficiency between the two methods.

The target losses are [0.2, 0.1, 0.8, 1.52] under CE, and [0.005, 0.005, 0.013, 0.0055] under MSE for the products, reddit, arXiv, and papers100M datasets, respectively. The corresponding target accuracies are [0.918, 0.962, 0.708, 0.599] under CE, and [0.89, 0.946, 0.676, 0.5] under MSE for the same datasets.

Figure 11 (under CE) and 12 (under MSE) illustrate time-to-accuracy on GraphSAGE across varying batch sizes and fan-out sizes for ogbn-products, ogbn-arxiv and ogbn-papers100M.

Figure 14 (under MSE) and 13 (under CE) illustrate time-to-accuracy on GraphSAGE across varying batch sizes and fan-out sizes for ogbn-products, ogbn-arxiv and ogbn-papers100M.

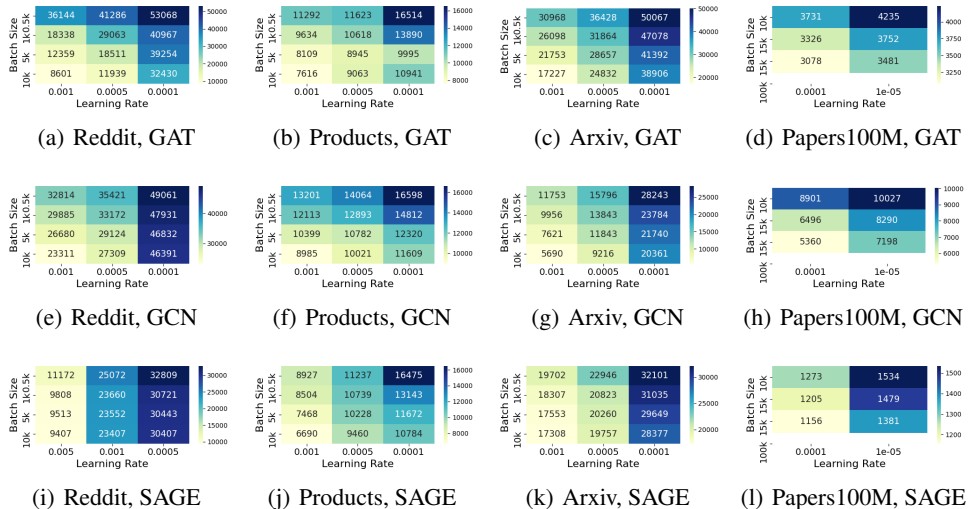

Figure 9: Iteration-to-loss for one-round real-world datasets for GAT, GCN, GraphSAGE across different batch sizes and learning rates under *CE*.

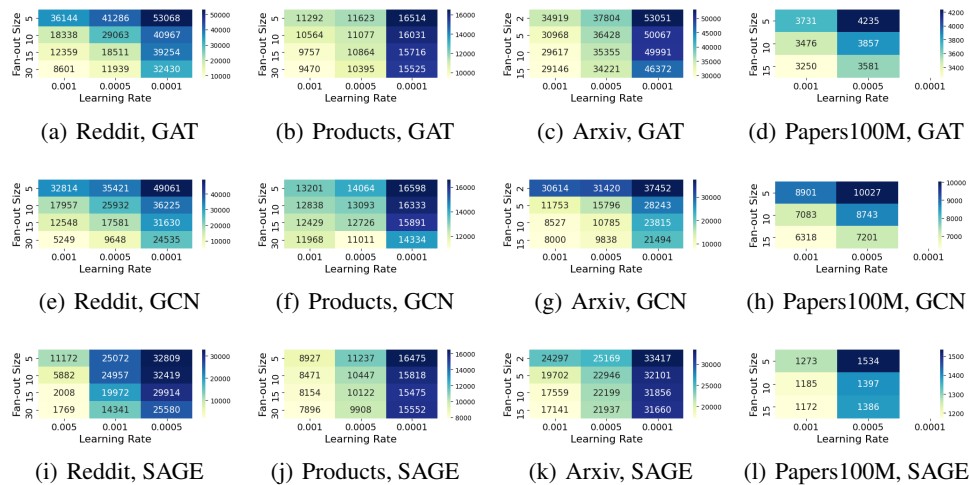

Figure 10: Iteration-to-loss for one-round real-world datasets for GAT, GCN, GraphSAGE across different fan-out sizes and learning rates under *CE*.

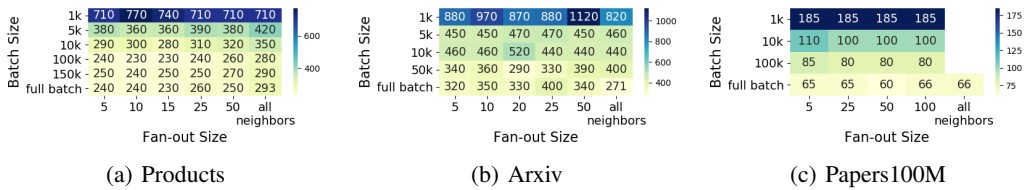

Figure 11: Iteration-to-acc of multi-layer GraphSAGE under *CE* across varying batch sizes and fan-out sizes.

## N.4 GENERALIZATION

**Generalization of one-round GNN trained with MSE.** For test accuracy, *the numbder of iterations are* $5 \times 10^5$ for GraphSAGE and GCN, or $1 \times 10^5$ for GAT, for ogbn-arxiv, ogbn-products, and reddit.

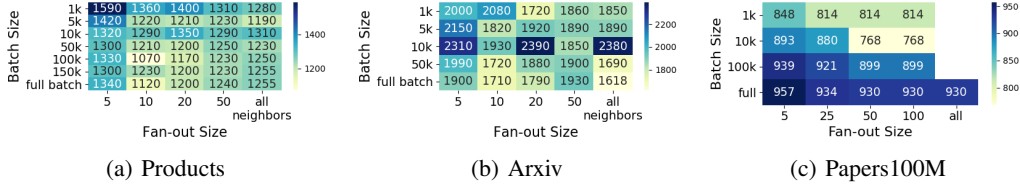

(a) Products           (b) Arxiv           (c) Papers100M

Figure 12: Iteration-to-acc of multi-layer GraphSAGE under *MSE* across varying batch sizes and fan-out sizes.

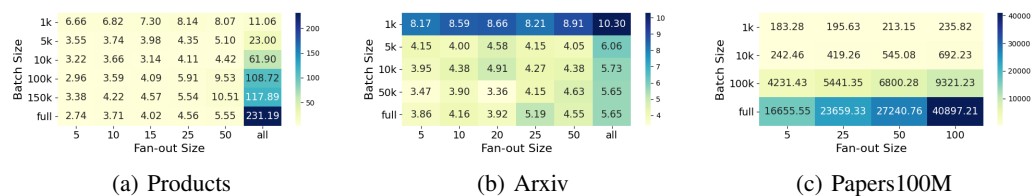

(a) Products           (b) Arxiv           (c) Papers100M

Figure 13: Time-to-accuracy (s) of multi-layer GraphSAGE under *CE* across varying batch sizes and fan-out sizes.

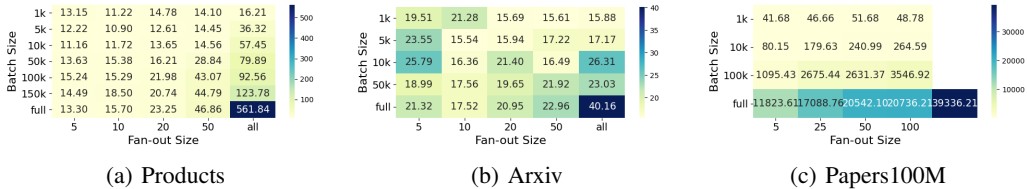

(a) Products           (b) Arxiv           (c) Papers100M

Figure 14: Time-to-accuracy (s) of multi-layer GraphSAGE under *MSE* across varying batch sizes and fan-out sizes.

And the number of iterations are $1 \times 10^4$ for ogbn-papers100M across all GNN models. The *learning rates* are [0.015,0.02,0.025] for ogbn-arxiv, ogbn-products, and reddit, and [0.00025, 0.0002] for ogbn-papers100M. The batch sizes and the fan-out sizes are consistent with the settings used in the experiments measuring time-to-accuracy. Other settings are the same as Appendix N.3.

Figure 15-16 shows the test accuracies for four datasets under GAT, GCN, and GraphSAGE trained with MSE across different learning rates and either varying batch sizes or varying fan-out sizes.

**Generalization in more general settings.** The settings are the same as the general settings in Appendix N.3.

Figure 17 (under CE) and 18 (under MSE) illustrate test accuracies on GraphSAGE across varying batch sizes and fan-out sizes for reddit, ogbn-arxiv and ogbn-papers100M.

### N.5 COMPUTATIONAL EFFICIENCY

The settings are the same as the general settings in Appendix N.3.

Figure 17 (under CE) and 18 (under MSE) illustrate training throughput as the number of processed nodes per second on GraphSAGE across varying batch sizes and fan-out sizes for reddit, ogbn-arxiv and ogbn-papers100M.

### N.6 FULL-GRAPH VS. MINI-BATCH TRAINING AFTER HYPERPARAMETER TUNING.

The settings are the same as the general settings in Appendix N.3.

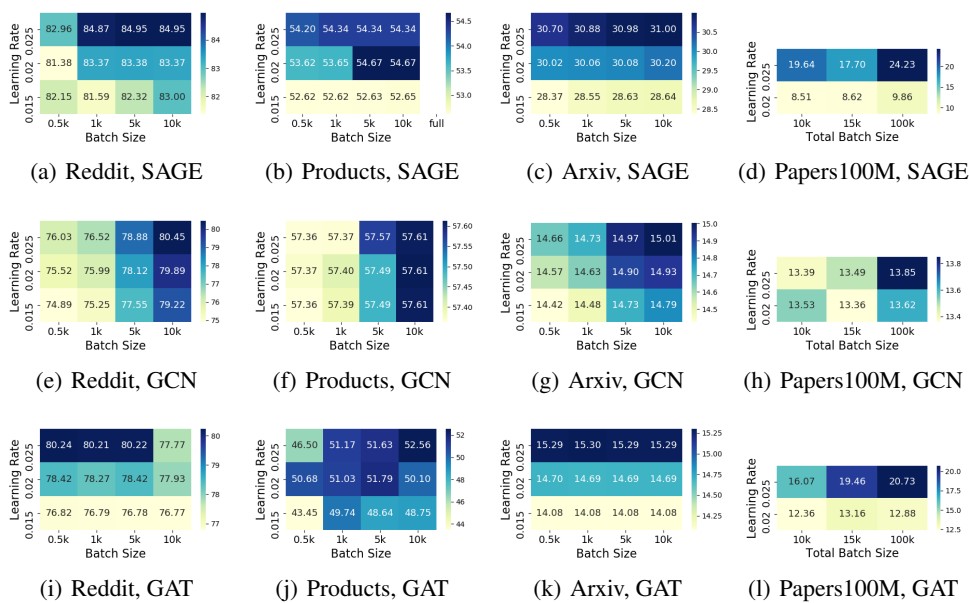

Figure 15: Test accuracy for real-world datasets for one-round GAT, GCN, GraphSAGE across different batch sizes and learning rates under MSE.

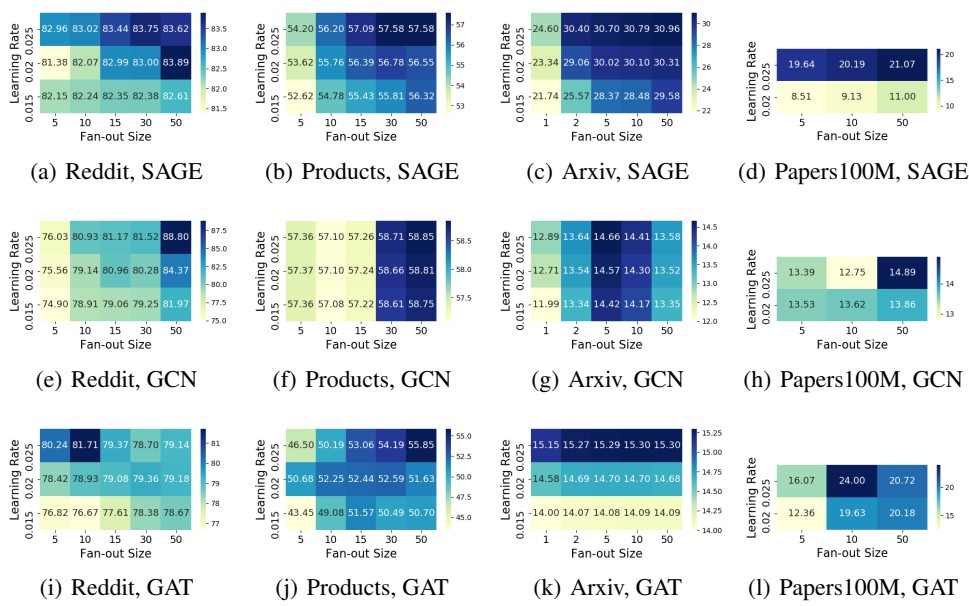

Figure 16: Test accuracy for real-world datasets for one-round GAT, GCN, GraphSAGE across different fan-out sizes and learning rates under MSE.

## N.7 Additional Runs for Key Experiments

The Tables 5-12 are as follows. We use $b$ as the batch size and $\beta$ as the fan-out size.

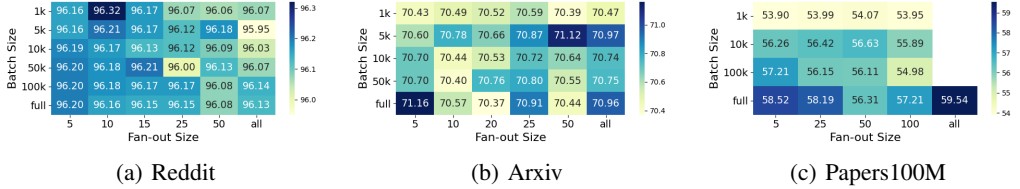

(a) Reddit      (b) Arxiv      (c) Papers100M

Figure 17: Test accuracies of multi-layer GraphSAGE trained with *CE* across varying batch sizes and fan-out sizes.

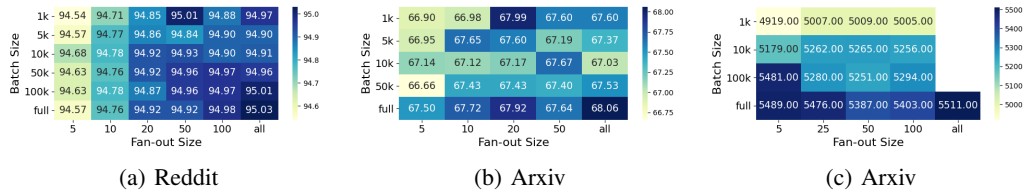

(a) Reddit      (b) Arxiv      (c) Arxiv

Figure 18: Test accuracies of multi-layer GraphSAGE trained with *MSE* across varying batch sizes and fan-out sizes.

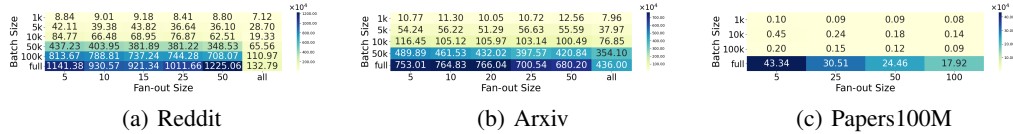

(a) Reddit      (b) Arxiv      (c) Papers100M

Figure 19: Training throughput (# nodes/s) of multi-layer GraphSAGE trained with *CE* across varying batch sizes and fan-out sizes.

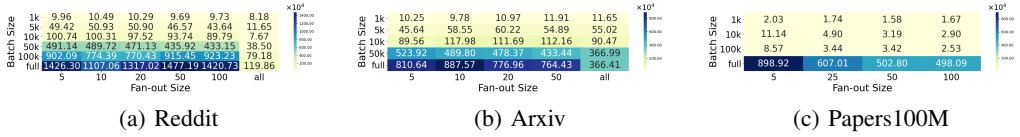

(a) Reddit      (b) Arxiv      (c) Papers100M

Figure 20: Training throughput (# nodes/s) of multi-layer GraphSAGE trained with *MSE* across varying batch sizes and fan-out sizes.

## O    RELATED WORK

For full-graph vs. mini-batch GNN training, the existing literature presents conflicting empirical findings on the GNN performance (i.e., convergence and generalization) and computational efficiency: some studies (Cai et al., 2021; Wan et al., 2022a;b; 2023) argue that full-graph training achieves higher model accuracy and faster convergence than mini-batch training, while others (Kaler et al., 2022; Zheng et al., 2022; Zhao et al., 2021; Bajaj et al., 2024) present contrasting findings. Furthermore, due to the message-passing process, performance insights from DNNs (Keskar et al., 2016; You et al., 2019; Smith, 2017; Golmant et al., 2018; Zou et al., 2020a; Bassily et al., 2018; Nabavinejad et al., 2021; Hauswald et al., 2015) cannot directly transfer to GNNs.

The only existing comparison work (Bajaj et al., 2024) between full-graph and mini-batch GNN training empirically evaluates overall performance but does not investigate the impact of key hyper-parameters (e.g., batch size and fan-out size) on model performance and computational efficiency, thereby overlooking the trade-offs achieved by tuning these hyperparameters. Recent efforts (Yuan et al., 2023; Hu et al., 2021) focus on these hyperparameters but remain limited. For instance, Yuan et al. (Yuan et al., 2023) lack theoretical support, consider only limited batch sizes and fan-out values that are far smaller than those of full-graph training, and overlook the interplay of the batch size and

Table 5: Run 1 for Figure 4(a).

| Test acc | $\beta = 5$ | $\beta = 10$ | $\beta = 15$ | $\beta = 50$ | all |
|---|---|---|---|---|---|
| $b = 1000$ | 747 | 671 | 618 | 427 | 481 |
| $b = 10000$ | 525 | 356 | 342 | 283 | 277 |
| $b = 100000$ | 416 | 308 | 291 | 245 | 232 |
| $b = 150000$ | 409 | 302 | 283 | 239 | 238 |
| full batch | 399 | 303 | 275 | 242 | 229 |

Table 6: Run 2 for Figure 4(a).

| Test acc | $\beta = 5$ | $\beta = 10$ | $\beta = 15$ | $\beta = 50$ | all |
|---|---|---|---|---|---|
| $b = 1000$ | 743 | 682 | 621 | 435 | 497 |
| $b = 10000$ | 515 | 355 | 342 | 291 | 278 |
| $b = 100000$ | 423 | 322 | 289 | 258 | 242 |
| $b = 150000$ | 417 | 315 | 287 | 246 | 240 |
| full batch | 387 | 306 | 279 | 232 | 210 |

the fan-out size. Hu et al. (Hu et al., 2021) rely on gradient variance to explain the role of batch size but do not consider fan-out size, thus their explanation conflicts with their empirical observations.

Existing theoretical analyses of GNN training typically focus on singular aspects (e.g., convergence, or generalization), overlooking key graph-related factors (e.g., irregular graphs with nodes of varying degrees, the difference between training and testing graphs in mini-batch settings) and the impact of non-linear activation on gradients. For *convergence* analysis, Yang et al. (Yang et al., 2023) and Lin et al. (Lin et al., 2023) apply the NTK framework by assuming infinite-width GNNs. Xu et al. (Xu et al., 2021) analyze multi-layer linear GNNs. Awasthi et al. (Awasthi et al., 2021) employ PL conditions to study one-round GNNs with ReLU activation, simplifying the analysis to regular graphs. All these convergence analyses are solely on full-graph training. For *generalization* analysis, full-graph GNN training has been studied (Scarselli et al., 2018; Vapnik & Chervonenkis, 2015; Garg et al., 2020; Lv, 2021; El-Yaniv & Pechyony, 2009; Oono & Suzuki, 2020; Koltchinskii, 2001; Cong et al., 2021b; Du et al., 2019; Liao et al., 2020) under the well-established frameworks (e.g., PAC-Bayesian framework (McAllester, 2003)), while the previous analyses of mini-batch training impractically assume the same graph structures used in training and testing (Tang & Liu, 2023; Verma & Zhang, 2019). The difference among graph structures in training and testing can result in generalization performance degradation or overfitting to graph structures used in training.

## P    EXTENSIONS AND FUTURE WORK

### P.1    EXTENSIONS

**Multi-layer GNN models in theoretical analysis.**    We focus on a one-layer GNN with ReLU activation in theoretical analysis. We discuss the extension of theoretical results to multi-layer settings in Appendix H, and conduct experiments using multi-layer GNNs in Sec 5 and Appendix N. The results validate that our key insights remain applicable in such settings. Therefore, our theoretical and empirical analyses support the multi-layer GNN settings.

**Sampling methods.**    We focus on uniform neighbor sampling before mini-batch training. There exist many other sampling methods (Hamilton et al., 2017; Chen et al., 2018; Zou et al., 2019; Chiang et al., 2019; Zeng et al., 2019) that have been proposed at the layer- or subgraph-level to enhance performance. Our core insights could extend to more sampling methods.

For example, compared to uniform neighbor sampling, the key difference in some advanced samplers lies in introducing specific constraints on the effective fan-out size by either assigning non-uniform sampling probabilities (Chen et al., 2018), or imposing layer-wise upper bounds on the number of neighbors per node (Zou et al., 2019). These specific constraints preserve the qualitative trend of the amount of aggregated information per node in the message-passing when varying fan-out sizes. In convergence analysis, following our analysis in Appendix G, increasing the effective fan-out size

Table 7: Run 1 for Figure 4(e).

| Test acc | $\beta = 5$ | $\beta = 10$ | $\beta = 20$ | $\beta = 50$ | all |
|---|---|---|---|---|---|
| $b = 1000$ | 1167 | 928 | 854 | 817 | 801 |
| $b = 10000$ | 1232 | 1028 | 991 | 907 | 861 |
| $b = 100000$ | 1250 | 1025 | 1005 | 919 | 902 |
| $b = 150000$ | 1256 | 1047 | 1013 | 928 | 909 |
| full batch | 1295 | 1035 | 1007 | 945 | 925 |

Table 8: Run 2 for Figure 4(e).

| Test acc | $\beta = 5$ | $\beta = 10$ | $\beta = 20$ | $\beta = 50$ | all |
|---|---|---|---|---|---|
| $b = 1000$ | 1169 | 943 | 872 | 809 | 787 |
| $b = 10000$ | 1222 | 1016 | 993 | 923 | 847 |
| $b = 100000$ | 1257 | 943 | 936 | 929 | 886 |
| $b = 150000$ | 1230 | 1037 | 978 | 923 | 902 |
| full batch | 1279 | 998 | 946 | 938 | 927 |

can enrich each target node's aggregated neighbors, improving embeddings and reducing gradient variance. Therefore, the mechanism "larger fan-out size $\rightarrow$ more iterations to convergence" still holds in GNN training under these samplers. For generalization, a larger fan-out size can reduce the Wasserstein distance $\Delta(\beta, b)$ under these constraints, which leads to improved generalization. While these advanced samplers may lessen the sensitivity of generalization to fan-out size, they cannot completely eliminate the effect of including unsampled but valid edges as fan-out increases (see Obs. 2). Consequently, generalization remains more sensitive to fan-out size than to batch size. Overall, our key insights remain applicable to these sampling methods.

On the other hand, we notice that some advanced works (Chen et al., 2017; Shi et al., 2023; Fey et al., 2021; Shi et al., 2025) use historical embeddings to incorporate nearly full-graph information at each iteration. Therefore, from a model performance perspective, these methods reduce the variance caused by different batch sizes and behave more like full-graph training. From a system design perspective, they also rely on additional memory to store historical embeddings, making them closer to full-graph training systems than typical mini-batch ones. In contrast, we preserve and study the effects of batch size and fan-out, rather than eliminating them. Hence, we adopt the standard neighbor-aggregation scheme that is commonly used in practice and do not consider these sampling methods.

**Link prediction tasks.** We focus on node classification tasks in GNN training, which can be easily extended to graph classification. Different from node classification, link prediction tasks use node pairs (connected and unconnected) for edge prediction, which can be transformed to node classification tasks using the line graph method in the graph theory. The new line graph $L(G)$ is constructed in the following way: for each edge in the original graph $G$, make a vertex in $L(G)$; for every two edges in $G$ that have a vertex in common, make an edge between their corresponding vertices in $L(G)$. Hence, our analyses and core insights naturally carry over to link prediction tasks.

**Graph classification tasks.** A graph-level prediction is obtained by first learning node representations and then pooling them into a single graph representation. Since the training dynamics before pooling are the same as in node classification, our analyses and key insights naturally carry over.

**Inductive GNN tasks.** We focus on transductive GNN tasks. Unlike transductive tasks, inductive tasks apply different graphs between testing and training. For convergence, our analysis can be applied to inductive tasks without considering the testing graphs. For generalization, our analysis can be easily extended to inductive tasks by revising $\delta_{i,j}^{\text{full}}$ in the Wasserstein distance to consider graph structure differences between testing and training graphs.

P.2 FUTURE WORK

**Different activations: GeLU and Tanh.** Our theoretical analysis readily extends to the GeLU and Tanh functions as the activation under our settings. The key difference lies in how the activation

Table 9: Run 1 for Figure 6(a).

| Test acc | $\beta = 5$ | $\beta = 15$ | $\beta = 25$ | $\beta = 50$ | all |
|---|---|---|---|---|---|
| $b = 1000$ | 0.7767 | 0.7832 | 0.7821 | 0.7810 | 0.7789 |
| $b = 5000$ | 0.7817 | 0.7846 | 0.7825 | 0.7803 | 0.7698 |
| $b = 10000$ | 0.7851 | 0.7818 | 0.7812 | 0.7775 | 0.7713 |
| $b = 100000$ | 0.7869 | 0.7823 | 0.7818 | 0.7783 | 0.7753 |
| $b = 150000$ | 0.7852 | 0.7818 | 0.7809 | 0.7781 | 0.7761 |
| full batch | 0.7868 | 0.7810 | 0.7778 | 0.7778 | 0.7803 |

Table 10: Run 2 for Figure 6(a).

| Test acc | $\beta = 5$ | $\beta = 15$ | $\beta = 25$ | $\beta = 50$ | all |
|---|---|---|---|---|---|
| $b = 1000$ | 0.7793 | 0.7840 | 0.7820 | 0.7818 | 0.7792 |
| $b = 5000$ | 0.7825 | 0.7842 | 0.7833 | 0.7817 | 0.7702 |
| $b = 10000$ | 0.7852 | 0.7821 | 0.7818 | 0.7771 | 0.7713 |
| $b = 100000$ | 0.7862 | 0.7825 | 0.7816 | 0.7780 | 0.7762 |
| $b = 150000$ | 0.7860 | 0.7818 | 0.7800 | 0.7768 | 0.7760 |
| full batch | 0.7864 | 0.7808 | 0.7778 | 0.7775 | 0.7808 |

affects the gradient norm bound. GeLU is a smooth approximation of ReLU and shares a similar upper bound, while Tanh is even smoother with bounded high-order derivatives that control the gradient norm. As a result, both our convergence and generalization methodology naturally translate to these activation functions.

Our core insights are clearly generalizable to GeLU due to its similarity with ReLU. However, whether the same insights hold for Tanh is less certain, as its bounded and more intricate derivative structure may affect the theoretical bounds in a nontrivial way.

**Heterogeneous graphs.** Different from homogeneous graphs, heterogeneous graphs require specialized handling to address different types of nodes and edges, involving distinct aggregation and transformation functions for each type, such as using separate neural networks for different edge types. This can be explored.

Table 11: Run 1 for Figure 6(b).

| Test acc | $\beta = 5$ | $\beta = 10$ | $\beta = 20$ | $\beta = 50$ | all |
|---|---|---|---|---|---|
| $b = 1000$ | 0.6617 | 0.6891 | 0.7117 | 0.7241 | 0.7242 |
| $b = 5000$ | 0.7113 | 0.7207 | 0.7336 | 0.7345 | 0.7369 |
| $b = 10000$ | 0.7209 | 0.7292 | 0.7341 | 0.7344 | 0.7362 |
| $b = 100000$ | 0.7318 | 0.7348 | 0.7373 | 0.7403 | 0.7415 |
| $b = 150000$ | 0.7329 | 0.7357 | 0.7372 | 0.7378 | 0.7401 |
| full batch | 0.7345 | 0.7391 | 0.7386 | 0.7384 | 0.7385 |

Table 12: Run 2 for Figure 6(b).

| Test acc | $\beta = 5$ | $\beta = 10$ | $\beta = 20$ | $\beta = 50$ | all |
|---|---|---|---|---|---|
| $b = 1000$ | 0.7295 | 0.7321 | 0.7344 | 0.7345 | 0.7341 |
| $b = 5000$ | 0.7307 | 0.7343 | 0.7361 | 0.7364 | 0.7371 |
| $b = 10000$ | 0.7326 | 0.7353 | 0.7366 | 0.7365 | 0.7381 |
| $b = 100000$ | 0.7342 | 0.7372 | 0.7392 | 0.7400 | 0.7411 |
| $b = 150000$ | 0.7343 | 0.7361 | 0.7385 | 0.7393 | 0.7405 |
| full batch | 0.7341 | 0.7396 | 0.7391 | 0.7389 | 0.7403 |

