# OpenReview forum: "Full-Graph vs. Mini-Batch Training: Comprehensive Analysis from a Batch Size and Fan-Out Size Perspective"
_ICLR.cc/2026/Conference — ICLR 2026 Poster_

### Official Review · Reviewer_463G · 2025-10-27

**Soundness:** 3
**Presentation:** 2
**Contribution:** 2
**Rating:** 4
**Confidence:** 4

**Summary:**

This paper investigates mini-batch training for GNNs. It provides theoretical analysis on both convergence and generalization within the mini-batch training framework, and conducts experiments to validate the proposed theory.

**Strengths:**

S1: Comprehensive theoretical analysis.

S2: The choice of datasets and models are reasonable.

S3: Results make sense.

**Weaknesses:**

W1: The results are somewhat plain. The first notable conclusion is that when trained with the MSE loss, larger batch size leads to slower convergence. But empirically, this is validated using node classification datasets. The second claim is that "mini-batch could leads to better generalization than full-graph", but in Fig. 4, the influence of batch size seems marginal. Reducing fan-out size is useful, but to the best of my knowledge, community is already aware of that, due to those DropEdge papers.

W2: Presentation could be further improved. Lots of statements in the experiments section refer to the Appendix N. If the information in the appendix is important enough, it should be presented in the main paper. Defer them to the appendix but continually mention them will not save space. For example, line 363, line 415, figures in the appendix is mentioned before figures in the main paper.

**Questions:**

Q1: May I ask what is the target loss in iteration-to-loss, the target accuracy in iteration-to-accuracy and time-to-accuracy? I try to look them in appendix N, but there is only description, not the actual value. If we change these value, will the conclusion change?

Q2: What is the setting and hyperparameters of Fig. 4(a)?

Q3: In the mini-batch training, different batch sizes means different amount of samples the model saw after one iteration. Is it fair to compare the number of iteration in such context?

Q4: The most relevant work is [1]. Could you briefly summarize the most important new insight of your paper in comparison to [1] (except using iteration-to-accuracy to avoid hardware influence)?

[1] Bajaj, Saurabh, et al. "Graph Neural Network Training Systems: A Performance Comparison of Full-Graph and Mini-Batch." Proceedings of the VLDB Endowment 18.4 (2024): 1196-1209.

---

> ### Author Response · Authors · 2025-11-21
> **Response (1/5)**
>
> ## W1: Difference between our work and the general understanding of training analyses
>
> Thank you for the thoughtful question. Below we clarify the key gaps in existing research, how our work fills these gaps, and how our theoretical findings connect to your observations (e.g., Fig. 4).
>
> As acknowledged by other reviewers, our main contributions consist of:
>
> - We provide the **first unified GNN analysis** that jointly characterizes the interplay between **batch size** and **fan-out size**.
> - We develop **theoretical settings that better reflect practical GNN training**, including irregular graphs, non-linear activations, and graph distribution shifts.
> - We uncover **non-isotropic effects** of batch size and fan-out size on **convergence** and **generalization**.
> - We offer **practical hyperparameter-tuning guidelines** and insights for **training paradigm selection**.
>
> Below we elaborate on the differences between our work and prior training analyses.
>
> ### 1). Existing gaps.
>
> Our goal is to systematically compare **full-graph vs. mini-batch training** by analyzing how **batch size** and **fan-out size** affect GNN optimization and generalization. However, existing analyses fall short in three main ways.
>
> **i). Conventional DNN training analyses have no concept of fan-out size**, as this concept arises specifically from neighbor sampling on graph-structured data.
>
> **ii). General GNN training analyses largely overlook the interplay between fan-out size and batch size**. For example, existing studies typically focus on ***individual parameters*** (e.g., batch size or fan-out size independently) [2,3] or ***singular aspects of evaluation*** (e.g., convergence or accuracy) [4,5], providing limited insights into the holistic trade-offs between full-graph and mini-batch paradigms.
>
> **iii). Most existing GNN theoretical analyses typically rely on strong simplifications**, such as infinite-width assumptions that average out per-neuron gradient noise [6] or linear models with convex losses that remove local optima [7], which obscure the effects of batch sizes or fan-out sizes on training dynamics.
>
> ### 2). Our main contributions
>
> Our main novelties are summarized as follows.
>
> **i). A unique GNN analysis of the interplay between batch size and fan-out size**. We characterize the roles of batch size and fan-out size in GNN optimization dynamics (Theorem 1 and 2) and generalization (Theorem 3), thereby addressing the existing gaps identified in points (i) and (ii) above. Therefore, **this analysis provides a principled and unified framework for understanding the GNN behaviors under full-graph and mini-batch training paradigms.**
>
> **ii). Theoretical settings that better reflect practical GNN training**. For convergence, we extend the settings to irregular graphs and GNNs with non-linear activations. For generalization, we use the Wasserstein distance to investigate the impact of graph structures, especially the fan-out size, where this distance can also measure graph structure differences between training and testing datasets. **These settings enable a sharper analysis to precisely characterize how the batch size and fan-out size affect training**, thereby addressing the existing gap identified in points (iii).
>
> **iii). Key observations**: We uncover **the non-isotropic impacts of the batch size and the fan-out size** in GNN convergence and generalization, where the batch size has a greater impact on GNN optimization dynamics (Obs.1), while the fan-out size more strongly affects GNN eneralization (Obs.2). **These findings cannot be captured by prior analyses that consider batch size and fan-out size independently, and provide practical guidelines in hyperparameter tuning and training paradigm selection.**

---

> > ### Author Response · Authors · 2025-11-21
> > **Response (2/5)**
> >
> > Furthermore, we would like to clarify your questions as below.
> >
> > - **Answering “in Fig. 4, the influence of batch size seems marginal”**: This observation is consistent with Obs.2 in Sec.4.2 and the performance degradation analysis in Sec 5.3 (i.e., Performance degradation is more severe with fan-out sizes than with batch sizes.). Therefore, ***your observation actually supports our main theoretical and empirical findings**.*
> > - **DropEdge vs. Reducing fan-out size:** We would like to clarify that ***DropEdge and reducing fan-out size are two different concepts*** in both their mechanism and applicable training paradigms. For the mechanism, DropEdge randomly sparsifies the entire graph structure (i.e., globally drops edges), whereas reducing the fan-out size means each training node samples neighbors without changing the entire graph structure. For example, in an undirected graph, if DropEdge removes the edge between nodes a and b, they are no longer one-hop neighbors; in contrast, reducing the fan-out size may result in node a not sampling node b as a neighbor, while node b could still sample node a. For applicable training paradigms, ***DropEdge analysis [8] primarily focuses on full-graph training*** to mitigate the over-smoothing problem, and ***thus does not address this comparison between full-graph and mini-batch training***. Therefore, ***DropEdge cannot serve as a substitute for studying the impact of fan-out size*** or batch size when comparing two paradigms.
> >
> > Therefore, different from the prior works, our work considers the interplay between batch size and fan-out size, provides theoretical analyses in practical settings, uncovers the isotropic impacts of these two hyperparameters, and offers practical guidelines.

---

> > > ### Author Response · Authors · 2025-11-21
> > > **Response (3/5)**
> > >
> > > ## W2: Figure presentation.
> > >
> > > Thanks for your suggestion. We would like to clarify that these referenced figures in Appendix N are largely supplementary and are not directly related to the convergence behaviors or generalization performance degradation analysis for multi-layer GNNs in the main text. For this reason, we originally placed them in the appendix to avoid redundancy.
> > >
> > > We have moved Figures 5(j), 6(j), 7(i), 8(j), 13(b) and 14(b) in the original version into the main text as Figures 2-3 in the new version to improve readability. We believe this is a minor presentation issue and does not undermine the core contributions of our work.
> > >
> > > ## Q1: Convergence settings.
> > >
> > > Thank you for the diligent review.
> > >
> > > We are sorry for forgetting to include specific values of target losses and target accuracies in the the first space. In the revised paper, we have added these values for three-layer GNNs in Appendix N.
> > >
> > > For reference, we also report them here: the target losses are [0.2, 0.1, 0.8, 1.52] under CE, and [0.005, 0.005, 0.013, 0.0055] under MSE for the products, reddit, arXiv, and papers100M datasets, respectively. The corresponding target accuracies are [0.918, 0.962, 0.708, 0.599] under CE, and [0.89, 0.946, 0.676, 0.5] under MSE for the same datasets.
> > >
> > > We agree that providing these details is important for transparency and reproducibility. Therefore, we also provide the code at the github repository in the abstract.
> > >
> > > Thank you again for pointing out this missed detail.
> > >
> > > ## Q2: Setting and hyperparameters of Fig. 4(a).
> > >
> > > Thanks for your question. We have reported the settings of Fig.4(a) in Appendix N (line 3347, 3393-3395, 3495-3496 in the original version).
> > >
> > > For reference, we also report them here: For Fig.4(a), we use a three-layer GraphSAGE model with hidden dimension of 256 for the ogbn-products dataset. The activation function is ReLU function. The optimizer is Adam with a learning rate of 0.001 and a weight decay of 0. The maximal number of iterations is 10000. The test accuracy is reported at the iteration corresponding to the best validation accuracy.
> > >
> > > For reproducibility, we provide the code at the github repository in the abstract.
> > >
> > > ## Q3: Iteration vs. Epoch.
> > >
> > > Thanks for your question. We would clarify that comparing the number of iterations is standard and appropriate in the context of convergent analysis with mini-batch training [5,9]. Specifically, ***each iteration corresponds to one stochastic gradient estimate followed by one parameter update***, regardless of the batch size. We would like to point out, if the mini-batch is constructed with sampling with replacement, each iteration may draw a potentially overlapping set of training nodes, and there is no fixed division of the dataset into complete passes; thereby, the concept of an epoch will not strictly exist in theory. Hence, the iteration serves as the fundamental unit of optimization progress.
> > >
> > > In addition, since our goal is to investigate how batch size and fan-out size influence the optimization dynamics, evaluating convergence per iteration provides a theoretically grounded measure. In contrast, using the number of epochs would confound the effect of the batch size and the number of gradient updates, making it difficult to isolate the impact of batch size on convergence behavior.

---

> > > > ### Author Response · Authors · 2025-11-21
> > > > **Response (4/5)**
> > > >
> > > > ## Q4: Difference between our work and [1]
> > > >
> > > > Thanks for your question. We would like to clarify that the difference between our work and the related paper [1] from our theoretical contributions and practical guidelines.
> > > >
> > > > ### 1). Theoretical contributions.
> > > >
> > > > **i). Theoretical characterization beyond empirical observations**. We characterize the roles of batch size and fan-out size in GNN optimization dynamics (Theorems 1 and 2) and generalization (Theorem 3). In contrast, ***the empirical results provided by [1] are largely observational and hardware- or environment-dependent***, limiting their generalizability. For example, the convergence time may flip conclusions depending on hardware (see discussion in Sec.5.1).
> > > >
> > > > Our analytical framework explicitly ***captures the interplay between batch size and fan-out size in GNN optimization dynamics***, which allows us to reveal the non-isotropic impacts of these two hyperparameters as follows.
> > > >
> > > > **ii). Key distinction**. We uncover ***the non-isotropic impacts of the batch size and the fan-out size in GNN convergence and generalization***, where the batch size has a greater impact on GNN optimization dynamics (Obs.1), while the fan-out size more strongly affects GNN eneralization (Obs.2). In contrast, ***[1] solely evaluates overall performance*** and ignores the interplay between batch size and fan-out size***,*** making it ***difficult to identify better trade-offs*** (e.g., faster convergence or improved generalization) under resource constraints. Therefore, ***[1] cannot provide practical guidelines for hyperparameter tuning or training paradigm selection*** under specific requirements.
> > > >
> > > > Our key observations can provide valuable practical guidelines as below.
> > > >
> > > > ### 2). Practical guidelines
> > > >
> > > > **i). Hyperparameter tuning**
> > > >
> > > > 1. **Insights for tuning hyperparameters.** For datasets with an average degree below 50, we suggest keeping the batch size below half of the training nodes and the fan-out size under 15 to avoid generalization degradation while balancing computational efficiency and model performance. In addition, the hardware-agnostic iteration-to-accuracy is more reliable to guide **early-stage configuration decisions** in practice than time-to-accuracy. For example, in a new hardware setup, practitioners can refer to known iteration-to-accuracy trends to narrow down the promising range of batch size and fan-out size, and perform only short runs to consider hardware-specific runtime, which reduces tuning overhead.
> > > > 2. **Contribution to adaptive optimizer.** Our findings also suggest opportunities for developing adaptive optimizers that dynamically adjust hyperparameter configurations during GNN training. For example, an adaptive fan-out controller could allocate larger fan-out sizes in the early stages to accelerate convergence and gradually reduce them to moderate levels later to improve generalization. Meanwhile, an adaptive batch-size controller may adjust the batch size based on whether computational efficiency or generalization is prioritized under a given resource budget.
> > > >
> > > > **ii). Training paradigm selection.** Our theoretical and empirical findings support that full-graph training does not always yield superior model performance or computational efficiency compared to smaller mini-batch settings. Therefore, under resource constraints, carefully tuning the batch size and fan-out size in mini-batch settings often leads to better trade-offs, such as faster convergence or improved generalization, than full-graph training.
> > > >
> > > > Therefore, compared to [1], we can provide theoretical characterization, uncover the isotropic-impacts of batch size and fan-out size in convergence and generalization, and thereby offer actionable guidelines.
> > > >
> > > > We will incorporate the above discussion in the revision.
> > > >
> > > > Thank you again for your diligent review, and we hope our response has satisfactorily addressed your questions and concerns.

---

> > > > > ### Author Response · Authors · 2025-11-21
> > > > > **Response (5/5)**
> > > > >
> > > > > [1] Bajaj, Saurabh, et al. "Graph Neural Network Training Systems: A Performance Comparison of Full-Graph and Mini-Batch." Proceedings of the VLDB Endowment 18.4 (2024): 1196-1209.
> > > > >
> > > > > [2] Yaochen Hu, Amit Levi, Ishaan Kumar, Yingxue Zhang, and Mark Coates. On batch-size selection
> > > > > for stochastic training for graph neural networks. 2021.
> > > > >
> > > > > [3] Hao Yuan, Yajiong Liu, Yanfeng Zhang, Xin Ai, Qiange Wang, Chaoyi Chen, Yu Gu, and Ge Yu. Comprehensive evaluation of gnn training systems: A data management perspective. arXiv preprint arXiv:2311.13279, 2023.
> > > > >
> > > > > [4] Huayi Tang and Yong Liu. Towards understanding generalization of graph neural networks. In
> > > > >
> > > > > International Conference on Machine Learning, pp. 33674–33719, 2023.
> > > > >
> > > > > [5] Pranjal Awasthi, Abhimanyu Das, and Sreenivas Gollapudi. A convergence analysis of gradient descent on graph neural networks. Advances in Neural Information Processing Systems, 34: 20385–20397, 2021.
> > > > >
> > > > > [6] Naganand Yadati. A convex formulation for graph convolutional training: Two layer case. In 2022 IEEE International Conference on Data Mining (ICDM), pp. 1281–1286. IEEE, 2022.
> > > > >
> > > > > [7] Yucong Lin, Silu Li, Jiaxing Xu, Jiawei Xu, Dong Huang, Wendi Zheng, Yuan Cao, and Junwei Lu. Graph over-parameterization: Why the graph helps the training of deep graph convolutional network. Neurocomputing, 534:77–85, 2023.
> > > > >
> > > > > [8] Rong, Y., Huang, W., Xu, T. and Huang, J., DropEdge: Towards Deep Graph Convolutional Networks on Node Classification. In International Conference on Learning Representations.
> > > > >
> > > > > [9] Difan Zou, Yuan Cao, Dongruo Zhou, and Quanquan Gu. Stochastic gradient descent optimizes
> > > > >
> > > > > over-parameterized deep relu networks. arXiv preprint arXiv:1811.08888, 2018.

---

### Official Review · Reviewer_VT8k · 2025-10-30

**Soundness:** 3
**Presentation:** 3
**Contribution:** 2
**Rating:** 4
**Confidence:** 2

**Summary:**

This paper aims to provide a systematic and comprehensive study on comparing the full-graph vs. mini-batch GNN training, focusing on the impact of batch size and fan-out size. Theoretical analysis shows that increasing batch size slows convergence under MSE but accelerates it under CE, while increasing fan-out size consistently improves convergence. The provided theories show batch size has a greater impact on GNN optimization dynamics, while the fan-out size more strongly affects GNN generalization. Empirical results validate these findings, using a hardware-agnostic iteration-to-accuracy metric. Experiments show that moderate fan-out size balances speed and efficiency, and overly large batch size and fan-out size degrades accuracy. Overall, well-tuned mini-batch training can match or surpass full-graph training in both performance and computational efficiency.

**Strengths:**

1. Both main observations—(Obs.1) convergence is more sensitive to batch size, and (Obs.2) generalization is more sensitive to fan-out size - are consistently supported by theoretical derivations and empirical results across multiple datasets and GNN models.
2. The paper addresses a practically important topic - understanding the differences between full-graph and mini-batch GNN training， which is beneficial to large-scale GNN training.
3. Code is provided for reproducibility.

**Weaknesses:**

1. The study only covers node-wise sampling; other major mini-batch paradigms (e.g., graph-level and layer-wise sampling) are omitted, which limits the comprehensiveness of the conclusions.
2. The main conclusion—that full-graph training does not always outperform well-tuned mini-batch training—has already been reported in prior empirical works (e.g., Bajaj et al., 2024). The novelty thus lies mainly in providing a formal theoretical framing rather than discovering a new phenomenon.

**Questions:**

1. Scope of Mini-Batch Paradigms:
The paper focuses on node-wise neighbor sampling when discussing mini-batch training. However, mini-batch GNNs also include subgraph-level (e.g., GraphSAINT, Cluster-GCN) and layer-wise (e.g., LADIES) sampling approaches, which may exhibit different trade-offs between variance, memory, and structural bias. Could the authors clarify whether their theoretical framework—especially the convergence and Wasserstein-based generalization analysis—can extend to graph-level or layer-wise mini-batching? Have the authors considered running at least small-scale comparisons (e.g., Cluster-GCN vs. neighbor sampling) to test whether the same sensitivity patterns still hold under these paradigms?


2. Conceptual Contribution and Practical Value:
The concluding message—“full-graph training does not always outperform well-tuned mini-batch training”—has been noted in prior empirical studies (e.g., Bajaj et al., 2024). What remains unclear is the new insight or practical benefit beyond formalizing known intuitions. The two key observations (Obs.1: convergence more sensitive to batch size; Obs.2: generalization more sensitive to fan-out size) are interesting, but they seem to reaffirm what many practitioners already do during hyper-parameter tuning. Could the authors elaborate on what new guidance or algorithmic implications their framework provides? How does this work go beyond offering “interpretive understanding” and translate into actionable or theoretical advancement for future GNN training design?

---

> ### Author Response · Authors · 2025-11-21
> **Response (1/4)**
>
> ## W1 and Q1: Extension to other mini-batch paradigms
>
> Thank you for the interesting question. We would like to briefly explain why we use node-level sampling settings and how our analysis extend to other mini-batch paradigms.
>
> ### 1). Justification for using node-level paradigms.
>
> We would like to clarify that node-level neighbor sampling is the most common mini-batch paradigm in the existing GNN studies (e.g., empirical benchmarks [1,3], theoretical analysis [3,8], system designs [9]).
>
> In addition, node-level neighbor sampling is also the most fundamental and unique paradigm. This is because layer-level and graph-level sampling methods are structured variants built on top of node-level sampling, relying on ***two elements*** that make full graph training a special case: **the batch size** (i.e., the number of training nodes per iteration) and **the effective fan-out size** (i.e., the effective number of neighbors chosen per node).
>
> For example, LADIES, a layer-level sampling method, draws neighbors from the union of the neighborhoods of all training nodes in a layer and uses the *lad-out size* as the upper bound on the number of neighbors sampled for that layer. Therefore, the effective fan out size can be roughly viewed as the lad-out size divided by the batch size. ClusterGCN, a graph-level sampling method, partitions the graph into multiple subgraphs. Increasing the number of partitions cuts more inter-subgraph edges, which leaves each subgraph with fewer preserved neighbors. Therefore, the effective fan out size can be roughly viewed as the training data size divided by the total number of partitions. ***The table below summarizes the concepts of batch size and effective fan out size with different samplers.***
>
> | node-level | batch size | fan-out size |
> | --- | --- | --- |
> | layer-level | batch size | lad-out size/batch size |
> | graph-level | number of partitions per iteration/total number of partitions* training datasize | training datasize/number of partitions |
>
> Therefore, our focus on node-level sampling is both representative of mainstream GNN training and well-suited to reveal the core factors of mini-batch paradigms.
>
> ### 2). Extension to other paradigms
>
> **i). Theoretical extension**.  Our theoretical analysis readily extends to layer-level and graph-level sampling methods. Specifically, the key difference between these two samplers and node-level sampler lies in introducing specific constraints on the effective fan-out size through non-uniform sampling probabilities. In convergence analysis, we bound the gradient norm using the amount of information aggregated during message passing. In generalization, we bound the performance using the Wasserstein distance, which is proportional to the structural difference between full-graph and mini-batch graphs (See discussion in Sec.4.2).
>
> Therefore, ***our theoretical framework and key insights can translate naturally to layer-level and graph-level samplers***.
>
> **ii). Empirical extension**: Thanks for your suggestion. We conduct experiments using LADIES and ClusterGCN below and vary the batch sizes $b$ and effective fan-out sizes $\beta$. For these experiments, we used a three-layer GraphSAGE model with ReLU function on the ogbn-product dataset under both CE and MSE losses.
>
> Due to the same dataset used in experiments with ClusterGCN, we use the number of partitions per iteration/total number of partitions to denote the batch size $b$ and use 1/number of partitions to denote the effective fan-out size $\beta$.
>
> ***The results below support our main observations in the paper***: 1. A larger batch size leads to opposite convergence trends under MSE and CE, while increasing the fan-out size exhibits a consistent trend. 2. Increasing both the fan-out size and the batch size generally improves the generalization performance.

---

> > ### Author Response · Authors · 2025-11-21
> > **Response (2/4)**
> >
> > ClusterGCN:
> >
> > CE loss:
> >
> > | Iter-to-loss | $\beta$: 1/8192 | $\beta$: 1/1024 | $\beta$: 1/128 |
> > | --- | --- | --- | --- |
> > | b: 1/16 | 5754 | 5741 | 4922 |
> > | b: 1/8 | 4570 | 4540 | 4414 |
> > | b: 1/4 | 4074 | 3930 | 4172 |
> >
> > | Iter-to-acc | $\beta$: 1/8192 | $\beta$: 1/1024 | $\beta$: 1/128 |
> > | --- | --- | --- | --- |
> > | b: 1/16 | 1500 | 1140 | 1220 |
> > | b: 1/8 | 870 | 810 | 740 |
> > | b: 1/4 | 540 | 630 | 410 |
> >
> > | Time-to-acc | $\beta$: 1/8192 | $\beta$: 1/1024 | $\beta$: 1/128 |
> > | --- | --- | --- | --- |
> > | b: 1/16 | 26.73 | 20.80 | 25.29 |
> > | b: 1/8 | 28.13 | 26.24 | 25.55 |
> > | b: 1/4 | 32.03 | 36.25 | 26.39 |
> >
> > | Test acc | $\beta$: 1/8192 | $\beta$: 1/1024 | $\beta$: 1/128 |
> > | --- | --- | --- | --- |
> > | b: 1/16 | 0.7464 | 0.7500 | 0.7585 |
> > | b: 1/8 | 0.7485 | 0.7553 | 0.7583 |
> > | b: 1/4 | 0.7577 | 0.7584 | 0.7585 |
> >
> > MSE loss:
> >
> > | Iter-to-loss | $\beta$: 1/8192 | $\beta$: 1/1024 | $\beta$: 1/128 |
> > | --- | --- | --- | --- |
> > | b: 1/16 | 7498 | 5334 | 3003 |
> > | b: 1/8 | 8076 | 5777 | 3849 |
> > | b: 1/4 | 8040 | 6181 | 4188 |
> >
> > | Iter-to-acc | $\beta$: 1/8192 | $\beta$: 1/1024 | $\beta$: 1/128 |
> > | --- | --- | --- | --- |
> > | b: 1/16 | 8300 | 6230 | 4090 |
> > | b: 1/8 | 7650 | 7100 | 6780 |
> > | b: 1/4 | 6110 | 5120 | 5100 |
> >
> > | Time-to-acc | $\beta$: 1/8192 | $\beta$: 1/1024 | $\beta$: 1/128 |
> > | --- | --- | --- | --- |
> > | b: 1/16 | 148.48 | 155.96 | 80.53 |
> > | b: 1/8 | 227.33 | 242.48 | 232.19 |
> > | b: 1/4 | 343.11 | 302.14 | 317.10 |
> >
> > | Test acc | $\beta$: 1/8192 | $\beta$: 1/1024 | $\beta$: 1/128 |
> > | --- | --- | --- | --- |
> > | b: 1/16 | 0.7159 | 0.7249 | 0.6999 |
> > | b: 1/8 | 0.7189 | 0.7357 | 0.7279 |
> > | b: 1/4 | 0.7333 | 0.7379 | 0.7333 |
> >
> > LADIES:
> >
> > CE loss:
> >
> > | Iter-to-loss | $\beta$=5 | $\beta$=10 | $\beta$=15 |
> > | --- | --- | --- | --- |
> > | b=1000 | 6701 | 3743 | 2722 |
> > | b=10000 | 1280 | 803 | 680 |
> > | b=10000 | 568 | 502 | 427 |
> >
> > | Iter-to-acc | $\beta$=5 | $\beta$=10 | $\beta$=15 |
> > | --- | --- | --- | --- |
> > | b=1000 | 7630 | 4830 | 3420 |
> > | b=10000 | 1400 | 1250 | 1060 |
> > | b=10000 | 1210 | 1040 | 800 |
> >
> > | Time-to-acc | $\beta$=5 | $\beta$=10 | $\beta$=15 |
> > | --- | --- | --- | --- |
> > | b=1000 | 82.04 | 50.02 | 35.52 |
> > | b=10000 | 15.21 | 14.17 | 12.16 |
> > | b=10000 | 14.82 | 13.51 | 11.56 |
> >
> > | Test acc | $\beta$=5 | $\beta$=10 | $\beta$=15 |
> > | --- | --- | --- | --- |
> > | b=1000 | 0.7765 | 0.7874 | 0.7903 |
> > | b=10000 | 0.7826 | 0.7825 | 0.7829 |
> > | b=10000 | 0.7838 | 0.7813 | 0.7800 |
> >
> > MSE loss:
> >
> > | Iter-to-loss | $\beta$=5 | $\beta$=10 | $\beta$=15 |
> > | --- | --- | --- | --- |
> > | b=1000 | 5566 | 4644 | 2660 |
> > | b=10000 | 8312 | 4980 | 3081 |
> > | b=10000 | 9354 | 7594 | 5128 |
> >
> > | Iter-to-acc | $\beta$=5 | $\beta$=10 | $\beta$=15 |
> > | --- | --- | --- | --- |
> > | b=1000 | 6230 | 5020 | 4290 |
> > | b=10000 | 4850 | 3460 | 3120 |
> > | b=10000 | 5690 | 4160 | 3610 |
> >
> > | Time-to-acc | $\beta$=5 | $\beta$=10 | $\beta$=15 |
> > | --- | --- | --- | --- |
> > | b=1000 | 64.28 | 52.07 | 46.13 |
> > | b=10000 | 53.16 | 38.92 | 35.99 |
> > | b=10000 | 64.61 | 49.12 | 45.80 |
> >
> > | Test acc | $\beta$=5 | $\beta$=10 | $\beta$=15 |
> > | --- | --- | --- | --- |
> > | b=1000 | 0.7070 | 0.7276 | 0.7105 |
> > | b=10000 | 0.7144 | 0.7634 | 0.7269 |
> > | b=10000 | 0.7568 | 0.7184 | 0.7200 |
> >
> > We will incorporate the above discussion in the revision.

---

> > > ### Author Response · Authors · 2025-11-21
> > > **Response (3/4)**
> > >
> > > ## W2 and Q2: Clarification on novelty and contributions
> > >
> > > Thanks for your question. We would like to clarify that ***our primary goal is to compare full-graph and mini-batch training paradigms by understanding how batch size and fan-out size affect GNN convergence and generalization.***
> > >
> > > As acknowledged by other reviewers, our main contributions consist of:
> > >
> > > - We provide the **first unified GNN analysis** that jointly characterizes the interplay between **batch size** and **fan-out size**.
> > > - We develop **theoretical settings that better reflect practical GNN training**, including irregular graphs, non-linear activations, and graph distribution shifts.
> > > - We uncover **non-isotropic effects** of batch size and fan-out size on **convergence** and **generalization**.
> > > - We offer **practical hyperparameter-tuning guidelines** and insights for **training paradigm selection**.
> > >
> > > We would like to further clarify why our contributions are novel by highlighting the existing gaps (especially with [1]), summarizing our main contributions, and outlining the practical guidelines derived from our analysis (including actionable algorithmic implications).
> > >
> > > ### 1). Existing gap.
> > >
> > > **i). The limitations of [1]**: Although a recent empirical study [1] has attempted comparisons between these two paradigms, there exist two limitations:
> > >
> > > - **Limited generalizability**. The results provided by [1] are ***largely observational and hardware- or environment-dependent***, limiting their generalizability. For example, the convergence time may flip conclusions depending on hardware: mini-batch training converges faster than full-graph training under lower bandwidth but becomes slower under higher bandwidth (see discussion in Sec.5.1).
> > > - **Limited practical guidance.** [1] ***solely evaluates overall performance*** and ******ignores the interplay between batch size and fan-out size***,*** making it ***difficult to identify better trade-offs*** (e.g., faster convergence or improved generalization) under resource constraints. Therefore, ***[1] cannot provide detailed practical guidelines*** for hyperparameter tuning or training paradigm selection under specific requirements.
> > >
> > > This requires us to characterize how batch size and fan-out size affect GNN convergence and generalization. However,
> > >
> > > **ii). Conventional DNN training analyses have no concept of fan-out size**, as this concept arises specifically from neighbor sampling on graph-structured data.
> > >
> > > **iii). General GNN training analyses largely overlook the interplay between fan-out size and batch size**. For example, existing studies typically focus on ***individual parameters*** (e.g., batch size or fan-out size independently) [2,3] or ***singular aspects of evaluation*** (e.g., convergence or accuracy) [4,5], providing limited insights into the holistic trade-offs between full-graph and mini-batch paradigms.
> > >
> > > **iv). Most existing GNN theoretical analyses typically rely on strong simplifications**, such as infinite-width assumptions that average out per-neuron gradient noise [6] or linear models with convex losses that remove local optima [7], which obscure the effects of batch sizes or fan-out sizes on training dynamics.
> > >
> > > ### 2). Our main contributions
> > >
> > > Our main novelties are summarized as follows.
> > >
> > > **i). A unique GNN analysis of the interplay between batch size and fan-out size**. We characterize the roles of batch size and fan-out size in GNN optimization dynamics (Theorem 1 and 2) and generalization (Theorem 3), thereby addressing the existing gaps identified in points (ii) and (iii) above. Therefore, **this analysis provides a principled and unified framework for understanding the GNN behaviors under full-graph and mini-batch training paradigms.**
> > >
> > > **ii). Theoretical settings that better reflect practical GNN training**. For convergence, we extend the settings to irregular graphs and GNNs with non-linear activations. For generalization, we use the Wasserstein distance to investigate the impact of graph structures, especially the fan-out size, where this distance can also measure graph structure differences between training and testing datasets. **These settings enable a sharper analysis to precisely characterize how the batch size and fan-out size affect training**, thereby addressing the existing gap identified in points (iv).
> > >
> > > **iii). Key distinction**: We uncover **the non-isotropic impacts of the batch size and the fan-out size** in GNN convergence and generalization, where the batch size has a greater impact on GNN optimization dynamics (Obs.1), while the fan-out size more strongly affects GNN eneralization (Obs.2). **These findings cannot be captured by prior analyses that consider batch size and fan-out size independently, and provide practical guidelines in hyperparameter tuning and training paradigm selection.**
> > >
> > > ***Our key observations can provide valuable practical guidelines as below***.

---

> > > > ### Author Response · Authors · 2025-11-21
> > > > **Response (4/4)**
> > > >
> > > > ### 3). Practical guidelines
> > > >
> > > > **i). Hyperparameter tuning**
> > > >
> > > > 1. **Insights for tuning hyperparameters.** For datasets with an average degree below 50, we suggest keeping the batch size below half of the training nodes and the fan-out size under 15 to avoid generalization degradation while balancing computational efficiency and model performance. In addition, the hardware-agnostic iteration-to-accuracy is more reliable to guide **early-stage configuration decisions** in practice than time-to-accuracy. For example, in a new hardware setup, practitioners can refer to known iteration-to-accuracy trends to narrow down the promising range of batch size and fan-out size, and perform only short runs to consider hardware-specific runtime, which reduces tuning overhead.
> > > > 2. **Contribution to adaptive optimizer.** Our findings also suggest opportunities for developing adaptive optimizers that dynamically adjust hyperparameter configurations during GNN training. For example, an adaptive fan-out controller could allocate larger fan-out sizes in the early stages to accelerate convergence and gradually reduce them to moderate levels later to improve generalization. Meanwhile, an adaptive batch-size controller may adjust the batch size based on whether computational efficiency or generalization is prioritized under a given resource budget.
> > > >
> > > > **ii). Training paradigm selection.** Our theoretical and empirical findings support that full-graph training does not always yield superior model performance or computational efficiency compared to smaller mini-batch settings. Therefore, under resource constraints, carefully tuning the batch size and fan-out size in mini-batch settings often leads to better trade-offs, such as faster convergence or improved generalization, than full-graph training.
> > > >
> > > > Therefore, compared to [1], we can provide theoretical characterization, uncover the isotropic-impacts of batch size and fan-out size in convergence and generalization, and thereby offer actionable guidelines.
> > > >
> > > > We have mentioned these points in the manuscript and will further emphasize them in the revision to make the novelty clearer.
> > > >
> > > > Thank you again for your diligent review, and we hope our response has satisfactorily addressed your questions and concerns.
> > > >
> > > > [1] Bajaj, Saurabh, et al. "Graph Neural Network Training Systems: A Performance Comparison of Full-Graph and Mini-Batch." Proceedings of the VLDB Endowment 18.4 (2024): 1196-1209.
> > > >
> > > > [2] Yaochen Hu, Amit Levi, Ishaan Kumar, Yingxue Zhang, and Mark Coates. On batch-size selection
> > > > for stochastic training for graph neural networks. 2021.
> > > >
> > > > [3] Hao Yuan, Yajiong Liu, Yanfeng Zhang, Xin Ai, Qiange Wang, Chaoyi Chen, Yu Gu, and Ge Yu. Comprehensive evaluation of gnn training systems: A data management perspective. arXiv preprint arXiv:2311.13279, 2023.
> > > >
> > > > [4] Huayi Tang and Yong Liu. Towards understanding generalization of graph neural networks. In
> > > >
> > > > International Conference on Machine Learning, pp. 33674–33719, 2023.
> > > >
> > > > [5] Pranjal Awasthi, Abhimanyu Das, and Sreenivas Gollapudi. A convergence analysis of gradient descent on graph neural networks. Advances in Neural Information Processing Systems, 34: 20385–20397, 2021.
> > > >
> > > > [6] Naganand Yadati. A convex formulation for graph convolutional training: Two layer case. In 2022 IEEE International Conference on Data Mining (ICDM), pp. 1281–1286. IEEE, 2022.
> > > >
> > > > [7] Yucong Lin, Silu Li, Jiaxing Xu, Jiawei Xu, Dong Huang, Wendi Zheng, Yuan Cao, and Junwei Lu. Graph over-parameterization: Why the graph helps the training of deep graph convolutional network. Neurocomputing, 534:77–85, 2023.
> > > >
> > > > [8] Saurabh Verma and Zhi-Li Zhang. Stability and generalization of graph convolutional neural networks. In Proceedings of the 25th ACM SIGKDD International Conference on Knowledge Discovery & Data Mining, pp. 1539–1548, 2019.
> > > >
> > > > [9] Zheng, Da, et al. "DistDGL: Distributed graph neural network training for billion-scale graphs." 2020 IEEE/ACM 10th Workshop on Irregular Applications: Architectures and Algorithms (IA3). IEEE, 2020.

---

### Official Review · Reviewer_wVrk · 2025-10-31

**Soundness:** 3
**Presentation:** 2
**Contribution:** 3
**Rating:** 6
**Confidence:** 3

**Summary:**

This paper presents a comparative study between full-batch and mini-batch training in GNNs, in terms of model convergence, generalization and computatioal efficiency. Based on their theoretical analysis and by using hardware-agnostic measures, the authors aim to guide practitioners on selecting the two key parameters of mini-batch training: batch size and fan-out size (i.e. the number of neighbors to be sampled for each node in the batch).

**Strengths:**

1. This work sheds light on factors such as selecting between mini and full batch training and tuning subsequent hyperparamers that are often overlooked and decided without much thought but that impact model performance in multiple ways.

2. Insights stem from theoretical analysis and are backed by comprehensive experiments.

3. The paper is well-organized.

**Weaknesses:**

1. The figures are a little difficult to read and follow. For e.g. in Fig 1, it would be better to place the legend as a single horizontal line above or below the subplots instead of repeating it in each subplot and covering the information that needs to be displayed. Also, it would make the comparison easier to read if the bar grouping is done based on dataset rather than training setting, for example blue represents mini-batch and orange represents full batch, and the each groups of bars represents a dataset. The later heatmaps are also a little overwhelming with lots of numbers thrown in. Perhaps only the numbers that reveal crucial information can be retained and the general trends can be left to be depicted by the heatmap color. This would make it easier for the reader to pick up key insights more easily than going through a lot of data.

2. The number of layers of the GNNs in the experiments is restricted to 3, hardly enough to be considered 'deep', so oversmoothing does not seem to be a plausible explanation on lines 368 and 369.

3. Regarding 'observed low variance' of experiment results repeated thrice, while I trust the authors on this claim, it would still be better to provide some sort of evidence for it, in table format perhaps, in the appendix atleast as standard scientific practice.

**Questions:**

1. Is there likely any role of the homophily level of the input graph in determining what batch and fanout sizes would be better for training a GNN for the task?

2. While the authors provide results for three GNN architectures of different natures in their experiment, could they also explicitly comment on whether the same trend was observed across them or was their any variation? For e.g. is a particular architecture (or type thereof) more suited to a training paradigm?

---

> ### Author Response · Authors · 2025-11-21
> **Response (1/2)**
>
> ## W1: Figure representation
>
> Thanks for your suggestion. We have updated Figure 1 in Sec. 5 in the revision: the legend is now shown once per figure rather than repeated in each subplot, where blue represents mini-batch and orange represents full-batch training.
>
> Regarding the heatmaps, removing the numbers is an easy fix; we currently keep them to preserve sufficient details. If you strongly recommend reorganizing the figures, we will remove the numbers in the final version. We believe this is a minor presentation issue and does not affect the core contributions of our work.
>
> ## W2: Clarification on a brief experimental remark
>
> Thanks for pointing it out.
>
> We would like to clarify that this remark about increasing the number of layers is not the focus of our paper, and we believe it does not affect the core contributions of our work.
>
> We acknowledge your concern and agree that oversmoothing may not be a very appropriate explanation for the behavior of three-layer GNNs. To avoid potential confusion, we will remove that part and keep only the discussion on “more complex optimization dynamics”.
>
> ## W3: Evidence for the reported low variance
>
> Thanks for your suggestion. We agree that providing evidence is beneficial for scientific completeness. We have added tables in Appendix N, reporting the results of two additional runs for the key experiments discussed in the main text.
>
> For instance, two additional runs of Figure 4(b) in the original version are shown below, and the standard deviation of the final accuracy among three runs is less than 3.17%. These experiments are conducted on three-layer GraphSAGE model with ReLU function on the ogbn-product dataset under the MSE loss.  We use b as the batch size and $\beta$ as the fan-out size in the table.
>
> Run1:
>
> | Test acc | $\beta$=5 | $\beta$=10 | $\beta$=20 | $\beta$=50 | all neighbors |
> | --- | --- | --- | --- | --- | --- |
> | b=1000 | 0.6617 | 0.6891 | 0.7117 | 0.7241 | 0.7242 |
> | b=5000 | 0.7113 | 0.7207 | 0.7336 | 0.7345 | 0.7369 |
> | b=10000 | 0.7209 | 0.7292 | 0.7341 | 0.7344 | 0.7362 |
> | b=100000 | 0.7318 | 0.7348 | 0.7373 | 0.7403 | 0.7415 |
> | b=150000 | 0.7329 | 0.7357 | 0.7372 | 0.7378 | 0.7401 |
> | full batch | 0.7345 | 0.7391 | 0.7386 | 0.7384 | 0.7385 |
>
> Run2:
>
> | Test acc | $\beta$=5 | $\beta$=10 | $\beta$=20 | $\beta$=50 | all neighbors |
> | --- | --- | --- | --- | --- | --- |
> | b=1000 | 0.7295 | 0.7321 | 0.7344 | 0.7345 | 0.7341 |
> | b=5000 | 0.7307 | 0.7343 | 0.7361 | 0.7364 | 0.7371 |
> | b=10000 | 0.7326 | 0.7353 | 0.7366 | 0.7365 | 0.7381 |
> | b=100000 | 0.7342 | 0.7372 | 0.7392 | 0.7400 | 0.7411 |
> | b=150000 | 0.7343 | 0.7361 | 0.7385 | 0.7393 | 0.7405 |
> | full batch | 0.7341 | 0.7396 | 0.7391 | 0.7389 | 0.7403 |

---

> > ### Author Response · Authors · 2025-11-21
> > **Response (2/2)**
> >
> > ## Q1: Extension to the homophily level
> >
> > Thanks for your insightful question. We would clarify that our analysis is agnostic to the homophily level of the input graph. While homophily affects the properties of the ground-truth model parameters $W^*$ and the underlying network graph structure, our analysis is designed for arbitrary graph structures and focuses on the process of optimization dynamics (e.g., gradient update) affected by batch size and fan-out size. These dynamics do not directly depend on the specific content of the ground-truth model parameters or graph structures, so our core insights naturally carry over across different homophily levels.
> >
> > Specifically, the key difference across homophily levels lies in the content of aggregated neighbors rather than the amount of information exchanged during message passing. In convergence analysis, we bound the gradient norm using the amount of information aggregated during message passing. In generalization, we bound the performance using the Wasserstein distance, which is proportional to the structural difference between full-graph and mini-batch graphs. Therefore, our analyses and core insights naturally carry over to different homophily levels.
> >
> > Empirically, we conduct experiments on real-world datasets with different homophily levels. For example, the homophily levels of ogbn-products, Reddit, and ogbn-arxiv are 0.8076, 0.7555, and 0.6551, respectively. The results validate that our key insights remain applicable in such settings.
> >
> > Therefore, ***our theoretical and empirical naturally carry over to different homophily level settings for homogeneous graphs.***
> >
> > This is a promising direction for future work, especially understanding GNN behavior across graphs with diverse homophily levels.
> >
> > ## Q2: Discussion on three GNN architectures
> >
> > Thanks for your interesting question. Our analysis generalizes across different GNN architectures and the overall trends of convergence and generalization are consistent across the three GNN architectures (see Figure 5–8, 13-14 in Appendix N in the original version), which aligns with our theoretical analysis.
> >
> > Furthermore, we also observe some subtle differences in performance. For instance, GCN tends to achieve slightly better generalization performance under fan-out sizes closer to full-graph training, whereas GraphSAGE benefits more from incremental increases in fan-out size in mini-batch sampling. These differences likely stem from the inherent neighborhood aggregation schemes: GCN uses fixed-weight averaging of neighbors, which relies on a more denser sampling, while GraphSAGE employs learnable aggregation functions, allowing it to benefit progressively from incremental increases in neighbor.
> >
> > This suggests a promising avenue for future research, particularly in investigating how GNN architecture designs interact with different training paradigms.
> >
> > Thank you again for your diligent review, and we hope our response has satisfactorily addressed your questions and concerns.

---

> > > ### Comment · Reviewer_wVrk · 2025-11-23
> > >
> > > I thank the authors for their detailed response. My concerns have been addressed adequately, hence I would like to raise my score in favour of acceptance.

---

> > > > ### Author Response · Authors · 2025-11-24
> > > >
> > > > We are very glad to hear that you found our response satisfactory.
> > > > Thank you again for your kind consideration and thoughtful feedback. We truly appreciate the time and effort you dedicated to reviewing our work.
> > > >
> > > > We noticed that the system may not have registered the score update yet. When convenient, could you kindly check whether the score has been updated? We would be very grateful.

---

### Official Review · Reviewer_DYBQ · 2025-11-02

**Soundness:** 3
**Presentation:** 3
**Contribution:** 3
**Rating:** 8
**Confidence:** 4

**Summary:**

This paper presents a systematic empirical and theoretical analysis comparing full-graph and mini-batch training for GNNs. The authors propose using the dual lenses of batch size and fan-out size as a unified framework for this comparison, noting that full-graph training is a special case with maximal values for both parameters. The work aims to address a significant gap in the literature by providing a holistic understanding of how these hyperparameters impact optimization dynamics convergence and generalization, moving beyond isolated or purely hardware-dependent comparisons. The central, counter-intuitive claim supported by the study is that full-graph training does not always outperform well-tuned mini-batch training in terms of performance or efficiency.

**Strengths:**

1. The paper's core strength is its unified framework for analyzing full-graph and mini-batch training through batch and fan-out sizes.

2. The topic is of high importance to the GNN community. As graph datasets grow larger, the question of which training paradigm to choose and how to configure it is critical. This paper provides a foundational study that can guide future research in optimized training algorithms and systems

3. This paper is easy to understand.

**Weaknesses:**

1. While the theoretical analysis is a key contribution, the paper would be strengthened by a more explicit discussion of the assumptions made in the theorems (e.g., Theorem 1 & 2 on optimization dynamics). The claim of "better aligning with practice" by considering irregular graphs and non-linear activations is commendable, but the specific limitations and the gap between the theoretical model and real-world GNN training dynamics should be clarified to assess the generality of the conclusions.

2. The empirical evaluation is currently focused on transductive node classification. While this is a standard and important task, the findings' applicability to other fundamental GNN tasks like inductive learning, link prediction, or graph classification remains unverified. Different tasks may exhibit different sensitivities to batch and fan-out sizes, and demonstrating the generality of the findings across tasks would significantly increase the impact.

**Questions:**

Do the same conclusions hold for other types of tasks?

---

> ### Author Response · Authors · 2025-11-21
> **Response (1/2)**
>
> ## W1: Limitation Discussion
>
> Thanks for your question. We briefly mention the limitations of the theoretical setting in line 132-136 in the original version, and provide a more detailed discussion in Appendix H.
>
> The main limitation lies in the use of one-layer settings in theoretical analysis. We would like to briefly explain why we use one-layer settings and how our analysis extend to multi-layer settings.
>
> **1). Theoretical justification for using one-layer GNNs**: We would like to clarify that existing theoretical studies commonly use one-layer settings during GNN training, especially for GNNs with finite width and nonlinear activations [1,2] in practice.
>
> While some theoretical studies have explored multi-layer GNNs, they rely on strong simplifications (e.g., assuming linear models that remove local optima [3] or infinite-width that average out per-neuron gradient noise [4,5]), which tend to obscure practical training dynamics. These assumptions make the analysis tractable but less accurate.
>
> In contrast, our goal is to **precisely characterize how the batch size $b$ and fan-out size $\beta$ affect training**, which is better achieved using a one-layer, finite-width, nonlinear setting.
>
> Therefore, ***our setting aligns with standard theoretical practice and enables a sharper analysis than current multi-layer studies.***
>
> **2).** **Extension to multi-layer GNNs**: Our theoretical analysis readily extends to multi-layer GNNs, as long as each layer introduces only one non-linearity (e.g., ReLU activation). In such settings, **the key difference** is that the output of each layer is recursively defined based on the previous layer.
>
> This recursive definition preserves the same message-passing structure at each layer. In convergence analysis, we bound the gradient norms layer by layer; in generalization analysis, the pre-training loss discrepancy propagates across layers. These recursive structures allow our convergence and generalization bounds to translate naturally to multi-layer GNNs.
>
> Our key theoretical insights (from the view of $\beta$ and $b$) are generalizable to multi-layer GNNs. This is because adding more layers simply nests the same operations, without changing the qualitative roles of $\beta$ and b. Hence, the analytical trends observed in the one-layer case remain consistent.
>
> Empirically, we conduct experiments using multi-layer GNNs (see Figures 2–4), and the results validate that our key insights remain applicable in such settings.
>
> Therefore, ***our theoretical and empirical analyses support the multi-layer GNN settings.***

---

> > ### Author Response · Authors · 2025-11-21
> > **Response (2/2)**
> >
> > ## W2 and Q1: Extension to different tasks
> >
> > **1). Theoretical extension**: Thanks for your insightful question. ***We would like to clarify that our analysis and core insights can extend to inductive learning, link prediction, and graph classification tasks.***
> >
> > **Inductive learning**: The key difference from the transductive setting is that inductive tasks apply different graphs between testing and training. For convergence, our analysis is naturally applied to inductive tasks as the testing graphs are not involved. For generalization, our analysis can be easily extended to inductive tasks by revising the term $\delta^\text{full}$ in the Wasserstein distance to consider graph structure differences between testing and training graphs. Hence, our analyses and core insights then naturally carry over to inductive learning.
> >
> > **Link prediction**: Link prediction tasks can be naturally transformed to node classification task using the line graph method in the graph theory. The new line graph L(G) is constructed in the following way: for each edge in the original graph G, make a vertex in L(G); for every two edges in G that have a vertex in common, make an edge between their corresponding vertices in L(G). Hence, our analyses and core insights then naturally carry over to link prediction tasks.
> >
> > **Graph classification**: A graph-level prediction is obtained by first learning node representations and then pooling them into a single graph representation. Since the training dynamics before pooling are the same as in node classification, our analyses and key insights naturally carry over.
> >
> > **2). Empirical extension:** Thanks for your suggestion. We only add link prediction tasks below and vary the batch sizes due to the time limit. For these experiments, we used a three-layer GraphSAGE model with ReLU function on the ogbl-collab dataset under both CE and MSE losses.
> >
> > Due to the link prediction task, we use hit@100 as the generalization metric, thereby changing the related convergence metrics as iteration-to-hit@100 and time-to-hit@100.
> >
> > ***The results below support our main observations in the paper***: a larger batch size leads to opposite convergence trends under MSE and CE, while increasing batch size improve the generalization performance.
> >
> > CE loss:
> >
> > | batch size | 1024 | 16384 | 32768 | 65536 |
> > | --- | --- | --- | --- | --- |
> > | iteration-to-loss | 10273 | 4526 | 4766 | 4203 |
> >
> > | batch size | 1024 | 16384 | 32768 | 65536 |
> > | --- | --- | --- | --- | --- |
> > | iteration-to-hit@100 | 12570 | 2860 | 2440 | 2110 |
> >
> > | batch size | 1024 | 16384 | 32768 | 65536 |
> > | --- | --- | --- | --- | --- |
> > | time-to-hit@100 | 379.51 | 98.78 | 76.97 | 65.88 |
> >
> > | batch size | 1024 | 16384 | 32768 | 65536 |
> > | --- | --- | --- | --- | --- |
> > | test hit@100 | 0.5143 | 0.5688 | 0.5722 | 0.5720 |
> >
> > MSE loss:
> >
> > | batch size | 1024 | 16384 | 32768 | 65536 |
> > | --- | --- | --- | --- | --- |
> > | iteration-to-loss | 15452 | 15345 | 15706 | 18031 |
> >
> > | batch size | 1024 | 16384 | 32768 | 65536 |
> > | --- | --- | --- | --- | --- |
> > | iteration-to-hit@100 | 18850 | 3520 | 3550 | 3670 |
> >
> > | batch size | 1024 | 16384 | 32768 | 65536 |
> > | --- | --- | --- | --- | --- |
> > | time-to-hit@100 | 574.48 | 108.29 | 109.41 | 115.34 |
> >
> > | batch size | 1024 | 16384 | 32768 | 65536 |
> > | --- | --- | --- | --- | --- |
> > | test hit@100 | 0.4876 | 0.5647 | 0.5616 | 0.5684 |
> >
> > We will incorporate the above discussion in the revision.
> >
> > Thank you again for your diligent review, and we hope our response has satisfactorily addressed your questions and concerns.
> >
> > [1] “A convergence analysis of gradient descent on graph neural networks.”, Neural Information Processing Systems.
> >
> > [2] “Fast learning of graph neural networks with guaranteed generalizability: one-hidden-layer case”, International Conference on Machine Learning.
> >
> > [3] “A convex formulation for graph convolutional training: Two layer case”, ICDM
> >
> > [4] “How graph neural networks learn: Lessons from training dynamics in function space”, arxiv.
> >
> > [5] “Graph over-parameterization: Why the graph helps the training of deep graph convolutional network”, Neurocomputing.

---

### Author Response · Authors · 2025-11-22
**Summary of Revisions**

We thank all reviewers for their time and constructive feedback. Below is a summary of the revisions made:

1. We added the specific target losses and target accuracies for three-layer GNNs in Appendix N.
2. We moved Figures 5(j), 6(j), 7(i), 8(j), 13(b), and 14(b) from the appendix into the main text (now Figures 2–3) to improve readability.
3. We updated Figure 1 in Section 5: the legend now appears once per figure rather than in every subplot, with blue denoting mini-batch and orange denoting full-batch training.
4. To avoid potential confusion, we removed the term *oversmoothing* and retained only the discussion of “more complex optimization dynamics” in Section 5.2.
5. We added tables in Appendix N reporting results from two additional runs for the key experiments discussed in the main text.

All changes are highlighted in red. We sincerely thank the reviewers again for their careful and thoughtful evaluation.

---

### Author Response · Authors · 2025-11-30
**Executive Summary for ACs (1/2)**

Dear AC/SACs:

We would like to express our sincere gratitude to you and the reviewers for the time and effort you have dedicated to evaluating our work. We recognize the significant pressure and time constraints placed on Area Chairs, and greatly appreciate your commitment to a careful and thorough assessment. To facilitate a more efficient review process, we provide the following overview: (1) a summary of the paper and (2) a concise summary of the reviewers’ questions and our responses. We truly hope this structure simplifies your navigation of the rebuttal and helps you efficiently verify the revisions we have implemented.

### Summary of the Paper

This paper systemically compares full-graph and mini-batch training paradigms by understanding how batch size and fan-out size affect GNN convergence and generalization. Our key contributions include: (1). We provide the first unified GNN analysis that jointly characterizes the interplay between batch size and fan-out size. (2). We develop theoretical settings that better reflect practical GNN training, including irregular graphs, non-linear activations, and graph distribution shifts. (3). We uncover non-isotropic impacts of batch size and fan-out size on convergence and generalization. (4). We offer practical hyperparameter-tuning guidelines and insights for training paradigm selection.

We also provide detailed theoretical derivations in the Appendix and conduct comprehensive experiments on four real-world datasets with three GNN models to validate our findings. All code is made publicly available to ensure reproducibility.

### Summary of Reviewers’ Questions and Our Responses

We are pleased that all reviewers recognized the significance, rigor and practicality of our work. We are especially grateful that Reviewer **wVrk** engaged with our rebuttal, found our clarifications satisfactory, and subsequently raised the score. Below we summarize the key concerns raised by the reviewers and how we addressed them.

### Common Questions

1. **Difference between our work and [1] (VT8k, 463G)**:
    - **Concern**: The reviewers asked for a clarification of the new insights our work provides beyond the relevant work [1].
    - **Response**: We clarified that [1] provides only hardware-dependent, observational results of overall performance without generalizable guidance. In contrast, our work provides the first unified GNN analysis of how batch size and fan-out size affect convergence and generalization, reveals their non-isotropic effects, extends the theoretical settings, and offers actionable guidance for hyperparameter tuning and training paradigm selection.

[1] Bajaj, Saurabh, et al. "Graph Neural Network Training Systems: A Performance Comparison of Full-Graph and Mini-Batch." Proceedings of the VLDB Endowment 18.4 (2024): 1196-1209.

### Reviewer-Specific Concerns

**Reviewer DYBQ:**

1. **Limitation discussion:**
    - **Concern**: The reviewer asked for a discussion of the gap between our theoretical settings and real-world GNN training.
    - **Response**: We explained that the main gap lies in the use of one-layer settings in theoretical analysis, and clarified why we use this setting (to enable a shaper analysis) and how our analysis extends to multi-layer settings. The discussion is included in the manuscript.
2. **Extension to different tasks**
    - **Concern**: The reviewer asked whether our conclusions hold for other types of tasks (e.g., inductive learning, link prediction, graph classification).
    - **Response**: We clarified how our analysis and core insights naturally extend to these tasks, and provided empirical results on link prediction to support this claim.

---

> ### Author Response · Authors · 2025-11-30
> **Executive Summary for ACs (2/2)**
>
> **Reviewer wVrk:**
>
> 1. **Figure presentation**
>     - **Concern**: The reviewer suggested improving figure representation (e.g., the legend placement and bar grouping in Fig.1, and the detail reduction in the heatmaps).
>     - **Response**: We updated Fig.1 to improve clarity, and retained the numbers in the heatmaps to preserve detail, which can be easily removed if preferred.
> 2. **Clarification on a brief experimental remark**
>     - **Concern**: The reviewer expressed concern about the use of oversmoothing in the brief experimental remark.
>     - **Response**: We clarified that this remark is not the focus of our paper, and have removed it to avoid confusion, keeping only the discussion on “more complex optimization dynamics.”
> 3. **Evidence for the reported low variance**
>     - **Concern**: The reviewer requested evidence supporting the reported low variance in our repeated experiment results.
>     - **Response**: We added tables of repeated results for some key experiments in the appendix to ensure scientific completeness.
> 4. **Extension to the homophily level**
>     - **Concern**: The reviewer asked whether the homophily level of the input graph affects the choice of batch and fan-out sizes for GNN training.
>     - **Response**: We clarified that our analysis is agnostic to the homophily level of the input graph, and explained how our conclusions extend to homogeneous graphs with different homophily levels.
> 5. **Discussion on three GNN architectures**
>     - **Concern**: The reviewer asked whether the trends we observed are consistent across different GNN architectures.
>     - **Response**: We clarified that our analysis generalizes across GNN architectures, with overall trends in convergence and generalization remaining consistent. We also discussed some subtle performance differences among these GNN architectures.
>
> **Reviewer VT8k:**
>
> 1. **Extension to other mini-batch paradigms**
>     - **Concern**: The reviewer asked whether our conclusions hold for other mini-batch paradigms (e.g., graph-level and layer-level sampling).
>     - **Response**: We clarified that the node-level sampling is a common and fundamental mini-batch paradigm, as layer- and graph-level sampling are structured variants with effective batch and fan-out sizes. We further explained how our analysis and core insights extend to these paradigms, and confirmed this claim with experiments on LADIES and ClusterGCN.
>
> **Reviewer 463G:**
>
> 1. **Difference between our work and the general understanding of training analyses**
>     - **Concern**: The reviewer questioned how our findings differ from general understanding of GNN training and existing observations (e.g., DropEdge).
>     - **Response**: We clarified that our work addresses gaps ignored by prior studies (e.g., the interplay between batch size and fan-out size), and provides a unified GNN analysis revealing the non-isotropic impacts of these two hyperparameters. We further explained Fig.4 supports our main findings, and DropEdge cannot serve as a substitute for studying the impact of fan-out size when comparing two paradigms.
> 2. **Figure presentation**
>     - **Concern**: The reviewer suggested including figures cited in the main text instead of only referring to the appendix.
>     - **Response**: We clarified that the referenced figures are supplementary and not central to the analysis in Sec. 5.2-5.3, and we moved key figures into the main text to improve readability.
> 3. **Convergence settings**
>     - **Concern**: The reviewer asked for the specific values of target loss and target accuracy.
>     - **Response**: We provided the missing target values in the Appendix to ensure the scientific completeness.
> 4. **Setting and hyperparameters of Fig. 4(a)**
>     - **Concern**: The reviewer asked about the setting and hyperparameters of Fig.4(a).
>     - **Response**: We clarified where the setting and hyperparameters for Fig. 4(a) are reported in the Appendix.
> 5. **Iteration vs. Epoch**
>     - **Concern**: The reviewer asked whether it is fair to compare convergence based on iterations instead of epochs.
>     - **Response**: We clarified that using iterations is standard in mini-batch training, as each iteration corresponds to one stochastic gradient update, and epochs may not be well-defined under sampling with replacement. Measuring convergence per iteration isolates the effects of batch size, which would be confounded by using epochs.
>
> This summary outlines how we have thoroughly addressed all reviewer concerns and strengthened the manuscript. We are grateful to all reviewers for their thoughtful questions, comments, and suggestions. The additional experiments, extended discussions, and clarifications substantially enhance the paper’s clarity, rigor, and practical impact.

---

### Meta-Review · Area_Chair_vTBP · 2026-01-07

**Summary:**

All reviewers acknowledged the value of the theoretical characterization, the support provided by the careful experimentation, and the practical importance. Several concerns were raised.

C1. The empirical evaluation could be improved and more thorough: (a) focuses entirely on transductive node classification; (b) only considers GNNs of at most 3 layers; (c) there are claims of low variance of experimental results but this is not clearly supported by quantitative results; (d) only considers node-wise sampling.

C2. There needs to be more extensive discussion of the assumptions underpinning the theorems. The justification for conducting theoretical analysis with a single layer GNN could be improved.

C3. The main results have already been established through empirical works, so the main contribution is the theoretical framing.

**Reviewer Concerns:**

The rebuttal provided thorough and thoughtful responses to the concerns raised by the reviewers. The extent to which the concerns are successfully addressed varies.

C1. (a) The authors argue in their response that the results extend to inductive settings, link prediction, and graph classification. While the arguments are reasonable, the response would be much more compelling if there actually was an extension. Claims that results can be "easily extended" are easy to make and yet there can often be complications when there is a genuine attempt to perform the extension. If the extension is so straightforward, why not actually provide the results?

C1. b) The authors argue that "increasing the number of layers is not the focus of our paper". If the goal is to conduct a study to understand the role of batch-size and fan-out, with the intent of eventually providing practical guidelines ("We offer practical hyperparameter-tuning guidelines and insights for training paradigm selection"), then there is signficant value in verifying performance for models of practical depth (e.g., for GraphSAGE on the analyzed datasets the best performance is usually obtained with 4-7 layers).

C1. c) Variability can be assessed in many ways. When reporting test accuracies (Table 1), it would be better scientific practice to calculate confidence intervals (estimated via bootstrap, for example). It is extremely unlikely that the 10-90 confidence intervals will be so small that they are meaningless to report.  This would allow proper assessment of whether any of these differences are meaningful.

C1. d) The authors provide compelling arguments that the analysis can be readily extended to other types of sampling.

C2 & C3. The response addresses these issues convincingly.

**Reviewer Scores:**

Reviewer DYBQ (Score 8). UNLIKELY TO CHANGE. The score is already high.

Reviewer wVrk (Score 6; indicated increase to 8). INDICATED INCREASE. The reviewer stated an intention to increase the score.

Reviewer VT8k (Score 4) REASONABLY LIKELY TO INCREASE. The authors provided a thorough response to the reviewer and revised the paper to take account of the concerns. There is a good chance of an increased score.

Reviewer 463G (Score 4) REASONABLY LIKELY TO INCREASE. The authors provided a thorough response to the reviewer and revised the paper to take account of the concerns. There is a good chance of an increased score.

---

### Decision · Program_Chairs · 2026-01-26

Accept (Poster)